# SINGLE-LOOP BYZANTINE-RESILIENT FEDERATED BILEVEL OPTIMIZATION

**Yangnan Li[1], Shenghui Song [1], Xuanyu Cao[2]**

[1] The Hong Kong University of Science and Technology    [2] Washington State University
```
{ylity}@connect.ust.hk, {eeshsong}@ust.hk,
{xuanyu.cao}@wsu.edu
```

## ABSTRACT

Federated bilevel optimization plays a crucial role in solving complex problems with nested optimization structures. However, its distributed nature makes it highly susceptible to faulty or Byzantine behaviors. Existing Byzantine-resilient approaches are either restricted to simple single-level optimization problems or rely on sub-loop updates that introduce significant computational and communication overhead. To address these limitations, we propose a family of Byzantine-resilient federated bilevel algorithms, which (i) operate within a single-loop structure, (ii) achieve optimal Byzantine resilience, and (iii) ensure computational and communication efficiency. The core of the proposed method, BR-FedBi, leverages an auxiliary variable that facilitates efficient hypergradient estimation while simultaneously solving the lower- and upper-level problems. Building on BR-FedBi, we further integrate the algorithm with *Polyak's momentum* and the probabilistic gradient estimator (PAGE) (Li et al., 2021b), resulting in provable optimal Byzantine resilience and optimal sample complexity. Both theoretical analysis and empirical results demonstrate the superior performance of the proposed algorithms.

## 1 INTRODUCTION

Federated Bilevel Optimization (FBO) (Tarzanagh et al., 2022; Xiao & Ji, 2023) has recently emerged as a powerful framework for tackling machine learning problems with hierarchical structures in distributed settings. By distributing the computational workload across multiple clients, it preserves local datasets' privacy and mitigates the computation burden on central server (Haddadpour et al., 2019; Li et al., 2020). Practical applications range from federated meta-learning (Finn et al., 2017; Hu et al., 2021) and hyperparameter optimization (Franceschi et al., 2018; Akiba et al., 2019) to imitation learning and federated reinforcement learning (Hussein et al., 2017; Li, 2017), demonstrating the versatility of bilevel formulations. The general formulation of the federated bilevel optimization problem is expressed as:

$$\min_{\mathbf{x} \in \mathbb{R}^p} \Phi(\mathbf{x}) = \frac{1}{n} \sum_{i=1}^{n} f_i(\mathbf{x}, \mathbf{y}^\star(\mathbf{x})) = \frac{1}{n} \sum_{i=1}^{n} \mathbb{E}_{\boldsymbol{\phi}_i}[F_i(\mathbf{x}, \mathbf{y}^\star(\mathbf{x}); \boldsymbol{\phi}_i)],$$

$$\text{subject to} \quad \mathbf{y}^\star(\mathbf{x}) = \arg\min_{\mathbf{y} \in \mathbb{R}^q} \frac{1}{n} \sum_{i=1}^{n} g_i(\mathbf{x}, \mathbf{y}) = \frac{1}{n} \sum_{i=1}^{n} \mathbb{E}_{\boldsymbol{\varphi}_i}[G_i(\mathbf{x}, \mathbf{y}; \boldsymbol{\varphi}_i)],$$

(1)

where $\mathbf{x} \in \mathbb{R}^p$ denotes the upper-level variable, $\mathbf{y} \in \mathbb{R}^q$ represents the lower-level variable, and $n$ denotes the number of clients. Each client $i$ possesses a local upper-level loss function $f_i(\mathbf{x}, \mathbf{y})$ and a lower-level loss function $g_i(\mathbf{x}, \mathbf{y})$. We consider objective functions expressed in expectation form, where $\boldsymbol{\phi}_i$ and $\boldsymbol{\varphi}_i$ represent random local data samples of client $i$ which usually follow unknown and heterogeneous distributions across different clients.

Despite its effectiveness, federated bilevel learning also faces critical challenges, including increased communication costs (Cao et al., 2023; He et al., 2024), privacy concerns (Wei et al., 2020), and vulnerability to attacks (Baruch et al., 2019; Xie et al., 2020). In this paper, we focus on a crucial aspect of federated bilevel learning: Byzantine resilience - the capability of distributed systems to maintain reliability in the presence of Byzantine attackers (Karimireddy et al., 2020a; Allouah

et al., 2024). Specifically, during the training process, some anomalous participants (i.e. Byzantine attackers) may submit corrupted or malicious information to the central server, thereby compromising the accuracy of the global model. These abnormal behaviors can result from hardware or software malfunctions (Armstrong, 2003), data poisoning (Zhang et al., 2020; Yerlikaya & Bahtiyar, 2022), or adversarial entities controlling parts of the system (Baruch et al., 2019).

Numerous prior works have focused on defending against Byzantine attackers, leading to the development of robust algorithms in both theoretical and practical contexts (Blanchard et al., 2017; Karimireddy et al., 2021; Gorbunov et al., 2022). However, most of the works primarily focused on single-level optimization, while advanced tasks in bilevel optimization involve inherently nested formulations, which extend beyond the capabilities of single-level approaches. Solving bilevel optimization problems in the presence of Byzantine attackers is challenging. In federated bilevel optimization, the computation of global hypergradient is not a straightforward aggregation of local hypergradients. Additionally, data heterogeneity unavoidably prevents any aggregation algorithm from reaching the optimal point with $\delta$-proportion of Byzantine attackers (Karimireddy et al., 2020a; Allouah et al., 2023). Due to the hierarchical structure of FBO, the asymptotic error becomes further amplified as the heterogeneity level increases in both the lower- and upper-level objectives.

In this paper, we aim to bridge the gap between federated bilevel optimization and Byzantine resilience by addressing the following research questions: *(i) What is the lower bound of the asymptotic error caused by Byzantine attackers in bilevel optimization? (ii) Can any bilevel algorithm achieve this lower bound and attain optimal Byzantine resilience?* To answer these questions, we propose a family of single-loop Byzantine-resilient bilevel algorithms that facilitate efficient hypergradient estimation while ensuring high efficiency in both communication and computation. Our key contributions are summarized as follows:

- **Lower bound for asymptotic error.** We derive an algorithm-independent lower bound on the asymptotic error induced by Byzantine attackers in bilevel optimization. This result highlights the fundamental limitations imposed by heterogeneity and Byzantine attackers, extending existing lower bounds from single-level optimization to the more complex bilevel setting.

- **Achievable optimal Byzantine resilience.** We propose a single-loop Byzantine-resilient bilevel algorithm, BR-FedBi, that eliminates the need for frequent gradient exchanges, significantly improving both efficiency and robustness. Building upon the framework of BR-FedBi, we further incorporate *Polyak's momentum* (Polyak, 1964) to match the established lower bound, achieving optimal Byzantine resilience and demonstrating the tightness of the theoretical lower bound.

- **Communication and computational efficiency.** BR-FedBi achieves a communication complexity of $\mathcal{O}(\epsilon^{-1})$ and demonstrates linear speedup with respect to the number of *honest* clients. Furthermore, by incorporating PAGE variance reduction techniques (Li et al., 2021b), the algorithm attains a sample complexity of $\mathcal{O}(\epsilon^{-1.5})$ under the mean-squared smoothness assumption, matching the optimal sample complexity in general expectation setting.

- **Experiment evaluation.** Experimental results on the hyper-representation task demonstrate that the proposed methods significantly outperform existing approaches under various Byzantine attack scenarios, in both homogeneous and heterogeneous settings.

## 1.1 RELATED WORK

**Asymptotic error for Byzantine-resilient distributed learning.** The convergence behavior of Byzantine-resilient distributed learning has been extensively studied over the years (Karimireddy et al., 2021; Farhadkhani et al., 2023; Allouah et al., 2024). More recently, research has focused on analyzing the convergence behavior in heterogeneous settings (Karimireddy et al., 2020a; Allouah et al., 2023; Liu et al., 2023; Murata et al., 2024). In this line of work, Karimireddy et al. (2020a) was the first to establish the lower bound on the asymptotic error, given by $\Omega\left(\delta\zeta^2\right)$. Building on this foundation, Allouah et al. (2023) proposed Nearest-Neighbor Mixing (NNM) algorithm, showing that most robust aggregation methods, when combined with NNM, are able to achieve an asymptotic error of $\mathcal{O}\left(\delta\zeta^2\right)$. Furthermore, Murata et al. (2024) introduced the Momentum Screening method and demonstrated that, under the assumption of maximum gradient heterogeneity $\zeta_{\max}$, this method achieves a smaller asymptotic error of $\mathcal{O}(\delta^2\zeta_{\max}^2)$ when the condition $\delta \leq (\zeta/\zeta_{\max})^2$ is satisfied. Despite these advancements, research on bilevel optimization remains limited.

**Federated bilevel optimization.** Unlike single-level optimization, bilevel optimization involves a nested structure, making hypergradient computation particularly challenging. Various techniques

have been developed to estimate the hypergradient, including approximate implicit differentiation (Domke, 2012; Grazzi et al., 2020), iterative differentiation (Maclaurin et al., 2015; Franceschi et al., 2018), and Neumann series expansion (Chen et al., 2023; Hong et al., 2023). Recent efforts have focused on designing efficient bilevel optimization algorithms for federated learning and decentralized learning. For instance, Tarzanagh et al. (2022) introduced FedNest, a federated bilevel algorithm based on approximate implicit differentiation (AID), while Xiao & Ji (2023) proposed an iterative differentiation (ITD) based approach that reduces communication costs. Dagréou et al. (2022) introduced SOBA, a novel single-loop framework for solving bilevel optimization problems, and Yang et al. (2024) extended their work to federated learning. Recent studies have incorporated communication compression (He et al., 2024) and differential privacy (Chen & Wang, 2024) into federated bilevel optimization. However, research on Byzantine-resilient federated bilevel learning remains scarce.

The most related work is (Abbas et al., 2024), which proposed a Byzantine-resilient federated bilevel optimization algorithm leveraging an AID-based method. However, this approach involves sub-loop computations of hypergradients and requires multiple communication rounds per iteration, leading to high communication costs and increased vulnerability to attackers. In contrast, our study introduces a single-loop algorithm with reduced communication and computational overhead, along with a theoretical guarantee for achieving optimal Byzantine resilience.

## 2 PRELIMINARIES

### 2.1 PROBLEM STATEMENT

We consider a distributed system with $n$ clients, $[n] = \{1, ..., n\}$, where at most $\delta < 1/2$ proportion of them are Byzantine attackers. Such attackers have the ability to access honest gradients and may send well-crafted malicious updates to the server. The objective of the honest clients is to minimize global *honest* loss of the bilevel optimization problem, defined as $f_{\mathcal{H}}(\mathbf{x}, \mathbf{y}^{\star}(\mathbf{x}))$, where $f_{\mathcal{H}}(\mathbf{x}, \mathbf{y}(\mathbf{x})) = \frac{1}{|\mathcal{H}|} \sum_{i \in \mathcal{H}} f_i(\mathbf{x}, \mathbf{y}(\mathbf{x}))$ and $\mathcal{H} \subseteq [n]$ represents the set of honest clients. Typically, we consider a Byzantine-resilient version of the problem (1):

$$\min_{\mathbf{x} \in \mathbb{R}^p} \Phi_{\mathcal{H}}(\mathbf{x}) = \frac{1}{|\mathcal{H}|} \sum_{i \in \mathcal{H}} f_i(\mathbf{x}, \mathbf{y}^{\star}(\mathbf{x})),$$
$$\text{subject to} \quad \mathbf{y}^{\star}(\mathbf{x}) = \arg \min_{\mathbf{y} \in \mathbb{R}^q} g_{\mathcal{H}}(\mathbf{x}, \mathbf{y}), \tag{2}$$

where $g_{\mathcal{H}}(\mathbf{x}, \mathbf{y}) := \frac{1}{|\mathcal{H}|} \sum_{i \in \mathcal{H}} g_i(\mathbf{x}, \mathbf{y})$, and $\Phi_i(\mathbf{x}) := f_i(\mathbf{x}, \mathbf{y}^{\star}(\mathbf{x}))$. The goal of Byzantine-resilient algorithms is to minimize honest global loss and achieve an $\epsilon$-stationary point, even when a $\delta$-proportion of the workers are Byzantine attackers. We formalize this concept in Definition 1, which introduces the notion of $(\delta, \epsilon)$-Byzantine-resilient.

**Definition 1 ( $(\delta, \epsilon)$-Byzantine-resilient).** *A bilevel algorithm is considered to be $(\delta, \epsilon)$-Byzantine-resilient if, despite the presence of $\delta$-proportion of Byzantine attackers, it outputs parameter $\hat{\mathbf{x}}$ such that*

$$\mathbb{E}\left[\|\nabla \Phi_{\mathcal{H}}(\hat{\mathbf{x}})\|^2\right] \leq \epsilon,$$

*where $\Phi_{\mathcal{H}}(\hat{\mathbf{x}}) := \frac{1}{|\mathcal{H}|} \sum_{i \in \mathcal{H}} \Phi_i(\hat{\mathbf{x}})$ denotes the global loss.*

The key to achieving Byzantine resilience lies in employing robust aggregators (Blanchard et al., 2017; Pillutla et al., 2022; Murata et al., 2024), which are designed to maintain comparable performance even when a fraction of the workers are Byzantine attackers. To evaluate the effectiveness of these robust aggregators, we adopt the $(\delta, \kappa)$-robustness criterion, as proposed by Allouah et al. (2023).

**Definition 2 ( $(\delta, \kappa)$-robustness).** *Let $\delta < \frac{1}{2}$ and $\kappa \geq 0$. A robust aggregation rule $AGG(\cdot) : \mathbb{R}^{d \times n} \to \mathbb{R}^d$ is said to be $(\delta, \kappa)$-robustness if, for any vectors $\mathbf{D}_1, \ldots, \mathbf{D}_n \in \mathbb{R}^d$ and any set $S \subseteq [n]$ of size $(1 - \delta)n$, the following holds:*

$$\left\|AGG(\mathbf{D}_1, \ldots, \mathbf{D}_n) - \bar{\mathbf{D}}_S\right\|^2 \leq \frac{\kappa}{|S|} \sum_{i \in S} \left\|\mathbf{D}_i - \bar{\mathbf{D}}_S\right\|^2,$$

*where $\bar{\mathbf{D}}_S = \frac{1}{|S|} \sum_{i \in S} \mathbf{D}_i$. The term $\kappa$ is referred to as the robustness coefficient.*

This criterion provides a tight convergence guarantee and is satisfied by most robust aggregation methods. Notably, $\kappa$ has a theoretical lower bound given by $\kappa > \frac{\delta}{1-2\delta}$, which implies that no robust aggregator can achieve arbitrarily close proximity to the average output of the honest workers.

## 2.2 ASSUMPTIONS

The following assumptions are used throughout this paper. They are standard assumptions made in the literature on bilevel optimization (Chu et al., 2024; He et al., 2024) and federated learning (Allouah et al., 2023).

**Assumption 1 (Lipschitz Continuity).** For any honest worker $i \in \mathcal{H}$, the upper-level loss function $f_i$ is $C_f$-Lipschitz continuous *w.r.t.* $\mathbf{y}$. The gradients and Hessian matrices $\nabla f_i, \nabla g_i, \nabla^2_{\mathbf{xy}} g_i, \nabla^2_{\mathbf{yy}} g_i$ are Lipschitz continuous with constants $L_f, L_g, L_g, L_g$, respectively.

**Assumption 2 (Strong Convexity).** For any honest worker $i \in \mathcal{H}$, the lower-level loss $g_i$ is $\mu_g$-strongly convex *w.r.t.* $\mathbf{y}$.

**Assumption 3 (Bounded Variance).** For any honest worker $i \in \mathcal{H}$, there exists $\sigma \geq 0$, such that the stochastic gradients and Hessians $\nabla F_i(\mathbf{x}, \mathbf{y}; \phi_i), \nabla_{\mathbf{y}} G_i(\mathbf{x}, \mathbf{y}; \varphi_i), \nabla^2_{\mathbf{xy}} G_i(\mathbf{x}, \mathbf{y}; \varphi_i)$ and $\nabla^2_{\mathbf{yy}} G_i(\mathbf{x}, \mathbf{y}; \varphi_i)$ are unbiased with bounded variance $\sigma^2$.

**Assumption 4 ($\zeta$-gradient/Hessian dissimilarity).** The local gradient/Hessian $\Lambda_i$ of the honest worker $i \in \mathcal{H}$ is said to satisfy $\zeta$-gradient/Hessian dissimilarity if

$$\frac{1}{|\mathcal{H}|} \sum_{i \in \mathcal{H}} \left\| \Lambda_i - \frac{1}{|\mathcal{H}|} \sum_{i \in \mathcal{H}} \Lambda_i \right\|^2 \leq \zeta^2.$$

- The gradients $\nabla \Phi(\mathbf{x})$ and $\nabla f_i(\mathbf{x}, \mathbf{y})$ of honest workers satisfy the $\zeta_f$-gradient dissimilarity assumption.
- The gradient/Hessians $\nabla_{\mathbf{y}} g_i(\mathbf{x}, \mathbf{y}), \nabla^2_{\mathbf{xy}} g_i(\mathbf{x}, \mathbf{y})$ and $\nabla^2_{\mathbf{yy}} g_i(\mathbf{x}, \mathbf{y})$ of honest workers satisfy the $\zeta_g$-gradient/Hessian dissimilarity assumption.

## 3 LOWER BOUND FOR ASYMPTOTIC ERROR

In Byzantine-resilient distributed learning, it is inherently impossible to fully eliminate the asymptotic error due to data heterogeneity. In this section, we derive the lower bound on the asymptotic error induced by Byzantine attackers in bilevel optimization, illustrating that both lower- and upper-level data heterogeneity contribute to the asymptotic error.

The analysis of the lower bound is performed with a slightly more general setting. Specifically, we replace Assumption 4 with a more general heterogeneity condition, namely the $(\zeta, B)$-gradient/Hessian dissimilarity assumption (Assumption 8 in Appendix E). When $B = 0$, Assumption 8 degenerates to Assumption 4, thus recovering the original setting as a special case.

**Theorem 1.** *For an arbitrary federated bilevel algorithm, there always exists a federated bilevel optimization problem satisfying Assumptions 1-3 and 8, such that:*

$$\|\nabla \Phi_{\mathcal{H}}(\hat{\mathbf{x}})\|^2 \geq \Omega \left( \frac{\delta \left( \zeta_f^2 + \zeta_g^2 \right)}{1 - \delta \left( 2 + B_f^2 \right)} \right), \tag{3}$$

*where $\delta < \min \left\{ \frac{1}{2+B_f^2}, \frac{1}{2+B_g^2} \right\}$ is the proportion of attackers and $\hat{\mathbf{x}}$ is the output of the algorithm.*

**Remark 1.** The above lower bound generalizes the results previously established for single-level optimization problems. Specifically, by removing the term related to $\zeta_g$, we obtain the bound $\Omega \left( \frac{\delta \zeta_f^2}{1-\delta(2+B_f^2)} \right)$, which matches the lower bound for single-level optimization in (Allouah et al., 2024). Moreover, under the $(\zeta, B)$-gradient dissimilarity assumption, the breakdown point for the proportional of attackers is given by $\delta < \min \left\{ \frac{1}{2+B_f^2}, \frac{1}{2+B_g^2} \right\}$, indicating that it is not a fixed threshold of $\frac{1}{2}$ but depends on the heterogeneity coefficients at both the lower- and upper-levels.

**Corollary 1.** *For an arbitrary federated bilevel algorithm, there always exists a federated bilevel optimization problem satisfying Assumptions 1-4, such that:*

$$\|\nabla \Phi_{\mathcal{H}}(\hat{\mathbf{x}})\|^2 \geq \Omega \Big( \delta \left( \zeta_f^2 + \zeta_g^2 \right) \Big), \tag{4}$$

where $\delta < \frac{1}{2}$ is the proportion of attackers and $\hat{\mathbf{x}}$ is the output of the algorithm.

**Remark 2.** Corollary 1 can be viewed as a special case of Theorem 1 by setting $B_f = B_g = 0$. Notably, Karimireddy et al. (2020a) established a lower bound of $\Omega(\delta\zeta^2)$ for single-level optimization. Corollary 1 extends this result to the bilevel optimization setting, highlighting that the inherent difficulty introduced by heterogeneity and Byzantine clients persists even in more complex hierarchical optimization problems.

## 4  SINGLE-LOOP BYZANTINE-RESILIENT FEDERATED BILEVEL ALGORITHM

In this section, we propose the first Single-Loop Byzantine-resilient Federated Bilevel Algorithm (BR-FedBi), which extends the SOBA framework (Dagréou et al., 2022). The algorithm employs an auxiliary variable to enable efficient hypergradient estimation while preserving the robustness of the distributed system against potential Byzantine attackers.

### 4.1  DESCRIPTION OF BR-FEDBI

A commonly employed strategy for bilevel optimization problems makes use of implicit differentiation (Griewank & Walther, 2008). In Byzantine-resilient federated bilevel optimization (BR-FBO), the expression of the hypergradient $\nabla\Phi_{\mathcal{H}}(\mathbf{x}, \mathbf{y}^{\star}(\mathbf{x}))$ can be written as:

$$\frac{1}{|\mathcal{H}|}\sum_{i\in\mathcal{H}}\nabla_{\mathbf{x}}f_i\left(\mathbf{x},\mathbf{y}^{\star}(\mathbf{x})\right) - \nabla_{\mathbf{xy}}^2 g_{\mathcal{H}}\left(\mathbf{x},\mathbf{y}^{\star}(\mathbf{x})\right)\cdot\left[\nabla_{\mathbf{yy}}^2 g_{\mathcal{H}}\left(\mathbf{x},\mathbf{y}^{\star}(\mathbf{x})\right)\right]^{-1}\cdot\frac{1}{|\mathcal{H}|}\sum_{i\in\mathcal{H}}\nabla_{\mathbf{y}}f_i\left(\mathbf{x},\mathbf{y}^{\star}(\mathbf{x})\right),$$

where the global Hessian matrices can be written as:

$$\nabla_{\mathbf{xy}}^2 g_{\mathcal{H}}\left(\mathbf{x},\mathbf{y}^{\star}(\mathbf{x})\right) = \frac{1}{|\mathcal{H}|}\sum_{i\in\mathcal{H}}\nabla_{\mathbf{xy}}^2 g_i(\mathbf{x},\mathbf{y}^{\star}(\mathbf{x})), \quad \nabla_{\mathbf{yy}}^2 g_{\mathcal{H}}\left(\mathbf{x},\mathbf{y}^{\star}(\mathbf{x})\right) = \frac{1}{|\mathcal{H}|}\sum_{i\in\mathcal{H}}\nabla_{\mathbf{yy}}^2 g_i(\mathbf{x},\mathbf{y}^{\star}(\mathbf{x})).$$

**Hypergradient estimation.** We extend the SOBA framework (Dagréou et al., 2022) to develop a single-loop solution for BR-FBO, which introduces an auxiliary variable $\mathbf{z}^{\star}(\mathbf{x})$ to approximate the Hessian-inverse-vector product:

$$\mathbf{z}^{\star}(\mathbf{x}) := -\left[\nabla_{\mathbf{yy}}^2 g_{\mathcal{H}}(\mathbf{x},\mathbf{y}^{\star})\right]^{-1}\cdot\frac{1}{|\mathcal{H}|}\sum_{i\in\mathcal{H}}\nabla_{\mathbf{y}}f_i\left(\mathbf{x},\mathbf{y}^{\star}\right),$$

where $\mathbf{z}^{\star}$ can be seen as the solution of the global linear system: $\nabla_{\mathbf{yy}}^2 g_{\mathcal{H}}(\mathbf{x},\mathbf{y}^{\star})\mathbf{z} + \nabla_{\mathbf{y}}f_{\mathcal{H}}(\mathbf{x},\mathbf{y}^{\star}) = 0$. Although computing the exact $\mathbf{z}^{\star}$ is infeasible in practice, an approximation can be obtained by minimizing the following objective:

$$\frac{1}{|\mathcal{H}|}\sum_{i\in\mathcal{H}}\left\{\frac{1}{2}\mathbf{z}^{\top}\nabla_{\mathbf{yy}}^2 g_i(\mathbf{x},\mathbf{y})\mathbf{z} + \mathbf{z}^{\top}\nabla_{\mathbf{y}}f_i(\mathbf{x},\mathbf{y})\right\}.$$

The auxiliary variable $\mathbf{z}$ enables the estimation of the global hypergradient using the expression: $\frac{1}{|\mathcal{H}|}\sum_{i\in\mathcal{H}}\nabla_{\mathbf{x}}f_i(\mathbf{x},\mathbf{y}) + \nabla_{\mathbf{xy}}^2 g_{\mathcal{H}}(\mathbf{x},\mathbf{y})\mathbf{z}$, where $\mathbf{y}$ represents the approximate solution to the lower-level problem.

As a result, hypergradient estimation can be decomposed into three steps: (i) solving the upper-level optimization problem, (ii) solving the lower-level optimization problem, and (iii) estimating the auxiliary variable of Hessian-inverse-vector product $\mathbf{z}$. Next, we provide the detailed procedure.

**Algorithm procedure.** The full procedure of BR-FedBi is summarized in Algorithm 1 in Appendix C. In the $k$-th communication round, the server distributes the global parameter $\{\mathbf{x}^k, \mathbf{y}^k, \mathbf{z}^k\}$ to the clients. Each honest client $i\in\mathcal{H}$ computes the gradients using their locally sampled data, $\phi_i^k\sim\mathcal{D}_{f_i}$ and $\varphi_i^k\sim\mathcal{D}_{g_i}$, and transmits them to the server:

$$\mathbf{D}_{\mathbf{x},i}^k = \nabla_{\mathbf{xy}}^2 G_i(\mathbf{x}^k,\mathbf{y}^k;\varphi_i^k)\mathbf{z}^k + \nabla_{\mathbf{x}}F_i(\mathbf{x}^k,\mathbf{y}^k;\phi_i^k), \tag{5a}$$

$$\mathbf{D}_{\mathbf{y},i}^k = \nabla_{\mathbf{y}}G_i(\mathbf{x}^k,\mathbf{y}^k;\varphi_i^k), \tag{5b}$$

$$\mathbf{D}_{\mathbf{z},i}^k = \nabla_{\mathbf{yy}}^2 G_i(\mathbf{x}^k,\mathbf{y}^k;\varphi_i^k)\mathbf{z}^k + \nabla_{\mathbf{y}}F_i(\mathbf{x}^k,\mathbf{y}^k;\phi_i^k). \tag{5c}$$

On the other hand, each Byzantine attacker may send an arbitrary value $*$ to the server. Upon receiving the gradients from all the clients (including honest clients and attackers), the server employs a robust aggregation rule $AGG(\cdot): \mathbb{R}^{d\times n}\to\mathbb{R}^d$ and updates the global parameters $\diamond\in\left\{\mathbf{x}^k,\mathbf{y}^k,\mathbf{z}^k\right\}$ with learning rate $\eta_{\mathbf{x}},\eta_{\mathbf{y}},\eta_{\mathbf{z}} > 0$:

$$\diamond^{k+1} = \diamond^k - \eta_\diamond AGG\left(\mathbf{D}_{\diamond,1}^k, \ldots, \mathbf{D}_{\diamond,n}^k\right). \tag{6}$$

Before broadcasting $\mathbf{z}^{k+1}$ to the local clients, a projection operation is applied to ensure $\mathbf{z}^{k+1}$ lies within an $\ell_2$-ball of radius $\rho$, which is defined as:

$$\mathcal{P}_\rho\left(\mathbf{z}^{k+1}\right) = \min\left\{1, \rho/\|\mathbf{z}^{k+1}\|_2\right\} \cdot \mathbf{z}^{k+1}.$$

This projection ensures smoothness of the linear system and maintains the boundedness of the gradient updates for the variables $\mathbf{x}$ and $\mathbf{z}$, both of which are essential for convergence analysis.

### 4.2 CONVERGENCE ANALYSIS

To establish convergence of BR-FedBi in Algorithm 1, the following additional assumptions are required. All proofs are relegated to Appendix G.

**Assumption 5** (Boundedness of gradient terms). There exists a constant $H_\mathbf{x} \geq 0$ such that, for any $\left(\mathbf{x}^k, \mathbf{y}^k, \mathbf{z}^k\right)$ produced during the iterations in Algorithm 1, the following inequality holds:

$$\left\|\nabla_{\mathbf{xy}}^2 g_\mathcal{H}\left(\mathbf{x}^k, \mathbf{y}^k\right)\mathbf{z}^k + \nabla_\mathbf{x} f_\mathcal{H}\left(\mathbf{x}^k, \mathbf{y}^k\right)\right\|_2^2 \leq H_\mathbf{x}^2.$$

Note that this assumption is milder than the Lipschitz continuity of $f_\mathcal{H}$ w.r.t. $\mathbf{x}$ in previous works (Lu et al., 2022) and is similar to the one in (He et al., 2024).

**Assumption 6.** Hessian matrices $\nabla_{\mathbf{xy}}^2 g, \nabla_{\mathbf{yy}}^2 g$ are $L_g$-smooth, and $\nabla^2 f$ is $L_f$-Lipschitz continuous.

**Theorem 2.** *Let $\delta < 1/2$. Suppose Assumptions 1-4 and 5-6 hold, and $AGG(\cdot)$ satisfies the $(\delta, \kappa)$-robustness criterion. Set the learning rates as: $\eta_\mathbf{x} = \min\left\{\alpha_1, \sqrt{\frac{\Delta_\Phi^0}{A_\sigma K}}\right\}, \eta_\mathbf{y} = c_\mathbf{y}\eta_\mathbf{x}, \eta_\mathbf{z} = c_\mathbf{z}\eta_\mathbf{x}$, where $A_\sigma := A_0 \cdot \sigma^2/((1-\delta)n)$ and $A_0, \alpha_1, c_\mathbf{y}, c_\mathbf{z}, \Delta_\Phi^0$ are some positive constants independent of $K, \delta$. Then, BR-FedBi converges as:*

$$\mathbb{E}\left[\|\nabla\Phi_\mathcal{H}(\hat{\mathbf{x}})\|^2\right] = \mathcal{O}\left(\sqrt{\frac{\sigma^2\Delta_\Phi^0}{(1-\delta)nK}}\right) + \mathcal{O}\left(\frac{\Delta_\Phi^0}{K}\right) + \mathcal{O}\left(\kappa\left(\zeta_f^2 + \zeta_g^2 + \sigma^2\right)\right), \tag{7}$$

*where the expectation is over the algorithm's randomness.*

**Sample Complexity.** BR-FedBi achieves a linear speedup with respect to the number of honest clients $(1 - \delta)n$. The dominant term, $\mathcal{O}(1/\sqrt{nK})$, demonstrates that increasing the number of participating honest workers improves the convergence rate. This leads to a sample complexity of $\mathcal{O}\left(\epsilon^{-2}n^{-1}\right)$, which aligns with the standard convergence rate of distributed stochastic gradient descent (D-SGD) for non-convex loss functions (Karimireddy et al., 2020b).

**Asymptotic error.** Theorem 2 shows a non-vanishing asymptotic error of $\mathcal{O}\left(\kappa\left(\zeta_f^2 + \zeta_g^2 + \sigma^2\right)\right)$ for BR-FBO, indicating that both lower-level and upper-level heterogeneity contribute to the aggregation error. Moreover, the error is influenced by a variance-related term, $\mathcal{O}\left(\kappa\sigma^2\right)$, where $\sigma^2$ is the noise variance of the stochastic gradients and Hessian matrices for both the upper-level and lower-level loss functions. This variance-related dependency limits the algorithm's ability to achieve optimal Byzantine resilience.

We further instantiate Theorem 2 by selecting the minimal robustness parameter $\kappa = \mathcal{O}(\delta)$, which yields an explicit bound on the optimization error $\epsilon$, as shown in Corollary 2.

**Corollary 2.** *Suppose $AGG(\cdot)$ satisfies the $(\delta, \kappa)$-robustness criterion with $\kappa = \mathcal{O}(\delta)$. Then, BR-FedBi is $(\delta, \epsilon)$-Byzantine-resilient with: $\epsilon = \mathcal{O}\left(\sqrt{\frac{1}{nK}} + \delta\left(\zeta_f^2 + \zeta_g^2 + \sigma^2\right)\right)$. The asymptotic error of BR-FedBi has a term proportional to $\delta\sigma^2$. Compared to the lower bound in Corollary 1, this additional term indicates that the algorithm achieves suboptimal Byzantine resilience.*

## 5 ACHIEVING OPTIMAL BYZANTINE RESILIENCE

In this section, we extend BR-FedBi by incorporating the Polyak's momentum (Polyak, 1964), a.k.a. heavy-ball momentum, resulting in a variant named BR-FedBiM. This improvement enables the algorithm to achieve optimal Byzantine resilience, i.e., achieving the lower bound in Corollary 1, while simultaneously relaxing the bounded gradient assumption (Assumption 5).

## 5.1 INCORPORATING POLYAK'S MOMENTUM

The full procedure of BR-FedBiM is summarized in Algorithm 1 in Appendix C. At each iteration $k \in [K]$, instead of directly transmitting the stochastic gradients, each honest client $i \in \mathcal{H}$ updates and sends local momentums of the parameters $\diamond \in \left\{ \mathbf{x}^k, \mathbf{y}^k, \mathbf{z}^k \right\}$ to the server,

$$\mathbf{m}_{\diamond,i}^k = \beta_\diamond \mathbf{m}_{\diamond,i}^{k-1} + (1 - \beta_\diamond) \mathbf{D}_{\diamond,i}^k, \tag{8}$$

where $\beta_{\mathbf{x}}, \beta_{\mathbf{y}}, \beta_{\mathbf{z}} \in (0, 1)$ are the momentum coefficients shared by all honest clients. After receiving the local momentum updates of all the clients (including honest clients and attackers), the server applies robust aggregation methods $AGG(\cdot)$ and updates the global parameter.

For an aggregation method $AGG(\cdot)$ that satisfies the $(\delta, \kappa)$-robustness criterion, the aggregation error is uniformly bounded by $\kappa$ times the drift between local gradients and the honest global gradients. The incorporation of momentum effectively mitigates this drift and eliminates the variance-related term in the asymptotic error. Specifically, for $\diamond \in \{\mathbf{x}, \mathbf{y}, \mathbf{z}\}$, the use of momentum replaces the drift term with $\mathbb{E}\left[\left\|\mathbf{m}_{\diamond,i}^k - \mathbf{m}_\diamond^k\right\|^2\right]$, where $\mathbf{m}_\diamond^k = \frac{1}{|\mathcal{H}|} \sum_{i \in \mathcal{H}} \mathbf{m}_{\diamond,i}^k$. It can be demonstrated that the application of momentum reduces the variance-related error to $\mathcal{O}\left(\kappa \left(1 - \beta_\diamond^2\right) \sigma^2\right)$. Consequently, by setting the momentum coefficient $\beta_\diamond \to 1$, the variance-related error can be eliminated.

## 5.2 CONVERGENCE ANALYSIS

In this subsection, we establish the convergence guarantee for BR-FedBiM.

**Theorem 3.** *Let $\delta < 1/2$. Suppose Assumptions 1-4 hold, and the robust aggregation rule $AGG(\cdot)$ satisfies the $(\delta, \kappa)$-robustness criterion. Set the learning rates as follows, $\beta_{\mathbf{x}} = \sqrt{1 - L_{\beta_{\mathbf{x}}} \eta_{\mathbf{x}}}$, $\beta_{\mathbf{y}} = \sqrt{1 - L_{\beta_{\mathbf{y}}} \eta_{\mathbf{x}}}$, $\beta_{\mathbf{z}} = \sqrt{1 - L_{\beta_{\mathbf{z}}} \eta_{\mathbf{x}}}$, $\eta_{\mathbf{x}} \leq \min\left\{\alpha_1, \sqrt{\frac{\Delta_\Phi^0}{A_\sigma K}}\right\}$, $\eta_{\mathbf{y}} = c_{\mathbf{y}} \eta_{\mathbf{x}}$, $\eta_{\mathbf{z}} = c_{\mathbf{z}} \eta_{\mathbf{x}}$, where $A_\sigma := A_0 \sigma^2 / (1 - \delta) n + \kappa A_3 \sigma^2$, $L_{\beta_{\mathbf{x}}}, L_{\beta_{\mathbf{y}}}, L_{\beta_{\mathbf{z}}}, c_{\mathbf{y}}, c_{\mathbf{z}}, \alpha_1, A_0, A_3, \Delta_\Phi^0$ are some positive constants independent of $K, \delta$. Then, BR-FedBiM converges as :*

$$\mathbb{E}\left[\left\|\nabla \Phi_{\mathcal{H}}(\hat{\mathbf{x}})\right\|^2\right] = \mathcal{O}\left(\sqrt{\frac{A_\sigma \Delta_\Phi^0}{K}}\right) + \mathcal{O}\left(\frac{\Delta_\Phi^0}{K}\right) + \mathcal{O}\left(\kappa \left(\zeta_f^2 + \zeta_g^2\right)\right), \tag{9}$$

*where the expectation is over the algorithm's randomness.*

We instantiate Theorem 3 using a specific choice of $\kappa = \mathcal{O}(\delta)$. Corollary 3 shows that BR-FedBiM achieves the optimal Byzantine resilience under heterogeneity.

**Corollary 3.** *Suppose $AGG(\cdot)$ satisfies the $(\delta, \kappa)$-robustness criterion with $\kappa = \mathcal{O}(\delta)$. Then, BR-FedBiM is $(\delta, \epsilon)$-Byzantine-resilient with: $\epsilon = \mathcal{O}\left(\sqrt{\frac{1}{K}\left(\frac{1}{n} + \delta\right)} + \delta\left(\zeta_f^2 + \zeta_g^2\right)\right)$. BR-FedBiM achieves optimal Byzantine resilience, as the upper bound of the asymptotic error matches the lower bound in Corollary 1.*

**Eliminating variance-related error.** If there exists no Byzantine attackers, i.e. $\delta = 0$, the convergence rate in Corollary 3 matches that of conventional FBO (Yang et al., 2024). However, if $\delta > 0$, the dominating term of convergence rate, $\mathcal{O}\left(\sqrt{\frac{1}{K}\left(\frac{1}{n} + \delta\right)}\right)$, is affected by Byzantine attackers. While BR-FedBiM cannot achieve linear speedup as BR-FedBi in the presence of Byzantine attackers, it eliminates the variance-related error term $\mathcal{O}\left(\kappa \sigma^2\right)$. Consequently, it is able to achieve optimal Byzantine resilience. Moreover, in the homogeneous setting, the BR-FedBiM algorithm is able to converge to a stationary point at a rate of $\mathcal{O}(\sqrt{1/K})$, even with a $\delta$-proportion of Byzantine attackers.

## 6 BYZANTINE-RESILIENT FEDERATED PROBABILISTIC BILEVEL ALGORITHM

In this section, we propose BR-FedBiP, a Byzantine-resilient and variance-reduced algorithm for federated bilevel optimization. In particular, we make Probabilistic Gradient Estimator (PAGE) (Li et al., 2021b), which is a variance-reduced method, applicable to the context of Byzantine-resilient bilevel federated learning to reduce the variance among honest workers. This enhancement allows BR-FedBiP to achieve optimal sample complexity, aligning with the results established in prior works (Chu et al., 2024).

## 6.1 PROBABILISTIC GRADIENT ESTIMATOR

The full procedure of BR-FedBiP is summarized in Algorithm 1 in Appendix C. At each iteration $k \in [K]$, the server samples $c_k \sim \text{Be}(p)$ from the Bernoulli distribution with probability $p$ and sends it to the clients. Then, each honest client $i \in \mathcal{H}$ employs the probabilistic gradient estimator to reduce the gradient variance:

$$\mathbf{v}_{\mathbf{x},i}^k = \boldsymbol{\pi}_k(\nabla_{\mathbf{xy}}^2 G_i \mathbf{z}; \boldsymbol{\varphi}_i^{k'}, \boldsymbol{\varphi}_i^k) + \boldsymbol{\pi}_k(\nabla_{\mathbf{x}} F_i; \boldsymbol{\phi}_i^{k'}, \boldsymbol{\phi}_i^k), \tag{10a}$$

$$\mathbf{v}_{\mathbf{y},i}^k = \boldsymbol{\pi}_k(\nabla_{\mathbf{y}} G_i; \boldsymbol{\varphi}_i^{k'}, \boldsymbol{\varphi}_i^k), \tag{10b}$$

$$\mathbf{v}_{\mathbf{z},i}^k = \boldsymbol{\pi}_k(\nabla_{\mathbf{yy}}^2 G_i \mathbf{z}; \boldsymbol{\varphi}_i^{k'}, \boldsymbol{\varphi}_i^k) + \boldsymbol{\pi}_k(\nabla_{\mathbf{y}} F_i; \boldsymbol{\phi}_i^{k'}, \boldsymbol{\phi}_i^k), \tag{10c}$$

where $\mathbf{u}^k = (\mathbf{x}^k, \mathbf{y}^k, \mathbf{z}^k)$. We define the operation $\boldsymbol{\pi}_k$ as:

$$\boldsymbol{\pi}_k(\boldsymbol{\psi}; b', b) = \begin{cases} \boldsymbol{\psi}(\mathbf{u}^k; b') & \text{if } c_k = 1, \\ \boldsymbol{\pi}_{k-1} + \boldsymbol{\psi}(\mathbf{u}^k; b) - \boldsymbol{\psi}(\mathbf{u}^{k-1}; b) & \text{if } c_k = 0, \end{cases} \tag{11}$$

where $\boldsymbol{\psi}$ represents one of the following quantities: $\{\nabla_{\mathbf{x}} F_i, \nabla_{\mathbf{y}} F_i, \nabla_{\mathbf{y}} G_i, \nabla_{\mathbf{xy}}^2 G_i \mathbf{z}, \nabla_{\mathbf{yy}}^2 G_i \mathbf{z}\}$, which is selected based on the specific context within $\boldsymbol{\pi}_k$.

In each iteration $k$, the gradient computation follows two cases: (i) if $c_k = 1$, PAGE uses vanilla mini-batch SGD with a larger batch size of $b'$, with $\boldsymbol{\phi}_i^{k'} \sim \mathcal{D}_{f_i}$ and $\boldsymbol{\varphi}_i^{k'} \sim \mathcal{D}_{g_i}$, and (ii) if $c_k = 0$, it reuses previous global updates with a small adjustment based on a smaller batch size $b$, with $\boldsymbol{\phi}_i^k \sim \mathcal{D}_{f_i}$ and $\boldsymbol{\varphi}_i^k \sim \mathcal{D}_{g_i}$. After collecting gradients from all clients, the server applies a robust aggregation method and updates the global parameters.

## 6.2 CONVERGENCE ANALYSIS

In this subsection, we establish a convergence guarantee for BR-FedBiP and show that BR-FedBiP achieves the optimal sample complexity. To establish this, we assume that the stochastic gradients satisfy the mean-squared smoothness property, as commonly assumed in existing works (Yang et al., 2021; Chu et al., 2024). All proofs are provided in Appendix I.

**Assumption 7** (Mean-squared smoothness). For any honest client $i \in \mathcal{H}$, the stochastic gradients and Hessians $\nabla F_i(\mathbf{x}, \mathbf{y}; \boldsymbol{\phi}_i), \nabla G_i(\mathbf{x}, \mathbf{y}; \boldsymbol{\varphi}_i), \nabla_{\mathbf{xy}}^2 G_i(\mathbf{x}, \mathbf{y}; \boldsymbol{\varphi}_i)$ and $\nabla_{\mathbf{yy}}^2 G_i(\mathbf{x}, \mathbf{y}; \boldsymbol{\varphi}_i)$ are $L_f, L_g, L_g$ and $L_g$ Lipschitz continuous, respectively.

**Theorem 4.** Let $\delta < 1/2$. Suppose Assumptions 1-4 and 7 hold, and the robust aggregation rule $AGG(\cdot)$ satisfies the $(\delta, \kappa)$-robustness criterion. Choose mini-batch size $b < \sqrt{b'}$ and the probability $p \in (0, 1]$. Set the learning rates as follows: $\eta_{\mathbf{x}} \leq \min\left\{\alpha_1, \frac{c}{1+\sqrt{\frac{1-p}{pb}}}\right\}, \eta_{\mathbf{y}} = c_{\mathbf{y}} \eta_{\mathbf{x}}, \eta_{\mathbf{z}} = c_{\mathbf{z}} \eta_{\mathbf{x}}$, where $\alpha_1, c, c_{\mathbf{y}}, c_{\mathbf{z}}$ are some positive constants independent of $K$. Then, the algorithm BR-FedBiP (1) converges as:

$$\mathbb{E}\left[\|\nabla \Phi_{\mathcal{H}}(\hat{\mathbf{x}})\|^2\right] = \mathcal{O}\left(\frac{\left(1 + \sqrt{\frac{1-p}{pb}} + \frac{1}{pb'}\right)}{K}\right) + \mathcal{O}\left(\frac{\sigma^2}{b'}\right) + \mathcal{O}\left(\kappa\left(\zeta_g^2 + \zeta_f^2 + \sigma^2\right)\right),$$

where the expectation is over the algorithm's randomness.

**Remark 3.** To achieve $(\delta, \epsilon)$-Byzantine resilience, we take $b' = \min\{\sigma^2 \epsilon^{-1}, M + N\}$, where $N$ and $M$ represent the numbers of data points for the upper-level and lower-level objectives held by each client, respectively. If $N + M$ is sufficiently large, we have $b' = \mathcal{O}(\epsilon^{-1})$. Assuming $b < \sqrt{b'}$ and choosing $p = \frac{b}{b'+b}$, the sample complexity of BR-FedBiP reduces to $\mathcal{O}(\epsilon^{-1.5})$, which matches the optimal sample complexity in the general expectation setting (Li et al., 2021b).

## 7 EXPERIMENTS

In this section, we evaluate the performance of the proposed algorithms on hyper-representation tasks. We compare these algorithms with BILANTINE (Abbas et al., 2024), a sub-loop Byzantine-resilient bilevel optimization algorithm under four different Byzantine attacks on MNIST (LeCun, 1998) and CIFAR-10 (Krizhevsky et al., 2009) datasets. Please refer to Appendix D for details of the implementation and experiment results.

Table 1: Hyper-representation on MNIST with heterogeneous data. "Worst" shows the worst test accuracy among all attacks.

| $AGG$ | Algorithm | ALIE | BF | IPM | RN | Worst | $AGG$ | Algorithm | ALIE | BF | IPM | RN | Worst |
|---|---|---|---|---|---|---|---|---|---|---|---|---|---|
| Median | BILANTINE | 73.74 | 46.02 | 49.85 | 63.12 | 46.02 | RFA | BILANTINE | 66.96 | 32.02 | 19.96 | 67.68 | 19.96 |
| | BR-FedBi | 83.11 | 82.00 | 81.28 | 82.65 | 81.28 | | BR-FedBi | 70.83 | 79.41 | 80.70 | 82.49 | 70.83 |
| | BR-FedBiP | **83.24** | 82.24 | 81.36 | **82.73** | **81.36** | | BR-FedBiP | 70.80 | 79.26 | 80.69 | **82.62** | 70.80 |
| | BR-FedBiM | 79.53 | **82.51** | **82.60** | 82.59 | 79.53 | | BR-FedBiM | **72.76** | **81.95** | **81.87** | 82.60 | **72.76** |
| TM | BILANTINE | 73.96 | 41.84 | 46.05 | 64.19 | 41.84 | CWMed | BILANTINE | 56.88 | 51.02 | 58.4 | 63.32 | 51.02 |
| | BR-FedBi | 83.27 | 82.61 | 82.54 | **82.66** | 82.54 | | BR-FedBi | 58.25 | 79.70 | 81.51 | 82.76 | 58.25 |
| | BR-FedBiP | **83.29** | 82.58 | 82.58 | 82.61 | **82.58** | | BR-FedBiP | **58.25** | 79.70 | 81.62 | **82.76** | 58.25 |
| | BR-FedBiM | 82.44 | **82.69** | **82.61** | 82.44 | 82.44 | | BR-FedBiM | 69.37 | **82.23** | **82.30** | 82.66 | **69.37** |

Table 2: Hyper-representation on CIFAR-10 with homogeneous data. "Worst" shows the worst test accuracy among all attacks.

| $AGG$ | Algorithm | ALIE | BF | IPM | RN | Worst | AGG | Algorithm | ALIE | BF | IPM | RN | Worst |
|---|---|---|---|---|---|---|---|---|---|---|---|---|---|
| Median | BR-FedBi | 60.61 | 59.36 | 63.22 | 43.70 | 43.70 | RFA | BR-FedBi | 51.91 | 57.52 | **63.13** | 43.02 | 43.02 |
| | BR-FedBiP | **63.39** | 62.04 | 59.36 | **62.86** | 59.36 | | BR-FedBiP | 49.36 | 52.67 | 57.52 | 62.91 | 49.36 |
| | BR-FedBiM | 63.12 | **62.68** | **63.40** | 62.67 | **62.67** | | BR-FedBiM | **54.83** | **62.83** | 62.47 | **63.09** | **54.83** |
| TM | BR-FedBi | 61.31 | 62.92 | 63.11 | 42.45 | 42.45 | CWTM | BR-FedBi | 48.84 | 54.83 | **63.14** | 42.01 | 42.01 |
| | BR-FedBiP | 63.22 | 62.09 | 62.92 | 63.22 | 62.09 | | BR-FedBiP | 37.70 | 48.84 | 54.83 | 61.40 | 37.70 |
| | BR-FedBiM | **63.64** | **63.34** | **63.18** | **63.74** | **63.18** | | BR-FedBiM | **51.92** | **62.21** | 60.37 | **62.92** | **51.92** |
| Krum | BR-FedBi | 63.49 | 63.16 | 63.06 | 43.60 | 43.60 | CWMed | BR-FedBi | **54.45** | 57.38 | **63.70** | 43.31 | 43.31 |
| | BR-FedBiP | **63.72** | **63.49** | **63.16** | 63.17 | **63.16** | | BR-FedBiP | 47.26 | 54.45 | 57.38 | **63.51** | 47.26 |
| | BR-FedBiM | 63.19 | 62.80 | 63.09 | **63.21** | 62.80 | | BR-FedBiM | 47.90 | 52.70 | 58.77 | 63.08 | **47.90** |

**Distributed system.** We consider a distributed system with $n = 25$ workers, among which $f = 7$ are Byzantine attackers. In the experiment, we test four representative attacks, which are A Little is Enough (ALIE) (Baruch et al., 2019), Inner Product Manipulation (IPM) (Xie et al., 2020), Bit Flipping (BF) (Allen-Zhu et al., 2020) and Random Noise (RN). To preserve Byzantine resilience, we implement six widely used robust aggregators, which are Krum (Blanchard et al., 2017), Median, Trimmed Mean, Coordinate-Wise Median (CWMed) (Yin et al., 2018), Coordinate-Wise Trimmed Mean (CWTM) (Yin et al., 2018), and RFA (Pillutla et al., 2022).

**Empirical results on MNIST.** Table 1 compares the performance of BR-FedBi, BR-FedBiM, and BR-FedBiP against BILANTINE on the heterogeneous MNIST dataset under four different Byzantine attacks. For each block (i.e. each robust aggregator), we highlight in **bold** the algorithm with the highest accuracy under each attack. It is observed that BR-FedBiP and BR-FedBiM achieve better performance on worst-case accuracy, and all the proposed single-loop bilevel algorithms significantly outperform the baseline BILANTINE. Additional experiments under homogeneous settings on MNIST dataset are presented in Appendix D.3.2.

**Empirical results on CIFAR-10.** Table 2 compares the performance of BR-FedBi, BR-FedBiM, and BR-FedBiP on CIFAR-10 dataset. The results for BILANTINE on CIFAR-10 dataset are not presented, as our extensive hyperparameter search revealed that BILANTINE failed to converge on CIFAR-10 dataset. The experimental results show that BR-FedBiM achieves better performance on CIFAR-10 dataset and attains higher worst-case accuracy when combined with multiple robust aggregation methods.

**Experiments on different numbers of attackers.** We further evaluate the efficacy of the proposed methods under different numbers of attackers $f = \{3, 5, 7, 10\}$ with $n = 25$ (see Appendix D.3.2 for details) on MNIST dataset. The results demonstrate that the proposed methods achieve better performance than the baseline BILANTINE algorithm with different numbers of attackers.

## 8 CONCLUSION

This work bridges the gap between federated bilevel optimization and Byzantine-resilient federated learning. We proposed BR-FedBi and its variants, which are a family of single-loop Byzantine-resilient bilevel algorithms that ensure both efficiency and robustness. We provided the first tight analysis of asymptotic error under Byzantine attacks in bilevel optimization. Experimental results demonstrate that the proposed methods significantly outperform existing approaches under various Byzantine attack scenarios. However, the convergence analysis relies on the $\zeta$-dissimilarity heterogeneity assumption (Assumption 4). Whether similar theoretical guarantees can be established under the milder $(\zeta, B)$-dissimilarity assumption (Assumption 8) remains an open question. We aim to address these limitations in future work.

ACKNOWLEDGMENTS

This work was supported in part by a grant from the NSFC/RGC Joint Research Scheme sponsored by the Research Grants Council of the Hong Kong Special Administrative Region, China and National Natural Science Foundation of China (Project No. N_HKUST656/22).

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

ORGANIZATION

## A  LLM USAGE STATEMENT

In this work, we used a large language model (LLM) solely for writing assistance (e.g. proofreading, rephrasing, stylistic polishing). We did not rely on it for scientific content generation, experiment design, idea generation, or analysis. All core research components—conceptualization, modeling, experiments, theoretical derivations, and interpretation—were performed by the authors. We take full responsibility for all content in the manuscript, including any modifications influenced by the LLM.

## B  MORE RELATED WORKS

**Distributed bilevel optimization.** Distributed optimization has emerged as a key paradigm for solving large-scale learning problems in which data are distributed across multiple agents. A major milestone in bilevel optimization was achieved by Ghadimi & Wang (2018), who provided the first non-asymptotic analysis of bilevel stochastic approximation methods, thereby motivating extensive research on more efficient bilevel optimization algorithms. This growing line of work includes AID-based methods (Domke, 2012; Pedregosa, 2016; Gould et al., 2016; Ghadimi & Wang, 2018; Grazzi et al., 2020; Ji et al., 2021), ITD-based approaches (Domke, 2012; Maclaurin et al., 2015; Franceschi et al., 2018; Grazzi et al., 2020; Ji et al., 2021), and Neumann-series-based techniques (Chen et al., 2021; Hong et al., 2020; Ji et al., 2021). Building on these foundations, Tarzanagh et al. (2022) proposed FedNest, a federated bilevel optimization algorithm based on approximate implicit differentiation, while Xiao & Ji (2023) introduced an iterative differentiation (ITD)–based method designed to reduce communication overhead in federated settings. More recently, Dagréou et al. (2022) developed SOBA, a single-loop framework for bilevel optimization, and Yang et al. (2024) extended this framework to the federated learning setting. Despite these advances, research on Byzantine-robust distributed bilevel learning remains limited.

**Decentralized bilevel optimization.** In contrast to distributed algorithms, decentralized approaches operate without a central server, offering increased robustness to communication network contingencies. However, the lack of a central coordinator necessitates peer-to-peer communication, which introduces additional challenges for convergence—especially in the presence of significant data heterogeneity. Kong et al. (2024) proposed a single-loop algorithm based on decentralized SOBA. To improve robustness against data heterogeneity, recent works have incorporated Gradient Tracking (GT) into both lower- and upper-level optimization procedures. While existing results such as (Dong et al., 2025; Chen et al., 2025) have primarily focused on deterministic settings, Niu et al. (2025) investigated personalized lower-level problems that relax the requirement for global consensus. Notably, approaches such as Chen et al. (2025; 2023) have relied on computationally intensive inner-loop GT updates. Furthermore, Zhu et al. (2024) presented a unified convergence analysis for GT-based methods and demonstrated that integrating GT achieves a smaller transient iteration complexity compared to D-SOBA for decentralized stochastic bilevel optimization. Despite these advances, research on Byzantine-robust decentralized bilevel learning remains limited.

**Byzantine-resilient distributed learning.** The goal of Byzantine-resilient learning is to protect the learning process from malicious attackers (Blanchard et al., 2017; Yin et al., 2018; Pillutla et al.,

2022). Most existing defenses replace the simple averaging step used in conventional FL frameworks (McMahan et al., 2017) with robust aggregation rules designed to compute a global update that is resistant to adversarial manipulation. Some robust aggregation rules aim to incorporate all honest gradients into the aggregation (Blanchard et al., 2017; Shejwalkar & Houmansadr, 2021), whereas others (Yin et al., 2018; Blanchard et al., 2017) purposely aggregate only a subset of gradients to ensure that adversarial updates are excluded. However, data heterogeneity makes Byzantine robustness substantially more challenging (Karimireddy et al., 2020a; Allouah et al., 2023). To mitigate this issue, Karimireddy et al. (2020a) proposed a pre-aggregation method, bucketing, which adapts existing robust aggregation rules to heterogeneous data distributions. Building on this direction, Allouah et al. (2023) developed the Nearest-Neighbor Mixing (NNM) algorithm, which strengthens standard robust distributed gradient descent methods and achieves optimal Byzantine resilience under heterogeneous settings. Despite these advancements, research on bilevel optimization remains limited.

## C  FULL ALGORITHM DESCRIPTION

In this section, we provide the full version of the proposed algorithm, which complements the abbreviated description given in the main paper.

A commonly employed strategy for bilevel optimization problems makes use of implicit differentiation (Griewank & Walther, 2008). In centralized bilevel optimization, under appropriate hypotheses, the function $\Phi$ is differentiable, and the chain rule and implicit function theorem give for any

$$\nabla\Phi(\mathbf{x}) = \nabla_{\mathbf{x}}f(\mathbf{x}, \mathbf{y}^\star(\mathbf{x})) - \nabla_{\mathbf{xy}}^2 g(\mathbf{x}, \mathbf{y}^\star(\mathbf{x})) \cdot \left[\nabla_{\mathbf{yy}}^2 g(\mathbf{x}, \mathbf{y}^\star(\mathbf{x}))\right]^{-1} \nabla_{\mathbf{y}}f(\mathbf{x}, \mathbf{y}^\star(\mathbf{x})).$$

By leveraging the results for centralized stochastic bilevel optimization, in Byzantine-resilient federated bilevel optimization (BR-FBO), the expression of the hypergradient $\nabla\Phi_{\mathcal{H}}(\mathbf{x}, \mathbf{y}^\star(\mathbf{x}))$ can be written as:

$$\frac{1}{|\mathcal{H}|}\sum_{i\in\mathcal{H}}\nabla_{\mathbf{x}}f_i\left(\mathbf{x}, \mathbf{y}^\star(\mathbf{x})\right) - \nabla_{\mathbf{xy}}^2 g_{\mathcal{H}}\left(\mathbf{x}, \mathbf{y}^\star(\mathbf{x})\right) \cdot \left[\nabla_{\mathbf{yy}}^2 g_{\mathcal{H}}\left(\mathbf{x}, \mathbf{y}^\star(\mathbf{x})\right)\right]^{-1} \cdot \frac{1}{|\mathcal{H}|}\sum_{i\in\mathcal{H}}\nabla_{\mathbf{y}}f_i\left(\mathbf{x}, \mathbf{y}^\star(\mathbf{x})\right),$$

where the global Hessian matrices can be written as:

$$\nabla_{\mathbf{xy}}^2 g_{\mathcal{H}}\left(\mathbf{x}, \mathbf{y}^\star(\mathbf{x})\right) = \frac{1}{|\mathcal{H}|}\sum_{i\in\mathcal{H}}\nabla_{\mathbf{xy}}^2 g_i(\mathbf{x}, \mathbf{y}^\star(\mathbf{x})), \quad \nabla_{\mathbf{yy}}^2 g_{\mathcal{H}}\left(\mathbf{x}, \mathbf{y}^\star(\mathbf{x})\right) = \frac{1}{|\mathcal{H}|}\sum_{i\in\mathcal{H}}\nabla_{\mathbf{yy}}^2 g_i(\mathbf{x}, \mathbf{y}^\star(\mathbf{x})).$$

We extend the SOBA framework (Dagréou et al., 2022) to develop a single-loop solution for BR-FBO, which introduces an auxiliary variable $\mathbf{z}^\star(\mathbf{x})$ to approximate the Hessian-inverse-vector product:

$$\mathbf{z}^\star(\mathbf{x}) := -\left[\nabla_{\mathbf{yy}}^2 g_{\mathcal{H}}(\mathbf{x}, \mathbf{y}^\star)\right]^{-1} \cdot \frac{1}{|\mathcal{H}|}\sum_{i\in\mathcal{H}}\nabla_{\mathbf{y}}f_i\left(\mathbf{x}, \mathbf{y}^\star\right). \tag{12}$$

Thus, the hypergradient $\nabla\Phi_{\mathcal{H}}(\mathbf{x})$ can be written as

$$\nabla\Phi_{\mathcal{H}}(\mathbf{x}) = \frac{1}{|\mathcal{H}|}\sum_{i\in\mathcal{H}}\nabla_{\mathbf{x}}f_i\left(\mathbf{x}, \mathbf{y}^\star(\mathbf{x})\right) + \nabla_{\mathbf{xy}}^2 g_{\mathcal{H}}\left(\mathbf{x}, \mathbf{y}^\star(\mathbf{x})\right) \cdot \mathbf{z}^\star(\mathbf{x}). \tag{13}$$

Approximating $\mathbf{z}^\star(\mathbf{x})$ in (12) amounts to letting each agent solve for the following equation:

$$\mathbf{z}^\star := \left(\sum_{i\in\mathcal{H}} H_i\right)^{-1}\left(\sum_{i\in\mathcal{H}} b_i\right) \text{ or } \left(\sum_{i\in\mathcal{H}} H_i\right)\mathbf{z}^\star = \left(\sum_{i\in\mathcal{H}} b_i\right), \tag{14}$$

where $H_i = \nabla_{\mathbf{yy}}^2 g_i(\mathbf{x}, \mathbf{y}^\star(\mathbf{x}))$ and $b_i = \nabla_{\mathbf{y}}f_i(\mathbf{x}, \mathbf{y}^\star(\mathbf{x}))$. Equality (14) is essentially the optimality condition of the following optimization problem:

$$\min_{\mathbf{z}\in\mathcal{R}^q}\frac{1}{|\mathcal{H}|}\sum_{i\in\mathcal{H}}\Psi_i, \quad \Psi_i = \frac{1}{2}\mathbf{z}^\top H_i\mathbf{z} + b_i^\top\mathbf{z}. \tag{15}$$

In the light of (12), (13) and (15), it turns out that the derivation of the gradient of $\Phi_{\mathcal{H}}$ at each iteration is cumbersome because it involves two subproblems: the resolution of the inner problem to find an approximation of $\mathbf{y}^\star(\mathbf{x})$ and the resolution of a linear system to find an approximation of $\mathbf{z}(\mathbf{x})$.

The auxiliary variable $\mathbf{z}$ enables the estimation of the global hypergradient using the expression: $\frac{1}{|\mathcal{H}|} \sum_{i \in \mathcal{H}} \nabla_{\mathbf{x}} f_i(\mathbf{x}, \mathbf{y}) + \nabla^2_{\mathbf{xy}} g_{\mathcal{H}}(\mathbf{x}, \mathbf{y})\mathbf{z}$, where $\mathbf{y}$ represents the approximate solution to the lower-level problem. Thus, the solution of the inner problem, the solution of the linear system (15), and the outer variable all evolve at the same time. Algorithm 1 outlines the unified framework for the proposed BR-FedBi, BR-FedBiM, and BR-FedBiP algorithms. For clarity, gradient computation and aggregation steps are marked with different colors corresponding to the respective variants.

---

**Algorithm 1** BR-FedBi, BR-FedBiM and BR-FedBiP

---

**Initialization: each honest worker** $i \in \mathcal{H}$ initializes $\mathbf{x}^0, \mathbf{y}^0, \mathbf{z}^0(\|\mathbf{z}^0\| \leq \rho) \in \mathbb{R}^d$, learning rates $\eta_x, \eta_y, \eta_z \in \mathbb{R}^+$ and a robust aggregation method $\mathcal{A} : \mathbb{R}^{n \times d} \rightarrow \mathbb{R}^d$.

**for** $k = 0, \cdots, K - 1$ **do**

    **Server:** Broadcast $\mathbf{x}^k, \mathbf{y}^k, \mathbf{z}^k$ to all workers;

    Get probability indicator: $c_k \sim \mathrm{Be}(p)$;

    **for** Each honest worker $i \in \mathcal{H}$**, in parallel: do**

        Locally compute gradients $D^k_{\mathbf{x},i}, D^k_{\mathbf{y},i}, D^k_{\mathbf{z},i}$ defined in eq.(5);

        Locally compute gradients $m^k_{\mathbf{x},i}, m^k_{\mathbf{y},i}, m^k_{\mathbf{z},i}$ defined in eq.(8);

        Locally compute gradients $v^k_{\mathbf{x},i}, v^k_{\mathbf{y},i}, v^k_{\mathbf{z},i}$ defined in eq.(10);

    **end for**

    **Server:** Received the gradients from all clients;

    **for** $\diamond \in \{\mathbf{x}, \mathbf{y}, \mathbf{z}\}$ **do**

        Server aggregates local estimators:    $\hat{D}^k_\diamond = AGG\left(D^k_{\diamond,1}, ..., D^k_{\diamond,n}\right)$;

        *// Similar statergy for the* $\hat{m}^k_\diamond$ *and* $\hat{v}^k_\diamond$.

        Server update global parameter:    $\diamond^{k+1} = \diamond^k - \eta_\diamond \hat{D}^k_\diamond$;

    **end for**

    Clip the norm of $\mathbf{z}^{k+1}$: $\mathbf{z}^{k+1} = \min\left\{1, \rho/\|\mathbf{z}^{k+1}\|_2\right\} \cdot \mathbf{z}^{k+1}$;

**end for**

**return** Sample $\hat{\mathbf{x}}$ uniformly at random from $\{\mathbf{x}^0, ..., \mathbf{x}^{K-1}\}$.

---

# D EXPERIMENTS

In this section, we evaluate the performance of the proposed methods on a hyper-representation task. Specifically, we compare the proposed Byzantine-resilient single-loop bilevel optimization algorithms with BILANTINE (Abbas et al., 2024), a sub-loop-based Byzantine-resilient bilevel optimization algorithm.

## D.1 ATTACKS AND ROBUST AGGREGATORS.

### D.1.1 ENVIRONMENTS

All methods were implemented using the PyTorch framework (Paszke et al., 2019) and executed on NVIDIA A100 GPUs with 40GB of memory. Notably, all experiments can be conducted with as little as 2GB of memory.

### D.1.2 ATTACKS.

In our experiments, the Byzantine workers execute four state-of-the-art gradient attacks, namely A Little is Enough (ALIE) (Baruch et al., 2019), Fall of Empires (Xie et al., 2020), Bit-flipping (BF) (Allen-Zhu et al., 2020), and Random noise. Let $a^k$ denote the attack direction at iteration $k$, and let

$\eta \geq 0$ be a fixed scalar. At each step $k$, every Byzantine worker sends to the server a manipulated vector $B^k = \bar{s}^k + \eta a^k$, where $\bar{s}^k$ is an estimate of the true gradient (or momentum). In our gradient descent (GD) experiments, we compute this estimate as $\bar{s}^k = \frac{1}{|\mathcal{H}|} \sum_{i \in \mathcal{H}} g_i^k$, where $g_i^k$ is the gradient computed by honest worker $w_i$ at iteration $k$.

- **ALIE:** The adversarial direction is chosen as $a^k = \sigma^k$, where $\sigma^k$ is the coordinate-wise standard deviation of $\bar{s}^k$.
- **IPM:** The adversary sets $a^k = -\bar{s}^k$. In this case, each Byzantine worker sends $(1 - \eta)\bar{s}^k$ to the server.
- **BF:** This attack also uses $a^k = -\bar{s}^k$, but fixes $\eta = 2$. Thus, Byzantine workers send $B^k = a^k = -\bar{s}^k$. at every step.
- **RN:** In this attack, the adversarial direction $a^k$ is sampled from a random Gaussian distribution. Formally, $a^k \sim \mathcal{N}(\mu, \sigma^2)$, so each Byzantine worker injects random Gaussian noise into its reported vector.

### D.1.3 ROBUST AGGREGATOR.

In our experiment, we use six widely adopted robust aggregators: Krum (Blanchard et al., 2017), Median (Acharya et al., 2022), Trimmed Mean, Coordinate-Wise Median (CWMed), Coordinate-Wise Trimmed Mean (CWTM) (Yin et al., 2018), and RFA (Pillutla et al., 2022). The following outlines how each method operates:

- **Krum.** Given input vectors $x_1, \ldots, x_n$, Krum selects one of them based on its proximity to its closest neighbors while ignoring the $f$ farthest vectors. For each $i \in [n]$, let $i_1, \ldots, i_{n-1}$ be the indices of the other vectors sorted by distance from $x_i$, and define the set of its $n - f - 1$ nearest neighbors as $C_i = \{i_1, \ldots, i_{n-f-1}\}$. Krum then chooses the vector whose neighbors are collectively the closest:

$$\text{Krum}(x_1, \ldots, x_n) = x_{i^*}, \quad i^* \in \arg\min_{i \in [n]} \sum_{j \in C_i} \|x_i - x_j\|^2.$$

- **Median.** Given vectors $x_1, \ldots, x_n \in \mathbb{R}^d$, the multivariate median is defined as the point $\text{Med}(x_1, \ldots, x_n) := \arg\min_{x \in \mathbb{R}^d} \sum_{i=1}^n \|x - x_i\|_2$. This estimator minimizes the sum of Euclidean distances to all input vectors. Although robust, it is computationally expensive in high-dimensional settings.
- **Trimmed Mean.** Given $x_1, \ldots, x_n \in \mathbb{R}^d$ and a trimming parameter $f$, let $x_{(1)}, \ldots, x_{(n)}$ denote the vectors sorted by their distance to the mean, $\bar{x} = \frac{1}{n} \sum_{i=1}^n x_i$, $\|x_{(1)} - \bar{x}\|_2 \leq \cdots \leq \|x_{(n)} - \bar{x}\|_2$. The trimmed mean removes the $f$ farthest and $f$ closest vectors and averages the remaining:

$$\text{TM}(x_1, \ldots, x_n) = \frac{1}{n - 2f} \sum_{j=f+1}^{n-f} x_{(j)}.$$

- **CWTM.** Given vectors $x_1, \ldots, x_n \in \mathbb{R}^d$, let $\tau_k$ be the permutation that sorts their $k$-th coordinates in non-decreasing order: $[x_{\tau_k(1)}]_k \leq \cdots \leq [x_{\tau_k(n)}]_k$. For a trimming parameter $f$, the CwTM is defined coordinate-wise by removing the largest and smallest $f$ values and averaging the rest:

$$\left[\text{CwTM}(x_1, \ldots, x_n)\right]_k = \frac{1}{n - 2f} \sum_{j=f+1}^{n-f} [x_{\tau_k(j)}]_k, \qquad k \in [d].$$

- **CWMed:** For a vector $x \in \mathbb{R}^d$, let $[x]_k$ denote its $k$-th coordinate. Instead of computing the full multivariate median, which is difficult in high dimensions—we take the median independently on each coordinate. Given input vectors $x_1, \ldots, x_n$, the coordinate-wise median $\text{CwMed}(x_1, \ldots, x_n)$ is the vector whose $k$-th coordinate is

$$\left[\text{CwMed}(x_1, \ldots, x_n)\right]_k = \text{Median}\left([x_1]_k, \ldots, [x_n]_k\right), \qquad k \in [d].$$

- **RFA:** Given vectors $x_1, \ldots, x_m \in \mathbb{R}^d$ and weights $\alpha_i > 0$, RFA computes a robust center via an iterative smoothed Weiszfeld update. Starting from $v^{(0)}$, for $r = 0, \ldots, R - 1$ it sets

$$\beta_i^{(r)} = \frac{\alpha_i}{\nu \vee \|v^{(r)} - x_i\|} \quad \text{and} \quad v^{(r+1)} = \frac{\sum_{i=1}^m \beta_i^{(r)} x_i}{\sum_{i=1}^m \beta_i^{(r)}}.$$

## D.2 Loss Function Tuning on Imbalanced Dataset

**Loss Function Tuning on Imbalanced Datasets.** We use bilevel optimization to tune a loss function for learning on an imbalanced MNIST dataset, aiming to maximize the class-balanced validation accuracy so as to better serve minority (tail) classes. Following the formulation in (Li et al., 2021a), we tune the VS-loss (Kini et al., 2021) in a federated setting, where each client $i$ has a local training set $\mathcal{D}_\tau^i$ and validation set $\mathcal{D}_v^i$. The bilevel problem is given by

$$\min_\alpha \; \mathcal{L}_{\text{val}}(\alpha) = \frac{1}{n} \sum_{i=1}^n \frac{1}{|\mathcal{D}_v^i|} \sum_{(x,y) \in \mathcal{D}_v^i} \ell_{\text{bal}}\big(y, x; f(\theta^*(\alpha))\big)$$

$$\text{s.t.} \quad \theta^*(\alpha) = \arg\min_\theta \frac{1}{n} \sum_{i=1}^n \frac{1}{|\mathcal{D}_\tau^i|} \sum_{(x,y) \in \mathcal{D}_\tau^i} \ell_{\text{VS}}\big(y, x; f_\theta; \alpha\big),$$

where $\ell_{\text{VS}}$ is the parametric VS-loss with hyperparameters $\alpha$ (class-dependent weights and logit adjustments), and $\ell_{\text{bal}}$ is a class-balanced validation loss. To create a long-tailed imbalance, we exponentially decrease the number of MNIST samples per class (Tarzanagh et al., 2022). For example, class 0 has 6000 samples, class 1 has 3597 samples, and class 9 has only 60 samples.

### D.2.1 Implement details

In our experiments, we use a 2-layer multilayer perceptron (MLP) with 200 hidden units. We set the number of communication rounds to be 1000 and ensure that all algorithms are run for a sufficient number of iterations to reach convergence. The batch size on each client is set to $b = 64$ for all algorithms, and $b' = 256$ for the BR-FedBiP algorithm. The learning rates are chosen as $\{\eta_y, \eta_z, \eta_x\} \in \{0.1, 0.02, 0.01\}$, and the momentum coefficient is set to $\beta = 0.8$ for all parameters $\{x, y, z\}$. The coefficient for BR-FedBiP is set to $p = 0.2$, which matches the theoretical value $p = \frac{b}{b'+b}$. Details of the model architectures and implementations are summarized in Table 3.

Table 3: Implementation details.

| Details | MNIST |
|---|---|
| Model type | 2-layer MLP |
| Model architecture | Input: 784, Hidden: 200, Output: 10 |
| Optimizer | SGD |
| Learning rate ($\eta$) | $\{\eta_y, \eta_z, \eta_x\} \in \{0.1, 0.02, 0.01\}$ |
| Batch size ($b$) | $b = 64$ |
| Momentum ($\beta$) | $\beta = 0.8$ |
| Probabilistic coefficient ($p$) | $p = 0.2$ |
| Server-side update (iteration) | 300 |
| Number of Byzantine workers | $f = 7$ |
| Number of honest workers | $n - f = 18$ |

### D.2.2 Experiment results.

Table 4 reports the test accuracy of BR-FedBi, BR-FedBiM, and BR-FedBiP compared with BILAN-TINE on the heterogeneous MNIST dataset under four types of Byzantine attacks. For each block corresponding to a robust aggregator, we mark in **bold** the method that attains the highest accuracy for a given attack. Overall, BR-FedBiP and BR-FedBiM consistently yield higher worst-case accuracy, and all three proposed single-loop bilevel methods substantially outperform the baseline BILANTINE.

**More results about wall-clock running time.** We also report the wall-clock running time of different algorithms for the MNIST datasets for 300 epochs, which shows that the single-loop algorithms less time for 1 training epoch. In terms of communication cost, the sub-loop algorithm requires 8 rounds per epoch, whereas the single-loop algorithm requires only 3 rounds per epoch. To further support the conclusion, we also present the wall-clock running time that different methods require to reach $50\%, 60\%, 75\%$ and $85\%$ of the top-1 test accuracy in Table 5.

Table 4: Imbalanced-Data Loss Tuning on MNIST with heterogeneous data. "Worst" shows the worst test accuracy among all attacks.

| $AGG$ | Algorithm | ALIE | BF | IPM | RN | Worst | $AGG$ | Algorithm | ALIE | BF | IPM | RN | Worst |
|---|---|---|---|---|---|---|---|---|---|---|---|---|---|
| Median | BILANTINE | 67.49 | 61.03 | 51.9 | 66.36 | 51.9 | RFA | BILANTINE | 30.29 | 30.52 | 27.46 | 50.63 | 27.46 |
| | BR-FedBi | 75.81 | 68.38 | 72.84 | 75.98 | 68.38 | | BR-FedBi | 78.25 | 68.61 | 65.7 | 76.16 | 65.7 |
| | BR-FedBiP | 76.02 | 68.37 | **73.15** | **76.18** | 68.37 | | BR-FedBiP | 78.05 | **68.68** | 65.88 | **76.41** | **65.88** |
| | BR-FedBiM | **76.27** | **69.97** | 71.17 | 76.03 | **69.97** | | BR-FedBiM | **78.26** | 67.35 | 65.19 | 76.01 | 65.19 |
| TM | BILANTINE | 57.14 | 52.78 | 52.38 | 62.47 | 52.38 | CWMed | BILANTINE | 69.5 | 55.29 | 54.09 | 62.31 | 54.09 |
| | BR-FedBi | 75.86 | 73.68 | 74.79 | 75.33 | 73.68 | | BR-FedBi | 78.38 | 67.12 | 60.69 | 75.7 | 60.69 |
| | BR-FedBiP | 76.08 | **73.76** | **74.97** | **76.02** | **73.76** | | BR-FedBiP | **78.52** | **67.38** | 60.99 | **75.75** | 60.99 |
| | BR-FedBiM | **76.59** | 71.06 | 72.48 | 75.79 | 71.06 | | BR-FedBiM | 78.35 | 65.49 | **61.67** | 75.1 | **61.67** |
| Krum | BILANTINE | 61.35 | 58.83 | 61.61 | 62.92 | 58.83 | CWTM | BILANTINE | 42.39 | 46.31 | 40.18 | 68.8 | 40.18 |
| | BR-FedBi | 75.88 | 72.91 | 73.96 | 75.96 | 72.91 | | BR-FedBi | 78.23 | 66.8 | 60.52 | 75.86 | 60.52 |
| | BR-FedBiP | 76.06 | **73.35** | **74.08** | 76.21 | **73.35** | | BR-FedBiP | **78.51** | **67.04** | 60.58 | **76.03** | 60.58 |
| | BR-FedBiM | **76.08** | 72.01 | 72.71 | **76.42** | 72.01 | | BR-FedBiM | 78.17 | 65.69 | **61.07** | 75.44 | **61.07** |

Table 5: Wall-clock running time (s) for different methods over 300 epochs. "X% of top-1 test acc." denotes the time required to reach X% of the final top-1 test accuracy.

| Clock-time | Algorithm | Krum | TM | Median | RFA | CWTM | CWMed |
|---|---|---|---|---|---|---|---|
| Total time | BR-FedBi | 5951.92 | 5890.92 | 5897.21 | 5929.72 | 5860.15 | 5837.08 |
| | BILANTINE | 7041.4429 | 6459.92 | 6516.94 | 7116.44 | 6523.39 | 6589.95 |
| 50% of top-1 test acc | BR-FedBi | 123.49 | 70.60 | 68.39 | 96.38 | 106.78 | 75.80 |
| | BILANTINE | 166.60 | 371.64 | 199.62 | 259.10 | 13.28 | 326.59 |
| 60% of top-1 test acc | BR-FedBi | 310.76 | 146.78 | 143.03 | 132.27 | 142.50 | 146.23 |
| | BILANTINE | 286.25 | 566.82 | 281.24 | 674.35 | 189.49 | 404.07 |
| 75% of top-1 test acc | BR-FedBi | 457.30 | 557.70 | 325.63 | 544.96 | 550.09 | 552.83 |
| | BILANTINE | 731.12 | 1301.09 | 359.70 | 1566.27 | 557.48 | 660.45 |
| 85% of top-1 test acc | BR-FedBi | 566.09 | 1825.56 | 584.79 | 1724.10 | 1591.81 | 1602.61 |
| | BILANTINE | 1315.46 | 1650.53 | 448.32 | 1856.44 | 766.82 | 876.82 |

## D.3   HYPER-REPRESENTATION

**Hyper-Representation.** Following the formulation presented in (Franceschi et al., 2018), the hyper-representation problem can be described as:

$$\min_{\lambda} \mathcal{L}(\lambda) = \frac{1}{n} \sum_{i=1}^{n} \frac{1}{|\mathcal{D}_v^i|} \sum_{\phi \in \mathcal{D}_v^i} \mathcal{L}\left(\theta^*(\lambda), \lambda; \phi\right)$$

$$\text{s.t.} \quad \theta^*(\lambda) = \arg\min_{\theta} \frac{1}{n} \sum_{i=1}^{n} \frac{1}{|\mathcal{D}_\tau^i|} \sum_{\varphi \in \mathcal{D}_\tau^i} \mathcal{L}(\theta, \lambda; \varphi),$$

where $\mathcal{D}_\tau^i$ and $\mathcal{D}_v^i$ are training and validation datasets of each client respectively. The last fully connected layer serves as the lower-level objective, and the remaining parameters serve as the upper-level objective. We conduct experiments on MNIST using a 2-layer multilayer perceptron and on CIFAR-10 using a CNN, where the last fully connected layer serves as the lower-level objective and the remaining parameters as the upper-level objective.

### D.3.1   IMPLEMENT DETAILS

In our experiments, we test two datasets: MNIST (LeCun, 1998) and CIFAR-10 (Krizhevsky et al., 2009). For MNIST, we use a 2-layer multilayer perceptron (MLP) with 200 hidden units. For CIFAR-10, we employ a convolutional neural network (CNN) with two convolutional layers, followed by two fully connected layers. In both models, we designate the fully connected layer as the lower-level objective, while the remaining parameters are treated as the upper-level objective. Details of the model architectures and datasets are summarized in Table 6.

**Wall-Clock Running Time Results.** We also present the wall-clock running time of the evaluated algorithms on the MNIST dataset over 200 training epochs. The results demonstrate that the single-loop algorithms have lower per-epoch training time. Regarding the communication workload, the sub-loop algorithm requires 8 communication operations per epoch, whereas the single-loop algorithm requires only 3, corresponding to its three variable updates. To further substantiate these observations,

Table 6: Implement details.

| Details | MNIST | CIFAR-10 |
|---|---|---|
| Model type | 2-layer MLP | CNN |
| Model architecture | Input: 784, Hidden: 200, Output: 10 | Conv1: $3 \times 6 \times 5$, Conv2: $6 \times 16 \times 5$, FC: $120 \to 84 \to 10$ |
| Optimizer | SGD | SGD |
| Learning rate ($\eta$) | $\{\eta_y, \eta_z, \eta_x\} \in \{0.1, 0.02, 0.01\}$ | $\{\eta_y, \eta_z, \eta_x\} \in \{0.1, 0.05, 0.03\}$ |
| Batch size ($b$) | $b = 64$ | $b = 64$ |
| Momentum ($\beta$) | $\beta \in \{0.5, 0.5, 0.5\}$ | $\beta \in \{0.5, 0.5, 0.5\}$ |
| Probabilistic coefficient ($p$) | $p = 0.5$ | $p = 0.5$ |
| Server-side update (iteration) | 200 | 1000 |
| Number of Byzantine workers | $f = 7$ | $f = 7$ |
| Number of honest workers | $n - f = 18$ | $n - f = 18$ |

we report the wall-clock time required for each method to reach $50\%$, $60\%$, $75\%$, and $85\%$ of the top-1 test accuracy in Table 7, Table 8, Table 9, and Table 10, respectively. As evidenced by the results, BR-FEDBI generally reaches the target accuracy in less time than BILANTINE.

Table 7: The wall clock running time (s) for different methods with 200 epochs on MNIST dataset.

| Algorithm | Krum | TM | Median | RFA | CWMed | CWTM |
|---|---|---|---|---|---|---|
| BILANTINE | 7701.42 | 7202.51 | 7229.27 | 7458.81 | 6590.10 | 6523.29 |
| BR-FedBi | 3138.29 | 3139.92 | 3102.36 | 3132.67 | 3138.48 | 3126.39 |
| BR-FedBiM | 3200.38 | 3148.26 | 3136.17 | 3173.29 | 3169.27 | 3153.02 |
| BR-FedBiP | 3158.74 | 3166.83 | 3165.91 | 3207.01 | 3238.28 | 3203.91 |

Table 8: The wall clock running time (s) for different methods to reach $50\%$ top-1 test accuracy on MNIST dataset under ALIE attack on MNIST dataset.

| Algorithm | Krum | TM | Median | RFA | CWMed | CWTM |
|---|---|---|---|---|---|---|
| BILANTINE | 871.26 | 1073.19 | 888.60 | 472.37 | 326.59 | 189.20 |
| BR-FedBi | 78.29 | 78.26 | 77.48 | 78.94 | 119.39 | 116.83 |
| BR-FedBiM | 128.38 | 129.36 | 126.37 | 126.22 | 140.37 | 140.38 |
| BR-FedBiP | 46.94 | 109.28 | 109.29 | 110.20 | 115.37 | 115.20 |

Table 9: The wall clock running time (s) for different methods to reach $60\%$ top-1 test accuracy on MNIST dataset under ALIE attack on MNIST dataset.

| Algorithm | Krum | TM | Median | RFA | CWMed | CWTM |
|---|---|---|---|---|---|---|
| BILANTINE | 1176.96 | 1215.51 | 1104.37 | 545.89 | 404.07 | 557.89 |
| BR-FedBi | 110.28 | 109.93 | 108.73 | 109.93 | 146.93 | 112.39 |
| BR-FedBiM | 128.38 | 129.36 | 126.37 | 126.22 | 140.37 | 148.30 |
| BR-FedBiP | 109.92 | 109.28 | 139.95 | 110.20 | 147.48 | 115.20 |

Table 10: The wall clock running time (s) for different methods to reach $75\%$ top-1 test accuracy on MNIST dataset under ALIE attack on MNIST dataset.

| Algorithm | Krum | TM | Median | RFA | CWMed | CWTM |
|---|---|---|---|---|---|---|
| BILANTINE | 1407.85 | 1466.47 | 1392.71 | 728.29 | 660.92 | 557.63 |
| BR-FedBi | 188.27 | 187.56 | 185.36 | 186.82 | 184.28 | 187.29 |
| BR-FedBiM | 255.37 | 254.17 | 254.73 | 253.22 | 276.39 | 279.40 |
| BR-FedBiP | 155.91 | 170.92 | 170.02 | 173.84 | 179.62 | 179.20 |

Table 11: The wall clock running time (s) for different methods to reach $85\%$ top-1 test accuracy on MNIST dataset under ALIE attack on MNIST dataset.

| Algorithm | Krum | TM | Median | RFA | CWMed | CWTM |
|---|---|---|---|---|---|---|
| BILANTINE | 1791.57 | 1574.27 | 1500.40 | 838.84 | 726.38 | 641.68 |
| BR-FedBi | 234.27 | 264.73 | 232.82 | 264.29 | 332.27 | 336.23 |
| BR-FedBiM | 652.36 | 379.27 | 376.39 | 380.36 | 360.29 | 357.29 |
| BR-FedBiP | 310.28 | 309.62 | 309.29 | 299.64 | 322.39 | 302.29 |

### D.3.2 ADDITIONAL EXPERIMENT RESULTS

In this subsection, we empirically compare the performance of the proposed Single-loop Byzantine-resilient Bilevel algorithms with BILANTINE (Abbas et al., 2024) on MNIST dataset. We compare the performance of the methods under different fractions of attackers on MNIST dataset. We choose the number of attackers, $f = \{3, 5, 7, 10\}$, out of $n = 25$. Under different numbers of clients, our proposed algorithms achieve better performance than the baseline method, which shows that our methods are more robust to the changing number of attackers.

Table 12: Comparison of top-1 test accuracy (%) for hyper-representation task on MNIST with heterogeneous Data, under BF attack for different numbers of attackers.

| Algorithm | RFA | | | | Algorithm | CWMed | | | |
|---|---|---|---|---|---|---|---|---|---|
| | $f = 3$ | $f = 5$ | $f = 7$ | $f = 10$ | | $f = 3$ | $f = 5$ | $f = 7$ | $f = 10$ |
| BILANTINE | 37.84% | 35.29% | 32.02% | 27.86% | BILANTINE | 69.20% | 53.48% | 51.02% | 35.72% |
| BR-FedBi | 82.84% | 82.08% | 79.41% | 49.75% | BR-FedBi | 82.83% | 82.40% | 79.71% | 55.23% |
| BR-FedBiM | 82.69% | 82.49% | 81.96% | 78.46% | BR-FedBiM | 82.68% | 82.50% | 82.24% | 80.98% |
| BR-FedBiP | 82.83% | 82.10% | 79.26% | 49.77% | BR-FedBiP | 82.84% | 82.37% | 79.71% | 55.31% |
| Algorithm | Median | | | | Algorithm | Krum | | | |
| | $f = 3$ | $f = 5$ | $f = 7$ | $f = 10$ | | $f = 3$ | $f = 5$ | $f = 7$ | $f = 10$ |
| BILANTINE | 66.85% | 61.82% | 46.02% | 47.17% | BILANTINE | 57.57% | 53.74% | 52.26% | 40.83% |
| BR-FedBi | 82.88% | 82.51% | 82.00% | 77.93% | BR-FedBi | 83.06% | 83.02% | 82.92% | 82.82% |
| BR-FedBiM | 82.91% | 82.55% | 82.51% | 82.15% | BR-FedBiM | 82.82% | 82.83% | 82.92% | 82.70% |
| BR-FedBiP | 82.86% | 82.59% | 82.25% | 77.93% | BR-FedBiP | 82.90% | 82.92% | 82.84% | 82.70% |
| Algorithm | CWTM | | | | Algorithm | TM | | | |
| | $f = 3$ | $f = 5$ | $f = 7$ | $f = 10$ | | $f = 3$ | $f = 5$ | $f = 7$ | $f = 10$ |
| BILANTINE | 10.20% | 45.68% | 10.90% | 21.11% | BILANTINE | 63.23% | 55.43% | 41.84% | 13.34% |
| BR-FedBi | 82.19% | 60.02% | 50.14% | 32.33% | BR-FedBi | 82.98% | 82.89% | 82.61% | 82.53% |
| BR-FedBiM | 82.51% | 81.65% | 78.64% | 20.69% | BR-FedBiM | 82.74% | 82.67% | 82.69% | 81.46% |
| BR-FedBiP | 82.16% | 59.96% | 50.95% | 34.71% | BR-FedBiP | 83.03% | 82.87% | 82.58% | 81.46% |

Table 13: Comparison of top-1 test accuracy (%) for hyper-representation task on MNIST with heterogeneous Data, under IPM attack for different numbers of attackers.

| Algorithm | RFA | | | | Algorithm | CWMed | | | |
|---|---|---|---|---|---|---|---|---|---|
| | $f = 3$ | $f = 5$ | $f = 7$ | $f = 10$ | | $f = 3$ | $f = 5$ | $f = 7$ | $f = 10$ |
| BILANTINE | 23.06% | 18.74% | 19.97% | 17.54% | BILANTINE | 67.72% | 60.26% | 58.35% | 48.75% |
| BR-FedBi | 82.78% | 82.20% | 80.70% | 51.68% | BR-FedBi | 82.84% | 82.28% | 81.51% | 65.18% |
| BR-FedBiM | 82.65% | 82.44% | 81.88% | 65.32% | BR-FedBiM | 82.63% | 82.52% | 82.31% | 79.67% |
| BR-FedBiP | 82.80% | 82.22% | 80.69% | 51.78% | BR-FedBiP | 82.85% | 82.29% | 81.62% | 63.75% |
| Algorithm | Median | | | | Algorithm | Krum | | | |
| | $f = 3$ | $f = 5$ | $f = 7$ | $f = 10$ | | $f = 3$ | $f = 5$ | $f = 7$ | $f = 10$ |
| BILANTINE | 67.33% | 55.85% | 49.85% | 48.64% | BILANTINE | 63.85% | 62.06% | 61.11% | 60.16% |
| BR-FedBi | 82.85% | 81.25% | 81.28% | 72.54% | BR-FedBi | 83.06% | 83.02% | 82.91% | 82.82% |
| BR-FedBiM | 82.70% | 82.76% | 82.60% | 81.71% | BR-FedBiM | 82.82% | 82.83% | 82.92% | 82.67% |
| BR-FedBiP | 82.85% | 81.63% | 81.36% | 72.54% | BR-FedBiP | 82.93% | 82.88% | 82.80% | 82.67% |
| Algorithm | CWTM | | | | Algorithm | TM | | | |
| | $f = 3$ | $f = 5$ | $f = 7$ | $f = 10$ | | $f = 3$ | $f = 5$ | $f = 7$ | $f = 10$ |
| BILANTINE | 30.94% | 36.09% | 11.35% | 19.58% | BILANTINE | 55.46% | 54.68% | 46.05% | 16.11% |
| BR-FedBi | 82.59% | 80.16% | 61.23% | 18.67% | BR-FedBi | 82.99% | 82.80% | 82.54% | 82.34% |
| BR-FedBiM | 82.52% | 81.22% | 72.68% | 26.21% | BR-FedBiM | 82.79% | 82.67% | 82.61% | 80.98% |
| BR-FedBiP | 82.60% | 80.14% | 61.40% | 18.67% | BR-FedBiP | 82.97% | 82.85% | 82.58% | 80.98% |

Table 14: Comparison of top-1 test accuracy (%) for hyper-representation task on MNIST with heterogeneous Data, under ALIE attack for different numbers of attackers.

| Algorithm | RFA | | | | Algorithm | CWMed | | | |
|---|---|---|---|---|---|---|---|---|---|
| | $f = 3$ | $f = 5$ | $f = 7$ | $f = 10$ | | $f = 3$ | $f = 5$ | $f = 7$ | $f = 10$ |
| BILANTINE | 81.36% | 76.78% | 66.96% | 62.95% | BILANTINE | 70.46% | 69.28% | 56.88% | 45.54% |
| BR-FedBi | 82.39% | 79.26% | 70.83% | 53.67% | BR-FedBi | 81.92% | 71.93% | 58.25% | 54.39 % |
| BR-FedBiM | 82.46% | 82.25% | 72.76% | 53.71% | BR-FedBiM | 82.32% | 74.03% | 69.38% | 54.31 % |
| BR-FedBiP | 82.40% | 79.25% | 70.80% | 53.68% | BR-FedBiP | 81.99% | 71.94% | 58.25% | 54.39 % |
| Algorithm | Median | | | | Algorithm | Krum | | | |
| | $f = 3$ | $f = 5$ | $f = 7$ | $f = 10$ | | $f = 3$ | $f = 5$ | $f = 7$ | $f = 10$ |
| BILANTINE | 65.69% | 74.73% | 73.74% | 72.44% | BILANTINE | 54.32% | 53.67% | 53.54% | 52.97 % |
| BR-FedBi | 84.02% | 83.56% | 83.11% | 82.98% | BR-FedBi | 83.06% | 83.02% | 82.92% | 82.82 % |
| BR-FedBiM | 83.37% | 83.16% | 79.53% | 82.99% | BR-FedBiM | 82.79% | 82.83% | 82.92% | 82.72 % |
| BR-FedBiP | 84.02% | 83.58% | 83.24% | 82.98% | BR-FedBiP | 83.01% | 82.92% | 82.90% | 82.72 % |
| Algorithm | CWTM | | | | Algorithm | TM | | | |
| | $f = 3$ | $f = 5$ | $f = 7$ | $f = 10$ | | $f = 3$ | $f = 5$ | $f = 7$ | $f = 10$ |
| BILANTINE | 76.63% | 72.22% | 56.88% | 50.30% | BILANTINE | 75.80% | 69.80% | 73.96% | 22.54 % |
| BR-FedBi | 82.13% | 56.71% | 58.25% | 31.09% | BR-FedBi | 83.31% | 83.35% | 83.27% | 82.90 % |
| BR-FedBiM | 82.19% | 65.06% | 59.52% | 31.95% | BR-FedBiM | 83.73% | 83.07% | 83.07% | 83.01 % |
| BR-FedBiP | 82.15% | 56.71% | 58.25% | 32.06% | BR-FedBiP | 83.31% | 83.35% | 83.27% | 82.90 % |

Table 15: Comparison of top-1 test accuracy (%) for hyper-representation task on MNIST with heterogeneous Data, under RN attack for different numbers of attackers.

| Algorithm | RFA | | | | Algorithm | CWMed | | | |
|---|---|---|---|---|---|---|---|---|---|
| | $f=3$ | $f=5$ | $f=7$ | $f=10$ | | $f=3$ | $f=5$ | $f=7$ | $f=10$ |
| BILANTINE | 71.32% | 69.95% | 67.68% | 9.21% | BILANTINE | 68.79% | 64.06% | 63.32% | 58.41% |
| BR-FedBi | 82.79% | 82.61% | 82.49% | 81.14% | BR-FedBi | 82.85% | 82.77% | 82.63% | 81.54% |
| BR-FedBiM | 82.61% | 82.53% | 82.60% | 81.24% | BR-FedBiM | 82.66% | 82.65% | 82.57% | 82.28% |
| BR-FedBiP | 82.83% | 82.60% | 82.62% | 80.35% | BR-FedBiP | 82.89% | 82.76% | 82.61% | 81.42% |
| Algorithm | Median | | | | Algorithm | Krum | | | |
| | $f=3$ | $f=5$ | $f=7$ | $f=10$ | | $f=3$ | $f=5$ | $f=7$ | $f=10$ |
| BILANTINE | 66.87% | 66.47% | 63.12% | 60.66% | BILANTINE | 57.57% | 57.25% | 45.62% | 44.64% |
| BR-FedBi | 82.85% | 82.66% | 82.65% | 80.78% | BR-FedBi | 82.87% | 82.57% | 82.78% | 82.45% |
| BR-FedBiM | 82.77% | 82.54% | 82.60% | 82.42% | BR-FedBiM | 82.70% | 82.50% | 82.50% | 82.51% |
| BR-FedBiP | 82.86% | 82.84% | 82.73% | 80.78% | BR-FedBiP | 82.90% | 82.57% | 82.85% | 82.51% |
| Algorithm | CWTM | | | | Algorithm | TM | | | |
| | $f=3$ | $f=5$ | $f=7$ | $f=10$ | | $f=3$ | $f=5$ | $f=7$ | $f=10$ |
| BILANTINE | 73.18% | 62.31% | 62.06% | 12.27% | BILANTINE | 71.36% | 65.33% | 64.19% | 15.36% |
| BR-FedBi | 82.69% | 82.28% | 81.65% | 75.29% | BR-FedBi | 82.82% | 82.50% | 82.66% | 82.11% |
| BR-FedBiM | 82.76% | 82.23% | 82.11% | 64.20% | BR-FedBiM | 82.46% | 82.28% | 82.44% | 82.34% |
| BR-FedBiP | 82.68% | 82.28% | 81.13% | 73.26% | BR-FedBiP | 82.78% | 82.52% | 82.61% | 82.34% |

**Impact of number of clients** We evaluate the performance of the methods under different numbers of clients, where we set $n = \{10, 25, 50\}$ and select the corresponding numbers of attackers as $f = \{3, 7, 14\}$. The top-1 test accuracy under these settings is presented in Figure 1. Across all configurations, the proposed algorithms consistently outperform the baseline methods. These empirical results demonstrate that the proposed methods maintain strong robustness even as the number of clients varies.

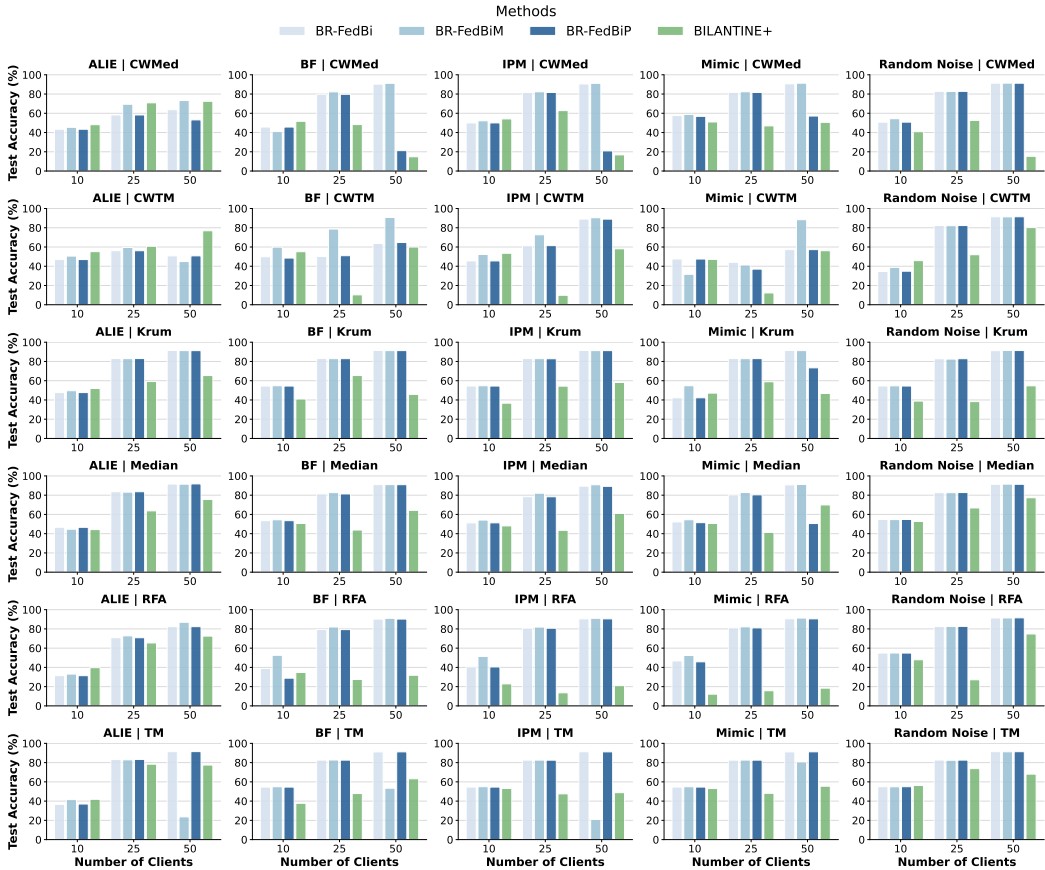

Figure 1: Impact of the number of participating clients on the top-1 test accuracy for the MNIST data hyper-representation task. The figure compares the performance of different aggregation methods as the number of clients varies.

**Performance under IID Setting.** We evaluate the methods on a homogeneous MNIST dataset with $f = 7$ Byzantine attackers among $n = 25$ workers. As shown in Table 16, the proposed approaches consistently outperform the baseline method (BILANTINE) under the IID setting. In particular, BR-FedBiM achieves the strongest worst-case performance across most aggregation rules, attaining 95.82% accuracy under Krum and 92.68% under CWTM during BF attacks. These results highlight the robustness and adaptability of the proposed methods in both IID and non-IID scenarios.

Table 16: Hyper-representation on MNIST with iid data. For each block, the algorithm with the highest test accuracy (%) under each attack is highlighted in **bold**. "Worst" shows the worst test accuracy among all attacks.

| AGG | Algorithm | ALIE | BF | IPM | RN | Worst | AGG | Algorithm | ALIE | BF | IPM | RN | Worst |
|-----|-----------|------|------|------|------|-------|-----|-----------|------|------|------|------|-------|
| Krum | BILANTINE | 84.52 | 84.88 | 24.47 | 85.11 | 24.47 | TM | BILANTINE | 84.63 | 81.52 | 81.08 | 84.94 | 81.08 |
| | BR-FedBi | 95.76 | 95.76 | 95.76 | 95.68 | 95.68 | | BR-FedBi | 95.88 | 95.24 | 95.25 | 95.71 | 95.24 |
| | BR-FedBiM | **95.82** | **95.82** | **95.82** | **95.93** | **95.82** | | BR-FedBiM | 95.86 | **95.65** | **95.65** | **95.92** | **95.65** |
| | BR-FedBiP | 95.75 | 95.75 | 95.76 | 95.70 | 95.70 | | BR-FedBiP | **95.89** | 95.25 | 95.24 | 95.74 | 95.24 |
| RFA | BILANTINE | 84.20 | 43.19 | 54.60 | 85.53 | 43.19 | CWMed | BILANTINE | 85.74 | 80.33 | 82.49 | 84.18 | 80.33 |
| | BR-FedBi | 95.69 | 94.81 | 94.87 | 95.67 | **94.81** | | BR-FedBi | 95.36 | 94.82 | 94.97 | 95.78 | 94.82 |
| | BR-FedBiM | **95.87** | **95.44** | 94.35 | **95.93** | 94.35 | | BR-FedBiM | **95.79** | **95.44** | **95.48** | **95.89** | **95.44** |
| | BR-FedBiP | 95.69 | 94.80 | **94.87** | 95.68 | 94.80 | | BR-FedBiP | 95.36 | 94.82 | 94.97 | 95.77 | 94.82 |
| Median | BILANTINE | 84.92 | 80.57 | 81.58 | 84.16 | 80.57 | CWTM | BILANTINE | 60.82 | 15.00 | 16.66 | 51.96 | 15.00 |
| | BR-FedBi | **95.93** | 94.98 | 94.90 | 95.73 | 94.90 | | BR-FedBi | 90.88 | 81.34 | 91.12 | 95.40 | 81.34 |
| | BR-FedBiM | 95.92 | **95.52** | **95.49** | **95.86** | **95.49** | | BR-FedBiM | **94.95** | **92.68** | 92.34 | **95.70** | **92.34** |
| | BR-FedBiP | **95.93** | 94.99 | 94.90 | 95.75 | 94.90 | | BR-FedBiP | 90.91 | 82.74 | **91.13** | 95.46 | 82.74 |

**Experiments with different seeds.** We evaluate the robustness and consistency of our methods under different random seeds by running each experiment with different seeds. Table 17 shows the performance on MNIST with heterogeneous data, while Table 18 reports results on CIFAR-10 with homogeneous data. Each block groups algorithms using the same aggregation rule, and we report the mean and standard deviation of the test accuracy under various Byzantine attacks. The "Worst" column highlights the lowest test accuracy across all attack types for each method. Our proposed BR-FedBiP and BR-FedBiM consistently outperform BR-FedBi and the baseline BILANTINE in most cases, showing both strong robustness and stability.

Table 17: Hyper-representation on MNIST with heterogeneous data. "Worst" shows the worst test accuracy among all attacks. Each block groups 4 algorithms under the same aggregation rule.

| AGG | Algorithm | ALIE | BF | IPM | RN | Worst |
|-----|-----------|------|------|------|------|-------|
| Median | BILANTINE | 73.74±1.46 | 46.02±2.36 | 49.85±1.83 | 63.12±1.87 | 46.02±2.36 |
| | BR-FedBi | 83.11±1.64 | 82.00±1.98 | 81.28±1.93 | 82.65±1.91 | 81.28±1.93 |
| | BR-FedBiP | **83.24±2.28** | 82.24±1.74 | 81.36±1.83 | **82.73±1.49** | **81.36±1.83** |
| | BR-FedBiM | 79.53±1.28 | **82.51±1.79** | **82.60±1.81** | 82.59±2.44 | 79.53±1.28 |
| RFA | BILANTINE | 66.96±1.73 | 32.02±2.75 | 19.96±1.57 | 67.68±1.81 | 19.96±1.57 |
| | BR-FedBi | 70.83±3.26 | 79.41±1.90 | 80.70±3.24 | 82.49±1.83 | 70.83±3.26 |
| | BR-FedBiP | 70.80±2.19 | 79.26±2.26 | 80.69±2.57 | **82.62±1.97** | 70.80±2.19 |
| | BR-FedBiM | **72.76±1.73** | **81.95±2.69** | **81.87±1.74** | 82.60±2.10 | **72.76±1.73** |
| TM | BILANTINE | 73.96±2.80 | 41.84±2.17 | 46.05±1.95 | 64.19±2.19 | 41.84±2.17 |
| | BR-FedBi | 83.27±1.89 | 82.61±2.14 | 82.54±1.76 | **82.66±2.00** | 82.54±1.76 |
| | BR-FedBiP | **83.29±2.28** | 82.58±1.77 | 82.58±1.83 | 82.61±1.59 | **82.58±1.83** |
| | BR-FedBiM | 82.44±3.44 | **82.69±1.77** | **82.61±1.81** | 82.44±2.44 | 82.44±3.44 |
| CWMed | BILANTINE | 56.88±1.81 | 51.02±1.53 | 58.40±1.78 | 63.32±1.87 | 51.02±1.53 |
| | BR-FedBi | 58.25±1.28 | 79.70±2.67 | 81.51±1.83 | 82.76±2.10 | 58.25±1.28 |
| | BR-FedBiP | **58.25±1.92** | 79.70±1.35 | 81.62±1.81 | **82.76±2.27** | 58.25±1.92 |
| | BR-FedBiM | 69.37±1.63 | **82.23±2.62** | **82.30±1.02** | 82.66±2.10 | **69.37±1.63** |

Table 18: Hyper-representation on CIFAR-10 with homogeneous data. "Worst" shows the worst test accuracy among all attacks. Each block groups 3 algorithms under the same aggregation rule.

| AGG | Algorithm | ALIE | BF | IPM | RN | Worst |
|---|---|---|---|---|---|---|
| Median | BR-FedBi | 60.61±1.39 | 59.36±1.03 | 63.22±1.32 | 43.70±1.12 | 43.70±1.12 |
| | BR-FedBiP | **63.39±1.55** | 62.04±1.44 | 59.36±1.07 | **62.86±1.35** | 59.36±1.07 |
| | BR-FedBiM | 63.12±1.52 | **62.68±1.09** | **63.40±1.48** | 62.67±0.98 | **62.67±0.98** |
| RFA | BR-FedBi | 51.59±1.09 | 57.52±1.36 | **63.13±1.13** | 43.02±1.62 | 43.02±1.62 |
| | BR-FedBiP | 49.36±1.45 | 52.67±1.04 | 57.52±1.56 | 62.91±1.36 | 49.36±1.45 |
| | BR-FedBiM | **54.83±1.38** | **62.83±1.12** | 62.47±1.25 | **63.09±1.48** | **54.83±1.48** |
| TM | BR-FedBi | 61.31±0.39 | 62.92±0.34 | 63.11±0.95 | 42.45±1.30 | 42.45±1.30 |
| | BR-FedBiP | 63.22±1.02 | 62.09±1.12 | 62.92±0.26 | 63.22±0.51 | 62.09±1.12 |
| | BR-FedBiM | **63.64±1.83** | **63.34±1.96** | **63.18±1.71** | **63.74±0.51** | **63.18±1.71** |
| Krum | BR-FedBi | 63.49±0.58 | 63.16±0.23 | 63.06±1.82 | 43.60±1.27 | 43.60±1.27 |
| | BR-FedBiP | **63.72±0.38** | **63.49±0.42** | **63.16±0.75** | 63.17±0.46 | **63.16±0.75** |
| | BR-FedBiM | 63.19±1.13 | 62.80±0.92 | 63.09±1.08 | **63.21±3.80** | 62.80±0.92 |
| CWTM | BR-FedBi | 48.84±0.64 | 54.83±0.82 | **63.14±0.55** | 42.01±1.01 | 42.01±1.01 |
| | BR-FedBiP | 37.70±1.60 | 48.84±0.67 | 54.83±0.51 | 61.40±0.51 | 37.70±1.60 |
| | BR-FedBiM | **51.92±0.57** | **62.21±1.03** | 60.37±0.63 | **62.92±0.33** | **51.92±0.57** |
| CWMed | BR-FedBi | **54.45±0.50** | **57.38±0.76** | **63.70±1.09** | 43.31±1.68 | 43.31±1.68 |
| | BR-FedBiP | 47.26±1.21 | 54.45±1.47 | 57.38±1.02 | **63.51±1.91** | 47.26±1.21 |
| | BR-FedBiM | 47.90±1.80 | 52.70±1.72 | 58.77±0.71 | 63.08±1.38 | **47.90±1.80** |

# E    LOWER BOUND FOR ASYMPTOTIC ERROR (THEOREM 1)

In this section, we establish the lower bound on the asymptotic error introduced by $\delta$-fraction of Byzantine attackers in distributed bilevel optimization. We assume that the gradients of the lower-level and upper-level loss functions satisfy the $(\zeta_g, B_g)$-gradient dissimilarity assumption and the $(\zeta_f, B_f)$-gradient dissimilarity assumption, respectively, as specified in Assumption 8.

**Assumption 8** (($\zeta, B$)-gradient/Hessian dissimilarity). *The local gradient/Hessian $\Lambda_i$ of the honest worker $i \in \mathcal{H}$ is said to satisfy $(\zeta, B)$-gradient/Hessian dissimilarity if*

$$\frac{1}{|\mathcal{H}|} \sum_{i \in \mathcal{H}} \|\Lambda_i - \Lambda_{\mathcal{H}}\|^2 \le \zeta^2 + B^2 \|\Lambda_{\mathcal{H}}\|^2,$$

*where $\Lambda_{\mathcal{H}} = \frac{1}{|\mathcal{H}|} \sum_{i \in \mathcal{H}} \Lambda_i$.*

- *The local gradients $\nabla \Phi_i(\mathbf{x}), \nabla f_i(\mathbf{x}, \mathbf{y})$ of the honest worker $i \in \mathcal{H}$ satisfy the $(\zeta_f, B_f)$-gradient dissimilarity assumption.*

- *The local gradient/Hessian $\nabla_{\mathbf{y}} g_i(\mathbf{x}, \mathbf{y}), \nabla^2_{yy} g_i(\mathbf{x}, \mathbf{y}), \nabla^2_{xy} g_i(\mathbf{x}, \mathbf{y})$ of honest worker $i \in \mathcal{H}$ satisfy the $(\zeta_g, B_g)$-gradient/ Hessian dissimilarity assumption.*

Our proof is inspired by the work of Allouah et al. (2024) and extends their work to bilevel optimization. Now, we restate Theorem 1.

**Theorem 1.** *For an arbitrary distributed bilevel algorithm, there always exists a federated bilevel optimization problem satisfying Assumptions 1-4, such that:*

$$\|\nabla \Phi_{\mathcal{H}}(\hat{\mathbf{x}})\|^2 \ge \Omega \left( \frac{\delta \left( \zeta_f^2 + \zeta_g^2 \right)}{1 - \delta \left( 2 + B_f^2 \right)} \right), \tag{16}$$

*where $\delta < \min \left\{ \frac{1}{2+B_f^2}, \frac{1}{2+B_g^2} \right\}$ is the proportion of attackers among clients and $\hat{\mathbf{x}}$ is the output of the algorithm.*

*Sketch of proof.* In the proof, we construct a set of quadratic bilevel loss functions for $n$ clients, denoted by $\{\Phi_1(\mathbf{x}), \dots, \Phi_n(\mathbf{x})\}$. We consider two quadratic FBO problems with attackers. In the first problem, the functions of the honest workers are denoted by $\Phi_{\mathcal{H}_1}(\mathbf{x}) = \{\Phi_1(\mathbf{x}), \dots, \Phi_{(1-\delta)n}(\mathbf{x})\}$, while the remaining $\{\Phi_{(1-\delta)n+1}(\mathbf{x}), \dots, \Phi_n(\mathbf{x})\}$ are the functions of attackers. In the second scenario, the functions of honest workers are $\Phi_{\mathcal{H}_2}(\mathbf{x}) = \{\Phi_{\delta n+1}(\mathbf{x}), \dots, \Phi_n(\mathbf{x})\}$, and the remaining are the functions of the attackers. The messages sent from the clients (including honest ones and attackers) to the server are the same for both problem settings. Regardless of the aggregation rule, the server, being unaware of the identity of the attackers, will produce the same output $\hat{\mathbf{x}}$ for the two FBO problems. However, the two problems have different optimal points and this single $\hat{\mathbf{x}}$ cannot be close to both points, which leads to our lower bound. □

We consider a distributed system consisting of $n$ clients, among which $\delta$-fraction of them are Byzantine attackers. For the proof, we construct a set of quadratic bilevel loss functions for the $n$ clients, denoted as $\{\Phi_1(\mathbf{x}), \dots, \Phi_n(\mathbf{x})\}$, each composed of a lower-level objective and an upper-level objective.

**Lower-level objective.** We consider the following lower-level objective for $n$ clients, where $\gamma_1, \gamma_2$ are positive constants, and $C_{\mathbf{x}}, C_{\mathbf{y}} \in \mathbb{R}^d$ are given positive vectors. For $i \in [n]$, the lower-level objective is defined as:

$$g_i(\mathbf{x}, \mathbf{y}) = \begin{cases} \frac{\gamma_1}{2} (\mathbf{x} - C_{\mathbf{x}})^2 + \frac{\gamma_1}{2} (\mathbf{y} - C_{\mathbf{y}})^2, & \text{for } i \in \{1, \dots, \delta n\}, \\ \frac{\gamma_2}{2} (\mathbf{x}^2 + \mathbf{y}^2), & \text{for } i \in \{\delta n + 1, \dots, (1-\delta)n\}, \\ \frac{\gamma_1}{2} (\mathbf{x}^2 + \mathbf{y}^2), & \text{for } i \in \{(1-\delta)n + 1, \dots, n\}. \end{cases} \tag{17}$$

**Upper-level objective.** We consider the following upper-level objective for $n$ clients, where $\alpha_1, \alpha_2$ are positive constants, and $C_{\mathbf{x}}, C_{\mathbf{y}} \in \mathbb{R}^d$ are vectors. For $i \in [n]$, the upper-level objective is defined

as:

$$f_i(\mathbf{x}, \mathbf{y}) = \begin{cases} \frac{\alpha_1}{2} (\mathbf{x} - C_{\mathbf{x}})^2 + \frac{\alpha_1}{2} (\mathbf{y} - C_{\mathbf{y}})^2 - \frac{\alpha_1}{\gamma_1} (\alpha_1\delta + \alpha_2(1 - 2\delta)) \, \mathbf{x} \cdot \mathbf{y} & \text{for } i \in \{1, \dots, \delta n\}, \\ \frac{\alpha_2}{2} (\mathbf{x}^2 + \mathbf{y}^2) - \frac{\alpha_1}{\gamma_1} (\alpha_1\delta + \alpha_2(1 - 2\delta)) \, \mathbf{x} \cdot \mathbf{y} & \text{for } i \in \{\delta n + 1, \dots, (1 - \delta)n\}, \\ \frac{\alpha_1}{2} (\mathbf{x}^2 + \mathbf{y}^2) - \frac{\alpha_1}{\gamma_1} (\alpha_1\delta + \alpha_2(1 - 2\delta)) \, \mathbf{x} \cdot \mathbf{y} & \text{for } i \in \{(1 - \delta)n + 1, \dots, n\}, \end{cases} \tag{18}$$

where $\Phi_i(\mathbf{x}) = f_i(\mathbf{x}, \mathbf{y}^\star(\mathbf{x}))$. To prove the theorem, we consider two scenarios, each corresponding to a different subset of honest workers and attackers in problem (18). In each scenario, $(1 - \delta)n$ clients are honest, while the remaining $\delta n$ clients are Byzantine attackers.

**The first scenario.** In the first scenario, we assume that honest workers have indices $\mathcal{H}_1 = \{1, \dots, (1 - \delta)n\}$, while the remaining are Byzantine attackers $\mathcal{B}_1 = \{(1 - \delta)n + 1, \dots, n\}$. The corresponding problem can be formulated as

$$\min_{\mathbf{x} \in \mathbb{R}^p} \ \Phi_{\mathcal{H}_1}(\mathbf{x}) = f_{\mathcal{H}_1}(\mathbf{x}, \mathbf{y}_1^\star(\mathbf{x})) = \frac{1}{|\mathcal{H}_1|} \sum_{i \in \mathcal{H}_1} f_i(\mathbf{x}, \mathbf{y}_1^\star(\mathbf{x})),$$

$$\text{subject to} \quad \mathbf{y}_1^\star(\mathbf{x}) = \arg\min_{\mathbf{y} \in \mathbb{R}^q} \frac{1}{|\mathcal{H}_1|} \sum_{i \in \mathcal{H}_1} g_i(\mathbf{x}, \mathbf{y}). \tag{19}$$

As defined in (17), the solution to the lower-level problem in (19) is given by

$$\mathbf{y}_1^\star(\mathbf{x}) = \frac{\gamma_1 \delta C_{\mathbf{y}}}{\gamma_1 \delta + \gamma_2 (1 - 2\delta)}. \tag{20}$$

As a result, the corresponding upper-level loss function in (19) for the $i$-th client is given by

$$f_i^1(\mathbf{x}, \mathbf{y}_1^\star(\mathbf{x})) = \begin{cases} \frac{\alpha_1}{2} (\mathbf{x} - C_{\mathbf{x}})^2 + \frac{\alpha_1}{2} (\mathbf{y}_1^\star(\mathbf{x}) - C_{\mathbf{y}})^2 - \alpha_1 \delta C_{\mathbf{y}} \mathbf{x}, & \text{for } i \in \{1, \dots, \delta n\}, \\ \frac{\alpha_2}{2} (\mathbf{x}^2 + (\mathbf{y}_1^\star(\mathbf{x}))^2) - \alpha_1 \delta C_{\mathbf{y}} \mathbf{x}, & \text{for } i \in \{\delta n + 1, \dots, (1 - \delta)n\}. \end{cases} \tag{21}$$

**The second scenario.** In the second setting, we assume that honest workers have indices $\mathcal{H}_2 = \{\delta n + 1, \dots, n\}$, while the remaining are Byzantine attackers $\mathcal{B}_2 = \{1, \dots, \delta n\}$. The corresponding problem is formulated as follows:

$$\min_{\mathbf{x} \in \mathbb{R}^p} \ \Phi_{\mathcal{H}_2}(\mathbf{x}) = f_{\mathcal{H}_2}(\mathbf{x}, \mathbf{y}_2^\star(\mathbf{x})) = \frac{1}{|\mathcal{H}_2|} \sum_{i \in \mathcal{H}_2} f_i(\mathbf{x}, \mathbf{y}_2^\star(\mathbf{x})),$$

$$\text{subject to} \quad \mathbf{y}_2^\star(\mathbf{x}) = \arg\min_{\mathbf{y} \in \mathbb{R}^q} \frac{1}{|\mathcal{H}_2|} \sum_{i \in \mathcal{H}_2} g_i(\mathbf{x}, \mathbf{y}). \tag{22}$$

As defined in (17), the solution to the lower-level problem in (22) is given by

$$\mathbf{y}_2^\star(\mathbf{x}) = 0. \tag{23}$$

As a result, the corresponding upper-level loss function in (22) for the honest clients is given by:

$$f_i^2(\mathbf{x}, \mathbf{y}_2^\star(\mathbf{x})) = \begin{cases} \frac{\alpha_2}{2} (\mathbf{x}^2 + \mathbf{y}_2^\star(\mathbf{x})^2), & \text{for } i \in \{\delta n + 1, \dots, (1 - \delta)n\}, \\ \frac{\alpha_1}{2} (\mathbf{x}^2 + \mathbf{y}_2^\star(\mathbf{x})^2), & \text{for } i \in \{(1 - \delta)n + 1, \dots, n\}. \end{cases} \tag{24}$$

Although these two sets of problems have different global optima, the server, unaware of the identity of the attackers, will produce the same result $\hat{\mathbf{x}}$ for the two FBO problems. However, the two problems have different optimal points, and this single $\hat{\mathbf{x}}$ cannot be close to both points, leading to our lower bound. This indicates that no aggregation algorithm can achieve an accurate model under Byzantine attacks.

### E.1 LOWER BOUND

In the following Lemma, we establish the minimal aggregation error.

**Lemma 1.** *Consider the two sets of bilevel optimization problems described above and assume that Assumption 4 holds. The asymptotic error is at least:*

$$\max_{j \in \{1,2\}} \Phi_{\mathcal{H}_j}(\hat{\mathbf{x}}) - \Phi_{\mathcal{H}_j}(\mathbf{x}_j^\star) \geq \frac{\delta^2 \alpha_1^2 \left(C_\mathbf{x}^2 + C_\mathbf{y}^2 + 2C_\mathbf{x}C_\mathbf{y}\right)}{8(1-\delta)(\alpha_1 \delta + \alpha_2(1-2\delta))}.$$

*Proof.* **The first scenario.** In the first setting, we assume that the indices of the honest workers are $\mathcal{H}_1 = \{1, \ldots, (1-\delta)n\}$. Recalling from (19), for any honest client $i \in \mathcal{H}_1$, the first set of lower-level loss functions is defined as:

$$g_i^1(\mathbf{x}, \mathbf{y}) = \begin{cases} \frac{\gamma_1}{2}(\mathbf{x} - C_\mathbf{x})^2 + \frac{\gamma_1}{2}(\mathbf{y} - C_\mathbf{y})^2 & \text{for } i \in \{1, ..., \delta n\}, \\ \frac{\gamma_2}{2}(\mathbf{x}^2 + \mathbf{y}^2) & \text{for } i \in \{\delta n + 1, ..., (1-\delta)n\}. \end{cases} \tag{25}$$

The solution of the lower level objective is given by $\mathbf{y}_1^\star(\mathbf{x}) = \frac{\delta \gamma_1 C_\mathbf{y}}{\gamma_1 \delta + \gamma_2(1-2\delta)}$. Recall from (19), for any honest client $i \in \mathcal{H}_1$, we consider the first set of the upper-level loss functions to be:

$$\Phi_i^1(\mathbf{x}) = f_i^1(\mathbf{x}, \mathbf{y}_1^\star(\mathbf{x})) = \begin{cases} \frac{\alpha_1}{2}(\mathbf{x} - C_\mathbf{x})^2 + \frac{\alpha_1}{2}(\mathbf{y}_1^\star(\mathbf{x}) - C_\mathbf{y})^2 - \alpha_1 \delta C_\mathbf{y}\mathbf{x} & \text{for } i \in \{1, ..., \delta n\}, \\ \frac{\alpha_2}{2}(\mathbf{x}^2 + \mathbf{y}_1^\star(\mathbf{x})^2) - \alpha_1 \delta C_\mathbf{y}\mathbf{x} & \text{for } i \in \{\delta n + 1, ..., n\}. \end{cases} \tag{26}$$

In the first setting, the global optima of the first set of loss functions (19) is given by

$$\mathbf{y}_1^\star(\mathbf{x}) = \frac{\delta \gamma_1 C_\mathbf{y}}{\gamma_1 \delta + \gamma_2(1-2\delta)}, \quad \mathbf{x}_1^\star = \frac{\delta \alpha_1 C_\mathbf{y} + \delta \alpha_1 C_\mathbf{x}}{\alpha_1 \delta + \alpha_2(1-2\delta)}. \tag{27}$$

**The second scenario.** In the second setting, we assume that the indices of the honest worker are $\mathcal{H}_2 = \{\delta n + 1, ..., n\}$. Recall from (22), for any honest client $i \in \mathcal{H}_2$, we consider the second set of the lower-level loss functions to be:

$$g_i^2(\mathbf{x}, \mathbf{y}) = \begin{cases} \frac{\gamma_2}{2}(\mathbf{x}^2 + \mathbf{y}^2) & \text{for } i \in \{\delta n + 1, ..., (1-\delta)n\}, \\ \frac{\gamma_1}{2}(\mathbf{x}^2 + \mathbf{y}^2) & \text{for } i \in \{(1-\delta)n + 1, ..., n\}. \end{cases} \tag{28}$$

The solution of the lower-level loss function is $\mathbf{y}_2^\star(\mathbf{x}) = 0$. Recall from (22), for any honest client $i \in \mathcal{H}_2$, we consider the second set of upper-level loss functions of the honest clients to be :

$$\Phi_i^2(\mathbf{x}) = f_i^2(\mathbf{x}, \mathbf{y}_2^\star(\mathbf{x})) = \begin{cases} \frac{\alpha_2}{2}(\mathbf{x}^2 + \mathbf{y}_2^\star(\mathbf{x})^2) & \text{for } i \in \{\delta n + 1, ..., (1-\delta)n\}, \\ \frac{\alpha_1}{2}(\mathbf{x}^2 + \mathbf{y}_2^\star(\mathbf{x})^2) & \text{for } i \in \{(1-\delta)n + 1, ..., n\}. \end{cases} \tag{29}$$

In the second setting, the global optima of the second set of loss functions (22) is given by

$$\mathbf{y}_2^\star(\mathbf{x}) = 0, \quad \mathbf{x}_2^\star = 0. \tag{30}$$

Although these two sets of problems have different global optima, the server, unaware of the identity of the attackers, will produce the same result $\hat{\mathbf{x}}$ for the two FBO problems. Then, we analyze the minimal error as follows. For the first scenario, we obtain that:

$$\Phi_{\mathcal{H}_1}(\hat{\mathbf{x}}) - \Phi_{\mathcal{H}_1}(\mathbf{x}_1^\star) = \frac{1}{2}\left(\frac{\delta}{1-\delta}\alpha_1 + \frac{1-2\delta}{1-\delta}\alpha_2\right)\|\hat{\mathbf{x}} - \mathbf{x}_1^\star\|^2 - \delta \alpha_1 C_\mathbf{y}(\hat{\mathbf{x}} - \mathbf{x}_1^\star)$$

$$\geq \frac{1}{2}\left(\frac{\delta}{1-\delta}\alpha_1 + \frac{1-2\delta}{1-\delta}\alpha_2\right)\|\hat{\mathbf{x}} - \mathbf{x}_1^\star\|^2. \tag{31}$$

where the inequality uses the output $\mathbf{x}_2^\star \leq \hat{\mathbf{x}} \leq \mathbf{x}_1^\star$, and $\mathbf{x}_1^\star$ is a positive vector. For the second set of loss functions, we can obtain:

$$\Phi_{\mathcal{H}_2}(\hat{\mathbf{x}}) - \Phi_{\mathcal{H}_2}(\mathbf{x}_2^\star) = \frac{1}{2}\left(\frac{\delta}{1-\delta}\alpha_1 + \frac{1-2\delta}{1-\delta}\alpha_2\right)\|\hat{\mathbf{x}}\|^2. \tag{32}$$

By using $\max\{a, b\} \geq \frac{a+b}{2}$, we have:

$$\max_{j \in \{1,2\}} \Phi_{\mathcal{H}_j}(\hat{\mathbf{x}}) - \Phi_{\mathcal{H}_j}(\mathbf{x}_j^\star) \geq \frac{1}{4}\left(\frac{\delta}{1-\delta}\alpha_1 + \frac{1-2\delta}{1-\delta}\alpha_2\right)\left(\|\hat{\mathbf{x}}\|^2 + \|\hat{\mathbf{x}} - \mathbf{x}_1^\star\|^2\right). \tag{33}$$

By Jensen's inequality, the following relationship holds:

$$\|\hat{\mathbf{x}}\|^2 + \|\hat{\mathbf{x}} - \mathbf{x}_1^\star\|^2 \geq \frac{1}{2}\|\mathbf{x}_1^\star\|^2. \tag{34}$$

Substituting this inequality in (34) into the expression (33), we obtain:

$$\max_{j \in \{1,2\}} \Phi_{\mathcal{H}_j}(\hat{\mathbf{x}}) - \Phi_{\mathcal{H}_j}(\mathbf{x}_j^\star) \geq \frac{1}{8}\left(\frac{\delta}{1-\delta}\alpha_1 + \frac{1-2\delta}{1-\delta}\alpha_2\right)\|\mathbf{x}_1^\star\|^2$$
$$\geq \frac{\delta^2 \alpha_1^2 \left(C_{\mathbf{x}}^2 + C_{\mathbf{y}}^2 + 2C_{\mathbf{x}}C_{\mathbf{y}}\right)}{8(1-\delta)(\alpha_1\delta + \alpha_2(1-2\delta))}. \tag{35}$$

Hence, the proof is complete. □

### E.2 SATISFACTION OF $(\zeta, B)$-GRADIENT HETEROGENEITY

We then establish the constraints for the constants and vectors $\{\gamma_1, \gamma_2, \alpha_1, \alpha_2, C_{\mathbf{x}}, C_{\mathbf{y}}\}$ in the bilevel loss function (17) and (18). These conditions ensure the gradients of the lower-level and upper-level loss functions satisfy the $(\zeta_g, B_g)$-gradient dissimilarity assumption and the $(\zeta_f, B_f)$-gradient dissimilarity assumption (Assumption 8), respectively.

---

**Lemma 2** (Satisfaction of $(\zeta, B)$-gradient heterogeneity). *Let us consider the two sets of bilevel optimization problems in Lemma 1. The condition for satisfying Assumption 4 is equivalent to the condition for the following inequalities to hold:*

$$A_{\mathbf{y},1} \leq 0, \quad A_{\mathbf{y},3} \leq 0 \quad and \quad \|A_{\mathbf{y},2}\|^2 - A_{\mathbf{y},1}A_{\mathbf{y},3} \leq 0,$$

*where*

$$A_{\mathbf{y},1} = \left(\frac{\delta(1-2\delta)}{(1-\delta)^2}(\gamma_1 - \gamma_2)^2 - B_g^2\left(\gamma_1\frac{\delta}{1-\delta} + \gamma_2\frac{1-2\delta}{1-\delta}\right)^2\right),$$

$$A_{\mathbf{y},2} = \left((1-2\delta)(\gamma_1 - \gamma_2) - B_g^2(\gamma_1\delta - \gamma_2(1-2\delta))\right)\frac{\delta}{(1-\delta)^2}\gamma_1 C_{\mathbf{y}},$$

$$A_{\mathbf{y},3} = \frac{\delta(1-2\delta)}{(1-\delta)^2}\gamma_1^2 C_{\mathbf{y}}^2 - B_g^2(\frac{\delta}{1-\delta})^2\gamma_1^2 C_{\mathbf{y}}^2 - \zeta_g^2.$$

*Similarly, we have:*

$$A_{\mathbf{x},1} \leq 0, \quad A_{\mathbf{x},3} \leq 0 \quad and \quad A_{\mathbf{x},2}^2 \leq A_{\mathbf{x},1}A_{\mathbf{x},3},$$

*where*

$$A_{\mathbf{x},1} = \left(\frac{\delta(1-2\delta)}{(1-\delta)^2}(\alpha_1 - \alpha_2)^2 - B_f^2\left(\alpha_1\frac{\delta}{1-\delta} + \alpha_2\frac{1-2\delta}{1-\delta}\right)^2\right),$$

$$A_{\mathbf{x},2} = \left(\frac{1-2\delta}{1-\delta}(\alpha_1 - \alpha_2)C_{\mathbf{x}} - B_f^2(\alpha_1\frac{\delta}{1-\delta} - \alpha_2\frac{1-2\delta}{1-\delta})(C_{\mathbf{x}} + C_{\mathbf{y}})\right)\frac{\delta}{1-\delta}\alpha_1,$$

$$A_{\mathbf{x},3} = \frac{\delta(1-2\delta)}{(1-\delta)^2}\alpha_1^2 C_{\mathbf{x}}^2 - B_f^2(\frac{\delta}{1-\delta})^2\alpha_1^2(C_{\mathbf{y}} + C_{\mathbf{x}})^2 - \zeta_f^2.$$

---

*Proof.* **The first scenario.** As defined in (19), in the first setting, we assume the indices of honest workers are $\mathcal{H}_1 = \{1, \ldots, (1-\delta)n\}$. For any honest client $i \in \mathcal{H}_1$, we define the first set of the lower-level loss functions as:

$$g_i^1(\mathbf{x}, \mathbf{y}) = \begin{cases} \frac{\gamma_1}{2}\left(\mathbf{x} - C_{\mathbf{x}}\right)^2 + \frac{\gamma_1}{2}\left(\mathbf{y} - C_{\mathbf{y}}\right)^2 & \text{for } i \in \{1, \ldots, \delta n\}, \\ \frac{\gamma_2}{2}\left(\mathbf{x}^2 + \mathbf{y}^2\right) & \text{for } i \in \{\delta n + 1, \ldots, (1-\delta)n\}. \end{cases} \tag{36}$$

The gradient of the global lower-level loss function in (36) is computed as follows:

$$\nabla_{\mathbf{y}}g_{\mathcal{H}_1}(\mathbf{x}, \mathbf{y}) = \frac{(\gamma_1\delta + \gamma_2(1-2\delta))\,\mathbf{y} - \delta\gamma_1 C_{\mathbf{y}}}{1 - \delta}. \tag{37}$$

Similarly, the gradient of the local lower-level loss function in (36) is computed as:

$$\nabla_{\mathbf{y}} g_i^1(\mathbf{x}, \mathbf{y}) = \begin{cases} \gamma_1 \mathbf{y} - \gamma_1 C_{\mathbf{y}} & \text{for } i \in \{1, ..., \delta n\}, \\ \gamma_2 \mathbf{y} & \text{for } i \in \{\delta n + 1, ..., (1 - \delta)n\}. \end{cases} \tag{38}$$

Combining (37) and (38), the gradient dissimilarity of the lower-level objectives can be bounded as:

$$\mathbb{E}\left[\left\|\nabla_{\mathbf{y}} g_i^1(\mathbf{x}, \mathbf{y}) - \nabla_{\mathbf{y}} g_{\mathcal{H}_1}(\mathbf{x}, \mathbf{y})\right\|^2\right] = \frac{\delta}{1 - \delta}\left((\gamma_1 - \gamma_2)\frac{1 - 2\delta}{1 - \delta}\mathbf{y} - \frac{1 - 2\delta}{1 - \delta}\gamma_1 C_{\mathbf{y}}\right)^2 \\ + \frac{1 - 2\delta}{1 - \delta}(\frac{\delta}{1 - \delta})^2\left((\gamma_2 - \gamma_1)\mathbf{y} + \gamma_1 C_{\mathbf{y}}\right)^2. \tag{39}$$

And the norm of the global gradient in (37) can be written as:

$$\mathbb{E}\left[\left\|\nabla_{\mathbf{y}} g_{\mathcal{H}_1}(\mathbf{x}, \mathbf{y})\right\|^2\right] = \left(\left(\gamma_1 \frac{\delta}{1 - \delta} + \gamma_2 \frac{1 - 2\delta}{1 - \delta}\right)\mathbf{y} - \frac{\delta}{1 - \delta}\gamma_1 C_{\mathbf{y}}\right)^2. \tag{40}$$

In order to satisfy $(\zeta, B)$-gradient heterogeneity in Assumption 4, we have:

$$\mathbb{E}\left[\left\|\nabla_{\mathbf{y}} g_i^1(\mathbf{x}, \mathbf{y}) - \nabla_{\mathbf{y}} g_{\mathcal{H}_1}(\mathbf{x}, \mathbf{y})\right\|^2\right] \leq B_g^2 \mathbb{E}\left[\left\|\nabla_{\mathbf{y}} g^1(\mathbf{x}, \mathbf{y})\right\|^2\right] + \zeta_g^2. \tag{41}$$

Combining equations (39) and (40) with (41), we have:

$$\mathbb{E}\left[\left\|\nabla_{\mathbf{y}} g_i^1(\mathbf{x}, \mathbf{y}) - \nabla_{\mathbf{y}} g_{\mathcal{H}_1}(\mathbf{x}, \mathbf{y})\right\|^2\right] - B_g^2 \mathbb{E}\left[\left\|\nabla_{\mathbf{y}} g_{\mathcal{H}_1}(\mathbf{x}, \mathbf{y})\right\|^2\right] - \zeta_g^2 \\ = \underbrace{\left(\frac{\delta(1 - 2\delta)}{(1 - \delta)^2}(\gamma_1 - \gamma_2)^2 - B_g^2\left(\gamma_1\frac{\delta}{1 - \delta} + \gamma_2\frac{1 - 2\delta}{1 - \delta}\right)^2\right)}_{A_{\mathbf{y},1}} \|\mathbf{y}\|^2 \\ - 2\underbrace{\left(\frac{1 - 2\delta}{1 - \delta}(\gamma_1 - \gamma_2) - B_g^2(\gamma_1\frac{\delta}{1 - \delta} - \gamma_2\frac{1 - 2\delta}{1 - \delta})\right)\frac{\delta}{1 - \delta}\gamma_1 C_{\mathbf{y}} \mathbf{y}}_{A_{\mathbf{y},2}} \\ + \underbrace{\frac{\delta(1 - 2\delta)}{(1 - \delta)^2}\gamma_1^2 C_{\mathbf{y}}^2 - B_g^2(\frac{\delta}{1 - \delta})^2\gamma_1^2 C_{\mathbf{y}}^2 - \zeta_g^2}_{A_{\mathbf{y},3}} \leq 0. \tag{42}$$

To ensure the validity of the above inequality, the following condition must be satisfied:

$$A_{\mathbf{y},1} \leq 0, \quad A_{\mathbf{y},3} \leq 0 \quad \text{and} \quad \|A_{\mathbf{y},2}\|^2 - A_{\mathbf{y},1} A_{\mathbf{y},3} \leq 0. \tag{43}$$

Recall from (21), for any honest worker $i \in \mathcal{H}_1$, we define the first set of the upper-level loss functions as:

$$f_i^1(\mathbf{x}, \mathbf{y}_1^\star(\mathbf{x})) = \begin{cases} \frac{\alpha_1}{2}\left(\mathbf{x} - C_{\mathbf{x}}\right)^2 + \frac{\alpha_1}{2}\left(\mathbf{y}_1^\star(\mathbf{x}) - C_{\mathbf{y}}\right)^2 - \alpha_1\delta C_{\mathbf{y}}\mathbf{x} & \text{for } i \in \{1, ..., \delta n\}, \\ \frac{\alpha_2}{2}\left(\mathbf{x}^2 + \mathbf{y}_1^\star(\mathbf{x})^2\right) - \alpha_1\delta C_{\mathbf{y}}\mathbf{x} & \text{for } i \in \{\delta n + 1, ..., n\}. \end{cases} \tag{44}$$

The gradient of the global upper-level objective in (44) is obtained as follows:

$$\nabla f_{\mathcal{H}_1}(\mathbf{x}, \mathbf{y}_1^\star(\mathbf{x})) = \frac{(\alpha_1\delta + \alpha_2(1 - 2\delta))\mathbf{x} - \delta\alpha_1 C_{\mathbf{y}} - \delta\alpha_1 C_{\mathbf{x}}}{1 - \delta}. \tag{45}$$

Similarly, the gradient of the local upper-level loss function in (44) is computed as:

$$\nabla f_i^1(\mathbf{x}, \mathbf{y}_1^\star(\mathbf{x})) = \begin{cases} \alpha_1\mathbf{x} - \alpha_1 C_{\mathbf{x}} - \delta\alpha_1 C_{\mathbf{y}} & \text{for } i \in \{1, ..., \delta n\}, \\ \alpha_2 x - \delta\alpha_1 C_{\mathbf{y}} & \text{for } i \in \{\delta n + 1, ..., (1 - \delta)n\}. \end{cases} \tag{46}$$

Combining (45) and (46), the gradient dissimilarity of the upper-level objectives can be bounded as:

$$
\mathbb{E}\left[\left\|\nabla f_i^1(\mathbf{x}) - \nabla f_{\mathcal{H}_1}(\mathbf{x})\right\|^2\right] = \frac{\delta(1-2\delta)^2}{(1-\delta)^3}\left((\alpha_1 - \alpha_2)\mathbf{x} - \alpha_1 C_{\mathbf{x}}\right)^2
$$
$$
+ \frac{(1-2\delta)\delta^2}{(1-\delta)^3}\left((\alpha_2 - \alpha_1)\mathbf{x} + \alpha_1 C_{\mathbf{x}}\right)^2.
\tag{47}
$$

And the norm of the global gradient in (45) can be written as:

$$
\mathbb{E}\left[\left\|\nabla f_{\mathcal{H}_1}(\mathbf{x})\right\|^2\right] = \left(\left(\alpha_1\frac{\delta}{1-\delta} + \alpha_2\frac{1-2\delta}{1-\delta}\right)\mathbf{x} - \frac{\delta}{1-\delta}\alpha_1 C_{\mathbf{y}} - \frac{\delta}{1-\delta}\alpha_1 C_{\mathbf{x}}\right)^2.
\tag{48}
$$

To satisfy the $(\zeta, B)$-gradient heterogeneity condition specified in Assumption 4, the following must hold:

$$
\mathbb{E}\left[\left\|\nabla f_i^1(\mathbf{x}) - \nabla f_{\mathcal{H}_1}(\mathbf{x})\right\|^2\right] \leq B_f^2 \mathbb{E}\left[\left\|\nabla f_{\mathcal{H}_1}(\mathbf{x})\right\|^2\right] + \zeta_f^2.
\tag{49}
$$

By combining equations (48) and (47), we obtain:

$$
\mathbb{E}\left[\left\|\nabla_{\mathbf{y}} f_i^1(\mathbf{x}) - \nabla_{\mathbf{y}} f_{\mathcal{H}_1}(\mathbf{x})\right\|^2\right] - B_f^2 \mathbb{E}\left[\left\|\nabla f_{\mathcal{H}_1}(\mathbf{x})\right\|^2\right] - \zeta_f^2
$$
$$
= \underbrace{\left(\frac{\delta(1-2\delta)}{(1-\delta)^2}(\alpha_1 - \alpha_2)^2 - B_f^2\left(\alpha_1\frac{\delta}{1-\delta} + \alpha_2\frac{1-2\delta}{1-\delta}\right)^2\right)}_{A_{\mathbf{x},1}}\|\mathbf{x}\|^2
\tag{50}
$$
$$
- 2\underbrace{\left(\frac{1-2\delta}{1-\delta}(\alpha_1 - \alpha_2)C_{\mathbf{x}} - B_f^2(\alpha_1\frac{\delta}{1-\delta} - \alpha_2\frac{1-2\delta}{1-\delta})(C_{\mathbf{x}} + C_{\mathbf{y}})\right)}_{A_{\mathbf{x},2}}\frac{\delta}{1-\delta}\alpha_1\mathbf{x}
$$
$$
+ \underbrace{\frac{\delta(1-2\delta)}{(1-\delta)^2}\alpha_1^2 C_{\mathbf{x}}^2 - B_f^2(\frac{\delta}{1-\delta})^2\alpha_1^2(C_{\mathbf{y}} + C_{\mathbf{x}})^2 - \zeta_f^2}_{A_{\mathbf{x},3}} \leq 0.
$$

To ensure the validity of the above inequality, the following condition must be satisfied:

$$
A_{\mathbf{x},1} \leq 0, \quad A_{\mathbf{x},3} \leq 0 \quad and \quad A_{\mathbf{x},2}^2 \leq A_{\mathbf{x},1}A_{\mathbf{x},3}.
\tag{51}
$$

Hence, the proof is complete. $\qquad\square$

### E.3  PROOF OF THE LOWER BOUND IN THEOREM 1

To satisfy the $(\zeta, B)$-gradient heterogeneity assumption, the constants and vectors $\{\gamma_1, \gamma_2, \alpha_1, \alpha_2, C_{\mathbf{x}}, C_{\mathbf{y}}\}$ in (17) and (18) have to satisfy the constraints in Lemma 2. Once these constants and vectors are chosen to meet the specified conditions, we derive the lower bound for the asymptotic error, which is influenced by the heterogeneity in both the lower-level and upper-level objectives.

---

**Lemma 3.** *Consider the two sets of bilevel optimization problems in Lemma 1, and Assumptions 1- 4 hold. If the constants and vectors $\alpha_1$, $\alpha_2$, $C_{\mathbf{x}}$ and $C_{\mathbf{y}}$ satisfy the constraints in Lemma 2, the asymptotic error is at least:*

$$
\max_{j\in\{1,2\}}\left\|\nabla\Phi_{\mathcal{H}_j}(\hat{\mathbf{x}})\right\|^2 \geq \frac{\delta\left(\zeta_f^2 + \zeta_g^2 + 2\zeta_f\zeta_g\right)}{4(1 - \delta(2 + B_f^2))}.
$$

---

*Proof.* Building on the analysis above, we select the constants $\alpha_1, \alpha_2$ and the vectors $C_{\mathbf{y}}, C_{\mathbf{x}}$ to satisfy the inequality specified in Lemma 2:

$$
\alpha_1 = \mu_f(1 + B_f^2), \quad \alpha_2 = \mu_f(1 - \frac{\delta}{1-2\delta}B_f^2).
\tag{52}
$$

The vectors $C_{\mathbf{y}}$ and $C_{\mathbf{x}}$ are set as follows:

$$C_{\mathbf{y}}^2 = \frac{(1-\delta)^2 \zeta_g^2}{\delta(1-2\delta)(1+B_f^2)\alpha_1\alpha_2}, \quad C_{\mathbf{x}}^2 = \frac{(1-\delta)^2 \zeta_f^2}{\delta(1-2\delta)(1+B_f^2)\alpha_1\alpha_2}. \tag{53}$$

Combined with equation (35) and the above parameter, we obtain that:

$$\max_{j\in\{1,2\}} \Phi_{\mathcal{H}_j}(\hat{\mathbf{x}}) - \Phi_{\mathcal{H}_j}(\mathbf{x}_j^\star) \geq \frac{\delta\left(\zeta_f^2 + \zeta_g^2 + 2\zeta_f\zeta_g\right)}{8\mu_f(1-\delta(2+B_f^2))}. \tag{54}$$

We demonstrate that the selected coefficients ensure the upper-level loss function satisfies $\mu_f$-strong convexity and $L_f$-smoothness:

$$\max_{j\in\{1,2\}} \|\nabla^2 f^j(\mathbf{x},\mathbf{y}_\star^j(\mathbf{x}))\|^2 = L_f = \mu_f + \frac{1-2\delta}{1-\delta}\alpha_2 \geq \mu_f. \tag{55}$$

Given that the upper-level objective satisfies the $\mu_f$-strong convexity assumption, the following inequality holds:

$$\|\nabla\Phi(\mathbf{x})\|^2 \geq 2\mu_f\left(\Phi(\mathbf{x}) - \Phi(\mathbf{x}^\star)\right). \tag{56}$$

By combining the results from (54) with (56), we derive the following lower bound:

$$\max_{j\in\{1,2\}} \left\|\nabla\Phi_{\mathcal{H}_j}(\hat{\mathbf{x}})\right\|^2 \geq \frac{\delta\left(\zeta_f^2 + \zeta_g^2 + 2\zeta_f\zeta_g\right)}{4(1-\delta(2+B_f^2))}. \tag{57}$$

Hence, the proof is complete. $\qquad\square$

# F    GENERAL LEMMAS FOR CONVERGENCE ANALYSIS

In this section, we provide some general lemmas that are frequently used throughout the proof.

**Lemma 4** ((Chen et al., 2024), Lemma B.2). *Under Assumptions 1, 2, there exist positive constants $L_{\nabla\Phi}, L_{\mathbf{y}^\star}, L_{\mathbf{z}^\star}$, such that $\nabla\Phi_{\mathcal{H}}(\mathbf{x}), \mathbf{y}^\star(\mathbf{x}), \mathbf{z}^\star(\mathbf{x})$ are $L_{\nabla\Phi}, L_{\mathbf{y}^\star}, L_{\mathbf{z}^\star}$-Lipschitz continuous, respectively. Moreover, we have $\|\mathbf{z}^\star(\mathbf{x})\| \leq \frac{C_f}{\mu_g}$ for all $\mathbf{x}\in\mathbb{R}^{d_{\mathbf{x}}}$.*

**Lemma 5** ((Dagréou et al., 2022), Lemma C.1 and C.2). *Under Assumptions 1, 2, 6, there exist positive constants $L_{\mathbf{yx}}$ and $L_{\mathbf{zx}}$, such that $\mathbf{y}^\star(\mathbf{x})$ and $\mathbf{z}^\star(\mathbf{x})$ are $L_{\mathbf{yx}}$ and $L_{\mathbf{zx}}$-smooth, respectively.*

**Lemma 6** ((Chen et al., 2024),Lemma B.1). *Under Assumptions 1, 2, if $\beta < \frac{2}{L_g+\mu_g}$, the following inequality holds:*

$$\left\|\mathbf{y}^k - \beta\nabla_{\mathbf{y}}g^k - \mathbf{y}_\star^k\right\| \leq (1-\beta\mu_g)\left\|\mathbf{y}^k - \mathbf{y}_\star^k\right\|.$$

**Lemma 7** ((He et al., 2024), Lemma B.3). *Under Assumption 3, we have the following variance bounds in distributed learning:*

$$\begin{aligned}
\mathrm{Var}\left[D_{\mathbf{y},i}^k \mid \mathcal{F}^k\right] &\leq \sigma^2, & \mathrm{Var}\left[D_{\mathbf{y}}^k \mid \mathcal{F}^k\right] &\leq \sigma^2/((1-\delta)n), \\
\mathrm{Var}\left[D_{\mathbf{x},i}^k \mid \mathcal{F}^k\right] &\leq \sigma_1^2, & \mathrm{Var}\left[D_{\mathbf{x}}^k \mid \mathcal{F}^k\right] &\leq \sigma_1^2/((1-\delta)n), \\
\mathrm{Var}\left[D_{\mathbf{z},i}^k \mid \mathcal{F}^k\right] &\leq \sigma_1^2, & \mathrm{Var}\left[D_{\mathbf{z}}^k \mid \mathcal{F}^k\right] &\leq \sigma_1^2/((1-\delta)n),
\end{aligned}$$

*where $\sigma_1^2 \triangleq \sigma^2\left(1+\rho^2\right)$.*

**Lemma 8** ((Khanduri et al., 2021), Lemma C.2). *If $\phi$ is $\eta_{\mathbf{x}}$-strongly convex and $\eta_{\mathbf{y}}$-smooth, then*

$$\langle\nabla\phi(\mathbf{x}) - \nabla\phi(\mathbf{y}), \mathbf{x} - \mathbf{y}\rangle \geq \frac{\eta_{\mathbf{x}}\eta_{\mathbf{y}}}{\eta_{\mathbf{x}}+\eta_{\mathbf{y}}}\|\mathbf{x}-\mathbf{y}\|^2 + \frac{1}{\eta_{\mathbf{x}}+\eta_{\mathbf{y}}}\|\nabla\phi(\mathbf{x})-\nabla\phi(\mathbf{y})\|^2.$$

# G    CONVERGENCE ANALYSIS FOR BR-FEDBI (THEOREM 2)

## G.1    PROOF OUTLINE

In this section, we prove the convergence results stated in Theorem 2. Our analysis of BR-FedBi in Algorithm 1, which follows the framework of (He et al., 2024), consists of four key elements:

(1) Aggregation error (Lemma 9),

(2) Lower-level convergence (Lemma 10),

(3) Gradient Update Bias (Lemma 11),

(4) Descent Lemma (Lemma 12).

By combining these elements, we derive the final convergence result presented in Theorem 2.

**Notation.** For convenience, we have the following notations used throughout the proof:

$$L_2^2 := L_f^2 + \rho^2 L_g^2, \quad L_1^2 := \frac{1}{2} L_f^2 + \rho^2 L_g^2, \quad \lambda_g := \frac{\mu_g L_g}{\mu_g + L_g}, \quad \sigma_1^2 := (1 + \rho^2)\sigma^2.$$

In the $k$-th communication round, the server distributes the global parameter $\{\mathbf{x}^k, \mathbf{y}^k, \mathbf{z}^k\}$ to the clients. Each honest client $i \in \mathcal{H}$ computes the gradients using their locally sampled data, $\phi_i^k \sim \mathcal{D}_{f_i}$ and $\varphi_i^k \sim \mathcal{D}_{g_i}$, and transmits them to the server:

$$
\begin{aligned}
\mathbf{D}_{\mathbf{x},i}^k &\triangleq \nabla_{xy}^2 G(\mathbf{x}^k, \mathbf{y}^k; \varphi_i^k)\mathbf{z}^k + \nabla_{\mathbf{x}} F(\mathbf{x}^k, \mathbf{y}^k; \phi_i^k), \\
\mathbf{D}_{\mathbf{y},i}^k &\triangleq \nabla_{\mathbf{y}} G(\mathbf{x}^k, \mathbf{y}^k; \varphi_i^k), \\
\mathbf{D}_{\mathbf{z},i}^k &\triangleq \nabla_{yy}^2 G(\mathbf{x}^k, \mathbf{y}^k; \varphi_i^k)\mathbf{z}^k + \nabla_{\mathbf{y}} F(\mathbf{x}^k, \mathbf{y}^k; \phi_i^k),
\end{aligned}
\tag{58}
$$

where $\mathbf{x}^k$ and $\mathbf{y}^k$ are the upper- and lower-level parameters, respectively. Here, $\varphi_i^k$ and $\phi_i^k$ denote datasets sampled from the distribution available to each honest client.

Upon receiving the gradients from all the clients (including honest clients and attackers), the server employs a Byzantine-resilient robust aggregation rule $AGG(\cdot) : \mathbb{R}^{d \times n} \to \mathbb{R}^d$ to aggregate the updates. The aggregated gradients are computed as follows:

$$
\begin{aligned}
\hat{\mathbf{D}}_{\mathbf{x}}^k &:= AGG(\mathbf{D}_{\mathbf{x},1}^k, \ldots, \mathbf{D}_{\mathbf{x},n}^k), \\
\hat{\mathbf{D}}_{\mathbf{y}}^k &:= AGG(\mathbf{D}_{\mathbf{y},1}^k, \ldots, \mathbf{D}_{\mathbf{y},n}^k), \\
\hat{\mathbf{D}}_{\mathbf{z}}^k &:= AGG(\mathbf{D}_{\mathbf{z},1}^k, \ldots, \mathbf{D}_{\mathbf{z},n}^k).
\end{aligned}
\tag{59}
$$

Finally, the server updates the global parameters $\{\mathbf{x}^k, \mathbf{y}^k, \mathbf{z}^k\}$ using the aggregated gradients and the learning rates $\eta_{\mathbf{x}}, \eta_{\mathbf{y}}, \eta_{\mathbf{z}} > 0$. The updates are performed as follows:

$$
\begin{aligned}
\mathbf{x}^{k+1} &= \mathbf{x}^k - \eta_{\mathbf{x}} \hat{\mathbf{D}}_{\mathbf{x}}^k, \\
\mathbf{y}^{k+1} &= \mathbf{y}^k - \eta_{\mathbf{y}} \hat{\mathbf{D}}_{\mathbf{y}}^k, \\
\mathbf{z}^{k+1} &= \mathbf{z}^k - \eta_{\mathbf{z}} \hat{\mathbf{D}}_{\mathbf{z}}^k.
\end{aligned}
\tag{60}
$$

## G.2 AGGREGATION ERROR

We define the aggregation error as the deviation between the robustly aggregated gradients and the average gradients of honest clients. Specifically, the aggregation error is expressed as:

$$
\begin{aligned}
\xi_{\mathbf{x}}^k &:= \hat{\mathbf{D}}_{\mathbf{x}}^k - \mathbf{D}_{\mathbf{x}}^k, \\
\xi_{\mathbf{y}}^k &:= \hat{\mathbf{D}}_{\mathbf{y}}^k - \mathbf{D}_{\mathbf{y}}^k, \\
\xi_{\mathbf{z}}^k &:= \hat{\mathbf{D}}_{\mathbf{z}}^k - \mathbf{D}_{\mathbf{z}}^k,
\end{aligned}
$$

where $\mathbf{D}_{\mathbf{x}}^k = \frac{1}{|\mathcal{H}|} \sum_{i \in \mathcal{H}} \mathbf{D}_{\mathbf{x},i}^k$, $\mathbf{D}_{\mathbf{y}}^k = \frac{1}{|\mathcal{H}|} \sum_{i \in \mathcal{H}} \mathbf{D}_{\mathbf{y},i}^k$ and $\mathbf{D}_{\mathbf{z}}^k = \frac{1}{|\mathcal{H}|} \sum_{i \in \mathcal{H}} \mathbf{D}_{\mathbf{z},i}^k$.

We then establish the following bounds for the aggregation error. The detailed proof is provided in Appendix J.1.1.

**Lemma 9.** *Suppose Assumptions 1-4 hold, and the robust aggregation rule $AGG(\cdot)$ satisfies the $(\delta, \kappa)$-robustness criterion. The aggregation errors in the BR-FedBi algorithm (Algorithm 1) satisfy the following inequalities:*

$$(1) \quad \mathbb{E}\left[\left\|\xi_{\mathbf{y}}^k\right\|^2\right] \leq \kappa\zeta_g^2 + \kappa\sigma^2\left(1 + \frac{1}{(1+\delta)n}\right),$$

$$(2) \quad \mathbb{E}\left[\left\|\xi_{\mathbf{x}}^k\right\|^2\right] \leq 2\kappa\left(\rho^2\zeta_g^2 + \zeta_f^2\right) + 2\kappa\sigma_1^2\left(1 + \frac{1}{(1+\delta)n}\right),$$

$$(3) \quad \mathbb{E}\left[\left\|\xi_{\mathbf{z}}^k\right\|^2\right] \leq 2\kappa\left(\rho^2\zeta_g^2 + \zeta_f^2\right) + 2\kappa\sigma_1^2\left(1 + \frac{1}{(1+\delta)n}\right),$$

*where $\sigma_1^2 = (1 + \rho^2)\sigma^2$ and $\rho = \frac{C_f}{\mu_g}$.*

### G.3 LOWER-LEVEL CONVERGENCE

We then analyze the convergence behavior of the lower-level variables in our bilevel algorithm. The detailed proof of this lemma is provided in Appendix J.1.2.

**Lemma 10.** *Under Assumptions 1-4, and 5-6, the following inequalities hold for variables $\mathbf{y}$ and $\mathbf{z}$:*

$$
\begin{aligned}
(1) \quad & \mathbb{E}\left[\left\|\mathbf{y}^{k+1} - \mathbf{y}_\star^{k+1}\right\|^2\right] \\
& \leq (1 - \frac{\eta_{\mathbf{y}}\mu_g}{4} + 2\eta_{\mathbf{y}}^2 L_g^2)\mathbb{E}\left[\left\|\mathbf{y}^k - \mathbf{y}_\star^k\right\|^2\right] + (3\eta_{\mathbf{y}}^2 + \frac{\eta_{\mathbf{y}}}{\mu_g})\mathbb{E}\left[\left\|\xi_{\mathbf{y}}^k\right\|^2\right] \\
& \quad + \left(6L_{\mathbf{y}^\star}^2\eta_{\mathbf{x}}^2 + \frac{4\eta_{\mathbf{x}}^2}{\eta_{\mathbf{y}}\mu_g}L_{\mathbf{y}^\star}^2\right)\left(\mathbb{E}\left[\left\|\mathbb{E}(\mathbf{D}_{\mathbf{x}}^k)\right\|^2\right] + \mathbb{E}\left[\left\|\xi_{\mathbf{x}}^k\right\|^2\right]\right) \\
& \quad + 3\eta_{\mathbf{z}}^2\frac{\sigma^2}{(1-\delta)n} + 6L_{\mathbf{y}^\star}^2\eta_{\mathbf{x}}^2\frac{\sigma_1^2}{(1-\delta)n},
\end{aligned}
$$

$$
\begin{aligned}
(2) \quad & \mathbb{E}\left[\left\|\mathbf{z}^{k+1} - \mathbf{z}_\star^{k+1}\right\|^2\right] \\
& \leq \left(1 - \frac{\eta_{\mathbf{z}}\mu_g}{2} + 6\eta_{\mathbf{z}}^2 L_g^2\right)\mathbb{E}\left[\left\|\mathbf{z}^k - \mathbf{z}_\star^k\right\|^2\right] + \left(\frac{4\eta_{\mathbf{z}}L_2^2}{\mu_g} + 6\eta_{\mathbf{z}}^2 L_2^2\right)\mathbb{E}\left[\left\|\mathbf{y}^k - \mathbf{y}_\star^k\right\|^2\right] \\
& \quad + \left(4L_{\mathbf{z}^\star}^2\eta_{\mathbf{x}}^2 + 4L_{zx}\rho\eta_{\mathbf{x}}^2 + \frac{4\eta_{\mathbf{x}}^2}{\eta_{\mathbf{z}}\mu_g}L_{\mathbf{z}^\star}^2\right)\left(\mathbb{E}\left[\left\|\mathbb{E}(\mathbf{D}_{\mathbf{x}}^k)\right\|^2\right] + \mathbb{E}\left[\left\|\xi_{\mathbf{x}}^k\right\|^2\right]\right) \\
& \quad + \left(3\eta_{\mathbf{z}}^2 + \frac{2\eta_{\mathbf{z}}}{\mu_g}\right)\mathbb{E}\left[\left\|\xi_{\mathbf{z}}^k\right\|^2\right] + \left(2\eta_{\mathbf{z}}^2 + 4L_{\mathbf{z}^\star}^2\eta_{\mathbf{x}}^2 + 4L_{zx}\rho\eta_{\mathbf{x}}^2\right)\frac{\sigma_1^2}{(1-\delta)n},
\end{aligned}
$$

*where $\mathbf{y}_\star^k = \mathbf{y}^\star(\mathbf{x}^k)$ and $\mathbf{z}_\star^k = \mathbf{z}^\star(\mathbf{x}^k)$.*

### G.4 GRADIENT UPDATE BIAS

Next, we then analyze the gradient update bias of BR-FedBi in Algorithm 1. The gradient update bias measures the distance between the hypergradient $\nabla\Phi_{\mathcal{H}}(\mathbf{x}^k)$ and the expectation of the update gradient $\mathbb{E}(\mathbf{D}_{\mathbf{x}}^k)$, where the expectation is over randomness.

**Lemma 11** ((He et al., 2024), Lemma B.4). *Suppose Assumption 1, 2 and 3 hold, if $\rho \geq C_f/\mu_g$, we have:*

$$\mathbb{E}\left[\left\|\mathbb{E}\left(\mathbf{D}_{\mathbf{x}}^k\right) - \nabla\Phi_{\mathcal{H}}\left(\mathbf{x}^k\right)\right\|^2\right] \leq 3L_2^2\mathbb{E}\left[\left\|\mathbf{y}^k - \mathbf{y}_\star^k\right\|^2\right] + 3L_g^2\mathbb{E}\left[\left\|\mathbf{z}^k - \mathbf{z}_\star^k\right\|^2\right],$$

*where $L_2^2 = L_f^2 + L_g^2\rho^2$.*

## G.5 DESCENT LEMMA

Finally, we analyze the progression of the upper-level loss function in BR-FedBi, as described in Algorithm 1. During each iteration, the updates for $\mathbf{x}^{k+1}$, $\mathbf{y}^{k+1}$, and $\mathbf{z}^{k+1}$ are given by:

$$\mathbf{x}^{k+1} = \mathbf{x}^k - \eta_{\mathbf{x}}\mathrm{AGG}\left(\mathbf{D}_{\mathbf{x},1}^k, \dots, \mathbf{D}_{\mathbf{x},n}^k\right) = \mathbf{x}^k - \eta_{\mathbf{x}}\hat{\mathbf{D}}_{\mathbf{x}}^k = \mathbf{x}^k - \eta_{\mathbf{x}}\mathbf{D}_{\mathbf{x}}^k - \eta_{\mathbf{x}}\xi_{\mathbf{x}}^k,$$

$$\mathbf{y}^{k+1} = \mathbf{y}^k - \eta_{\mathbf{y}}\mathrm{AGG}\left(\mathbf{D}_{\mathbf{y},1}^k, \dots, \mathbf{D}_{\mathbf{y},n}^k\right) = \mathbf{y}^k - \eta_{\mathbf{y}}\hat{\mathbf{D}}_{\mathbf{y}}^k = \mathbf{y}^k - \eta_{\mathbf{y}}\mathbf{D}_{\mathbf{y}}^k - \eta_{\mathbf{y}}\xi_{\mathbf{y}}^k,$$

$$\mathbf{z}^{k+1} = \mathbf{z}^k - \eta_{\mathbf{z}}\mathrm{AGG}\left(\mathbf{D}_{\mathbf{z},1}^k, \dots, \mathbf{D}_{\mathbf{z},n}^k\right) = \mathbf{z}^k - \eta_{\mathbf{z}}\hat{\mathbf{D}}_{\mathbf{z}}^k = \mathbf{z}^k - \eta_{\mathbf{z}}\mathbf{D}_{\mathbf{z}}^k - \eta_{\mathbf{z}}\xi_{\mathbf{z}}^k.$$

The detailed proof of Lemma 12 is provided in Appendix J.1.3.

---

**Lemma 12.** *Suppose that assumptions 1-4 hold true. Recall in Lemma 4 that the upper-level loss function is $L_{\nabla\Phi}$-smooth, for all $k \in [K]$, we can obtain*

$$\mathbb{E}\left[\Phi_{\mathcal{H}}\left(\mathbf{x}^{k+1}\right)\right] \le \mathbb{E}\left[\Phi_{\mathcal{H}}\left(\mathbf{x}^k\right)\right] - \frac{\eta_{\mathbf{x}}}{4}\mathbb{E}\left[\left\|\Phi_{\mathcal{H}}\left(\mathbf{x}^k\right)\right\|^2\right] + \frac{\eta_x}{2}\mathbb{E}\left[\left\|\nabla\Phi_{\mathcal{H}}(\mathbf{x}^k) - \mathbb{E}(\mathbf{D}_{\mathbf{x}}^k)\right\|^2\right]$$

$$+ \left(L_{\nabla\Phi}\eta_{\mathbf{x}}^2 - \frac{\eta_{\mathbf{x}}}{2}\right)\mathbb{E}\left[\left\|\mathbb{E}(\mathbf{D}_{\mathbf{x}}^k)\right\|^2\right] + \left(4\eta_{\mathbf{x}} + L_{\nabla\Phi}\eta_{\mathbf{x}}^2\right)\mathbb{E}\left[\left\|\xi_{\mathbf{x}}^k\right\|^2\right] + \frac{\sigma_1^2 L_{\nabla\Phi}\eta_{\mathbf{x}}^2}{(1-\delta)n},$$

*where $\sigma_1^2 = \left(1 + \rho^2\right)\sigma^2$.*

---

## G.6 PROOF OF THEOREM 2

**Theorem 5.** *Suppose that Assumptions 1-4 and 5-6 hold, and that the robust aggregation rule $AGG(\cdot)$ satisfies the $(\delta, \kappa)$-robustness criterion. There exist positive constants $c$, $c_{\mathbf{y}}$, and $c_{\mathbf{z}}$ such that*

$$\eta_{\mathbf{y}} = c_{\mathbf{y}}\eta_{\mathbf{x}}, \quad \eta_{\mathbf{z}} = c_{\mathbf{z}}\eta_{\mathbf{x}},$$

*and the step size $\eta_{\mathbf{x}}$ satisfies*

$$\eta_{\mathbf{x}} = \min\left\{\alpha_1, \sqrt{\frac{\Delta_\Phi^0}{KA_\sigma}}\right\}.$$

*Here, we have*

$$A_\sigma := \frac{A_0}{(1-\delta)n}\sigma^2, \alpha_1 := \frac{1}{4\left(L_{\nabla\Phi} + 6L_{\mathbf{y}^\star}^2 + 4L_{\mathbf{z}^\star}^2 + 4L_{zx}\rho\right)},$$

*and $A_0$ is a positive constant independent of $n$ and $K$. Then, for $K \ge 1$, the algorithm converges as:*

$$\frac{1}{K}\sum_{k=0}^{K-1}\mathbb{E}\left[\left\|\nabla\Phi_{\mathcal{H}}(\mathbf{x}^k)\right\|^2\right] = \mathcal{O}\left(\frac{\Delta_\Phi^0}{K} + \sqrt{\frac{A_\sigma\Delta_\Phi^0}{K}}\right) + \mathcal{O}\left(\kappa\left(\zeta_f^2 + \zeta_g^2 + \sigma^2\right)\right), \tag{61}$$

*where $\Delta_\Phi^0 = \Phi_{\mathcal{H}}(\mathbf{x}^0) + \|\mathbf{y}^0 - \mathbf{y}_\star^0\|^2 + \|\mathbf{z}^0 - \mathbf{z}_\star^0\|^2$, and the expectation is taken over the randomness of the algorithm.*

*Proof.* In order to obtain the final convergence result, we combine the above Lemmas. To obtain the convergence result we define the Lyapunov function to be

$$\mathcal{L}_k = \Phi_{\mathcal{H}}(\mathbf{x}^k) + \mathbb{E}\left[\left\|\mathbf{y}^k - \mathbf{y}_\star^k\right\|^2\right] + \mathbb{E}\left[\left\|\mathbf{z}^k - \mathbf{z}_\star^k\right\|^2\right]. \tag{62}$$

Consider the Lyapunov function in (62). By combining Lemmas 10, 11, and 12, we obtain:

$$\begin{aligned}
&\mathcal{L}_{k+1} - \mathcal{L}_k \\
&\leq -\frac{\eta_{\mathbf{x}}}{4} \mathbb{E}\left[\left\|\nabla\Phi_{\mathcal{H}}\left(\mathbf{x}^k\right)\right\|^2\right] \\
&\quad + \underbrace{\left(\frac{3\eta_{\mathbf{x}}L_2^2}{2} - \frac{\eta_{\mathbf{y}}\mu_g}{4} + 2\eta_{\mathbf{y}}^2 L_g^2 + \frac{4\eta_{\mathbf{z}}L_2^2}{\mu_g} + 6\eta_{\mathbf{z}}^2 L_2^2\right)}_{T_1} \mathbb{E}\left[\left\|\mathbf{y}^k - \mathbf{y}_\star^k\right\|^2\right] \\
&\quad + \underbrace{\left(\frac{3\eta_{\mathbf{x}}L_g^2}{2} - \frac{\eta_{\mathbf{z}}\mu_g}{2} + 6\eta_{\mathbf{z}}^2 L_g^2\right)}_{T_2} \mathbb{E}\left[\left\|\mathbf{z}^k - \mathbf{z}_\star^k\right\|^2\right] \\
&\quad + \underbrace{\left(-\frac{\eta_{\mathbf{x}}}{2} + L_{\nabla\Phi}\eta_{\mathbf{x}}^2 + 6L_{\mathbf{y}^\star}^2\eta_{\mathbf{x}}^2 + \frac{4\eta_{\mathbf{x}}^2}{\eta_{\mathbf{y}}\mu_g}L_{\mathbf{y}^\star}^2 + 4L_{\mathbf{z}^\star}^2\eta_{\mathbf{x}}^2 + 4L_{zx}\rho\eta_{\mathbf{x}}^2 + \frac{4\eta_{\mathbf{x}}^2}{\eta_{\mathbf{z}}\mu_g}L_{\mathbf{z}^\star}^2\right)}_{T_3} \mathbb{E}\left[\left\|\mathbb{E}(\mathbf{D}_{\mathbf{x}}^k)\right\|^2\right] \\
&\quad + \left(2\eta_{\mathbf{z}}^2 + 6L_{\mathbf{y}^\star}^2\eta_{\mathbf{x}}^2 + 4L_{\mathbf{z}^\star}^2\eta_{\mathbf{x}}^2 + 4L_{zx}\rho\eta_{\mathbf{x}}^2 + L_{\nabla\Phi}\right)\frac{\sigma_1^2}{(1-\delta)n} + 2\eta_{\mathbf{y}}^2\frac{\sigma^2}{(1-\delta)n} \\
&\quad + \left(4\eta_{\mathbf{x}} + \frac{L_{\nabla\Phi}\eta_{\mathbf{x}}^2}{2} + \frac{4\eta_{\mathbf{x}}^2}{\eta_{\mathbf{y}}\mu_g}L_{\mathbf{y}^\star}^2 + \frac{4\eta_{\mathbf{x}}^2}{\eta_{\mathbf{z}}\mu_g}L_{\mathbf{z}^\star}^2 + 6L_{\mathbf{y}^\star}^2\eta_{\mathbf{x}}^2 + 4L_{\mathbf{z}^\star}^2\eta_{\mathbf{x}}^2 + 4L_{zx}\rho\eta_{\mathbf{x}}^2\right)\mathbb{E}\left[\left\|\xi_{\mathbf{x}}^k\right\|^2\right] \\
&\quad + \left(3\eta_{\mathbf{z}}^2 + \frac{2\eta_{\mathbf{z}}}{\mu_g}\right)\mathbb{E}\left[\left\|\xi_{\mathbf{z}}^k\right\|^2\right] + \left(3\eta_{\mathbf{y}}^2 + \frac{\eta_{\mathbf{y}}}{\mu_g}\right)\mathbb{E}\left[\left\|\xi_{\mathbf{y}}^k\right\|^2\right].
\end{aligned}$$

$$(63)$$

Under the assumptions that

$$\eta_{\mathbf{y}} \leq \frac{1}{2\mu_g}, \quad \eta_{\mathbf{z}} \leq \frac{1}{2\mu_g}, \quad \eta_{\mathbf{x}} \leq \frac{1}{4\left(L_{\nabla\Phi} + 6L_{\mathbf{y}^\star}^2 + 4L_{\mathbf{z}^\star}^2 + 4L_{zx}\rho\right)}, \quad \eta_{\mathbf{x}} \leq \frac{\mu_g\eta_{\mathbf{y}}}{32L_{\mathbf{y}^\star}^2}, \quad \eta_{\mathbf{x}} \leq \frac{\mu_g\eta_{\mathbf{z}}}{32L_{\mathbf{z}^\star}^2},$$

$$(64)$$

and invoking Lemma 9, we can implies

$$\begin{aligned}
&\mathcal{L}_{k+1} - \mathcal{L}_k \\
&\leq -\frac{\eta_{\mathbf{x}}}{2} \mathbb{E}\left[\left\|\nabla\Phi_{\mathcal{H}}\left(\mathbf{x}^k\right)\right\|^2\right] \\
&\quad + \underbrace{\left(\frac{3\eta_{\mathbf{x}}L_2^2}{2} - \frac{\eta_{\mathbf{y}}\mu_g}{4} + 2\eta_{\mathbf{y}}^2 L_g^2 + \frac{4\eta_{\mathbf{z}}L_2^2}{\mu_g} + 6\eta_{\mathbf{z}}^2 L_2^2\right)}_{T_1} \mathbb{E}\left[\left\|\mathbf{y}^k - \mathbf{y}_\star^k\right\|^2\right] \\
&\quad + \underbrace{\left(\frac{3\eta_{\mathbf{x}}L_g^2}{2} - \frac{\eta_{\mathbf{z}}\mu_g}{2} + 6\eta_{\mathbf{z}}^2 L_g^2\right)}_{T_2} \mathbb{E}\left[\left\|\mathbf{z}^k - \mathbf{z}_\star^k\right\|^2\right] \\
&\quad + \underbrace{\left(-\frac{\eta_{\mathbf{x}}}{2} + L_{\nabla\Phi}\eta_{\mathbf{x}}^2 + 6L_{\mathbf{y}^\star}^2\eta_{\mathbf{x}}^2 + \frac{4\eta_{\mathbf{x}}^2}{\eta_{\mathbf{y}}\mu_g}L_{\mathbf{y}^\star}^2 + 4L_{\mathbf{z}^\star}^2\eta_{\mathbf{x}}^2 + 4L_{zx}\rho\eta_{\mathbf{x}}^2 + \frac{4\eta_{\mathbf{x}}^2}{\eta_{\mathbf{z}}\mu_g}L_{\mathbf{z}^\star}^2\right)}_{T_3} \mathbb{E}\left[\left\|\mathbb{E}(\mathbf{D}_{\mathbf{x}}^k)\right\|^2\right] \\
&\quad + \left(2\eta_{\mathbf{z}}^2 + 6L_{\mathbf{y}^\star}^2\eta_{\mathbf{x}}^2 + 4L_{\mathbf{z}^\star}^2\eta_{\mathbf{x}}^2 + 4L_{zx}\rho\eta_{\mathbf{x}}^2 + L_{\nabla\Phi}\right)\frac{\sigma_1^2}{(1-\delta)n} + 2\eta_{\mathbf{y}}^2\frac{\sigma^2}{(1-\delta)n} \\
&\quad + \left(\left(\frac{8\eta_{\mathbf{z}}}{\mu_g} + 10\eta_{\mathbf{x}}\right)\cdot\kappa\sigma_1^2 + \frac{3\eta_{\mathbf{y}}}{\mu_g}\cdot\kappa\sigma^2\right)\left(1 + \frac{1}{(1+\delta)n}\right) \\
&\quad + \left(\frac{8\eta_{\mathbf{z}}}{\mu_g} + 10\eta_{\mathbf{x}}\right)\kappa(\rho^2\zeta_g^2 + \zeta_f^2) + \frac{3\eta_{\mathbf{y}}}{\mu_g}\cdot\kappa\zeta_g^2.
\end{aligned}$$

$$(65)$$

We now analyze each term on the R.H.S. to further bound them.

**Analysis of the coefficient for $\mathbb{E}\left[\left\|\mathbf{y}^k - \mathbf{y}_\star^k\right\|^2\right]$.**

Under the assumptions that

$$\eta_{\mathbf{y}} \le \frac{\mu_g}{16L_g^2}, \quad \eta_{\mathbf{z}} \le \frac{1}{6\mu_g}, \tag{66}$$

we can implies

$$2\eta_{\mathbf{y}}^2 L_g^2 \le \frac{\eta_{\mathbf{y}}\mu_g}{8}, \quad 6\eta_{\mathbf{z}}^2 L_2^2 \le \frac{\eta_{\mathbf{z}} L_2^2}{\mu_g}. \tag{67}$$

Then, we can bound the term $T_1$ in the following manner:

$$T_1 \le \frac{3\eta_{\mathbf{x}} L_2^2}{2} - \frac{\eta_{\mathbf{y}}\mu_g}{8} + \frac{5\eta_{\mathbf{z}} L_2^2}{\mu_g} \le \frac{3\eta_{\mathbf{x}} L_2^2}{2} - \frac{\eta_{\mathbf{y}}\mu_g}{16} \le 0, \tag{68}$$

where the first inequality follows from (66). The last two inequalities uses $\eta_{\mathbf{z}} \le \frac{\mu_g \eta_{\mathbf{y}}}{80L_2^2}$ and $\eta_{\mathbf{x}} \le \frac{\mu_g \eta_{\mathbf{y}}}{24L_2^2}$.

**Analysis of the coefficient for $\mathbb{E}\left[\left\|\mathbf{z}^k - \mathbf{z}_\star^k\right\|^2\right]$.**

Under the assumptions that

$$\eta_{\mathbf{z}} \le \frac{\mu_g}{24L_g^2}, \quad \eta_{\mathbf{x}} \le \frac{\mu_g \eta_{\mathbf{z}}}{6L_g^2}, \tag{69}$$

Then, we can bound the term $T_2$ in the following manner:

$$T_2 = \frac{3\eta_{\mathbf{x}} L_g^2}{2} - \frac{\eta_{\mathbf{z}}\mu_g}{2} + 6\eta_{\mathbf{z}}^2 L_g^2 \le \frac{3\eta_{\mathbf{x}} L_g^2}{2} - \frac{\eta_{\mathbf{z}}\mu_g}{4} \le 0. \tag{70}$$

**Analysis of the coefficient for $\mathbb{E}\left[\left\|\mathbb{E}(\mathbf{D}_{\mathbf{x}}^k)\right\|^2\right]$.**

Given the assumption that

$$\eta_{\mathbf{x}} \le \frac{1}{4\left(L_{\nabla\Phi} + 6L_{\mathbf{y}^\star}^2 + 4L_{\mathbf{z}^\star}^2 + 4L_{zx}\rho\right)}, \tag{71}$$

we can deduce

$$L_{\nabla\Phi}\eta_{\mathbf{x}}^2 + 6L_{\mathbf{y}^\star}^2\eta_{\mathbf{x}}^2 + 4L_{\mathbf{z}^\star}^2\eta_{\mathbf{x}}^2 + 4L_{zx}\rho\eta_{\mathbf{x}}^2 - \frac{\eta_{\mathbf{x}}}{4} \le 0. \tag{72}$$

Then, using the assumptions that

$$\eta_{\mathbf{x}} \le \frac{\mu_g \eta_{\mathbf{y}}}{32L_{\mathbf{y}^\star}^2}, \quad \eta_{\mathbf{x}} \le \frac{\mu_g \eta_{\mathbf{z}}}{32L_{\mathbf{z}^\star}^2}, \tag{73}$$

and inequality (72), we can bound the term $T_3$ in the following manner:

$$T_3 \le \frac{4\eta_{\mathbf{x}}^2}{\eta_{\mathbf{y}}\mu_g}L_{\mathbf{y}^\star}^2 + \frac{4\eta_{\mathbf{x}}^2}{\eta_{\mathbf{z}}\mu_g}L_{\mathbf{z}^\star}^2 - \frac{\eta_{\mathbf{x}}}{4} \le 0. \tag{74}$$

Combining the above inequality (68), (70) and (74), we then drive the convergence result. For simplicity, we define:

$$A_0 := \left(2c_{\mathbf{z}}^2 + 6L_{\mathbf{y}^\star}^2 + 4L_{\mathbf{z}^\star}^2 + 4L_{zx}\rho + L_{\nabla\Phi}\right)(1 + \rho^2) + 2c_{\mathbf{y}}^2, \tag{75}$$

$$A_1 := \left(\left(\frac{8c_{\mathbf{z}}}{\mu_g} + 10\right)(1 + \rho^2) + \frac{3c_{\mathbf{y}}}{\mu_g}\right)\left(1 + \frac{1}{(1 + \delta)n}\right), \tag{76}$$

and

$$A_2 := \frac{8c_{\mathbf{z}}}{\mu_g} + 10, \quad A_3 := \frac{3c_{\mathbf{y}}}{\mu_g} + \frac{8c_{\mathbf{z}}\rho^2}{\mu_g} + 2\rho^2. \tag{77}$$

Thus, substituting the above inequality into (65), re-arranging the R.H.S. in (65), and combining the term $A_0, A_1, A_2, A_3$ from (75), (76) and (77), we obtain

$$\mathcal{L}_{k+1} - \mathcal{L}_k \le -\frac{\eta_{\mathbf{x}}}{4}\mathbb{E}\left[\left\|\nabla\Phi_{\mathcal{H}}\left(\mathbf{x}^k\right)\right\|^2\right] + \eta_{\mathbf{x}}\left(\frac{A_0}{(1-\delta)n}\sigma^2\eta_{\mathbf{x}} + \kappa A_1\sigma^2 + \kappa A_2\zeta_f^2 + \kappa A_3\zeta_g^2\right). \tag{78}$$

For simplicity, we define:
$$A_\sigma := \frac{A_0}{(1-\delta)n}\sigma^2 \tag{79}$$

Rearranging the terms in (78), we obtain
$$\frac{\eta_{\mathbf{x}}}{4} \cdot \mathbb{E}\left[\left\|\nabla\Phi_{\mathcal{H}}\left(\mathbf{x}^k\right)\right\|^2\right] \le \mathcal{L}_k - \mathcal{L}_{k+1} + \eta_{\mathbf{x}}\left(A_\sigma\eta_{\mathbf{x}} + \kappa A_1\sigma^2 + \kappa A_2\zeta_f^2 + \kappa A_3\zeta_g^2\right). \tag{80}$$

As the above inequality (80) is true for an arbitrary $k \in [K]$, by taking summation on both sides from $k = 0$ to $k = K - 1$, we can obtain
$$\sum_{k=0}^{K-1}\mathbb{E}\left[\left\|\nabla\Phi_{\mathcal{H}}\left(\mathbf{x}^k\right)\right\|^2\right] \le 4\left(\frac{\mathcal{L}_0 - \mathcal{L}_K}{\eta_{\mathbf{x}}} + A_\sigma\eta_{\mathbf{x}}K + \kappa A_1\sigma^2 K + \kappa A_2\zeta_f^2 K + \kappa A_3\zeta_g^2 K\right). \tag{81}$$

Dividing both sides of (81) by $K$, we can obtain
$$\frac{1}{K}\sum_{k=0}^{K-1}\mathbb{E}\left[\left\|\nabla\Phi_{\mathcal{H}}\left(\mathbf{x}^k\right)\right\|^2\right] \le 4\left(\frac{\mathcal{L}_0 - \mathcal{L}_K}{\eta_{\mathbf{x}}} + A_\sigma\eta_{\mathbf{x}}K + \kappa A_1\sigma^2 + \kappa A_2\zeta_f^2 + \kappa A_3\zeta_g^2\right). \tag{82}$$

**Analyzing $\mathcal{L}_0 - \mathcal{L}_K$.** Recall the definition of Lyapunov function in (62), we can obtain

$$\mathcal{L}_0 - \mathcal{L}_K \le \Phi_{\mathcal{H}}(\mathbf{x}^0) + \|\mathbf{y}^0 - \mathbf{y}_\star^0\|^2 + \|\mathbf{z}^0 - \mathbf{z}_\star^0\|^2. \tag{83}$$

For simplicity, we define $\Delta_\Phi^0 = \Phi_{\mathcal{H}}(\mathbf{x}^0) + \|\mathbf{y}^0 - \mathbf{y}_\star^0\|^2 + \|\mathbf{z}^0 - \mathbf{z}_\star^0\|^2$.

Invoking (83) into (82), we can obtain
$$\frac{1}{K}\sum_{k=0}^{K-1}\mathbb{E}\left[\left\|\nabla\Phi_{\mathcal{H}}\left(\mathbf{x}^k\right)\right\|^2\right] \le 4\left(\frac{\Delta_\Phi^0}{\eta_{\mathbf{x}}K} + A_\sigma\eta_{\mathbf{x}} + \kappa A_1\sigma^2 + \kappa A_2\zeta_f^2 + \kappa A_3\zeta_g^2\right). \tag{84}$$

Recall that the step size $\eta_{\mathbf{x}}$ satisfies:
$$\eta_{\mathbf{x}} = \min\left\{\alpha_1, \sqrt{\frac{\Delta_\Phi^0}{KA_\sigma}}\right\}, \tag{85}$$

where
$$\alpha_1 := \frac{1}{4\left(L_{\nabla\Phi} + 6L_{\mathbf{y}^\star}^2 + 4L_{\mathbf{z}^\star}^2 + 4L_{zx}\rho\right)}, \quad A_\sigma := \frac{A_0}{(1-\delta)n}\sigma^2 \tag{86}$$

Thus, we have
$$\frac{1}{\eta_{\mathbf{x}}} = \max\left\{\frac{1}{\alpha_1}, \sqrt{\frac{KA_\sigma}{\Delta_\Phi^0}}\right\} \le \frac{1}{\alpha_1} + \sqrt{\frac{KA_\sigma}{\Delta_\Phi^0}}. \tag{87}$$

Therefore, substituting the value of $\frac{1}{\eta_{\mathbf{x}}}$ in (87) into (84), we can obtain
$$\frac{1}{K}\sum_{k=0}^{K-1}\mathbb{E}\left[\left\|\nabla\Phi_{\mathcal{H}}(\mathbf{x}^k)\right\|^2\right] \le 4\left(\frac{\Delta_\Phi^0}{\alpha_1 K} + 2\sqrt{\frac{A_\sigma\Delta_\Phi^0}{K}}\right) + 4\kappa\left(A_1\sigma^2 + A_2\zeta_f^2 + A_3\zeta_g^2\right), \tag{88}$$

which simplifies to:
$$\frac{1}{K}\sum_{k=0}^{K-1}\mathbb{E}\left[\left\|\nabla\Phi_{\mathcal{H}}(\mathbf{x}^k)\right\|^2\right] = \mathcal{O}\left(\frac{\Delta_\Phi^0}{K} + \sqrt{\frac{A_\sigma\Delta_\Phi^0}{K}}\right) + \mathcal{O}\left(\kappa\left(\zeta_f^2 + \zeta_g^2 + \sigma^2\right)\right). \tag{89}$$

As $\hat{\mathbf{x}}^{(i)} \sim \mathcal{U}\left\{\mathbf{x}_0^{(i)}, \ldots, \mathbf{x}_{K-1}^{(i)}\right\}$, combing with (89), we finally get:
$$\mathbb{E}\left[\|\nabla\Phi_{\mathcal{H}}(\hat{\mathbf{x}})\|^2\right] = \mathcal{O}\left(\frac{\Delta_\Phi^0}{K} + \sqrt{\frac{A_\sigma\Delta_\Phi^0}{K}}\right) + \mathcal{O}\left(\kappa\left(\zeta_f^2 + \zeta_g^2 + \sigma^2\right)\right). \tag{90}$$

Hence, the proof is complete. $\qquad\square$

# H CONVERGENCE ANALYSIS FOR BR-FEDBIM (THEOREM 3)

In this section, we drive the convergence analysis of BR-FedBiM, which shows that incorporating momentum helps mitigate the impact of local variance on asymptotic error by reducing the drift between local and global parameters.

## H.1 PROOF OUTLINE

Our convergence analysis for BR-FedBiM in Algorithm 1, consists of five elements:

(1) Aggregation error for momentum (Lemma 13),

(2) Lower-level convergence (Lemma 14),

(3) Gradient Update Bias (Lemma 15),

(4) Momentum Deviation Lemma (Lemma 16),

(5) Descent lemma (Lemma 17).

**Notation.** Recall that, at each iteration $k$, Byzantine clients send carefully crafted malicious gradients to the server with the intention of degrading the global model's performance. Meanwhile, each honest client $i$ computes the momentum term as:

$$
\begin{aligned}
\mathbf{m}_{\mathbf{x},i}^k &= \beta_{\mathbf{x}} m_{\mathbf{x},i}^{k-1} + (1 - \beta_{\mathbf{x}}) \mathbf{D}_{\mathbf{x},i}^k, \\
\mathbf{m}_{\mathbf{y},i}^k &= \beta_{\mathbf{y}} m_{\mathbf{y},i}^{k-1} + (1 - \beta_{\mathbf{y}}) \mathbf{D}_{\mathbf{y},i}^k, \\
\mathbf{m}_{\mathbf{z},i}^k &= \beta_{\mathbf{z}} m_{\mathbf{z},i}^{k-1} + (1 - \beta_{\mathbf{z}}) \mathbf{D}_{\mathbf{z},i}^k.
\end{aligned}
\tag{91}
$$

After receiving the gradients, the server aggregates the gradients using robust aggregation rules $AGG(\cdot)$, ensuring robustness by satisfying the $(\delta, \kappa)$ criteria in Definition 2. The aggregated momentum terms for $\mathbf{x}$, $\mathbf{y}$, and $\mathbf{z}$ are computed as follows:

$$
\begin{aligned}
\hat{\mathbf{m}}_{\mathbf{x}}^k &:= AGG(\mathbf{m}_{\mathbf{x},1}^k, \ldots, \mathbf{m}_{\mathbf{x},n}^k), \\
\hat{\mathbf{m}}_{\mathbf{y}}^k &:= AGG(\mathbf{m}_{\mathbf{y},1}^k, \ldots, \mathbf{m}_{\mathbf{y},n}^k), \\
\hat{\mathbf{m}}_{\mathbf{z}}^k &:= AGG(\mathbf{m}_{\mathbf{z},1}^k, \ldots, \mathbf{m}_{\mathbf{z},n}^k).
\end{aligned}
\tag{92}
$$

## H.2 AGGREGATION ERROR

By using the $(f, \kappa)$-robustness criteria in Definition 2 and the bounded heterogeneity Assumption 4, we define the error for BR-FedBiM to be the distance between aggregated momentum and *True* update:

$$
\begin{aligned}
\xi_{\mathbf{x}}^k &:= \hat{\mathbf{m}}_{\mathbf{x}}^k - \mathbf{m}_{\mathbf{x}}^k \\
\xi_{\mathbf{y}}^k &:= \hat{\mathbf{m}}_{\mathbf{y}}^k - \mathbf{m}_{\mathbf{y}}^k \\
\xi_{\mathbf{z}}^k &:= \hat{\mathbf{m}}_{\mathbf{z}}^k - \mathbf{m}_{\mathbf{z}}^k
\end{aligned}
\tag{93}
$$

where $\mathbf{m}_{\mathbf{x}}^k = \frac{1}{|\mathcal{H}|} \sum_{i \in \mathcal{H}} \mathbf{m}_{\mathbf{x},i}^k$, $\quad \mathbf{m}_{\mathbf{y}}^k = \frac{1}{|\mathcal{H}|} \sum_{i \in \mathcal{H}} \mathbf{m}_{\mathbf{y},i}^k$, $\quad \mathbf{m}_{\mathbf{z}}^k = \frac{1}{|\mathcal{H}|} \sum_{i \in \mathcal{H}} \mathbf{m}_{\mathbf{z},i}^k$.

We get the following aggregation error bound, the detailed proof can be found in Appendix J.2.1

**Lemma 13.** *Suppose Assumptions 1-4 hold,and the robust aggregation rule $AGG(\cdot)$ satisfies the $(\delta, \kappa)$-robustness criterion. The aggregation errors in the BR-FedBiM algorithm (Algorithm 1) satisfy the following inequalities:*

$$
(1) \quad \mathbb{E}\left[\left\|\xi_{\mathbf{y}}^k\right\|^2\right] \le 3\kappa \left(\frac{1 - \beta_{\mathbf{y}}}{1 + \beta_{\mathbf{y}}}\right) \left(1 + \frac{1}{(1 - \delta)n}\right) \sigma^2 + 3\kappa \zeta_g^2,
$$

$$
(2) \quad \mathbb{E}\left[\left\|\xi_{\mathbf{z}}^k\right\|^2\right] \le 6\kappa \left(\frac{1 - \beta_{\mathbf{z}}}{1 + \beta_{\mathbf{z}}}\right) \left(1 + \frac{1}{(1 - \delta)n}\right) \sigma_1^2 + 6\kappa \left(\rho^2 \zeta_g^2 + \zeta_f^2\right),
$$

$$
(3) \quad \mathbb{E}\left[\left\|\xi_{\mathbf{x}}^k\right\|^2\right] \le 6\kappa \left(\frac{1 - \beta_{\mathbf{x}}}{1 + \beta_{\mathbf{x}}}\right) \left(1 + \frac{1}{(1 - \delta)n}\right) \sigma_1^2 + 6\kappa \left(\rho^2 \zeta_g^2 + \zeta_f^2\right),
$$

*where $\sigma_1^2 = \left(1 + \rho^2\right) \sigma^2$ and $\rho = C_f / \mu_g$.*

## H.3 Lower-Level Convergence

In this subsection, we analyze the lower-level convergence behavior of our bilevel algorithm. The detailed proof of this lemma is provided in J.2.2.

**Lemma 14.** *Under Assumptions 1, 2, 3, and 4, and assuming $\eta_{\mathbf{y}} \le \min\left\{\lambda_g, \frac{1}{\mu_g + L_g}\right\}$ and $\eta_{\mathbf{z}} \le \min\left\{\lambda_g, \frac{1}{\mu_g + L_g}\right\}$, the following holds:*

$$
\begin{aligned}
(1) \quad \mathbb{E}\left[\left\|\mathbf{y}^{k+1} - \mathbf{y}_\star^{k+1}\right\|^2\right] &\le \left(1 - \frac{\mu_g L_g \eta_{\mathbf{y}}}{2\left(\mu_g + L_g\right)}\right) \mathbb{E}\left[\left\|\mathbf{y}^k - \mathbf{y}_\star^k\right\|^2\right] - \frac{\eta_{\mathbf{y}}}{\mu_g L_g} \mathbb{E}\left[\left\|\mathbb{E}(\mathbf{D}_{\mathbf{y}}^k)\right\|^2\right] \\
&\quad + \frac{12\left(\mu_g + L_g\right)\eta_{\mathbf{y}}}{\mu_g L_g} \left(\mathbb{E}\left[\left\|\mathbf{m}_{\mathbf{y}}^k - \mathbb{E}(\mathbf{D}_{\mathbf{y}}^k)\right\|^2\right] + \mathbb{E}\left[\left\|\xi_{\mathbf{y}}^k\right\|^2\right]\right) \\
&\quad + \frac{2\left(\mu_g + L_g\right) L_{\mathbf{y}^\star}^2 \eta_{\mathbf{x}}^2}{\mu_g L_g \eta_{\mathbf{y}}} \mathbb{E}\left[\left\|\hat{\mathbf{m}}_{\mathbf{x}}^k\right\|^2\right],
\end{aligned}
$$

$$
\begin{aligned}
(2) \quad \mathbb{E}\left[\left\|\mathbf{z}^{k+1} - \mathbf{z}_\star^{k+1}\right\|^2\right] &\le \left(1 - \frac{\mu_g L_g \eta_{\mathbf{z}}}{2\left(\mu_g + L_g\right)}\right) \mathbb{E}\left[\left\|\mathbf{z}^k - \mathbf{z}_\star^k\right\|^2\right] - \frac{\eta_{\mathbf{z}}}{\mu_g L_g} \mathbb{E}\left[\left\|\mathbb{E}(\mathbf{D}_{\mathbf{z}}^k)\right\|^2\right] \\
&\quad + \frac{12\left(\mu_g + L_g\right)\eta_{\mathbf{z}}}{\mu_g L_g} \left(\mathbb{E}\left[\left\|\mathbf{m}_{\mathbf{z}}^k - \mathbb{E}(\mathbf{D}_{\mathbf{z}}^k)\right\|^2\right] + \mathbb{E}\left[\left\|\xi_{\mathbf{z}}^k\right\|^2\right]\right) \\
&\quad + \frac{2\left(\mu_g + L_g\right) L_{\mathbf{z}^\star}^2 \eta_{\mathbf{x}}^2}{\mu_g L_g \eta_{\mathbf{z}}} \mathbb{E}\left[\left\|\hat{\mathbf{m}}_{\mathbf{x}}^k\right\|^2\right].
\end{aligned}
$$

## H.4 Gradient Update Bias

Next, we analyze the gradient update bias of BR-FedBiM in Algorithm 1. The proof details of this lemma can be found in J.2.3

**Lemma 15.** *Suppose Assumption 1, 2 and 3 hold, if $\rho \ge C_f / \mu_g$, we have:*

$$
\begin{aligned}
\mathbb{E}\left[\left\|\nabla \Phi_{\mathcal{H}}\left(\mathbf{x}^k\right) - \mathbf{m}_{\mathbf{x}}^k\right\|^2\right] &\le 6L_2^2 \mathbb{E}\left[\left\|\mathbf{y}^k - \mathbf{y}_\star^k\right\|^2\right] + 6L_g^2 \mathbb{E}\left[\left\|\mathbf{z}^k - \mathbf{z}_\star^k\right\|^2\right] \\
&\quad + 2\mathbb{E}\left[\left\|\mathbf{m}_{\mathbf{x}}^k - \mathbb{E}\left(\mathbf{D}_{\mathbf{x}}^k\right)\right\|^2\right],
\end{aligned}
$$

*where $L_2^2 = L_f^2 + L_g^2 \rho^2$.*

## H.5 MOMENTUM DEVIATION

We now analyze the momentum deviation, which is defined as the distance between the average momentum and the true gradient. The proof of this lemma can be found in Appendix J.2.4.

**Lemma 16.** *Suppose Assumptions 1-4 hold. Considering Algorithm BR-FedBiM in 1, for each $k \in [K]$, the following bounds hold for $\mathbf{m}_{\mathbf{y}}^{k+1}$, $\mathbf{m}_{\mathbf{x}}^{k+1}$, and $\mathbf{m}_{\mathbf{z}}^{k+1}$:*

(1) $\mathbb{E}\left[\left\|\mathbf{m}_{\mathbf{y}}^{k+1} - \mathbb{E}\left(\mathbf{D}_{\mathbf{y}}^{k+1}\right)\right\|^2\right]$

$$\leq \beta_{\mathbf{y}}^2 \left(1 + L_g \eta_{\mathbf{y}}\right)\left(1 + 3\eta_{\mathbf{y}} L_g\right) \mathbb{E}\left[\left\|\mathbf{m}_{\mathbf{y}}^k - \mathbb{E}\left(\mathbf{D}_{\mathbf{y}}^k\right)\right\|^2\right] + L_g^2 \eta_{\mathbf{x}}^2 \beta_{\mathbf{y}}^2 \left(1 + \frac{1}{\eta_{\mathbf{y}} L_g}\right) \mathbb{E}\left[\left\|\hat{\mathbf{m}}_{\mathbf{x}}^k\right\|^2\right]$$

$$+ 3\beta_{\mathbf{y}}^2 \left(1 + \frac{1}{\eta_{\mathbf{y}} L_g}\right)\left(\eta_{\mathbf{y}}^2 L_g^4 \mathbb{E}\left[\left\|\mathbf{y}_k - \mathbf{y}^*\left(\mathbf{x}^k\right)\right\|^2\right] + \eta_{\mathbf{y}}^2 L_g^2 \mathbb{E}\left[\left\|\xi_{\mathbf{y}}^k\right\|^2\right]\right) + (1 - \beta_{\mathbf{y}})^2 \frac{\sigma^2}{(1-\delta)n}.$$

(2) $\mathbb{E}\left[\left\|\mathbf{m}_{\mathbf{x}}^{k+1} - \mathbb{E}\left(\mathbf{D}_{\mathbf{x}}^{k+1}\right)\right\|^2\right]$

$$\leq \beta_{\mathbf{x}}^2 \left(1 + L_g \eta_{\mathbf{z}}\right) \mathbb{E}\left[\left\|\mathbf{m}_{\mathbf{x}}^k - \mathbb{E}\left(\mathbf{D}_{\mathbf{x}}^k\right)\right\|^2\right] + 4\beta_{\mathbf{x}}^2 \left(1 + \frac{1}{\eta_{\mathbf{z}} L_1}\right) L_1^2 \eta_{\mathbf{x}}^2 \mathbb{E}\left[\left\|\hat{\mathbf{m}}_{\mathbf{x}}^k\right\|^2\right]$$

$$+ 12\beta_{\mathbf{x}}^2 \left(1 + \frac{1}{\eta_{\mathbf{z}} L_1}\right) L_1^2 \eta_{\mathbf{y}}^2 \mathbb{E}\left[\left\|\mathbf{m}_{\mathbf{y}}^k - \mathbb{E}\left(\mathbf{D}_{\mathbf{y}}^k\right)\right\|^2\right]$$

$$+ 12\beta_{\mathbf{x}}^2 \left(1 + \frac{1}{\eta_{\mathbf{z}} L_g}\right) L_g^2 \eta_{\mathbf{z}}^2 \mathbb{E}\left[\left\|\mathbf{m}_{\mathbf{z}}^k - \mathbb{E}\left(\mathbf{D}_{\mathbf{z}}^k\right)\right\|^2\right]$$

$$+ 12\beta_{\mathbf{x}}^2 \left(1 + \frac{1}{\eta_{\mathbf{z}} L_g}\right) \left(L_1^2 L_g^2 \eta_{\mathbf{y}}^2 + 3L_g^2 L_2^2 \eta_{\mathbf{z}}^2\right) \mathbb{E}\left[\left\|\mathbf{y}_k - \mathbf{y}^*\left(\mathbf{x}^k\right)\right\|^2\right]$$

$$+ 36\beta_{\mathbf{x}}^2 \left(1 + \frac{1}{\eta_{\mathbf{z}} L_g}\right) L_g^4 \eta_{\mathbf{z}}^2 \mathbb{E}\left[\left\|\mathbf{z}_k - \mathbf{z}^*\left(\mathbf{x}^k\right)\right\|^2\right]$$

$$+ 12\beta_{\mathbf{x}}^2 \left(1 + \frac{1}{\eta_{\mathbf{z}} L_g}\right) L_g^2 \eta_{\mathbf{z}}^2 \mathbb{E}\left[\left\|\xi_{\mathbf{z}}^k\right\|^2\right] + 12\beta_{\mathbf{x}}^2 \left(1 + \frac{1}{\eta_{\mathbf{z}} L_g}\right) L_1^2 \eta_{\mathbf{y}}^2 \mathbb{E}\left[\left\|\xi_{\mathbf{y}}^k\right\|^2\right] + (1 - \beta_{\mathbf{x}})^2 \frac{\sigma_1^2}{(1-\delta)n}.$$

(3) $\mathbb{E}\left[\left\|\mathbf{m}_{\mathbf{z}}^{k+1} - \mathbb{E}\left(\mathbf{D}_{\mathbf{z}}^{k+1}\right)\right\|^2\right]$

$$\leq \beta_{\mathbf{z}}^2 \left(1 + L_g \eta_{\mathbf{z}}\right)\left(1 + 12 L_g \eta_{\mathbf{z}}\right) \mathbb{E}\left[\left\|\mathbf{m}_{\mathbf{z}}^k - \mathbb{E}\left(\mathbf{D}_{\mathbf{z}}^k\right)\right\|^2\right] + 4\beta_{\mathbf{z}}^2 \left(1 + \frac{1}{\eta_{\mathbf{z}} L_g}\right) L_1^2 \eta_{\mathbf{x}}^2 \mathbb{E}\left[\left\|\hat{\mathbf{m}}_{\mathbf{x}}^k\right\|^2\right]$$

$$+ 12\beta_{\mathbf{z}}^2 \left(1 + \frac{1}{\eta_{\mathbf{z}} L_g}\right) L_1^2 \eta_{\mathbf{y}}^2 \mathbb{E}\left[\left\|\mathbf{m}_{\mathbf{y}}^k - \mathbb{E}\left(\mathbf{D}_{\mathbf{y}}^k\right)\right\|^2\right]$$

$$+ 12\beta_{\mathbf{z}}^2 \left(1 + \frac{1}{\eta_{\mathbf{z}} L_g}\right) \left(L_1^2 L_g^2 \eta_{\mathbf{y}}^2 + 3L_g^2 L_2^2 \eta_{\mathbf{z}}^2\right) \mathbb{E}\left[\left\|\mathbf{y}_k - \mathbf{y}^*\left(\mathbf{x}^k\right)\right\|^2\right]$$

$$+ 36\beta_{\mathbf{z}}^2 \left(1 + \frac{1}{\eta_{\mathbf{z}} L_g}\right) L_g^4 \eta_{\mathbf{z}}^2 \mathbb{E}\left[\left\|\mathbf{z}_k - \mathbf{z}^*\left(\mathbf{x}^k\right)\right\|^2\right]$$

$$+ 12\beta_{\mathbf{z}}^2 \left(1 + \frac{1}{\eta_{\mathbf{z}} L_g}\right) L_1^2 \eta_{\mathbf{y}}^2 \mathbb{E}\left[\left\|\xi_{\mathbf{y}}^k\right\|^2\right] + 12\beta_{\mathbf{z}}^2 \left(1 + \frac{1}{\eta_{\mathbf{z}} L_g}\right) L_g^2 \eta_{\mathbf{z}}^2 \mathbb{E}\left[\left\|\xi_{\mathbf{z}}^k\right\|^2\right] + (1 - \beta_{\mathbf{z}})^2 \frac{\sigma_1^2}{(1-\delta)n}.$$

## H.6 DESCENT LEMMA

In this section, we analyze the progression of the upper-level loss function in BR-FedBiM as described in Algorithm 1. At each iteration, we have the following updates:

$$\mathbf{x}^{k+1} = \mathbf{x}^k - \eta_{\mathbf{x}} \text{AGG}\left(\mathbf{m}_{\mathbf{x},1}^k, \ldots, \mathbf{m}_{\mathbf{x},n}^k\right) = \mathbf{x}^k - \eta_{\mathbf{x}} \hat{\mathbf{m}}_{\mathbf{x}}^k = \mathbf{x}^k - \eta_{\mathbf{x}} \mathbf{m}_{\mathbf{x}}^k - \eta_{\mathbf{x}} \xi_{\mathbf{x}}^k,$$

$$\mathbf{y}^{k+1} = \mathbf{y}^k - \eta_{\mathbf{y}} \text{AGG}\left(\mathbf{m}_{\mathbf{y},1}^k, \ldots, \mathbf{m}_{\mathbf{y},n}^k\right) = \mathbf{y}^k - \eta_{\mathbf{y}} \hat{\mathbf{m}}_{\mathbf{y}}^k = \mathbf{y}^k - \eta_{\mathbf{y}} \mathbf{m}_{\mathbf{y}}^k - \eta_{\mathbf{y}} \xi_{\mathbf{y}}^k,$$

$$\mathbf{z}^{k+1} = \mathbf{z}^k - \eta_{\mathbf{z}} \text{AGG}\left(\mathbf{m}_{\mathbf{z},1}^k, \ldots, \mathbf{m}_{\mathbf{z},n}^k\right) = \mathbf{z}^k - \eta_{\mathbf{z}} \hat{\mathbf{m}}_{\mathbf{z}}^k = \mathbf{z}^k - \eta_{\mathbf{z}} \mathbf{m}_{\mathbf{z}}^k - \eta_{\mathbf{z}} \xi_{\mathbf{z}}^k,$$

where $\text{AGG}(\cdot)$ denotes the aggregation operator, $\mathbb{E}(\cdot)$ represents the expected value over randomness, and $\xi_{\mathbf{x}}^k$, $\xi_{\mathbf{y}}^k$, and $\xi_{\mathbf{z}}^k$ denote the aggregation error terms.

The detailed proof of Lemma 17 can be found in Appendix J.2.5.

---

**Lemma 17.** *Assuming that the upper-level loss function $\Phi_{\mathcal{H}}(\cdot)$ is $L_{\nabla\Phi}$-smooth, we have:*

$$\mathbb{E}\left[\Phi_{\mathcal{H}}\left(\mathbf{x}^{k+1}\right)\right] \leq \mathbb{E}\left[\Phi_{\mathcal{H}}\left(\mathbf{x}^{k}\right)\right] - \frac{\eta_{\mathbf{x}}}{2}\mathbb{E}\left[\left\|\nabla\Phi_{\mathcal{H}}\left(\mathbf{x}^{k}\right)\right\|^2\right] - \left(\frac{\eta_{\mathbf{x}}}{2} - \frac{L_{\nabla\Phi}\eta_{\mathbf{x}}^2}{2}\right)\mathbb{E}\left[\left\|\hat{\mathbf{m}}_{\mathbf{x}}^k\right\|^2\right]$$
$$+ \eta_{\mathbf{x}}\mathbb{E}\left[\left\|\nabla\Phi_{\mathcal{H}}\left(\mathbf{x}^{k}\right) - m_{\mathbf{x}}^k\right\|^2\right] + \eta_{\mathbf{x}}\mathbb{E}\left[\left\|\xi_{\mathbf{x}}^k\right\|^2\right].$$

---

### H.7 PROOF OF THEOREM 3

**Theorem 6.** *Suppose that Assumptions 1-4 hold, and that the robust aggregation rule $\text{AGG}(\cdot)$ satisfies the $(\delta, \kappa)$-robustness criterion. There exist positive constants $c$, $c_{\mathbf{y}}$, and $c_{\mathbf{z}}$ such that*

$$\eta_{\mathbf{y}} = c_{\mathbf{y}}\eta_{\mathbf{x}}, \quad \eta_{\mathbf{z}} = c_{\mathbf{z}}\eta_{\mathbf{x}},$$

*and step size $\eta_{\mathbf{x}}$ satisfies,*

$$\eta_{\mathbf{x}} \leq \min\left\{\alpha_1, \sqrt{\frac{\Delta_\Phi^0}{KA_\sigma}}\right\},$$

*where*

$$\alpha_1 = \frac{1}{8L_{\nabla\Phi}}, A_\sigma = \frac{A_0'}{(1-\delta)n}\sigma^2 + \kappa A_3\sigma^2,$$

*and $A_0'$ and $A_3$ are positive constants independent of $n$ and $K$. Then, for $K \geq 1$, the algorithm converges as:*

$$\mathbb{E}\left[\left\|\nabla\Phi_{\mathcal{H}}(\hat{\mathbf{x}})\right\|^2\right] = \mathcal{O}\left(\frac{\Delta_\Phi^0}{K} + \sqrt{\frac{A_\sigma\Delta_\Phi^0}{K}}\right) + \mathcal{O}\left(\kappa\left(\zeta_f^2 + \zeta_g^2 + \zeta_g^2\rho^2\right)\right), \tag{94}$$

*where $\Delta_\Phi^0 = \Phi_{\mathcal{H}}(\mathbf{x}^0) + P_k\left\|\mathbf{y}^0 - \mathbf{y}_\star^0\right\|^2 + Q_k\left\|\mathbf{z}^0 - \mathbf{z}_\star^0\right\|^2 + A_k\left\|\mathbb{E}(\mathbf{D}_{\mathbf{x}}^0)\right\|^2 + B_k\left\|\mathbb{E}(\mathbf{D}_{\mathbf{y}}^0)\right\|^2 + C_k\left\|\mathbb{E}(\mathbf{D}_{\mathbf{z}}^0)\right\|^2$, and $P_k, Q_k, A_k, B_k, C_k$ are positive constants. The expectation is over the randomness of the algorithm.*

*Proof.* To obtain the final convergence result, we build upon the above lemma. To obtain the convergence result we define the Lyapunov function to be

$$\mathcal{L}_k = \Phi_{\mathcal{H}}(\mathbf{x}^k) + P_k\mathbb{E}\left[\left\|\mathbf{y}_k - \mathbf{y}_\star^k\right\|^2\right] + Q_k\mathbb{E}\left[\left\|\mathbf{z}_k - \mathbf{z}_\star^k\right\|^2\right]$$
$$+ A_k\mathbb{E}\left[\left\|m_{\mathbf{x}}^k - \mathbb{E}(\mathbf{D}_{\mathbf{x}}^k)\right\|^2\right] + B_k\mathbb{E}\left[\left\|\mathbf{m}_{\mathbf{y}}^k - \mathbb{E}(\mathbf{D}_{\mathbf{y}}^k)\right\|^2\right] + C_k\mathbb{E}\left[\left\|\mathbf{m}_{\mathbf{z}}^k - \mathbb{E}(\mathbf{D}_{\mathbf{z}}^k)\right\|^2\right]. \tag{95}$$

The coefficients are chosen to be

$$P_k = \frac{8L_g^2L_2}{L_1\lambda_g} + \frac{2L_g^2}{\lambda_g}, Q_k = \frac{8L_g^4}{L_1^2\lambda_g}, A_k = \frac{L_g}{36L_1^2}, B_k = \frac{1}{36L_g}, C_k = \frac{L_g}{36L_1^2}. \tag{96}$$

By invoking Lemmas 14, 15, 16 and 17 into (95), we can obtain:

$$\mathcal{L}_{k+1} - \mathcal{L}_k$$
$$\leq -\frac{\eta_{\mathbf{x}}}{2}\mathbb{E}\left[\left\|\nabla\Phi_{\mathcal{H}}(\mathbf{x}^k)\right\|^2\right]$$
$$+ \underbrace{\left(6\eta_{\mathbf{x}}L_2^2 - \frac{\lambda_g\eta_{\mathbf{y}}}{2}P_k + 12A_k\beta_{\mathbf{x}}^2\left(1 + \frac{1}{\eta_{\mathbf{z}}L_g}\right)\left(L_1^2L_g^2\eta_{\mathbf{y}}^2 + 3L_g^2L_2^2\eta_{\mathbf{z}}^2\right)\right.}_{T_1(1)}$$

$$+3B_k\beta_{\mathbf{y}}^2\left(1+\frac{1}{\eta_{\mathbf{y}}L_g}\right)\eta_{\mathbf{y}}^2L_g^4+12C_k\beta_{\mathbf{z}}^2\left(1+\frac{1}{\eta_{\mathbf{z}}L_g}\right)\left(L_1^2L_g^2\eta_{\mathbf{y}}^2+3L_g^2L_2^2\eta_{\mathbf{z}}^2\right)\Bigg)\underbrace{\mathbb{E}\left[\left\|\mathbf{y}_k-\mathbf{y}^\star(\mathbf{x}^k)\right\|^2\right]}_{}$$
$$\underbrace{\phantom{+3B_k\beta_{\mathbf{y}}^2\left(1+\frac{1}{\eta_{\mathbf{y}}L_g}\right)\eta_{\mathbf{y}}^2L_g^4+12C_k\beta_{\mathbf{z}}^2\left(1+\frac{1}{\eta_{\mathbf{z}}L_g}\right)}}_{T_1(2)}$$

$$+\underbrace{\left(6\eta_{\mathbf{x}}L_g^2-\frac{\lambda_g\eta_{\mathbf{z}}}{2}Q_k+36A_k\beta_{\mathbf{x}}^2\left(1+\frac{1}{\eta_{\mathbf{z}}L_g}\right)L_g^4\eta_{\mathbf{z}}^2+36C_k\beta_{\mathbf{z}}^2\left(1+\frac{1}{\eta_{\mathbf{z}}L_g}\right)L_g^4\eta_{\mathbf{z}}^2\right)}_{T_2}\mathbb{E}\left[\left\|\mathbf{z}_k-\mathbf{z}^\star(\mathbf{x}^k)\right\|^2\right]$$

$$+\underbrace{\left(2\eta_{\mathbf{x}}+A_k\left(\beta_{\mathbf{x}}^2(1+L_g\eta_{\mathbf{z}})-1\right)\right)}_{T_3}\mathbb{E}\left[\left\|\mathbf{m}_{\mathbf{x}}^k-\mathbb{E}(\mathbf{D}_{\mathbf{x}}^k)\right\|^2\right]$$

$$+\underbrace{\left(B_k\left(\beta_{\mathbf{y}}^2(1+L_g\eta_{\mathbf{y}})(1+3L_g\eta_{\mathbf{y}})-1\right)+12A_k\beta_{\mathbf{x}}^2\left(1+\frac{1}{\eta_{\mathbf{z}}L_g}\right)L_1^2\eta_{\mathbf{y}}^2\right.}_{T_4(1)}$$

$$\underbrace{+12C_k\beta_{\mathbf{z}}^2\left(1+\frac{1}{\eta_{\mathbf{z}}L_g}\right)L_1^2\eta_{\mathbf{y}}^2+\frac{12\eta_{\mathbf{y}}}{\lambda_g}P_k\Bigg)}_{T_4(2)}\mathbb{E}\left[\left\|\mathbf{m}_{\mathbf{y}}^k-\mathbb{E}(\mathbf{D}_{\mathbf{y}}^k)\right\|^2\right]$$

$$+\underbrace{\left(C_k\left(\beta_{\mathbf{z}}^2(1+L_g\eta_{\mathbf{z}})(1+12L_g\eta_{\mathbf{z}})-1\right)+12A_k\beta_{\mathbf{x}}^2\left(1+\frac{1}{\eta_{\mathbf{z}}L_g}\right)L_g^2\eta_{\mathbf{z}}^2+\frac{12\eta_{\mathbf{z}}}{\lambda_g}Q_k\right)}_{T_5}\mathbb{E}\left[\left\|\mathbf{m}_{\mathbf{z}}^k-\mathbb{E}(\mathbf{D}_{\mathbf{z}}^k)\right\|^2\right]$$

$$+\underbrace{\left(-\frac{\eta_{\mathbf{x}}}{2}+\frac{L_{\nabla\Phi}\eta_{\mathbf{x}}^2}{2}+\frac{2L_{\mathbf{z}^\star}^2\eta_{\mathbf{x}}^2}{\lambda_g\eta_{\mathbf{z}}}Q_k+\frac{2L_{\mathbf{y}^\star}^2\eta_{\mathbf{x}}^2}{\lambda_g\eta_{\mathbf{y}}}P_k+4A_k\beta_{\mathbf{x}}^2\left(1+\frac{1}{\eta_{\mathbf{z}}L_g}\right)L_1^2\eta_{\mathbf{x}}^2\right.}_{T_6(1)}\tag{97}$$

$$\underbrace{+B_k\beta_{\mathbf{y}}^2\left(1+\frac{1}{\eta_{\mathbf{y}}L_g}\right)L_g^2\eta_{\mathbf{x}}^2+4C_k\beta_{\mathbf{z}}^2\left(1+\frac{1}{\eta_{\mathbf{z}}L_g}\right)L_1^2\eta_{\mathbf{x}}^2\Bigg)}_{T_6(2)}\mathbb{E}\left[\left\|\hat{\mathbf{m}}_{\mathbf{x}}^k\right\|^2\right]$$

$$+A_k(1-\beta_{\mathbf{x}})^2\frac{\sigma_1^2}{(1-\delta)n}+B_k(1-\beta_{\mathbf{y}})^2\frac{\sigma^2}{(1-\delta)n}+C_k(1-\beta_{\mathbf{z}})^2\frac{\sigma_1^2}{(1-\delta)n}$$

$$+\eta_{\mathbf{x}}\mathbb{E}\left[\left\|\xi_{\mathbf{x}}^k\right\|^2\right]+A_1\eta_{\mathbf{x}}\mathbb{E}\left[\left\|\xi_{\mathbf{y}}^k\right\|^2\right]+A_2\eta_{\mathbf{x}}\mathbb{E}\left[\left\|\xi_{\mathbf{z}}^k\right\|^2\right].$$

For simplicity, we define:

$$A_1=\left(\frac{1}{8}+\frac{3c_{\mathbf{z}}}{4c_{\mathbf{y}}}+\frac{96L_2L_g^2}{\lambda_g^2L_1}+\frac{24L_g^2}{\lambda_g^2}\right)c_{\mathbf{y}},\tag{98}$$

and

$$A_2=\left(\frac{3L_g^2}{4L_1^2}+\frac{96L_g^4}{\lambda_g^2L_1^2}\right)c_{\mathbf{z}}.\tag{99}$$

We now analyze each term on the R.H.S. to further bound them.

**Analysis of the Coefficient for $\mathbb{E}\left[\left\|\mathbf{y}_k-\mathbf{y}^\star(\mathbf{x}^k)\right\|^2\right]$.**

By assuming

$$\eta_{\mathbf{y}}\le\frac{1}{18L_g},\quad\eta_{\mathbf{z}}\le\frac{1}{18L_g},\tag{100}$$

incorporating $P_k = \frac{8L_g^2 L_2}{L_1 \lambda_g} + \frac{2L_g^2}{\lambda_g}$, $A_k = C_k = \frac{L_g}{36L_1^2}$ and $B_k = \frac{1}{36L_g}$ as in (96), we can bound the term $T_1$:

$$T_1 \leq 6\eta_\mathbf{x} L_2^2 - \eta_\mathbf{y} \left( \frac{4L_g^2 L_2}{L_1} + L_g^2 \right) + \frac{1}{3} L_g^2 \eta_\mathbf{y} + 3L_g^2 \frac{L_2^2}{L_1^2} \eta_\mathbf{z} + \frac{2}{3} L_g^2 \frac{\eta_\mathbf{y}^2}{\eta_\mathbf{z}}. \tag{101}$$

By assuming

$$\eta_\mathbf{y} \leq \frac{4}{3} \frac{L_1}{L_2} \eta_\mathbf{z}, \quad \eta_\mathbf{z} \leq 6 \frac{L_2}{L_1} \eta_\mathbf{y}, \quad \eta_\mathbf{x} \leq \frac{L_g^2}{9L_2^2} \eta_\mathbf{y}, \tag{102}$$

we further bound the term $T_1$ in the following manner:

$$T_1 \leq 6\eta_\mathbf{x} L_2^2 - \eta_\mathbf{y} \left( \frac{4L_g^2 L_2}{L_1} + L_g^2 \right) + \frac{1}{3} L_g^2 \eta_\mathbf{y} + 3L_g^2 \frac{L_2^2}{L_1^2} \eta_\mathbf{z} + \frac{2}{3} L_g^2 \frac{\eta_\mathbf{y}^2}{\eta_\mathbf{z}}$$

$$\leq 6\eta_\mathbf{x} L_2^2 - \frac{2}{3} L_g^2 \eta_\mathbf{y} \leq 0. \tag{103}$$

**Analysis of the Coefficient for** $\mathbb{E}\left[ \left\| \mathbf{z}_k - \mathbf{z}^\star(\mathbf{x}^k) \right\|^2 \right]$.

Under the assumptions that,

$$\eta_\mathbf{x} \leq \frac{L_g^2}{6L_1^2} \eta_\mathbf{z}, \tag{104}$$

and incorporating $A_k = C_k = \frac{L_g}{36L_1^2}$ and $Q_k = \frac{8L_g^4}{L_1^2 \lambda_g}$ as in (96), we further bound the term $T_2$ in the following manner

$$T_2 = 6\eta_\mathbf{x} L_g^2 - \frac{4L_g^4}{L_1^2} \eta_\mathbf{z} + \frac{2L_g^5}{L_1^2} \eta_\mathbf{z}^2 + \frac{2L_g^4}{L_1^2} \eta_\mathbf{z} \leq 6\eta_\mathbf{x} L_g^2 - \frac{L_g^4}{L_1^2} \eta_\mathbf{z} \leq 0. \tag{105}$$

**Analysis of the Coefficient for** $\mathbb{E}\left[ \left\| \mathbf{m}_\mathbf{x}^k - \mathbb{E}(\mathbf{D}_\mathbf{x}^k) \right\|^2 \right]$.

We analyze $T_3$ by rewriting it in a form that separates the contributions from the momentum parameter $\beta_\mathbf{x}$ and the step size $\eta_\mathbf{x}$:

$$T_3 = 2\eta_\mathbf{x} + A_k \left( \beta_\mathbf{x}^2 (1 + L_g \eta_\mathbf{z}) - 1 \right) = -(1 - \beta_\mathbf{x}^2) A_k + \eta_\mathbf{x} \left( 2 + \beta_\mathbf{x}^2 L_g A_k c_\mathbf{z} \right). \tag{106}$$

Using the fact that $\beta_\mathbf{x}^2 \leq 1$, and incorporating $A_k = \frac{L_g}{36L_1^2}$ as in (96), we bound the term $T_3$:

$$T_3 = -(1 - \beta_\mathbf{x}^2) A_k + \eta_\mathbf{x} \left( 2 + \beta_\mathbf{x}^2 L_g A_k c_\mathbf{z} \right) \leq \frac{L_g}{36L_1^2} \left( -(1 - \beta_\mathbf{x}^2) + \eta_\mathbf{x} \left( \frac{72L_1^2}{L_g} + L_g c_\mathbf{z} \right) \right) = 0, \tag{107}$$

where the equality uses

$$1 - \beta_\mathbf{x}^2 = \eta_\mathbf{x} \frac{72L_1^2}{L_g} + \eta_\mathbf{z} L_g. \tag{108}$$

**Analysis of the Coefficient for** $\mathbb{E}\left[ \left\| \mathbf{m}_\mathbf{y}^k - \mathbb{E}(\mathbf{D}_\mathbf{y}^k) \right\|^2 \right]$.

We analyze $T_4$ by rewriting it in a form that separates the contributions from the momentum parameter $\beta_\mathbf{y}$ and the step size $\eta_\mathbf{y}$:

$$T_4 = -\left(1 - \beta_\mathbf{y}^2\right) B_k + \eta_\mathbf{y} \left( \frac{12}{\lambda_g} P_k + 4B_k \beta_\mathbf{y}^2 L_g + 3B_k \beta_\mathbf{y}^2 L_g^2 \eta_\mathbf{y} \right)$$

$$+ \eta_\mathbf{y} \left( 12A_k \beta_\mathbf{x}^2 \left( 1 + \frac{1}{\eta_\mathbf{z} L_g} \right) L_1^2 \eta_\mathbf{y} + 12C_k \beta_\mathbf{z}^2 \left( 1 + \frac{1}{\eta_\mathbf{z} L_g} \right) L_1^2 \eta_\mathbf{y} \right). \tag{109}$$

Using the facts $\beta_{\mathbf{x}}^2, \beta_{\mathbf{z}}^2, \beta_{\mathbf{y}}^2 \leq 1$, $\eta_{\mathbf{x}}, \eta_{\mathbf{y}}, \eta_{\mathbf{z}} \leq \frac{1}{18L_g}$, $A_k = C_k = \frac{L_g}{36L_1^2}$, and $B_k = \frac{1}{36L_g}$ as in (96), we can bound the term $T_4$:

$$T_4 \leq \frac{1}{36L_g}\left(-\left(1-\beta_{\mathbf{y}}^2\right) + \eta_{\mathbf{y}}\left(\frac{432L_g}{\lambda_g}P_k + 6L_g + 24L_g\frac{c_{\mathbf{y}}}{c_{\mathbf{z}}}\right)\right) = 0. \tag{110}$$

Here, the last equality follows from

$$1 - \beta_{\mathbf{y}}^2 = \eta_{\mathbf{y}}\left(\frac{432L_g}{\lambda_g}P_k + 6L_g\right) + 24\eta_{\mathbf{z}}L_g. \tag{111}$$

**Analysis of the Coefficient for $\mathbb{E}\left[\left\|\mathbf{m}_{\mathbf{z}}^k - \mathbb{E}(\mathbf{D}_{\mathbf{z}}^k)\right\|^2\right]$.**

We analyze $T_5$ by rewriting it in a form that separates the contributions from the momentum parameter $\beta_{\mathbf{z}}$ and the step size $\eta_{\mathbf{z}}$:

$$T_5 = -\left(1-\beta_{\mathbf{z}}^2\right)C_k + \eta_{\mathbf{z}}\left(\frac{12}{\lambda_g}Q_k + 13C_k\beta_{\mathbf{z}}^2L_g + 12C_k\beta_{\mathbf{z}}^2L_g^2\eta_{\mathbf{z}} + A_k\beta_{\mathbf{x}}^2L_g^2\eta_{\mathbf{z}} + 12A_k\beta_{\mathbf{x}}^2L_g\right). \tag{112}$$

Using the facts $\beta_{\mathbf{x}}^2, \beta_{\mathbf{z}}^2 \leq 1$, $\eta_{\mathbf{z}} \leq \frac{1}{18L_g}$, and $A_k = C_k = \frac{L_g}{36L_1^2}$ as in (96), we can further bound the term $T_5$:

$$T_5 \leq \frac{L_g}{36L_1^2}\left(-\left(1-\beta_{\mathbf{z}}^2\right) + \eta_{\mathbf{z}}\left(\frac{432L_1^2}{\lambda_g L_g}Q_k + 14L_g + 12L_gc_{\mathbf{z}}\right)\right) = 0, \tag{113}$$

where the last equality uses

$$1 - \beta_{\mathbf{z}}^2 = \eta_{\mathbf{z}}\left(\frac{432L_1^2}{\lambda_g L_g}Q_k + 26L_g\right). \tag{114}$$

**Analysis of the Coefficient for $\mathbb{E}\left[\|\hat{\mathbf{m}}_{\mathbf{x}}^k\|^2\right]$.**

Under the assumption that

$$\eta_{\mathbf{x}} \leq \frac{1}{8L_{\nabla\Phi}}, \quad \eta_{\mathbf{y}} \leq \frac{1}{18L_g}, \quad \eta_{\mathbf{z}} \leq \frac{1}{18L_g}, \quad \eta_{\mathbf{x}} \leq \frac{\eta_{\mathbf{z}}}{\frac{16Q_kL_{\mathbf{z}^\star}^2}{\lambda_g} + \frac{32}{9}}, \quad \eta_{\mathbf{x}} \leq \frac{\eta_{\mathbf{y}}}{\frac{16P_kL_{\mathbf{y}^\star}^2}{\lambda_g} + \frac{4}{9}}, \tag{115}$$

and incorporating $A_k = C_k = \frac{L_g}{36L_1^2}$ and $B_k = \frac{1}{36L_g}$, we can bound the term $T_6$ in the following manner:

$$T_6 \leq -\frac{\eta_{\mathbf{x}}}{2} + \frac{\eta_{\mathbf{x}}}{8} + \frac{\eta_{\mathbf{x}}}{36} + Q_k\frac{2L_{\mathbf{z}^\star}^2\eta_{\mathbf{x}}^2}{\lambda_g\eta_{\mathbf{z}}} + P_k\frac{2L_{\mathbf{y}^\star}^2\eta_{\mathbf{x}}^2}{\lambda_g\eta_{\mathbf{y}}} + \frac{\eta_{\mathbf{x}}^2}{18\eta_{\mathbf{y}}} + \frac{4\eta_{\mathbf{x}}^2}{9\eta_{\mathbf{z}}} \leq 0. \tag{116}$$

Next, we analyze the **variance-related aggregation error** in Lemma 13, using the chosen momentum coefficients in (108), (111),and (114). We start by analyzing the relationship between the momentum coefficient and $\eta_{\mathbf{x}}$,

$$(1-\beta_{\mathbf{x}})^2 = \frac{(1-\beta_{\mathbf{x}}^2)^2}{(1+\beta_{\mathbf{x}})^2} \leq (1-\beta_{\mathbf{x}}^2)^2 \leq \left(\frac{144L_1^2}{L_g} + c_{\mathbf{z}}L_g\right)^2\eta_{\mathbf{x}}^2 = L_{\beta_{\mathbf{x}}}^2\eta_{\mathbf{x}}^2,$$

$$(1-\beta_{\mathbf{y}})^2 = \frac{(1-\beta_{\mathbf{y}}^2)^2}{(1+\beta_{\mathbf{y}})^2} \leq (1-\beta_{\mathbf{y}}^2)^2 \leq \left(\left(\frac{432(\mu_g+L_g)}{\mu_g}P_k + 6L_g\right)c_{\mathbf{y}} + 24c_{\mathbf{z}}L_g\right)^2\eta_{\mathbf{x}}^2 = L_{\beta_{\mathbf{y}}}^2\eta_{\mathbf{x}}^2,$$

$$(1-\beta_{\mathbf{z}})^2 = \frac{(1-\beta_{\mathbf{z}}^2)^2}{(1+\beta_{\mathbf{z}})^2} \leq (1-\beta_{\mathbf{z}}^2)^2 \leq \left(\frac{432(\mu_g+L_g)L_1^2}{\mu_gL_g^2}Q_k + 26L_g\right)^2c_{\mathbf{z}}^2\eta_{\mathbf{x}}^2 = L_{\beta_{\mathbf{z}}}^2\eta_{\mathbf{x}}^2. \tag{117}$$

Based on the above analysis, we proceed to examine the terms in Lemma 13. Observing that $\frac{1-\beta}{1+\beta} = \frac{1-\beta^2}{(1+\beta)^2} \le 1 - \beta^2$, combined with (117), we can derive an upper bound for the aggregation error associated with the parameters $\mathbf{x}$, $\mathbf{y}$, and $\mathbf{z}$ as follows.

For parameter $\mathbf{y}$, we have:

$$\mathbb{E}\left[\left\|\xi_{\mathbf{y}}^k\right\|^2\right] \le 3\kappa L_{\beta_{\mathbf{y}}}\eta_{\mathbf{x}}\left(1 + \frac{1}{(1-\delta)n}\right)\sigma^2 + 3\kappa\zeta_g^2. \tag{118}$$

For parameter $\mathbf{x}$, we derive:

$$\mathbb{E}\left[\left\|\xi_{\mathbf{x}}^k\right\|^2\right] \le 6\kappa L_{\beta_{\mathbf{x}}}\eta_{\mathbf{x}}\left(1 + \frac{1}{(1-\delta)n}\right)\sigma^2 + 6\kappa\zeta_g^2\rho^2 + 6\kappa\zeta_f^2. \tag{119}$$

Similarly, for parameter $\mathbf{z}$, we have:

$$\mathbb{E}\left[\left\|\xi_{\mathbf{z}}^k\right\|^2\right] \le 6\kappa L_{\beta_{\mathbf{z}}}\eta_{\mathbf{x}}\left(1 + \frac{1}{(1-\delta)n}\right)\sigma^2 + 6\kappa\zeta_g^2\rho^2 + 6\kappa\zeta_f^2. \tag{120}$$

These bounds highlight the influence of the momentum coefficients $\beta_{\mathbf{x}}$, $\beta_{\mathbf{y}}$, and $\beta_{\mathbf{z}}$ on the variance-related error terms. By appropriately selecting $\beta \to 1$, the impact of the variance terms $\sigma^2$ on the aggregation errors can be mitigated.

Combining the above inequality (103), (105), (106),(110), (112) and (116), we then drive the convergence result. For simplicity, we define:

$$A_0 := \frac{L_g\left(L_{\beta_{\mathbf{x}}}^2 + L_{\beta_{\mathbf{z}}}^2\right)}{36L_1^2}(1 + \rho^2) + \frac{L_{\beta_{\mathbf{y}}}^2}{36L_g}, \tag{121}$$

$$A_1 := \left(\frac{1}{8} + \frac{3c_{\mathbf{z}}}{4c_{\mathbf{y}}} + \frac{96L_2L_g^2}{\lambda_g^2 L_1} + \frac{24L_g^2}{\lambda_g^2}\right)c_{\mathbf{y}}, \tag{122}$$

$$A_2 := \left(\frac{3L_g^2}{4L_1^2} + \frac{96L_g^4}{\lambda_g^2 L_1^2}\right)c_{\mathbf{z}}, \tag{123}$$

$$A_3 := \left(6(1 + \rho^2)L_{\beta_{\mathbf{x}}} + 6A_2(1 + \rho^2)L_{\beta_{\mathbf{z}}} + 3A_1 L_{\beta_{\mathbf{y}}}\right)\left(1 + \frac{1}{(1-\delta)n}\right), \tag{124}$$

$$A_4 := 3A_1 + (6A_2 + 6)\rho^2, \tag{125}$$

and

$$A_5 := 6A_2 + 6. \tag{126}$$

Thus, substituting the above inequality into (97) and combining the term $A_0, A_1, A_2, A_3, A_4, A_5$, we can obtain

$$\mathcal{L}_{k+1} - \mathcal{L}_k \le -\frac{\eta_{\mathbf{x}}}{2}\mathbb{E}\left[\left\|\nabla\Phi_{\mathcal{H}}\left(\mathbf{x}^k\right)\right\|^2\right] + \frac{A_0\sigma^2\eta_{\mathbf{x}}^2}{(1-\delta)n} + \eta_{\mathbf{x}}^2\kappa A_3\sigma^2 + \kappa\left(A_4\zeta_g^2 + A_5\zeta_f^2\right)\eta_{\mathbf{x}}. \tag{127}$$

Rearranging the term in (127), we can obtain

$$\frac{\eta_{\mathbf{x}}}{2}\cdot\mathbb{E}\left[\left\|\nabla\Phi_{\mathcal{H}}\left(\mathbf{x}^k\right)\right\|^2\right] \le \mathcal{L}_k - \mathcal{L}_{k+1} + \frac{A_0\sigma^2\eta_{\mathbf{x}}^2}{(1-\delta)n} + \kappa A_3\sigma^2\eta_{\mathbf{x}}^2 + \kappa\left(A_4\zeta_g^2 + A_5\zeta_f^2\right)\eta_{\mathbf{x}}. \tag{128}$$

**Analyzing $\mathcal{L}_0 - \mathcal{L}_K$.** Recall the definition of Lyapunov function in (96), we can obtain

$$\begin{aligned}\mathcal{L}_0 - \mathcal{L}_K \le {}& \Phi_{\mathcal{H}}(\mathbf{x}^0) + P_k\left\|\mathbf{y}^0 - \mathbf{y}_\star^0\right\|^2 + Q_k\left\|\mathbf{z}^0 - \mathbf{z}_\star^0\right\|^2 \\ & + A_k\left\|\mathbf{m}_{\mathbf{x}}^0 - \mathbb{E}(\mathbf{D}_{\mathbf{x}}^0)\right\|^2 + B_k\left\|\mathbf{m}_{\mathbf{y}}^0 - \mathbb{E}(\mathbf{D}_{\mathbf{y}}^0)\right\|^2 + C_k\left\|\mathbf{m}_{\mathbf{z}}^0 - \mathbb{E}(\mathbf{D}_{\mathbf{z}}^0)\right\|^2.\end{aligned} \tag{129}$$

Using the definition of momentum in (91) and set the initial momentum $\mathbf{m_y^{-1}} = 0$, we can obtain

$$\mathbb{E}\left[\left\|\mathbf{m_y^0} - \mathbb{E}(\mathbf{D_y^0})\right\|^2\right] = \mathbb{E}\left[\left\|(1 - \beta_\mathbf{y})\nabla_\mathbf{y}G(\mathbf{x^0}, \mathbf{y^0}) - \mathbb{E}(\mathbf{D_y^0})\right\|^2\right]. \tag{130}$$

Under Assumption 3, we have

$$\mathbb{E}\left[\left\|\nabla_\mathbf{y}G(\mathbf{x^0}, \mathbf{y^0})\right\|^2\right] = \mathbb{E}(\mathbf{D_y^0}), \quad \mathbb{E}\left[\left\|\nabla_\mathbf{y}G(\mathbf{x^0}, \mathbf{y^0}) - \mathbb{E}(\mathbf{D_y^0})\right\|^2\right] \leq \sigma^2/(1 - \delta)n.$$

Thus, we can obtain

$$\mathbb{E}\left[\left\|\mathbf{m_y^0} - \mathbb{E}(\mathbf{D_y^0})\right\|^2\right] = (1 - \beta_\mathbf{y})^2 \frac{\sigma^2}{(1 - \delta)n} + \beta_\mathbf{y}^2\left\|\nabla_\mathbf{y}g(\mathbf{x^0}, \mathbf{y^0})\right\|^2, \tag{131}$$

using inequality (117) and $\beta_\mathbf{y}^2 < 1$, we can obtain

$$\mathbb{E}\left[\left\|\mathbf{m_y^0} - \mathbb{E}(\mathbf{D_y^0})\right\|^2\right] \leq L_{\beta_\mathbf{y}}^2\eta_\mathbf{x}^2\frac{\sigma^2}{(1 - \delta)n} + \left\|\nabla_\mathbf{y}g(\mathbf{x^0}, \mathbf{y^0})\right\|^2. \tag{132}$$

We can obtain similar results for $\mathbf{x}, \mathbf{z}$,

$$\mathbb{E}\left[\left\|\mathbf{m_x^0} - \mathbb{E}(\mathbf{D_x^0})\right\|^2\right] \leq L_{\beta_\mathbf{x}}^2\eta_\mathbf{x}^2\frac{\sigma_1^2}{(1 - \delta)n} + \left\|\mathbb{E}(\mathbf{D_x^0})\right\|^2, \tag{133}$$

and

$$\mathbb{E}\left[\left\|\mathbf{m_z^0} - \mathbb{E}(\mathbf{D_z^0})\right\|^2\right] \leq L_{\beta_\mathbf{z}}^2\eta_\mathbf{x}^2\frac{\sigma_1^2}{(1 - \delta)n} + \left\|\mathbb{E}(\mathbf{D_z^0})\right\|^2. \tag{134}$$

Combining the above inequalities, we can obtain

$$\mathcal{L}_0 - \mathcal{L}_K \leq \Delta_\Phi^0 + \eta_\mathbf{x}^2\frac{\sigma^2}{(1 - \delta)n}\left((1 + \rho^2)(A_kL_{\beta_\mathbf{x}}^2 + C_kL_{\beta_\mathbf{z}}^2) + B_kL_{\beta_\mathbf{y}}^2\right). \tag{135}$$

For simplicity, we define

$$\begin{aligned}\Delta_\Phi^0 = &\Phi_\mathcal{H}(\mathbf{x^0}) + P_k\left\|\mathbf{y^0} - \mathbf{y_\star^0}\right\|^2 + Q_k\left\|\mathbf{z^0} - \mathbf{z_\star^0}\right\|^2 \\ &+ A_k\left\|\mathbb{E}(\mathbf{D_x^0})\right\|^2 + B_k\left\|\mathbb{E}(\mathbf{D_y^0})\right\|^2 + C_k\left\|\mathbb{E}(\mathbf{D_z^0})\right\|^2.\end{aligned} \tag{136}$$

As the above inequality (128) is true for an arbitrary $k \in [K]$, by taking summation on both sides from $k = 0$ to $k = K - 1$, and defining $A_\sigma := \frac{A_0 + (1 + \rho^2)(A_kL_{\beta_\mathbf{x}}^2 + C_kL_{\beta_\mathbf{z}}^2) + B_kL_{\beta_\mathbf{y}}^2}{(1 - \delta)n}\sigma^2 + \kappa A_3\sigma^2$, we can obtain

$$\sum_{k=0}^{K-1}\mathbb{E}\left[\left\|\nabla\Phi_\mathcal{H}\left(\mathbf{x^k}\right)\right\|^2\right] \leq 2\left(\frac{\Delta_\Phi^0}{\eta_\mathbf{x}} + A_\sigma\eta_\mathbf{x}K\right) + 2\left(\kappa A_4\zeta_g^2K + \kappa A_5\zeta_f^2K\right). \tag{137}$$

Dividing both sides by $K$, we can obtain

$$\frac{1}{K}\sum_{k=0}^{K-1}\mathbb{E}\left[\left\|\nabla\Phi_\mathcal{H}\left(\mathbf{x^k}\right)\right\|^2\right] \leq 2\left(\frac{\Delta_\Phi^0}{\eta_\mathbf{x}K} + A_\sigma\eta_\mathbf{x}\right) + 2\left(\kappa A_4\zeta_g^2 + \kappa A_5\zeta_f^2\right). \tag{138}$$

**Final step.** Recall that the step size $\eta_\mathbf{x}$ satisfies:

$$\eta_\mathbf{x} \leq \min\left\{\alpha_1, \sqrt{\frac{\Delta_\Phi^0}{KA_\sigma}}\right\},$$

where

$$A_0' := A_0 + (1 + \rho^2)(A_kL_{\beta_\mathbf{x}}^2 + C_kL_{\beta_\mathbf{z}}^2) + B_kL_{\beta_\mathbf{y}}^2, \quad A_\sigma = A_0'\sigma^2 + \kappa A_3\sigma^2, \quad \alpha_1 = \frac{1}{8L_{\nabla\Phi}}. \tag{139}$$

This implies that

$$\frac{1}{\eta_{\mathbf{x}}} = \max\left\{8L_{\nabla\Phi}, \sqrt{\frac{KA_{\sigma}}{\Delta_{\Phi}^0}}\right\} \leq 8L_{\nabla\Phi} + \sqrt{\frac{KA_{\sigma}}{\Delta_{\Phi}^0}}. \tag{140}$$

Consequently, using this upper bound for $\frac{1}{\eta_{\mathbf{x}}}$ in (140), we can obtain

$$\frac{1}{K}\sum_{k=0}^{K-1}\mathbb{E}\left[\left\|\nabla\Phi_{\mathcal{H}}(\mathbf{x}^k)\right\|^2\right] \leq 2\left(\frac{8L_{\nabla\Phi}\Delta_{\Phi}^0}{K} + 2\sqrt{\frac{A_{\sigma}\Delta_{\Phi}^0}{K}}\right) \\ + 2\kappa\left(A_4\zeta_g^2 + A_5\zeta_f^2\right). \tag{141}$$

This simplifies to:

$$\frac{1}{K}\sum k = 0^{K-1}\mathbb{E}\left[\left\|\nabla\Phi_{\mathcal{H}}(\mathbf{x}^k)\right\|^2\right] = \mathcal{O}\left(\frac{\Delta_{\Phi}^0}{K} + \sqrt{\frac{A_{\sigma}\Delta_{\Phi}^0}{K}}\right) + \mathcal{O}\left(\kappa(\zeta_f^2 + \zeta_g^2 + \zeta_g^2\rho^2)\right). \tag{142}$$

Finally, as $\hat{\mathbf{x}}^{(i)} \sim \mathcal{U}\{\mathbf{x}_0^{(i)}, \ldots, \mathbf{x}_{K-1}^{(i)}\}$, combing with (142), we have

$$\mathbb{E}\left[\left\|\nabla\Phi_{\mathcal{H}}(\hat{\mathbf{x}})\right\|^2\right] = \mathcal{O}\left(\frac{\Delta_{\Phi}^0}{K} + \sqrt{\frac{A_{\sigma}\Delta_{\Phi}^0}{K}}\right) + \mathcal{O}\left(\kappa(\zeta_f^2 + \zeta_g^2 + \zeta_g^2\rho^2)\right). \tag{143}$$

Hence, the proof is complete. $\qquad\square$

# I CONVERGENCE ANALYSIS FOR BR-FEDBIP (THEOREM 4)

## I.1 PROOF OUTLINE

Our convergence analysis for BR-FedBiP in Algorithm 1 is similar to BR-FedBiM and BR-FedBi, the only additional lemma that we have to prove is the deviation of PAGE (Lemma 21).

**Notation.** Recall that for each iteration $k$, each malicious client send well-craft malicious gradient, and each honest client $i$ computes:

$$\mathbf{v}_{\mathbf{x},i}^k = \boldsymbol{\pi}_k(\nabla_{xy}^2 G_i z; \boldsymbol{\varphi}_i^{k'}, \boldsymbol{\varphi}_i^k) + \boldsymbol{\pi}_k(\nabla_{\mathbf{x}} F_i; \boldsymbol{\phi}_i^{k'}, \boldsymbol{\phi}_i^k),$$
$$\mathbf{v}_{\mathbf{y},i}^k = \boldsymbol{\pi}_k(\nabla_{\mathbf{y}} G_i; \boldsymbol{\varphi}_i^{k'}, \boldsymbol{\varphi}_i^k),$$
$$\mathbf{v}_{\mathbf{y},i}^k = \boldsymbol{\pi}_k(\nabla_{yy}^2 G_i z; \boldsymbol{\varphi}_i^{k'}, \boldsymbol{\varphi}_i^k) + \boldsymbol{\pi}_k(\nabla_{\mathbf{y}} F_i; \boldsymbol{\phi}_i^{k'}, \boldsymbol{\phi}_i^k).$$

where $\mathbf{u}^k = (\mathbf{x}^k, \mathbf{y}^k, \mathbf{y}^k)$ and we define the operation $\boldsymbol{\pi}_k$ as:

$$\boldsymbol{\pi}_k(\boldsymbol{\psi}; b', b) = \begin{cases} \boldsymbol{\psi}(\mathbf{u}^k; b'), & \text{if } c_k = 1, \\ \boldsymbol{\pi}_{k-1} + \boldsymbol{\psi}(\mathbf{u}^k; b) - \boldsymbol{\psi}(\mathbf{u}^k; b), & \text{if } c_k = 0. \end{cases}$$

The server receives the gradient from the clients and performs robust aggregation methods $AGG(\cdot)$:

$$\hat{\mathbf{v}}_{\mathbf{x}}^k := AGG(\mathbf{v}_{\mathbf{x},1}^k, ..., \mathbf{v}_{\mathbf{x},n}^k),$$
$$\hat{\mathbf{v}}_{\mathbf{y}}^k := AGG(\mathbf{v}_{\mathbf{y},1}^k, ..., \mathbf{v}_{\mathbf{y},n}^k),$$
$$\hat{\mathbf{v}}_{\mathbf{z}}^k := AGG(\mathbf{v}_{\mathbf{y},1}^k, ..., \mathbf{v}_{\mathbf{y},n}^k).$$

## I.2 AGGREGATION ERROR

We define the aggregation error as the deviation between the robustly aggregated gradients and the average gradients of honest clients. Specifically, the aggregation error is expressed as:

$$\xi_{\mathbf{x}}^k := \hat{\mathbf{v}}_{\mathbf{x}}^k - \mathbf{v}_{\mathbf{x}}^k,$$
$$\xi_{\mathbf{y}}^k := \hat{\mathbf{v}}_{\mathbf{y}}^k - \mathbf{v}_{\mathbf{y}}^k,$$
$$\xi_{\mathbf{z}}^k := \hat{\mathbf{v}}_{\mathbf{z}}^k - \mathbf{v}_{\mathbf{z}}^k,$$

where $\mathbf{v}_{\mathbf{x}}^k = \frac{1}{|\mathcal{H}|} \sum_{i \in \mathcal{H}} \mathbf{v}_{\mathbf{x},i}^k$, $\mathbf{v}_{\mathbf{y}}^k = \frac{1}{|\mathcal{H}|} \sum_{i \in \mathcal{H}} \mathbf{v}_{\mathbf{y},i}^k$ and $\mathbf{v}_{\mathbf{z}}^k = \frac{1}{|\mathcal{H}|} \sum_{i \in \mathcal{H}} \mathbf{v}_{\mathbf{y},i}^k$.

The detailed proof follows a similar approach as that provided in Appendix J.3.1.

---

**Lemma 18.** *Suppose Assumptions 3 and 4 hold, and the robust aggregation rule $AGG(\cdot)$ satisfies the $(\delta, \kappa)$-robustness criterion. The aggregation errors in the BR-FedBi algorithm (Algorithm 1) satisfy the following inequalities:*

$$(1) \quad \mathbb{E}\left[\left\|\xi_{\mathbf{y}}^k\right\|^2\right] \leq 4\kappa\zeta_g^2 + 4\kappa\left(1 + \frac{1}{(1-\delta)n}\right)\sigma^2,$$

$$(2) \quad \mathbb{E}\left[\left\|\xi_{\mathbf{x}}^k\right\|^2\right] \leq 8\kappa\left(\rho^2\zeta_g^2 + \zeta_f^2\right) + 8\kappa\left(1 + \frac{1}{(1-\delta)n}\right)\sigma_1^2,$$

$$(3) \quad \mathbb{E}\left[\left\|\xi_{\mathbf{z}}^k\right\|^2\right] \leq 8\kappa\left(\rho^2\zeta_g^2 + \zeta_f^2\right) + 8\kappa\left(1 + \frac{1}{(1-\delta)n}\right)\sigma_1^2,$$

*where $\sigma_1^2 = (1 + \rho^2)\sigma^2$ and $\rho = \frac{C_f}{\mu_g}$.*

---

## I.3 LOWER-LEVEL CONVERGENCE

In this subsection, we analyze the lower-level convergence behavior of our bilevel algorithm. Specifically, we examine the distances between $\mathbf{y}^k$ and $\mathbf{y}_\star^k$, as well as $\mathbf{y}^k$ and $\mathbf{z}_\star^k$. The proof of this lemma is similar to Lemma 14.

**Lemma 19.** *Under Assumptions 1, 2, 3, and 4, and assuming $\eta_{\mathbf{y}} \leq \min\left\{\lambda_g, \frac{1}{\mu_g + L_g}\right\}$ and $\eta_{\mathbf{z}} \leq \min\left\{\lambda_g, \frac{1}{\mu_g + L_g}\right\}$, the following holds:*

$$(1) \quad \mathbb{E}\left[\|\mathbf{y}^{k+1} - \mathbf{y}_\star^{k+1}\|\right] \leq \left(1 - \frac{\mu_g L_g \eta_{\mathbf{y}}}{2(\mu_g + L_g)}\right)\mathbb{E}\left[\|\mathbf{y}^k - \mathbf{y}_\star^k\|^2\right] - \frac{\eta_{\mathbf{y}}}{\mu_g L_g}\mathbb{E}\left[\|\mathbb{E}(\mathbf{D}_{\mathbf{y}}^k)\|^2\right]$$
$$+ \frac{12(\mu_g + L_g)\eta_{\mathbf{y}}}{\mu_g L_g}\left(\mathbb{E}\left[\|\mathbf{v}_{\mathbf{y}}^k - \mathbb{E}(\mathbf{D}_{\mathbf{y}}^k)\|^2\right] + \mathbb{E}\left[\|\xi_{\mathbf{y}}^k\|^2\right]\right)$$
$$+ \frac{2(\mu_g + L_g)L_{\mathbf{y}^\star}^2 \eta_{\mathbf{x}}^2}{\mu_g L_g \eta_{\mathbf{y}}}\mathbb{E}\left[\|\hat{\mathbf{v}}_{\mathbf{x}}^k\|^2\right].$$

$$(2) \quad \mathbb{E}\left[\|\mathbf{z}^{k+1} - \mathbf{z}_\star^{k+1}\|\right] \leq \left(1 - \frac{\mu_g L_g \eta_{\mathbf{z}}}{2(\mu_g + L_g)}\right)\mathbb{E}\left[\|\mathbf{z}^k - \mathbf{z}_\star^k\|^2\right] - \frac{\eta_{\mathbf{z}}}{\mu_g L_g}\mathbb{E}\left[\|\mathbb{E}(\mathbf{D}_{\mathbf{z}}^k)\|^2\right]$$
$$+ \frac{12(\mu_g + L_g)\eta_{\mathbf{z}}}{\mu_g L_g}\left(\mathbb{E}\left[\|\mathbf{v}_{\mathbf{z}}^k - \mathbb{E}(\mathbf{D}_{\mathbf{z}}^k)\|^2\right] + \mathbb{E}\left[\|\xi_{\mathbf{z}}^k\|^2\right]\right)$$
$$+ \frac{2(\mu_g + L_g)L_{\mathbf{y}^\star}^2 \eta_{\mathbf{x}}^2}{\mu_g L_g \eta_{\mathbf{z}}}\mathbb{E}\left[\|\hat{\mathbf{v}}_{\mathbf{x}}^k\|^2\right].$$

### I.4 GRADIENT UPDATE BIAS

Next, we analyze the gradient update bias of BR-FedBiP in Algorithm 1. The proof details of this Lemma are similar to J.2.3.

**Lemma 20.** *Suppose Assumption 1, 2 and 3 hold, if $\rho \geq C_f/\mu_g$, we have:*

$$\mathbb{E}\left[\|\nabla \Phi_{\mathcal{H}}\left(\mathbf{x}^k\right) - \mathbf{v}_{\mathbf{x}}^k\|^2\right] \leq 6L_2^2 \mathbb{E}\left[\|\mathbf{y}^k - \mathbf{y}_\star^k\|^2\right] + 6L_g^2 \mathbb{E}\left[\|\mathbf{y}^k - \mathbf{y}_\star^k\|^2\right]$$
$$+ 2\mathbb{E}\left[\|\mathbf{v}_{\mathbf{x}}^k - \mathbb{E}\left(\mathbf{D}_{\mathbf{x}}^k\right)\|^2\right],$$

*where $L_2^2 = L_f^2 + L_g^2 \rho^2$.*

### I.5 PAGE DEVIATION

Next, we study the deviation, which is the distance between the average PAGE gradient and the true gradient. Proof of this Lemma can be found in J.3.2.

**Lemma 21.** *Suppose Assumption 2, 3 and 7 hold. Considering Algorithm BR-FedBiP in Algorithm 1, we have*

$$(1) \quad \mathbb{E}\left[\|\mathbf{v}_{\mathbf{y}}^{k+1} - \mathbb{E}\left(\mathbf{D}_{\mathbf{y}}^{k+1}\right)\|^2\right] \leq p\frac{\sigma^2}{b'(1-\delta)n} + (1-p)\mathbb{E}\left[\|\mathbf{v}_{\mathbf{y}}^k - \mathbb{E}(\mathbf{D}_{\mathbf{y}}^k)\|^2\right]$$
$$+ \frac{(1-p)L_g^2}{b}\left(\eta_{\mathbf{x}}^2 \mathbb{E}\left[\|\hat{\mathbf{v}}_{\mathbf{x}}^k\|^2\right] + 3\eta_{\mathbf{y}}^2 \mathbb{E}\left[\|\mathbf{v}_{\mathbf{y}}^k - \mathbb{E}\left(\mathbf{D}_{\mathbf{y}}^k\right)\|^2\right]\right)$$
$$+ \frac{(1-p)L_g^2}{b}\left(3L_g^2 \eta_{\mathbf{y}}^2 \mathbb{E}\left[\|\mathbf{y}^k - \mathbf{y}_\star^k\|^2\right] + 3\eta_{\mathbf{y}}^2 \mathbb{E}\left[\|\xi_{\mathbf{y}}^k\|^2\right]\right),$$

(2) $\mathbb{E}\left[\|\mathbf{v}_{\mathbf{x}}^{k+1} - \mathbb{E}\left(\mathbf{D}_{\mathbf{x}}^{k+1}\right)\|^2\right]$

$$\leq p\frac{\sigma_1^2}{b'(1-\delta)n} + (1-p)\mathbb{E}\left[\|\mathbf{v}_{\mathbf{x}}^k - \mathbb{E}(\mathbf{D}_{\mathbf{x}}^k)\|^2\right] + 4\frac{(1-p)}{b}L_1^2\eta_{\mathbf{x}}^2\mathbb{E}\left[\|\hat{\mathbf{v}}_{\mathbf{x}}^k\|^2\right]$$

$$+ 12\frac{(1-p)}{b}L_1^2\eta_{\mathbf{y}}^2\left(\mathbb{E}\left[\|\xi_{\mathbf{y}}^k\|^2\right] + \mathbb{E}\left[\|\mathbf{v}_{\mathbf{y}}^k - \mathbb{E}\left(\mathbf{D}_{\mathbf{y}}^k\right)\|^2\right] + L_g^2\mathbb{E}\left[\|\mathbf{y}^k - \mathbf{y}_\star^k\|^2\right]\right)$$

$$+ \frac{12(1-p)}{b}L_g^2\eta_{\mathbf{z}}^2\left(\mathbb{E}\left[\|\xi_{\mathbf{z}}^k\|^2\right] + \mathbb{E}\left[\|\mathbf{v}_{\mathbf{z}}^k - \mathbb{E}\left(\mathbf{D}_{\mathbf{z}}^k\right)\|^2\right]\right)$$

$$+ \frac{12(1-p)}{b}L_g^2\eta_{\mathbf{z}}^2\left(3L_g^2\mathbb{E}\left[\|\mathbf{z}^k - \mathbf{z}_\star^k\|^2\right] + 3L_2^2\mathbb{E}\left[\|\mathbf{y}^k - \mathbf{y}_\star^k\|^2\right]\right),$$

(3) $\mathbb{E}\left[\|\mathbf{v}_{\mathbf{z}}^{k+1} - \mathbb{E}\left(\mathbf{D}_{\mathbf{z}}^{k+1}\right)\|^2\right]$

$$\leq p\frac{\sigma_1^2}{b'(1-\delta)n} + (1-p)\mathbb{E}\left[\|\mathbf{v}_{\mathbf{z}}^k - \mathbb{E}(\mathbf{D}_{\mathbf{z}}^k)\|^2\right] + 4\frac{(1-p)}{b}L_1^2\eta_{\mathbf{x}}^2\mathbb{E}\left[\|\hat{\mathbf{v}}_{\mathbf{x}}^k\|^2\right]$$

$$+ 12\frac{(1-p)}{b}L_1^2\eta_{\mathbf{y}}^2\left(\mathbb{E}\left[\|\xi_{\mathbf{y}}^k\|^2\right] + \mathbb{E}\left[\|\mathbf{v}_{\mathbf{y}}^k - \mathbb{E}\left(\mathbf{D}_{\mathbf{y}}^k\right)\|^2\right] + L_g^2\mathbb{E}\left[\|\mathbf{y}^k - \mathbf{y}_\star^k\|^2\right]\right)$$

$$+ \frac{12(1-p)}{b}L_g^2\eta_{\mathbf{z}}^2\left(\mathbb{E}\left[\|\xi_{\mathbf{z}}^k\|^2\right] + \mathbb{E}\left[\|\mathbf{v}_{\mathbf{z}}^k - \mathbb{E}\left(\mathbf{D}_{\mathbf{z}}^k\right)\|^2\right] + 3L_g^2\mathbb{E}\left[\|\mathbf{z}^k - \mathbf{z}_\star^k\|^2\right]\right)$$

$$+ \frac{12(1-p)}{b}L_g^2\eta_{\mathbf{z}}^2\left(3L_2^2\mathbb{E}\left[\|\mathbf{y}^k - \mathbf{y}_\star^k\|^2\right]\right).$$

## I.6 DESCENT LEMMA

In this section, we analyze the progression of the upper-level loss function in BR-FedBiP as described in Algorithm 1. The proof this Lemma is similar to Lemma 17.

---

**Lemma 22.** *Assuming that the upper-level loss function $\Phi_{\mathcal{H}}(\cdot)$ is $L_{\nabla\Phi}$-smooth. For each step $k \in [K]$, we have:*

$$\mathbb{E}\left[\Phi_{\mathcal{H}}\left(\mathbf{x}^{k+1}\right)\right] \leq \mathbb{E}\left[\Phi_{\mathcal{H}}\left(\mathbf{x}^k\right)\right] - \frac{\eta_{\mathbf{x}}}{2}\mathbb{E}\left[\|\nabla\Phi_{\mathcal{H}}\left(\mathbf{x}^k\right)\|^2\right] - \left(\frac{\eta_{\mathbf{x}}}{2} - \frac{L_{\nabla\Phi}\eta_{\mathbf{x}}^2}{2}\right)\mathbb{E}\left[\|\hat{\mathbf{v}}_{\mathbf{x}}^k\|^2\right]$$
$$+ \eta_{\mathbf{x}}\mathbb{E}\left[\|\nabla\Phi_{\mathcal{H}}\left(\mathbf{x}^k\right) - \mathbf{v}_{\mathbf{x}}^k\|^2\right] + \eta_{\mathbf{x}}\mathbb{E}\left[\|\xi_{\mathbf{x}}^k\|^2\right].$$

---

## I.7 PROOF OF THEOREM 4

**Theorem 7.** *Suppose that Assumptions 1-4 and 7 hold, and that the robust aggregation rule $AGG(\cdot)$ satisfies the $(\delta, \kappa)$-robustness criterion. There exist positive constants $c$, $c_{\mathbf{y}}$, and $c_{\mathbf{z}}$ such that if*

$$\eta_{\mathbf{y}} = c_{\mathbf{y}}\eta_{\mathbf{x}}, \quad \eta_{\mathbf{z}} = c_{\mathbf{z}}\eta_{\mathbf{x}}$$

*and step size $\eta_{\mathbf{x}}$ satisfies,*

$$\eta_{\mathbf{x}} \leq \min\left\{\frac{1}{8L_{\nabla\Phi}}, \frac{c}{1 + \sqrt{\frac{1-p}{bp}}}\right\},$$

*the algorithm converges as :*

$$\mathbb{E}\left[\|\nabla\Phi_{\mathcal{H}}(\hat{\mathbf{x}})\|^2\right] = \mathcal{O}\left(\frac{1 + \sqrt{\frac{1-p}{pb}}}{K} + \frac{1}{Kpb'} + \frac{\sigma_0^2}{b'(1-\delta)n}\right) \tag{144}$$
$$+ \mathcal{O}\left(\kappa\left(\zeta_g^2 + \zeta_g^2\rho^2 + \zeta_f^2 + \sigma^2\right)\right),$$

*where the expectation is over the randomness of the algorithm.*

*Proof.* We consider the Lyapunov function

$$
\begin{aligned}
\mathcal{L}_k = {}& \frac{\Phi_{\mathcal{H}}(\mathbf{x}^k)}{2} + \mathbb{E}\left[\left\|\mathbf{y}^k - \mathbf{y}_\star^k\right\|^2\right] + \mathbb{E}\left[\left\|\mathbf{z}^k - \mathbf{z}_\star^k\right\|^2\right] + \frac{\eta_{\mathbf{x}}}{p}\mathbb{E}\left[\left\|\mathbf{v}_{\mathbf{x}}^k - \mathbb{E}(\mathbf{D}_{\mathbf{x}}^k)\right\|^2\right] \\
& + \frac{\eta_{\mathbf{x}}}{p}\mathbb{E}\left[\left\|\mathbf{v}_{\mathbf{y}}^k - \mathbb{E}(\mathbf{D}_{\mathbf{y}}^k)\right\|^2\right] + \frac{\eta_{\mathbf{x}}}{p}\mathbb{E}\left[\left\|\mathbf{v}_{\mathbf{z}}^k - \mathbb{E}(\mathbf{D}_{\mathbf{z}}^k)\right\|^2\right].
\end{aligned}
\tag{145}
$$

Using above equation (145), and Lemmas 19, 20, 21 and 22, we can obtain that

$$
\begin{aligned}
& \mathcal{L}_{k+1} - \mathcal{L}_k \\
& \leq -\frac{\eta_{\mathbf{x}}}{4}\mathbb{E}\left[\left\|\nabla\Phi_{\mathcal{H}}\left(\mathbf{x}^k\right)\right\|^2\right] \\
& + \underbrace{\left(3L_2^2\eta_{\mathbf{x}} - \frac{\lambda_g\eta_{\mathbf{y}}}{2} + \frac{3(1-p)L_g^4\eta_{\mathbf{y}}^2\eta_{\mathbf{x}}}{bp} + \frac{24(1-p)L_1^2L_g^2\eta_{\mathbf{y}}^2\eta_{\mathbf{x}}}{bp} + \frac{72(1-p)L_g^2L_2^2\eta_{\mathbf{z}}^2\eta_{\mathbf{x}}}{bp}\right)}_{T_1}\mathbb{E}\left[\left\|\mathbf{y}^k - \mathbf{y}_\star^k\right\|^2\right] \\
& + \underbrace{\left(3L_g^2\eta_{\mathbf{x}} - \frac{\lambda_g\eta_{\mathbf{z}}}{2} + \frac{72(1-p)L_g^4\eta_{\mathbf{z}}^2\eta_{\mathbf{x}}}{bp}\right)}_{T_2}\mathbb{E}\left[\left\|\mathbf{z}^k - \mathbf{z}_\star^k\right\|^2\right] \\
& + \left(-\eta_{\mathbf{x}} + \eta_{\mathbf{x}}\right)\mathbb{E}\left[\left\|\mathbf{v}_{\mathbf{x}}^k - \mathbb{E}(\mathbf{D}_{\mathbf{x}}^k)\right\|^2\right] \\
& + \underbrace{\left(-\eta_{\mathbf{x}} + \frac{12\eta_{\mathbf{y}}}{\lambda_g} + \frac{3(1-p)L_g^2\eta_{\mathbf{y}}^2\eta_{\mathbf{x}}}{bp} + \frac{24(1-p)L_1^2\eta_{\mathbf{y}}^2}{bp}\eta_{\mathbf{x}}\right)}_{T_3}\mathbb{E}\left[\left\|\mathbf{v}_{\mathbf{y}}^k - \mathbb{E}(\mathbf{D}_{\mathbf{y}}^k)\right\|^2\right] \\
& + \underbrace{\left(-\eta_{\mathbf{x}} + \frac{12\eta_{\mathbf{z}}}{\lambda_g} + \frac{24(1-p)L_g^2\eta_{\mathbf{z}}^2}{bp}\eta_{\mathbf{x}}\right)}_{T_4}\mathbb{E}\left[\left\|\mathbf{v}_{\mathbf{z}}^k - \mathbb{E}(\mathbf{D}_{\mathbf{z}}^k)\right\|^2\right] \\
& + \underbrace{\left(-\frac{\eta_{\mathbf{x}}}{4} + \frac{L_{\nabla\Phi}\eta_{\mathbf{x}}^2}{4} + \frac{2L_{\mathbf{y}^\star}^2\eta_{\mathbf{x}}^2}{\lambda_g\eta_{\mathbf{y}}} + \frac{2L_{\mathbf{y}^\star}^2\eta_{\mathbf{x}}^2}{\lambda_g\eta_{\mathbf{z}}} + \frac{(1-p)L_g^2\eta_{\mathbf{x}}^3}{bp} + \frac{8(1-p)L_1^2\eta_{\mathbf{x}}^3}{bp}\right)}_{T_5}\mathbb{E}\left[\left\|\hat{\mathbf{v}}_{\mathbf{x}}^k\right\|^2\right] \\
& + \eta_{\mathbf{x}}\left(\frac{12c_{\mathbf{y}}}{\lambda_g} + \frac{3(1-p)L_g^2c_{\mathbf{y}}^2}{bp}\eta_x + \frac{24(1-p)L_1^2c_{\mathbf{y}}^2}{bp}\eta_x\right)\mathbb{E}\left[\left\|\xi_{\mathbf{y}}^k\right\|^2\right] \\
& + \eta_{\mathbf{x}}\left(\frac{12c_{\mathbf{z}}}{\lambda_g} + \frac{12(1-p)L_g^2c_{\mathbf{z}}^2}{bp}\eta_x\right)\mathbb{E}\left[\left\|\xi_{\mathbf{z}}^k\right\|^2\right] + \frac{\eta_{\mathbf{x}}}{2}\mathbb{E}\left[\left\|\xi_{\mathbf{x}}^k\right\|^2\right] \\
& + \eta_{\mathbf{x}}\frac{\sigma^2}{b'(1-\delta)n} + 2\eta_{\mathbf{x}}\frac{\sigma_1^2}{b'(1-\delta)n}.
\end{aligned}
\tag{146}
$$

Under the assumptions that

$$
\eta_y \leq \frac{4bp}{(1-p)(L_g^2 + 8L_1^2)\lambda_g}, \quad \eta_z \leq \frac{bp}{(1-p)L_g^2\lambda_g},
\tag{147}
$$

and invoking Lemma 18 into 146 ,we can implies

$$
\begin{aligned}
& \mathcal{L}_{k+1} - \mathcal{L}_k \\
& \leq -\frac{\eta_{\mathbf{x}}}{4}\mathbb{E}\left[\left\|\nabla\Phi_{\mathcal{H}}\left(\mathbf{x}^k\right)\right\|^2\right] \\
& + \underbrace{\left(3L_2^2\eta_{\mathbf{x}} - \frac{\lambda_g\eta_{\mathbf{y}}}{2} + \frac{3(1-p)L_g^4\eta_{\mathbf{y}}^2\eta_{\mathbf{x}}}{bp} + \frac{24(1-p)L_1^2L_g^2\eta_{\mathbf{y}}^2\eta_{\mathbf{x}}}{bp} + \frac{72(1-p)L_g^2L_2^2\eta_{\mathbf{z}}^2\eta_{\mathbf{x}}}{bp}\right)}_{T_1}\mathbb{E}\left[\left\|\mathbf{y}^k - \mathbf{y}_\star^k\right\|^2\right]
\end{aligned}
$$

$$+ \underbrace{\left(3L_g^2\eta_{\mathbf{x}} - \frac{\lambda_g\eta_{\mathbf{z}}}{2} + \frac{72(1-p)L_g^4\eta_{\mathbf{z}}^2\eta_{\mathbf{x}}}{bp}\right)}_{T_2} \mathbb{E}\left[\left\|\mathbf{z}^k - \mathbf{z}_\star^k\right\|^2\right]$$

$$+ \left(-\eta_{\mathbf{x}} + \eta_{\mathbf{x}}\right)\mathbb{E}\left[\left\|\mathbf{v}_{\mathbf{x}}^k - \mathbb{E}(\mathbf{D}_{\mathbf{x}}^k)\right\|^2\right]$$

$$+ \underbrace{\left(-\eta_{\mathbf{x}} + \frac{12\eta_{\mathbf{y}}}{\lambda_g} + \frac{3(1-p)L_g^2\eta_{\mathbf{y}}^2\eta_{\mathbf{x}}}{bp} + \frac{24(1-p)L_1^2\eta_{\mathbf{y}}^2}{bp}\eta_{\mathbf{x}}\right)}_{T_3} \mathbb{E}\left[\left\|\mathbf{v}_{\mathbf{y}}^k - \mathbb{E}(\mathbf{D}_{\mathbf{y}}^k)\right\|^2\right]$$

$$+ \underbrace{\left(-\eta_{\mathbf{x}} + \frac{12\eta_{\mathbf{z}}}{\lambda_g} + \frac{24(1-p)L_g^2\eta_{\mathbf{z}}^2}{bp}\eta_{\mathbf{x}}\right)}_{T_4} \mathbb{E}\left[\left\|\mathbf{v}_{\mathbf{z}}^k - \mathbb{E}(\mathbf{D}_{\mathbf{z}}^k)\right\|^2\right]$$

$$+ \underbrace{\left(-\frac{\eta_{\mathbf{x}}}{4} + \frac{L_{\nabla\Phi}\eta_{\mathbf{x}}^2}{4} + \frac{2L_{\mathbf{y}^\star}^2\eta_{\mathbf{x}}^2}{\lambda_g\eta_{\mathbf{y}}} + \frac{2L_{\mathbf{y}^\star}^2\eta_{\mathbf{x}}^2}{\lambda_g\eta_{\mathbf{z}}} + \frac{(1-p)L_g^2\eta_{\mathbf{x}}^3}{bp} + \frac{8(1-p)L_1^2\eta_{\mathbf{x}}^3}{bp}\right)}_{T_5} \mathbb{E}\left[\left\|\hat{\mathbf{v}}_{\mathbf{x}}^k\right\|^2\right]$$

$$+ \left(\frac{96c_y}{\lambda_g} + \left(\frac{192c_z}{\lambda_g} + 4\right)\rho^2\right)\zeta_g^2\kappa\eta_x + \left(\frac{192c_z}{\lambda_g} + 4\right)\zeta_f^2\kappa\eta_x$$

$$+ \left(\left(\frac{192c_z}{\lambda_g} + 4\right)(1+\rho^2) + \frac{96c_y}{\lambda_g}\right)\left(1 + \frac{1}{(1-\delta)n}\right)\sigma^2\kappa\eta_x$$

$$+ \eta_{\mathbf{x}}\frac{\sigma^2}{b'(1-\delta)n} + 2\eta_{\mathbf{x}}\frac{\sigma_1^2}{b'(1-\delta)n}. \tag{148}$$

We now analyze each term on the R.H.S. to further bound them.

**Analysis of the Coefficient for** $\mathbb{E}\left[\left\|\mathbf{y}^k - \mathbf{y}_\star^k\right\|^2\right]$**.**

Due to the assumption that

$$\eta_{\mathbf{x}} \leq \frac{\lambda_g\eta_{\mathbf{y}}}{12L_2^2}, \qquad c^2 \leq \frac{\lambda_g c_{\mathbf{y}}}{4\Delta_1^2}, \qquad \Delta_1^2 = \left(3L_g^4c_{\mathbf{y}}^2 + 24L_1^2 + L_g^2c_{\mathbf{y}}^2 + 72L_g^2L_2^2c_{\mathbf{z}}^2\right), \tag{149}$$

we can bound the term $T_1$ in the following manner

$$\begin{aligned}
T_1 &\leq 3L_2^2\eta_{\mathbf{x}} - \frac{\lambda_g\eta_{\mathbf{y}}}{2} + \frac{3(1-p)L_g^4\eta_{\mathbf{y}}^2\eta_{\mathbf{x}}}{bp} + \frac{24(1-p)L_1^2L_g^2\eta_{\mathbf{y}}^2\eta_{\mathbf{x}}}{bp} + \frac{72(1-p)L_g^2L_2^2\eta_{\mathbf{z}}^2\eta_{\mathbf{x}}}{bp} \\
&\leq 3L_2^2\eta_{\mathbf{x}} - \frac{\lambda_g\eta_{\mathbf{y}}}{2} + \frac{(1-p)}{bp}\eta_{\mathbf{x}}^3\left(3L_g^4c_{\mathbf{y}}^2 + 24L_1^2 + L_g^2c_{\mathbf{y}}^2 + 72L_g^2L_2^2c_{\mathbf{z}}^2\right) \\
&\leq 3L_2^2\eta_{\mathbf{x}} - \frac{\lambda_g\eta_{\mathbf{y}}}{2} + \frac{\lambda_g\eta_{\mathbf{y}}}{4} \leq 0.
\end{aligned} \tag{150}$$

**Analysis of the Coefficient for** $\mathbb{E}\left[\left\|\mathbf{z}^k - \mathbf{z}_\star^k\right\|^2\right]$**.**

Due to the assumption that

$$\eta_{\mathbf{x}} \leq \frac{\lambda_g\eta_{\mathbf{z}}}{12L_g^2}, \qquad c^2 \leq \frac{\lambda_g}{288L_g^4c_{\mathbf{z}}}, \tag{151}$$

we can bound the term $T_2$ in the following manner

$$T_2 = 3L_g^2\eta_{\mathbf{x}} - \frac{\lambda_g\eta_{\mathbf{z}}}{2} + \frac{72(1-p)L_g^4\eta_{\mathbf{z}}^2\eta_{\mathbf{x}}}{bp} \leq 3L_g^2\eta_{\mathbf{x}} - \frac{\lambda_g\eta_{\mathbf{z}}}{2} + \frac{\lambda_g\eta_{\mathbf{z}}}{4} \leq 0. \tag{152}$$

**Analysis of the Coefficient for** $\mathbb{E}\left[\left\|\mathbf{v}_{\mathbf{y}}^k - \mathbb{E}(\mathbf{D}_{\mathbf{y}}^k)\right\|^2\right]$**.**

Due to the assumption that

$$\eta_{\mathbf{y}} \leq \frac{\mu_g L_g \eta_{\mathbf{x}}}{24(\mu_g + L_g)}, \qquad c^2 \leq \frac{1}{2\left(3L_g^2 c_{\mathbf{y}}^2 + 24L_1^2 c_{\mathbf{y}}^2\right)}, \tag{153}$$

we can bound the term $T_3$ in the following manner

$$\begin{aligned}
T_3 &= -\eta_{\mathbf{x}} + \frac{12\eta_{\mathbf{y}}}{\lambda_g} + \frac{3(1-p)L_g^2 \eta_{\mathbf{y}}^2 \eta_{\mathbf{x}}}{bp} + \frac{24(1-p)L_1^2 \eta_{\mathbf{y}}^2 \eta_{\mathbf{x}}}{bp} \\
&\leq -\eta_{\mathbf{x}} + \frac{12\eta_{\mathbf{y}}}{\lambda_g} + \frac{(1-p)}{bp}\eta_{\mathbf{x}}^3 \left(3L_g^2 c_{\mathbf{y}}^2 + 24L_1^2 c_{\mathbf{y}}^2\right) \leq 0.
\end{aligned} \tag{154}$$

**Analysis of the Coefficient for** $\mathbb{E}\left[\left\|\mathbf{v}_{\mathbf{z}}^k - \mathbb{E}(\mathbf{D}_{\mathbf{z}}^k)\right\|^2\right]$**.**

Due to the assumption that

$$\eta_{\mathbf{z}} \leq \frac{\lambda_g \eta_{\mathbf{x}}}{24}, \qquad c^2 \leq \frac{1}{24c_{\mathbf{z}}^2 L_g^2}, \tag{155}$$

we can bound the term $T_4$ in the following manner

$$T_4 = -\eta_{\mathbf{x}} + \frac{12\eta_{\mathbf{z}}}{\lambda_g} + \frac{12(1-p)L_g^2 \eta_{\mathbf{z}}^2 \eta_{\mathbf{x}}}{bp} \leq 0. \tag{156}$$

**Analysis of the Coefficient for** $\mathbb{E}\left[\left\|\hat{\mathbf{v}}_{\mathbf{x}}^k\right\|^2\right]$**.**

Due to the assumption that

$$\eta_{\mathbf{x}} \leq \frac{c}{\sqrt{\frac{1-p}{bp}}}, \qquad c^2 \leq \frac{1}{16\left(L_g^2 + 8L_1^2\right)}, \tag{157}$$

we can bound the term $T_5$ in the following manner

$$\begin{aligned}
T_5 &= -\frac{\eta_{\mathbf{x}}}{4} + \frac{L_{\nabla\Phi}\eta_{\mathbf{x}}^2}{4} + \frac{2L_{\mathbf{y}^\star}^2 \eta_{\mathbf{x}}^2}{\lambda_g \eta_{\mathbf{y}}} + \frac{2L_{\mathbf{y}^\star}^2 \eta_{\mathbf{x}}^2}{\lambda_g \eta_{\mathbf{z}}} + \frac{(1-p)}{bp}\eta_{\mathbf{x}}^3(L_g^2 + 8L_1^2) \\
&\leq -\frac{\eta_{\mathbf{x}}}{4} + \frac{L_{\nabla\Phi}\eta_{\mathbf{x}}^2}{4} + \frac{2L_{\mathbf{y}^\star}^2 \eta_{\mathbf{x}}^2}{\lambda_g \eta_{\mathbf{y}}} + \frac{2L_{\mathbf{y}^\star}^2 \eta_{\mathbf{x}}^2}{\lambda_g \eta_{\mathbf{z}}} + \frac{\eta_{\mathbf{x}}}{16}.
\end{aligned} \tag{158}$$

Due to the assumption that

$$\eta_{\mathbf{x}} \leq \frac{1}{8L_{\nabla\Phi}}, \qquad \eta_{\mathbf{x}} \leq \frac{\lambda_g}{32L_{\mathbf{y}}^\star}\eta_{\mathbf{y}}, \qquad \eta_{\mathbf{x}} \leq \frac{\lambda_g}{32L_{\mathbf{z}}^\star}\eta_{\mathbf{z}}, \tag{159}$$

we further bound the term $T_2$ in the following manner

$$T_5 \leq -\frac{\eta_{\mathbf{x}}}{4} + \frac{L_{\nabla\Phi}\eta_{\mathbf{x}}^2}{4} + \frac{2L_{\mathbf{y}^\star}^2 \eta_{\mathbf{x}}^2}{\lambda_g \eta_{\mathbf{y}}} + \frac{2L_{\mathbf{y}^\star}^2 \eta_{\mathbf{x}}^2}{\lambda_g \eta_{\mathbf{z}}} + \frac{\eta_{\mathbf{x}}}{16} \leq 0. \tag{160}$$

Combining the above inequality (150), (152), (154), (156), (158) and (160), we then drive the convergence result. For simplify, we define

$$A_0 = 1 + 2(1 + \rho^2), \tag{161}$$

$$A_1 = \left(\left(\frac{192c_z}{\lambda_g} + 4\right)(1 + \rho^2) + \frac{96c_y}{\lambda_g}\right)\left(1 + \frac{1}{(1-\delta)n}\right), \tag{162}$$

$$A_2 = \frac{96c_y}{\lambda_g} + \left(\frac{192c_z}{\lambda_g} + 4\right)\rho^2, \tag{163}$$

and

$$A_3 = \frac{192c_z}{\lambda_g} + 4. \tag{164}$$

Thus, substituting the above inequality into (97) and combining the term $A_1, A_2, A_3, A_4, A_5$, we can obtain

$$\mathcal{L}_{k+1} - \mathcal{L}_k \leq \frac{\eta_{\mathbf{x}}}{4} \cdot \mathbb{E}\left[\left\|\nabla\Phi_{\mathcal{H}}\left(\mathbf{x}^k\right)\right\|^2\right]$$
$$+ A_0 \frac{\sigma^2}{b'(1-\delta)n}\eta_x + \kappa A_1 \eta_{\mathbf{x}}\sigma^2 + \kappa A_2 \eta_{\mathbf{x}}\zeta_g^2 + \kappa A_3 \eta_{\mathbf{x}}\zeta_f^2. \tag{165}$$

Rearranging the term in (165), we can obtain

$$\mathbb{E}\left[\left\|\nabla\Phi_{\mathcal{H}}\left(\mathbf{x}^k\right)\right\|^2\right] \leq 4\left(\frac{\mathcal{L}_k - \mathcal{L}_{k+1}}{\eta_{\mathbf{x}}} + A_0\frac{\sigma^2}{b'(1-\delta)n} + \kappa A_1\sigma^2 + \kappa A_2\zeta_g^2 + \kappa A_3\zeta_f^2\right). \tag{166}$$

As the above inequality (166) is true for an arbitrary $k \in [K]$, by taking summation on both sides from $k = 0$ to $k = K - 1$, and Dividing both sides by $K$ we can obtain

$$\frac{1}{K}\sum_{k=0}^{K-1}\mathbb{E}\left[\left\|\nabla\Phi_{\mathcal{H}}\left(\mathbf{x}^k\right)\right\|^2\right] \leq 4\left(\frac{\mathcal{L}_0 - \mathcal{L}_K}{\eta_{\mathbf{x}}K} + A_0\frac{\sigma^2}{b'(1-\delta)n} + \kappa A_1\sigma^2 + \kappa A_2\zeta_g^2 + \kappa A_3\zeta_f^2\right). \tag{167}$$

**Analyzing $\mathcal{L}_0 - \mathcal{L}_K$.** Recall the definition of Lyapunov function in (145), we can obtain

$$\mathcal{L}_0 - \mathcal{L}_K \leq \frac{\Phi(\mathbf{x}^0)}{2} + \mathbb{E}\left[\left\|\mathbf{y}^0 - \mathbf{y}_\star^0\right\|^2\right] + \mathbb{E}\left[\left\|\mathbf{z}^0 - \mathbf{z}_\star^0\right\|^2\right] + \frac{\eta_{\mathbf{x}}\sigma^2}{pb'(1-\delta)n}. \tag{168}$$

**Final step.** Recall that the step size $\eta_{\mathbf{x}}$ satisfies:

$$\eta_{\mathbf{x}} \leq \min\left\{\frac{c}{1 + \sqrt{\frac{1-p}{pb}}}, \frac{1}{8L_{\nabla\Phi}}\right\}, \quad \eta_{\mathbf{y}} = c_{\mathbf{y}}\eta_{\mathbf{x}}, \quad \eta_{\mathbf{z}} = c_{\mathbf{z}}\eta_{\mathbf{x}}. \tag{169}$$

Combing (168) and (167), the iterates satisfy

$$\frac{1}{K}\sum_{k=0}^{K}\mathbb{E}\left[\left\|\nabla\Phi_{\mathcal{H}}\left(\mathbf{x}^k\right)\right\|^2\right] = \mathcal{O}\left(\frac{1 + \sqrt{\frac{1-p}{pb}}}{K} + \frac{1}{Kpb'} + \frac{\sigma^2}{b'(1-\delta)n}\right)$$
$$+ \mathcal{O}\left(\kappa\left(\zeta_g^2 + \zeta_g^2\rho^2 + \zeta_f^2 + \sigma^2\right)\right). \tag{170}$$

Finally, as $\hat{\mathbf{x}}^{(i)} \sim \mathcal{U}\{\mathbf{x}_0^{(i)}, \ldots, \mathbf{x}_{K-1}^{(i)}\}$, we have

$$\mathbb{E}\left[\left\|\nabla\Phi_{\mathcal{H}}(\hat{\mathbf{x}})\right\|^2\right] = \mathcal{O}\left(\frac{1 + \sqrt{\frac{1-p}{pb}}}{K} + \frac{1}{Kpb'} + \frac{\sigma^2}{b'(1-\delta)n}\right)$$
$$+ \mathcal{O}\left(\kappa\left(\zeta_g^2 + \zeta_g^2\rho^2 + \zeta_f^2 + \sigma^2\right)\right). \tag{171}$$

Hence, the proof is complete. $\qquad\square$

**Corollary 4** (Analysis for sample complexity). *In general expectation cases, choose $b' = \min\left\{\frac{\sigma^2}{\epsilon}, N + M\right\}$, where $N$ and $M$ are the number of data points of upper and lower-level problem. If $N + M$ is very large, then $b' = \mathcal{O}(\epsilon^{-1})$. In each iteration, it uses $pb' + (1-p)b$ samples on expectation. Let $p = \frac{b}{b'+b}$, $b' = \mathcal{O}\left(\epsilon^{-1}\right)$ and $b \leq \sqrt{b'}$. Then the number of iterations performed by BR-FedBiP sufficient for finding an $\epsilon$-approximate solution in the general non-convex case can be bounded by*

$$K = \mathcal{O}\left(\epsilon^{-1}\left(1 + \frac{\sqrt{b'}}{b} + \frac{b'+b}{b'b}\right)\right). \tag{172}$$

*Thus, the total sample complexity is*

$$K\left(pb' + (1-p)b\right)$$
$$= \mathcal{O}\left(\epsilon^{-1}\left(1 + \frac{\sqrt{b'}}{b} + \frac{b'+b}{b'b}\right)\frac{2b'b}{b'+b}\right) \tag{173}$$
$$= \mathcal{O}\left(\sqrt{b'}\epsilon^{-1}\right) = \mathcal{O}\left(\epsilon^{-1.5}\right).$$

## J    PROOF OF SUPPORTING LEMMAS

### J.1    PROOF OF SUPPORTING LEMMAS FOR BR-FEDBI

#### J.1.1    PROOF OF LEMMA 9

**Lemma 9.** *Suppose Assumptions 3 and 4 hold,and the robust aggregation rule $AGG(\cdot)$ satisfies the $(\delta, \kappa)$-robustness criterion. The aggregation errors in the BR-FedBi algorithm (Algorithm 1) satisfy the following inequalities:*

$$(1) \quad \mathbb{E}\left[\left\|\xi_{\mathbf{y}}^k\right\|^2\right] \le \kappa\zeta_g^2 + \kappa\sigma^2\left(1 + \frac{1}{(1-\delta)n}\right),$$

$$(2) \quad \mathbb{E}\left[\left\|\xi_{\mathbf{x}}^k\right\|^2\right] \le 2\kappa\left(\rho^2\zeta_g^2 + \zeta_f^2\right) + 2\kappa\sigma_1^2\left(1 + \frac{1}{(1-\delta)n}\right), \qquad (174)$$

$$(3) \quad \mathbb{E}\left[\left\|\xi_{\mathbf{z}}^k\right\|^2\right] \le 2\kappa\left(\rho^2\zeta_g^2 + \zeta_f^2\right) + 2\kappa\sigma_1^2\left(1 + \frac{1}{(1-\delta)n}\right).$$

*Proof.* Now we analyze the aggregation error of $\mathbf{y}$ and $\mathbf{z}$ under Assumption 4.

**Analysis for $\mathbb{E}\left[\left\|\xi_{\mathbf{y}}^k\right\|^2\right]$.** Recall that

$$\xi_{\mathbf{y}}^k = \hat{\mathbf{D}}_{\mathbf{y}}^k - \mathbf{D}_{\mathbf{y}}^k.$$

For an arbitrary step $k > 1$, we obtain that

$$\mathbb{E}\left[\left\|\hat{\mathbf{D}}_{\mathbf{y}}^k - \mathbf{D}_{\mathbf{y}}^k\right\|^2\right]$$
$$\le \frac{\kappa}{(1-\delta)n}\sum_{i\in\mathcal{H}}\mathbb{E}\left[\left\|\mathbf{D}_{\mathbf{y},i}^k - \mathbf{D}_{\mathbf{y}}^k\right\|^2\right]$$
$$= \frac{\kappa}{(1-\delta)n}\sum_{i\in\mathcal{H}}\mathbb{E}\left[\left\|\nabla_{\mathbf{y}}g_i(\mathbf{x}^k,\mathbf{y}^k) - \nabla_{\mathbf{y}}g(\mathbf{x}^k,\mathbf{y}^k) + \mathbf{D}_{\mathbf{y},i}^k - g_i(\mathbf{x}^k,\mathbf{y}^k) + \nabla_{\mathbf{y}}g(\mathbf{x}^k,\mathbf{y}^k) - \mathbf{D}_{\mathbf{y}}^k\right\|^2\right]$$
$$\le \frac{\kappa}{(1-\delta)n}\sum_{i\in\mathcal{H}}\mathbb{E}\left[\left\|\nabla_{\mathbf{y}}g_i(\mathbf{x}^k,\mathbf{y}^k) - \nabla_{\mathbf{y}}g(\mathbf{x}^k,\mathbf{y}^k)\right\|^2\right] + \sigma^2\left(1 + \frac{1}{(1-\delta)n}\right)$$
$$\le \frac{\kappa}{(1-\delta)n}\sum_{i\in\mathcal{H}}\zeta_g^2 + \sigma^2\left(1 + \frac{1}{(1-\delta)n}\right)$$
$$\le \kappa\zeta_g^2 + \kappa\sigma^2\left(1 + \frac{1}{(1-\delta)n}\right).$$
$$(175)$$

The first inequality holds due to the $(\delta, \kappa)$-robustness of $AGG(\cdot)$ as defined in Definition 2. The third inequality is obtained using Assumptions 3, 4, and Lemma 7.

**Analysis for $\mathbb{E}\left[\left\|\xi_{\mathbf{x}}^k\right\|^2\right]$.** Recall that

$$\xi_{\mathbf{x}}^k = \hat{\mathbf{D}}_{\mathbf{x}}^k - \mathbf{D}_{\mathbf{x}}^k.$$

For an arbitrary step $k > 1$, using the Definition 2, we obtain that

$$\mathbb{E}\left[\left\|\hat{\mathbf{D}}_{\mathbf{x}}^k - \mathbf{D}_{\mathbf{x}}^k\right\|^2\right]$$
$$\le \frac{\kappa}{(1-\delta)n}\sum_{i\in\mathcal{H}}\mathbb{E}\left[\left\|\mathbf{D}_{\mathbf{x},i}^k - \mathbf{D}_{\mathbf{x}}^k\right\|^2\right]$$
$$\le \frac{\kappa}{(1-\delta)n}\sum_{i\in\mathcal{H}}\left(\mathbb{E}\left[\left\|\nabla_{\mathbf{xy}}^2 g_i\left(\mathbf{x}^k,\mathbf{y}^k\right)\mathbf{z}^k + \nabla_{\mathbf{x}}f_i\left(\mathbf{x}^k,\mathbf{y}^k\right) - \nabla_{\mathbf{xy}}^2 g\left(\mathbf{x}^k,\mathbf{y}^k\right)\mathbf{z}^k - \nabla_{\mathbf{x}}f(\mathbf{x}^k,\mathbf{y}^k)\right\|^2\right]\right.$$
$$\left. + \sigma_1^2\left(1 + \frac{1}{(1-\delta)n}\right)\right). \qquad (176)$$

Using the Cauchy-Schwarz inequality, we then obtain that

$$
\begin{aligned}
&\mathbb{E}\left[\left\|\hat{\mathbf{D}}_{\mathbf{x}}^k - \mathbf{D}_{\mathbf{x}}^k\right\|^2\right] \\
&\leq \frac{2\kappa}{(1-\delta)n}\sum_{i\in\mathcal{H}}\left(\mathbb{E}\left[\left\|\nabla_{\mathbf{xy}}^2 g_i\left(\mathbf{x}^k,\mathbf{y}^k\right) - \nabla_{\mathbf{xy}}^2 g\left(\mathbf{x}^k,\mathbf{y}^k\right)\right\|^2\right]\left\|\mathbf{z}^k\right\|^2\right) \\
&\quad + \frac{2\kappa}{(1-\delta)n}\sum_{i\in\mathcal{H}}\left(\mathbb{E}\left[\left\|\nabla_{\mathbf{x}}f_i\left(\mathbf{x}^k,\mathbf{y}^k\right) - \nabla_{\mathbf{x}}f(\mathbf{x}^k,\mathbf{y}^k)\right\|^2\right] + \sigma_1^2\left(1+\frac{1}{(1-\delta)n}\right)\right) \\
&\leq \frac{\kappa}{(1-\delta)n}\sum_{i\in\mathcal{H}} 2\rho^2\zeta_g^2 + 2\zeta_f^2 + 2\sigma_1^2\left(1+\frac{1}{(1-\delta)n}\right) \\
&\leq \kappa\left(2\rho^2\zeta_g^2 + 2\zeta_f^2\right) + 2\kappa\sigma_1^2\left(1+\frac{1}{(1-\delta)n}\right),
\end{aligned}
\tag{177}
$$

where inequality is derived using Assumptions 3, 4, and Lemma 7.

**Analysis for** $\mathbb{E}\left[\left\|\xi_{\mathbf{z}}^k\right\|^2\right]$**.** Recall that

$$
\xi_{\mathbf{z}}^k = \hat{\mathbf{D}}_{\mathbf{z}}^k - \mathbf{D}_{\mathbf{z}}^k.
$$

For an arbitrary step $k > 1$, follows Definition 2, we obtain that

$$
\begin{aligned}
&\mathbb{E}\left[\left\|\hat{\mathbf{D}}_{\mathbf{z}}^k - \mathbf{D}_{\mathbf{z}}^k\right\|^2\right] \\
&\leq \frac{\kappa}{(1-\delta)n}\sum_{i\in\mathcal{H}}\mathbb{E}\left[\left\|\mathbf{D}_{\mathbf{z},i}^k - \mathbf{D}_{\mathbf{z}}^k\right\|^2\right] \\
&\leq \frac{\kappa}{(1-\delta)n}\sum_{i\in\mathcal{H}}\left(\mathbb{E}\left[\left\|\nabla_{\mathbf{yy}}^2 g_i\left(\mathbf{x}^k,\mathbf{y}^k\right)\mathbf{z}^k + \nabla_{\mathbf{y}}f_i\left(\mathbf{x}^k,\mathbf{y}^k\right) - \nabla_{\mathbf{yy}}^2 g\left(\mathbf{x}^k,\mathbf{y}^k\right)\mathbf{z}^k - \nabla_{\mathbf{y}}f(\mathbf{x}^k,\mathbf{y}^k)\right\|^2\right]\right. \\
&\quad \left. + \sigma_1^2\left(1+\frac{1}{(1-\delta)n}\right)\right).
\end{aligned}
\tag{178}
$$

Using the the Cauchy-Schwarz inequality, we obtain that

$$
\begin{aligned}
&\mathbb{E}\left[\left\|\hat{\mathbf{D}}_{\mathbf{z}}^k - \mathbf{D}_{\mathbf{z}}^k\right\|^2\right] \\
&\leq \frac{2\kappa}{(1-\delta)n}\sum_{i\in\mathcal{H}}\mathbb{E}\left[\left\|\nabla_{\mathbf{yy}}^2 g_i\left(\mathbf{x}^k,\mathbf{y}^k\right) - \nabla_{\mathbf{yy}}^2 g\left(\mathbf{x}^k,\mathbf{y}^k\right)\right\|^2\right]\left\|\mathbf{z}^k\right\|^2 \\
&\quad + \frac{2\kappa}{(1-\delta)n}\sum_{i\in\mathcal{H}}\mathbb{E}\left[\left\|\nabla_{\mathbf{y}}f_i\left(\mathbf{x}^k,\mathbf{y}^k\right) - \nabla_{\mathbf{y}}f(\mathbf{x}^k,\mathbf{y}^k)\right\|^2\right] + \sigma_1^2\left(1+\frac{1}{(1-\delta)n}\right) \\
&\leq \frac{\kappa}{(1-\delta)n}\sum_{i\in\mathcal{H}} 2\rho^2\zeta_g^2 + 2\zeta_f^2 + 2\sigma_1^2\left(1+\frac{1}{(1-\delta)n}\right) \\
&\leq \kappa\left(2\rho^2\zeta_g^2 + 2\zeta_f^2\right) + 2\kappa\sigma_1^2\left(1+\frac{1}{(1-\delta)n}\right),
\end{aligned}
\tag{179}
$$

where the inequality is derived using Assumptions 3, 4, and Lemma 7. $\qquad\square$

### J.1.2 PROOF OF LEMMA 10

We now restate the Lemma 10 and present its proof.

**Lemma 10.** *Under Assumptions 1, 2, 3, 4, and 5, the following inequalities hold for* $\mathbf{y}$ *and* $\mathbf{z}$:

(1) $\mathbb{E}\left[\left\|\mathbf{y}^{k+1}-\mathbf{y}_\star^{k+1}\right\|^2\right]$

$$\leq (1 - \frac{\eta_{\mathbf{y}}\mu_g}{4} + 2\eta_{\mathbf{y}}^2 L_g^2)\mathbb{E}\left[\left\|\mathbf{y}^k-\mathbf{y}_\star^k\right\|^2\right] + (3\eta_{\mathbf{y}}^2 + \frac{\eta_{\mathbf{y}}}{\mu_g})\mathbb{E}\left[\left\|\xi_{\mathbf{y}}^k\right\|^2\right]$$

$$+ (6L_{\mathbf{y}^\star}^2\eta_{\mathbf{x}}^2 + \frac{4\eta_{\mathbf{x}}^2}{\eta_{\mathbf{y}}\mu_g}L_{\mathbf{y}^\star}^2)\left(\mathbb{E}\left[\left\|\mathbb{E}(\mathbf{D}_{\mathbf{x}}^k)\right\|^2\right] + \mathbb{E}\left[\left\|\xi_{\mathbf{x}}^k\right\|^2\right]\right)$$

$$+ 2\eta_{\mathbf{y}}^2\frac{\sigma^2}{(1-\delta)n} + 3L_{\mathbf{y}^\star}^2\eta_{\mathbf{x}}^2\frac{\sigma_1^2}{(1-\delta)n},$$

(180)

(2) $\mathbb{E}\left[\left\|\mathbf{z}^{k+1}-\mathbf{z}_\star^{k+1}\right\|^2\right]$

$$\leq \left(1 - \frac{\eta_{\mathbf{z}}\mu_g}{2} + 6\eta_{\mathbf{z}}^2 L_g^2\right)\mathbb{E}\left[\left\|\mathbf{z}^k-\mathbf{z}_\star^k\right\|^2\right] + \left(\frac{4\eta_{\mathbf{z}}L_2^2}{\mu_g} + 6\eta_{\mathbf{z}}^2 L_2^2\right)\mathbb{E}\left[\left\|\mathbf{y}^k-\mathbf{y}_\star^k\right\|^2\right]$$

$$+ \left(4L_{\mathbf{z}^\star}^2\eta_{\mathbf{x}}^2 + 4L_{zx}\rho\eta_{\mathbf{x}}^2 + \frac{4\eta_{\mathbf{x}}^2}{\eta_{\mathbf{z}}\mu_g}L_{\mathbf{z}^\star}^2\right)\left(\mathbb{E}\left[\left\|\mathbb{E}(\mathbf{D}_{\mathbf{x}}^k)\right\|^2\right] + \mathbb{E}\left[\left\|\xi_{\mathbf{x}}^k\right\|^2\right]\right)$$

$$+ \left(3\eta_{\mathbf{z}}^2 + \frac{2\eta_{\mathbf{z}}}{\mu_g}\right)\mathbb{E}\left[\left\|\xi_{\mathbf{z}}^k\right\|^2\right] + \left(2\eta_{\mathbf{z}}^2 + 4L_{\mathbf{z}^\star}^2\eta_{\mathbf{x}}^2 + 4L_{zx}\rho\eta_{\mathbf{x}}^2\right)\frac{\sigma_1^2}{(1-\delta)n},$$

where $\mathbf{y}_\star^k = \mathbf{y}^\star(\mathbf{x}^k)$, and $\mathbf{z}_\star^k = \mathbf{z}^\star(\mathbf{x}^k)$.

*Proof.* **Inequality for** $\mathbf{y}$.

We first separate the $\mathbb{E}\left[\left\|\mathbf{y}^{k+1}-\mathbf{y}_\star^{k+1}\right\|^2\right]$ into five parts. Due to iteratively updating $\mathbf{y}^{k+1}$, we have

$$\mathbb{E}\left[\left\|\mathbf{y}^{k+1}-\mathbf{y}_\star^{k+1}\right\|^2\right]$$

$$=\mathbb{E}\left[\left\|\mathbf{y}^{k+1}-\mathbf{y}_\star^k\right\|^2\right] + \mathbb{E}\left[\left\|\mathbf{y}_\star^{k+1}-\mathbf{y}_\star^k\right\|^2\right] - 2\mathbb{E}\left[\langle\mathbf{y}^{k+1}-\mathbf{y}_\star^k, \mathbf{y}_\star^{k+1}-\mathbf{y}_\star^k\rangle\right]$$

$$=\mathbb{E}\left[\left\|\mathbf{y}^{k+1}-\mathbf{y}_\star^k\right\|^2\right] + \mathbb{E}\left[\left\|\mathbf{y}_\star^{k+1}-\mathbf{y}_\star^k\right\|^2\right] - 2\mathbb{E}\left[\langle\mathbf{y}^k-\mathbf{y}_\star^k, \mathbf{y}_\star^{k+1}-\mathbf{y}_\star^k\rangle\right] + 2\eta_{\mathbf{y}}\mathbb{E}\left[\langle\hat{\mathbf{D}}_{\mathbf{y}}^k, \mathbf{y}_\star^{k+1}-\mathbf{y}_\star^k\rangle\right]$$

$$=\mathbb{E}\left[\left\|\mathbf{y}^{k+1}-\mathbf{y}_\star^k\right\|^2\right] + \mathbb{E}\left[\left\|\mathbf{y}_\star^{k+1}-\mathbf{y}_\star^k\right\|^2\right] - 2\mathbb{E}\left[\langle\mathbf{y}^k-\mathbf{y}_\star^k, \nabla\mathbf{y}^\star(\mathbf{x}^k)(\mathbf{x}^{k+1}-\mathbf{x}^k)\rangle\right]$$

$$- 2\mathbb{E}\left[\langle\mathbf{y}^k-\mathbf{y}_\star^k, \mathbf{y}_\star^{k+1}-\mathbf{y}_\star^k - \nabla\mathbf{y}^\star(\mathbf{x}^k)(\mathbf{x}^{k+1}-\mathbf{x}^k)\rangle\right] + 2\eta_{\mathbf{y}}\mathbb{E}\left[\langle\hat{\mathbf{D}}_{\mathbf{y}}^k, \mathbf{y}_\star^{k+1}-\mathbf{y}_\star^k\rangle\right],$$

(181)

where the existence of $\nabla\mathbf{y}^\star(\mathbf{x}^k)$ is guaranteed by Lemma 4.

For the first part in the R.H.S. of (181), by the Assumption 3, $\mathbf{D}_{\mathbf{y}}^k$ is the unbiased estimation of $\mathbb{E}(\mathbf{D}_{\mathbf{y}}^k)$, then we have

$$\mathbb{E}\left[\left\|\mathbf{y}^{k+1}-\mathbf{y}_\star^k\right\|^2\right]$$

$$= \mathbb{E}\left[\left\|\mathbf{y}^k - \eta_{\mathbf{y}}\hat{\mathbf{D}}_{\mathbf{y}}^k - \mathbf{y}_\star^k\right\|^2\right] = \mathbb{E}\left[\left\|(\mathbf{y}^k-\mathbf{y}_\star^k - \eta_{\mathbf{y}}\nabla_{\mathbf{y}}g^k) - \eta_{\mathbf{y}}\left(\hat{\mathbf{D}}_{\mathbf{y}}^k - \nabla_{\mathbf{y}}g^k\right)\right\|^2\right]$$

$$= \mathbb{E}\left[\left\|\mathbf{y}^k-\mathbf{y}_\star^k - \eta_{\mathbf{y}}\nabla_{\mathbf{y}}g^k\right\|^2\right] + \mathbb{E}\left[\left\|\eta_{\mathbf{y}}\left(\hat{\mathbf{D}}_{\mathbf{y}}^k - \nabla_{\mathbf{y}}g^k\right)\right\|^2\right] - 2\eta_{\mathbf{y}}\mathbb{E}\left[\langle\mathbf{y}^k-\mathbf{y}^\star - \eta_{\mathbf{y}}\nabla_{\mathbf{y}}g^k, \hat{\mathbf{D}}_{\mathbf{y}}^k - \nabla_{\mathbf{y}}g^k\rangle\right]$$

$$\leq \mathbb{E}\left[\left\|\mathbf{y}^k-\mathbf{y}_\star^k - \eta_{\mathbf{y}}\nabla_{\mathbf{y}}g^k\right\|^2\right] + \mathbb{E}\left[\left\|\eta_{\mathbf{y}}\left(\hat{\mathbf{D}}_{\mathbf{y}}^k - \nabla_{\mathbf{y}}g^k\right)\right\|^2\right] - 2\eta_{\mathbf{y}}\mathbb{E}\left[\langle\mathbf{y}^k-\mathbf{y}^\star - \eta_{\mathbf{y}}\nabla_{\mathbf{y}}g^k, \hat{\mathbf{D}}_{\mathbf{y}}^k - \mathbf{D}_{\mathbf{y}}^k\rangle\right]$$

$$\leq (1 - \eta_{\mathbf{y}}\mu_g)^2\mathbb{E}\left[\left\|\mathbf{y}^k-\mathbf{y}_\star^k\right\|^2\right] + \eta_{\mathbf{y}}^2\mathbb{E}\left[\left\|\left(\hat{\mathbf{D}}_{\mathbf{y}}^k - \nabla_{\mathbf{y}}g^k\right)\right\|^2\right] - 2\eta_{\mathbf{y}}\mathbb{E}\left[\langle\mathbf{y}^k-\mathbf{y}^\star - \eta_{\mathbf{y}}\nabla_{\mathbf{y}}g^k, \hat{\mathbf{D}}_{\mathbf{y}}^k - \mathbf{D}_{\mathbf{y}}^k\rangle\right].$$

(182)

The first equality follows the unbiased estimation of $\mathbf{D}_{\mathbf{y}}^k$ in Assumption 3, and the second inequality is derived using Lemma 6.

Then, using Young's inequality $2\langle a, b\rangle \leq ca^2 + \frac{1}{c}b^2$, we have

$$
\begin{aligned}
&- 2\eta_{\mathbf{y}}\mathbb{E}\left[\left\langle \mathbf{y}^k - \mathbf{y}^\star - \eta_{\mathbf{y}}\nabla_{\mathbf{y}}g^k, \hat{\mathbf{D}}_{\mathbf{y}}^k - \mathbf{D}_{\mathbf{y}}^k\right\rangle\right] \\
&\leq \frac{\eta_{\mathbf{y}}\mu_g}{1 - \eta_{\mathbf{y}}\mu_g}\mathbb{E}\left[\left\|\mathbf{y}^k - \mathbf{y}^\star - \eta_{\mathbf{y}}\nabla_{\mathbf{y}}g^k\right\|^2\right] + \frac{\eta_{\mathbf{y}}}{\mu_g}(1 - \eta_{\mathbf{y}}\mu_g)\mathbb{E}\left[\left\|\xi_{\mathbf{y}}^k\right\|^2\right] \\
&\leq \eta_{\mathbf{y}}\mu_g(1 - \eta_{\mathbf{y}}\mu_g)\mathbb{E}\left[\left\|\mathbf{y}^k - \mathbf{y}_\star^k\right\|^2\right] + \frac{\eta_{\mathbf{y}}}{\mu_g}(1 - \eta_{\mathbf{y}}\mu_g)\mathbb{E}\left[\left\|\xi_{\mathbf{y}}^k\right\|^2\right],
\end{aligned}
\tag{183}
$$

where the last inequality is derived using Lemma 6.

Combining (183) into (182), and use the Cauchy–Schwarz inequality, we have

$$
\begin{aligned}
&\mathbb{E}\left[\left\|\mathbf{y}^{k+1} - \mathbf{y}_\star^k\right\|^2\right] \\
&\leq (1 - \eta_{\mathbf{y}}\mu_g)^2\mathbb{E}\left[\left\|\mathbf{y}^k - \mathbf{y}_\star^k\right\|^2\right] + \eta_{\mathbf{y}}^2\mathbb{E}\left[\left\|\left(\hat{\mathbf{D}}_{\mathbf{y}}^k - \nabla_{\mathbf{y}}g^k\right)\right\|^2\right] \\
&\quad + \eta_{\mathbf{y}}\mu_g(1 - \eta_{\mathbf{y}}\mu_g)\mathbb{E}\left[\left\|\mathbf{y}^k - \mathbf{y}_\star^k\right\|^2\right] + \frac{\eta_{\mathbf{y}}}{\mu_g}(1 - \eta_{\mathbf{y}}\mu_g)\mathbb{E}\left[\left\|\xi_{\mathbf{y}}^k\right\|^2\right] \\
&\leq (1 - \eta_{\mathbf{y}}\mu_g)\mathbb{E}\left[\left\|\mathbf{y}^k - \mathbf{y}_\star^k\right\|^2\right] + \left(\eta_{\mathbf{y}}^2 + \frac{\eta_{\mathbf{y}}}{\mu_g}\right)\mathbb{E}\left[\left\|\xi_{\mathbf{y}}^k\right\|^2\right] + \frac{2\eta_{\mathbf{y}}^2\sigma^2}{(1 - \delta)n}.
\end{aligned}
\tag{184}
$$

For the second part in the R.H.S. of (181), using Lemma 4, we can imply that

$$
\mathbb{E}\left[\left\|\mathbf{y}_\star^{k+1} - \mathbf{y}_\star^k\right\|^2\right] \leq L_{\mathbf{y}^\star}^2\eta_{\mathbf{x}}^2\mathbb{E}\left[\left\|\hat{\mathbf{D}}_{\mathbf{x}}^k\right\|^2\right].
\tag{185}
$$

For the third part in the R.H.S. of (181), using the fact that $\mathbf{D}_{\mathbf{y}}^k$ is the unbiased estimation of $\mathbb{E}(\mathbf{D}_{\mathbf{y}}^k)$, we have

$$
\begin{aligned}
&- 2\mathbb{E}\left[\left\langle \mathbf{y}^k - \mathbf{y}_\star^k, \nabla\mathbf{y}^\star\left(\mathbf{x}^k\right)\left(\mathbf{x}^{k+1} - \mathbf{x}^k\right)\right\rangle\right] \\
&= 2\eta_{\mathbf{x}}\mathbb{E}\left[\left\langle \mathbf{y}^k - \mathbf{y}_\star^k, \nabla\mathbf{y}^\star\left(\mathbf{x}^k\right)\left(\mathbf{D}_{\mathbf{x}}^k + \xi_{\mathbf{x}}^k\right)\right\rangle\right] \\
&= 2\eta_{\mathbf{x}}\mathbb{E}\left[\left\langle \mathbf{y}^k - \mathbf{y}_\star^k, \nabla\mathbf{y}^\star\left(\mathbf{x}^k\right)\left(\mathbb{E}(\mathbf{D}_{\mathbf{x}}^k) + \xi_{\mathbf{x}}^k\right)\right\rangle\right] \\
&\leq \frac{\eta_{\mathbf{y}}\mu_g}{2}\mathbb{E}\left[\left\|\mathbf{y}^k - \mathbf{y}_\star^k\right\|^2\right] + \frac{4\eta_{\mathbf{x}}^2}{\eta_{\mathbf{y}}\mu_g}L_{\mathbf{y}^\star}^2\mathbb{E}\left[\left\|\mathbb{E}(\mathbf{D}_{\mathbf{x}}^k)\right\|^2\right] + \frac{4\eta_{\mathbf{x}}^2}{\eta_{\mathbf{y}}\mu_g}L_{\mathbf{y}^\star}^2\mathbb{E}\left[\left\|\xi_{\mathbf{x}}^k\right\|^2\right],
\end{aligned}
\tag{186}
$$

where the inequality uses Young's inequality that $2ab \leq \frac{1}{c}a^2 + cb^2$ and Lemma 4.

For the fourth part in the R.H.S. of (181), using the Cauchy–Schwarz inequality, we have

$$
\begin{aligned}
&- 2\mathbb{E}\left[\left\langle \mathbf{y}^k - \mathbf{y}_\star^k, \mathbf{y}_\star^{k+1} - \mathbf{y}_\star^k - \nabla\mathbf{y}^\star\left(\mathbf{x}^k\right)\left(\mathbf{x}^{k+1} - \mathbf{x}^k\right)\right\rangle\right] \\
&\leq 2\mathbb{E}\left[\left\|\mathbf{y}^k - \mathbf{y}_\star^k\right\| \cdot \left\|\mathbf{y}_\star^{k+1} - \mathbf{y}_\star^k - \nabla\mathbf{y}^\star\left(\mathbf{x}^k\right)\left(\mathbf{x}^{k+1} - \mathbf{x}^k\right)\right\|\right].
\end{aligned}
$$

Then, using Taylor's expansion, and Lemma 5, we have

$$
\begin{aligned}
&2\mathbb{E}\left[\left\|\mathbf{y}^k - \mathbf{y}_\star^k\right\| \cdot \left\|\mathbf{y}_\star^{k+1} - \mathbf{y}_\star^k - \nabla\mathbf{y}^\star\left(\mathbf{x}^k\right)\left(\mathbf{x}^{k+1} - \mathbf{x}^k\right)\right\|\right] \\
&\leq L_{\mathbf{yx}}\mathbb{E}\left[\left\|\mathbf{y}^k - \mathbf{y}_\star^k\right\| \left\|\mathbf{x}^{k+1} - \mathbf{x}^k\right\|^2\right].
\end{aligned}
\tag{187}
$$

Further, applies Young's inequality $2ab \leq \frac{1}{c}a^2 + cb^2$, we have

$$
\begin{aligned}
&L_{\mathbf{yx}}\mathbb{E}\left[\left\|\mathbf{y}^k - \mathbf{y}_\star^k\right\| \left\|\mathbf{x}^{k+1} - \mathbf{x}^k\right\|^2\right] \\
&\leq \frac{L_{\mathbf{yx}}^2}{4L_{\mathbf{y}^\star}^2}\mathbb{E}\left[\left\|\mathbf{y}^k - \mathbf{y}_\star^k\right\|^2 \left\|\mathbf{x}^{k+1} - \mathbf{x}^k\right\|^2\right] + L_{\mathbf{y}^\star}^2\mathbb{E}\left[\left\|\mathbf{x}^{k+1} - \mathbf{x}^k\right\|^2\right] \\
&= \frac{L_{\mathbf{yx}}^2\eta_{\mathbf{x}}^2}{4L_{\mathbf{y}^\star}^2}\mathbb{E}\left[\left\|\mathbf{y}^k - \mathbf{y}_\star^k\right\|^2\right]\mathbb{E}\left[\left\|\hat{\mathbf{D}}_{\mathbf{x}}^k\right\|^2\right] + L_{\mathbf{y}^\star}^2\eta_{\mathbf{x}}^2\mathbb{E}\left[\left\|\hat{\mathbf{D}}_{\mathbf{x}}^k\right\|^2\right].
\end{aligned}
\tag{188}
$$

Then, using the following inequality

$$
\begin{aligned}
\mathbb{E}\left[\left\|\hat{\mathbf{D}}_{\mathbf{x}}^k\right\|^2\right] &= \mathbb{E}\left[\left\|\hat{\mathbf{D}}_{\mathbf{x}}^k - \mathbf{D}_{\mathbf{x}}^k + \mathbf{D}_{\mathbf{x}}^k\right\|^2\right] \\
&\leq 2\mathbb{E}\left[\left\|\hat{\mathbf{D}}_{\mathbf{x}}^k - \mathbf{D}_{\mathbf{x}}^k\right\|^2\right] + 2\mathbb{E}\left[\left\|\mathbf{D}_{\mathbf{x}}^k\right\|^2\right] \\
&\leq 2\mathbb{E}\left[\left\|\mathbf{D}_{\mathbf{x}}^k\right\|^2\right] + 2\mathbb{E}\left[\left\|\hat{\mathbf{D}}_{\mathbf{x}}^k - \mathbf{D}_{\mathbf{x}}^k\right\|^2\right] \\
&\leq 2H_{\mathbf{x}}^2 + 2\mathbb{E}\left[\left\|\xi_{\mathbf{x}}^k\right\|^2\right] + 2\frac{\sigma^2}{(1-\delta)n} \leq 2H_{\mathbf{x}}'^2,
\end{aligned} \tag{189}
$$

where the third inequality uses Assumption 5 and we define $H_x'$ as follows

$$
H_{\mathbf{x}}' := H_{\mathbf{x}}^2 + 2\kappa(\zeta_g^2\rho^2 + \zeta_f^2) + 2\kappa\sigma_1^2\left(1 + \frac{1}{(1+\delta)n}\right) + \frac{\sigma^2}{(1-\delta)n}.
$$

Finally, choosing the learning rate that $\eta_{\mathbf{x}} \leq \sqrt{\frac{\mu_g L_{\mathbf{y}^\star}^2 \eta_{\mathbf{y}}}{2L_{\mathbf{yx}}^2 H_{\mathbf{x}}'^2}}$, we can bound the fourth term in the R.H.S of (181) as follows

$$
-2\mathbb{E}\left[\left\langle \mathbf{y}^k - \mathbf{y}_\star^k, \mathbf{y}_\star^{k+1} - \mathbf{y}_\star^k - \nabla\mathbf{y}^\star\left(\mathbf{x}^k\right)\left(\mathbf{x}^{k+1} - \mathbf{x}^k\right)\right\rangle\right] \tag{190}
$$

$$
\leq \frac{\eta_{\mathbf{y}}\mu_g}{4}\mathbb{E}\left[\left\|\mathbf{y}^k - \mathbf{y}_\star^k\right\|^2\right] + L_{\mathbf{y}^\star}^2\eta_{\mathbf{x}}^2\mathbb{E}\left[\left\|\hat{\mathbf{D}}_{\mathbf{x}}^k\right\|^2\right]. \tag{191}
$$

For the fifth part in the R.H.S. of (181), by Young's inequality and Lemma 4, we have

$$
2\eta_{\mathbf{y}}\mathbb{E}\left[\left\langle \hat{\mathbf{D}}_{\mathbf{y}}^k, \mathbf{y}_\star^{k+1} - \mathbf{y}_\star^k\right\rangle\right] \leq \eta_{\mathbf{y}}^2\mathbb{E}\left[\left\|\hat{\mathbf{D}}_{\mathbf{y}}^k\right\|^2\right] + L_{\mathbf{y}^\star}^2\eta_{\mathbf{x}}^2\mathbb{E}\left[\left\|\hat{\mathbf{D}}_{\mathbf{x}}^k\right\|^2\right]. \tag{192}
$$

By substituting the above inequality into (181) and use Cauchy–Schwarz inequality and Lemma 24, we have

$$
\begin{aligned}
&\mathbb{E}\left[\left\|\mathbf{y}^{k+1} - \mathbf{y}_\star^{k+1}\right\|^2\right] \\
&\leq (1 - \frac{\eta_{\mathbf{y}}\mu_g}{4})\mathbb{E}\left[\left\|\mathbf{y}^k - \mathbf{y}_\star^k\right\|^2\right] + \left(6L_{\mathbf{y}^\star}^2\eta_{\mathbf{x}}^2 + \frac{4\eta_{\mathbf{x}}^2}{\eta_{\mathbf{y}}\mu_g}L_{\mathbf{y}^\star}^2\right)\left(\left[\left\|\mathbb{E}(\mathbf{D}_{\mathbf{x}}^k)\right\|^2\right] + \mathbb{E}\left[\left\|\xi_{\mathbf{x}}^k\right\|^2\right]\right) \\
&+ \eta_{\mathbf{y}}^2\mathbb{E}\left[\left\|\hat{\mathbf{D}}_{\mathbf{y}}^k\right\|^2\right] + \left(\eta_{\mathbf{y}}^2 + \frac{\eta_{\mathbf{y}}}{\mu_g}\right)\mathbb{E}\left[\left\|\xi_{\mathbf{y}}^k\right\|^2\right] + 3L_{\mathbf{y}^\star}^2\eta_{\mathbf{x}}^2\frac{\sigma_1^2}{(1-\delta)n}.
\end{aligned} \tag{193}
$$

By substituting Lemmas 24 into equation (193), we obtain

$$
\begin{aligned}
\mathbb{E}\left[\left\|\mathbf{y}^{k+1} - \mathbf{y}_\star^{k+1}\right\|^2\right] &\leq (1 - \frac{\eta_{\mathbf{y}}\mu_g}{4} + 2\eta_{\mathbf{y}}^2 L_g^2)\mathbb{E}\left[\left\|\mathbf{y}^k - \mathbf{y}_\star^k\right\|^2\right] + (3\eta_{\mathbf{y}}^2 + \frac{\eta_{\mathbf{y}}}{\mu_g})\mathbb{E}\left[\left\|\xi_{\mathbf{y}}^k\right\|^2\right] \\
&+ \left(6L_{\mathbf{y}^\star}^2\eta_{\mathbf{x}}^2 + \frac{4\eta_{\mathbf{x}}^2}{\eta_{\mathbf{y}}\mu_g}L_{\mathbf{y}^\star}^2\right)\left(\mathbb{E}\left[\left\|\mathbb{E}(\mathbf{D}_{\mathbf{x}}^k)\right\|^2\right] + \mathbb{E}\left[\left\|\xi_{\mathbf{x}}^k\right\|^2\right]\right) \\
&+ 2\eta_{\mathbf{y}}^2\frac{\sigma^2}{(1-\delta)n} + +3L_{\mathbf{y}^\star}^2\eta_{\mathbf{x}}^2\frac{\sigma_1^2}{(1-\delta)n}.
\end{aligned} \tag{194}
$$

Thus, the proof is finished.

**Inequality for z.** Similarly, We separate $\mathbb{E}\left[\left\|\mathbf{z}^{k+1} - \mathbf{z}_\star^{k+1}\right\|^2\right]$ into five parts. Due to iteratively updating $\mathbf{z}^{k+1}$ , we have

$$
\begin{aligned}
&\mathbb{E}\left[\left\|\mathbf{z}^{k+1} - \mathbf{z}_\star^{k+1}\right\|^2\right] \\
=&\mathbb{E}\left[\left\|\mathbf{z}^{k+1} - \mathbf{z}_\star^k\right\|^2\right] + \mathbb{E}\left[\left\|\mathbf{z}_\star^{k+1} - \mathbf{z}_\star^k\right\|^2\right] - 2\mathbb{E}\left[\left\langle \mathbf{z}^{k+1} - \mathbf{z}_\star^k, \mathbf{z}_\star^{k+1} - \mathbf{z}_\star^k \right\rangle\right] \\
=&\mathbb{E}\left[\left\|\mathbf{z}^{k+1} - \mathbf{z}_\star^k\right\|^2\right] + \mathbb{E}\left[\left\|\mathbf{z}_\star^{k+1} - \mathbf{z}_\star^k\right\|^2\right] - 2\mathbb{E}\left[\left\langle \mathbf{z}^k - \mathbf{z}_\star^k, \mathbf{z}_\star^{k+1} - \mathbf{z}_\star^k \right\rangle\right] + 2\eta_{\mathbf{z}}\mathbb{E}\left[\left\langle \hat{\mathbf{D}}_{\mathbf{z}}^k, \mathbf{z}_\star^{k+1} - \mathbf{z}_\star^k \right\rangle\right] \\
=&\mathbb{E}\left[\left\|\mathbf{z}^{k+1} - \mathbf{z}_\star^k\right\|^2\right] + \mathbb{E}\left[\left\|\mathbf{z}_\star^{k+1} - \mathbf{z}_\star^k\right\|^2\right] - 2\mathbb{E}\left[\left\langle \mathbf{z}^k - \mathbf{z}_\star^k, \nabla \mathbf{z}^\star\left(\mathbf{x}^k\right)\left(\mathbf{x}^{k+1} - \mathbf{x}^k\right) \right\rangle\right] \\
&- 2\mathbb{E}\left[\left\langle \mathbf{z}^k - \mathbf{z}_\star^k, \mathbf{z}_\star^{k+1} - \mathbf{z}_\star^k - \nabla \mathbf{z}^\star\left(\mathbf{x}^k\right)\left(\mathbf{x}^{k+1} - \mathbf{x}^k\right) \right\rangle\right] + 2\eta_{\mathbf{z}}\mathbb{E}\left[\left\langle \hat{\mathbf{D}}_{\mathbf{z}}^k, \mathbf{z}_\star^{k+1} - \mathbf{z}_\star^k \right\rangle\right] .
\end{aligned}
\tag{195}
$$

where the existence of $\nabla \mathbf{z}^\star\left(\mathbf{x}^k\right)$ is guaranteed by Lemma 4.

For the first part in the R.H.S. of (195), we separate each term and analysis, and we have

$$
\begin{aligned}
&\mathbb{E}\left[\left\|\mathbf{z}^{k+1} - \mathbf{z}_\star^k\right\|^2\right] \\
=& \mathbb{E}\left[\left\|\mathbf{z}^k - \eta_{\mathbf{z}}\mathbb{E}\left(\mathbf{D}_{\mathbf{z}}^k\right) - \mathbf{z}_\star^k + \eta_{\mathbf{z}}(\mathbb{E}\left(\mathbf{D}_{\mathbf{z}}^k\right) - \hat{\mathbf{D}}_{\mathbf{z}}^k)\right\|^2\right] \\
=& \mathbb{E}\left[\left\|\left(\mathbf{z}^k - \mathbf{z}_\star^k\right) - \eta_{\mathbf{z}}\nabla_{\mathbf{yy}}^2 g^k\left(\mathbf{z}^k - \mathbf{z}_\star^k\right) - \eta_{\mathbf{z}}\left(\nabla_{\mathbf{yy}}^2 g^k \mathbf{z}_\star^k + \nabla_{\mathbf{y}} f^k\right) + \eta_{\mathbf{z}}(\mathbb{E}\left(\mathbf{D}_{\mathbf{z}}^k\right) - \hat{\mathbf{D}}_{\mathbf{z}}^k)\right\|^2\right] \\
=& \mathbb{E}\left[\left\|\left(\mathbf{z}^k - \mathbf{z}_\star^k\right) - \eta_{\mathbf{z}}\nabla_{\mathbf{yy}}^2 g^k\left(\mathbf{z}^k - \mathbf{z}_\star^k\right)\right\|^2\right] + \eta_{\mathbf{z}}^2\mathbb{E}\left[\left\|\mathbb{E}(\mathbf{D}_{\mathbf{z}}^k) - \hat{\mathbf{D}}_{\mathbf{z}}^k\right\|^2\right] + \mathbb{E}\left[\left\|\eta_{\mathbf{z}}\left(\nabla_{\mathbf{yy}}^2 g^k \mathbf{z}_\star^k + \nabla_{\mathbf{y}} f^k\right)\right\|^2\right] \\
&+ 2\eta_{\mathbf{z}}^2\mathbb{E}\left[\left\langle \left(\mathbf{z}^k - \mathbf{z}_\star^k\right) - \eta_{\mathbf{z}}\nabla_{\mathbf{yy}}^2 g^k\left(\mathbf{z}^k - \mathbf{z}_\star^k\right), \mathbb{E}(\mathbf{D}_{\mathbf{z}}^k) - \hat{\mathbf{D}}_{\mathbf{z}}^k \right\rangle\right] \\
&- 2\eta_{\mathbf{z}}\mathbb{E}\left[\left\langle \left(\nabla_{\mathbf{yy}}^2 g^k \mathbf{z}_\star^k + \nabla_{\mathbf{y}} f^k\right), \mathbb{E}(\mathbf{D}_{\mathbf{z}}^k) - \hat{\mathbf{D}}_{\mathbf{z}}^k \right\rangle\right] \\
&- 2\eta_{\mathbf{z}}\mathbb{E}\left[\left\langle \left(\nabla_{\mathbf{yy}}^2 g^k \mathbf{z}_\star^k + \nabla_{\mathbf{y}} f^k\right), \left(\mathbf{z}^k - \mathbf{z}_\star^k\right) - \eta_{\mathbf{z}}\nabla_{\mathbf{yy}}^2 g^k\left(\mathbf{z}^k - \mathbf{z}_\star^k\right) \right\rangle\right] .
\end{aligned}
\tag{196}
$$

Then, we start to analyze the **last three terms** in the R.H.S. of (196).

For the first term, using Young's inequality $2\langle a, b\rangle \le ca^2 + \frac{1}{c}b^2$ and $\left\|I - \eta_{\mathbf{z}}\nabla_{\mathbf{yy}}^2 g^k\right\| \le 1 - \eta_{\mathbf{z}}\mu_g$, we have

$$
\begin{aligned}
&2\eta_{\mathbf{z}}\mathbb{E}\left[\left\langle \left(\mathbf{z}^k - \mathbf{z}_\star^k\right) - \eta_{\mathbf{z}}\nabla_{\mathbf{yy}}^2 g^k\left(\mathbf{z}^k - \mathbf{z}_\star^k\right), \mathbb{E}(\mathbf{D}_{\mathbf{z}}^k) - \hat{\mathbf{D}}_{\mathbf{z}}^k \right\rangle\right] \\
\le&\ 2\eta_{\mathbf{z}}\mathbb{E}\left[\left\|\left(\mathbf{z}^k - \mathbf{z}_\star^k\right) - \eta_{\mathbf{z}}\nabla_{\mathbf{yy}}^2 g^k\left(\mathbf{z}^k - \mathbf{z}_\star^k\right)\right\| \cdot \left\|\mathbf{D}_{\mathbf{z}}^k - \hat{\mathbf{D}}_{\mathbf{z}}^k\right\|\right] \\
\le&\ \frac{\eta_{\mathbf{z}}\mu_g}{2(1 - \eta_{\mathbf{z}}\mu_g)}\mathbb{E}\left[\left\|\left(\mathbf{z}^k - \mathbf{z}_\star^k\right) - \eta_{\mathbf{z}}\nabla_{\mathbf{yy}}^2 g^k\left(\mathbf{z}^k - \mathbf{z}_\star^k\right)\right\|^2\right] + \frac{2\eta_{\mathbf{z}}}{\mu_g}(1 - \eta_{\mathbf{z}}\mu_g)\mathbb{E}\left[\left\|\xi_{\mathbf{z}}^k\right\|^2\right] \\
\le&\ \frac{\eta_{\mathbf{z}}\mu_g(1 - \eta_{\mathbf{z}}\mu_g)}{2}\mathbb{E}\left[\left\|\mathbf{z}^k - \mathbf{z}_\star^k\right\|^2\right] + \frac{2\eta_{\mathbf{z}}}{\mu_g}(1 - \eta_{\mathbf{z}}\mu_g)\mathbb{E}\left[\left\|\xi_{\mathbf{z}}^k\right\|^2\right] ,
\end{aligned}
\tag{197}
$$

where the two inequality uses Young's inequality, $\left\|I - \eta_{\mathbf{z}}\nabla_{\mathbf{yy}}^2 g^k\right\| \le 1 - \eta_{\mathbf{z}}\mu_g$ and Lemma 4.

For the second term, using the Assumption 1 and Cauchy–Schwarz inequality, we have

$$
\mathbb{E}\left[\left\|\left(\nabla_{\mathbf{yy}}^2 g^k - \nabla_{\mathbf{yy}}^2 g_\star^k\right)\mathbf{z}_\star^k + \left(\nabla_{\mathbf{y}} f^k - \nabla_{\mathbf{y}} f_\star^k\right)\right\|^2\right] \le 2\left(L_f^2 + L_g^2\rho^2\right)\mathbb{E}\left[\left\|\mathbf{y}^k - \mathbf{y}_\star^k\right\|^2\right] .
\tag{198}
$$

Then, using (198) and define $L_2^2 := L_f^2 + L_g^2\rho^2$, we obtain that

$$
\begin{aligned}
&- 2\eta_{\mathbf{z}}^2\mathbb{E}\left[\left\langle \left(\nabla_{\mathbf{yy}}^2 g^k \mathbf{z}_\star^k + \nabla_{\mathbf{y}} f^k\right), \mathbf{D}_{\mathbf{z}}^k - \hat{\mathbf{D}}_{\mathbf{z}}^k \right\rangle\right] \\
&\le 2\eta_{\mathbf{z}}^2 L_2^2\mathbb{E}\left[\left\|\mathbf{y}^k - \mathbf{y}_\star^k\right\|^2\right] + \eta_{\mathbf{z}}^2\mathbb{E}\left[\left\|\xi_{\mathbf{z}}^k\right\|^2\right] ,
\end{aligned}
\tag{199}
$$

where the inequality uses $\nabla_{yy}^2 g_\star^k \mathbf{z}_\star^k + \nabla_{\mathbf{y}} f_\star^k = 0$ and unbiased estimation of $\mathbf{D}_{\mathbf{z}}^k$.

For the third term, using Young's inequality $2\langle a, b\rangle \le ca^2 + \frac{1}{c}b^2$ and $\left\|I - \eta_{\mathbf{z}}\nabla^2_{\mathbf{yy}}g^k\right\| \le 1 - \eta_{\mathbf{z}}\mu_g$, we have

$$
-2\eta_{\mathbf{z}}\mathbb{E}\left[\left\langle\left(\nabla^2_{\mathbf{yy}}g^k\mathbf{z}^k_\star + \nabla_{\mathbf{y}}f^k\right), \left(\mathbf{z}^k - \mathbf{z}^k_\star\right) - \eta_{\mathbf{z}}\nabla^2_{\mathbf{yy}}g^k\left(\mathbf{z}^k - \mathbf{z}^k_\star\right)\right\rangle\right]
$$
$$
\le \frac{\eta_{\mathbf{z}}\mu_g}{2(1 - \eta_{\mathbf{z}}\mu_g)}\mathbb{E}\left[\left\|\left(\mathbf{z}^k - \mathbf{z}^k_\star\right) - \eta_{\mathbf{z}}\nabla^2_{\mathbf{yy}}g^k\left(\mathbf{z}^k - \mathbf{z}^k_\star\right)\right\|^2\right] + \frac{2\eta_{\mathbf{z}}}{\mu_g}(1 - \eta_{\mathbf{z}}\mu_g)\mathbb{E}\left[\left\|\nabla^2_{\mathbf{yy}}g^k\mathbf{z}^k_\star + \nabla_{\mathbf{y}}f^k\right\|^2\right]
$$
$$
\le \frac{\eta_{\mathbf{z}}\mu_g(1 - \eta_{\mathbf{z}}\mu_g)}{2}\mathbb{E}\left[\left\|\mathbf{z}^k - \mathbf{z}^k_\star\right\|^2\right] + 4\frac{\eta_{\mathbf{z}}}{\mu_g}(1 - \eta_{\mathbf{z}}\mu_g)L^2_2\mathbb{E}\left[\left\|\mathbf{y}^k - \mathbf{y}^k_\star\right\|^2\right],
$$
(200)

where the two inequality uses Young's inequality, $\left\|I - \eta_{\mathbf{z}}\nabla^2_{\mathbf{yy}}g^k\right\| \le 1 - \eta_{\mathbf{z}}\mu_g$ and (198).

Substituting the above inequalities in to (196), we obtain that

$$
\mathbb{E}\left[\left\|\mathbf{z}^{k+1} - \mathbf{z}^k_\star\right\|^2\right]
$$
$$
\le \mathbb{E}\left[\left\|\mathbf{z}^k - \mathbf{z}^k_\star - \eta_{\mathbf{z}}\nabla^2_{yy}g^k(\mathbf{z}^k - \mathbf{z}^k_\star)\right\|^2\right] + \eta^2_{\mathbf{z}}\mathbb{E}\left[\left\|\left(\nabla^2_{\mathbf{yy}}g^k\mathbf{z}^k_\star + \nabla_{\mathbf{y}}f^k\right)\right\|^2\right] + \eta^2_{\mathbf{z}}\mathbb{E}\left[\left\|\mathbb{E}\left(\mathbf{D}^k_{\mathbf{z}}\right) - \hat{\mathbf{D}}^k_{\mathbf{z}}\right\|^2\right]
$$
$$
+ (1 - \eta_{\mathbf{z}}\mu_g)\eta_{\mathbf{z}}\mu_g\mathbb{E}\left[\left\|\mathbf{z}^k - \mathbf{z}^k_\star\right\|^2\right] + (2\frac{\eta_{\mathbf{z}}}{\mu_g} - \eta^2_{\mathbf{z}})\mathbb{E}\left[\left\|\xi^k_{\mathbf{z}}\right\|^2\right] + (2\frac{\eta_{\mathbf{z}}}{\mu_g} - \eta^2_{\mathbf{z}}) \cdot 2L^2_2\mathbb{E}\left[\left\|\mathbf{y}^k - \mathbf{y}^k_\star\right\|^2\right]
$$
$$
\le (1 - \eta_{\mathbf{z}}\mu_g)\mathbb{E}\left[\left\|\mathbf{z}^k - \mathbf{z}^k_\star\right\|^2\right] + \frac{4\eta_{\mathbf{z}}L^2_2}{\mu_g}\mathbb{E}\left[\left\|\mathbf{y}^k - \mathbf{y}^k_\star\right\|^2\right] + \left(\eta^2_{\mathbf{z}} + \frac{2\eta_{\mathbf{z}}}{\mu_g}\right)\mathbb{E}\left[\left\|\xi^k_{\mathbf{z}}\right\|^2\right],
$$
(201)

where the second inequality uses (198) and $\left\|I - \eta_{\mathbf{z}}\nabla^2_{\mathbf{yy}}g^k\right\| \le 1 - \eta_{\mathbf{z}}\mu_g$.

For the second part in the R.H.S of (195), using Lemma 4, we have

$$
\mathbb{E}\left[\left\|\mathbf{z}^{k+1}_\star - \mathbf{z}^k_\star\right\|^2\right] \le L^2_{\mathbf{z}^\star}\eta^2_{\mathbf{x}}\mathbb{E}\left[\left\|\hat{\mathbf{D}}^k_{\mathbf{x}}\right\|^2\right].
$$
(202)

For the third part in the R.H.S of (195), using Young's inequality and Lemma 4, we have

$$
-2\mathbb{E}\left[\left\langle\mathbf{z}^k - \mathbf{z}^k_\star, \nabla\mathbf{z}^\star\left(\mathbf{x}^k\right)\left(\mathbf{x}^{k+1} - \mathbf{x}^k\right)\right\rangle\right]
$$
$$
= -2\mathbb{E}\left[\left\langle\mathbf{z}^k - \mathbf{z}^k_\star, \nabla\mathbf{z}^\star\left(\mathbf{x}^k\right)\left(\mathbf{D}^k_{\mathbf{x}} + \xi^k_{\mathbf{x}}\right)\right\rangle\right]
$$
$$
\le \frac{\eta_{\mathbf{z}}\mu_g}{2}\mathbb{E}\left[\left\|\mathbf{z}^k - \mathbf{z}^k_\star\right\|^2\right] + \frac{4\eta^2_{\mathbf{x}}}{\eta_{\mathbf{z}}\mu_g}L^2_{\mathbf{z}^\star}\mathbb{E}\left[\left\|\mathbb{E}(\mathbf{D}^k_{\mathbf{x}})\right\|^2\right] + \frac{4\eta^2_{\mathbf{x}}}{\eta_{\mathbf{z}}\mu_g}L^2_{\mathbf{z}^\star}\mathbb{E}\left[\left\|\xi^k_{\mathbf{x}}\right\|^2\right],
$$
(203)

where the last inequality uses $2ab \le \frac{1}{c}a^2 + cb^2$ and Assumption 3.

For the fourth part in the R.H.S of (195), we have

$$
-2\mathbb{E}\left[\left\langle\mathbf{z}^k - \mathbf{z}^k_\star, \mathbf{z}^{k+1}_\star - \mathbf{z}^k_\star - \nabla\mathbf{z}^\star\left(\mathbf{x}^k\right)\left(\mathbf{x}^{k+1} - \mathbf{x}^k\right)\right\rangle\right]
$$
$$
\le 2\mathbb{E}\left[\left\|\mathbf{z}^k - \mathbf{z}^k_\star\right\| \cdot \left\|\mathbf{z}^{k+1}_\star - \mathbf{z}^k_\star - \nabla\mathbf{z}^\star\left(\mathbf{x}^k\right)\left(\mathbf{x}^{k+1} - \mathbf{x}^k\right)\right\|\right]
$$
$$
\le L_{\mathbf{zx}}\mathbb{E}\left[\left\|\mathbf{z}^k - \mathbf{z}^k_\star\right\|\left\|\mathbf{x}^{k+1} - \mathbf{x}^k\right\|^2\right]
$$
$$
\le 2L_{\mathbf{zx}}\rho\eta^2_{\mathbf{x}}\mathbb{E}\left[\left\|\hat{\mathbf{D}}^k_{\mathbf{x}}\right\|^2\right].
$$
(204)

The first inequality is derived using Cauchy's inequality. The second inequality is derived using Taylor's expansion, and Lemma 5, the third inequality uses the boundness of $\mathbf{z}$, $\|\mathbf{z}\| \le \rho$.

For the fifth part in the R.H.S of (195), by Young's inequality and Lemma 4, we have

$$
2\eta_{\mathbf{z}}\mathbb{E}\left[\left\langle\hat{\mathbf{D}}^k_{\mathbf{z}}, \mathbf{z}^{k+1}_\star - \mathbf{z}^k_\star\right\rangle\right] \le \eta^2_{\mathbf{z}}\mathbb{E}\left[\left\|\hat{\mathbf{D}}^k_{\mathbf{z}}\right\|^2\right] + L^2_{\mathbf{z}^\star}\eta^2_{\mathbf{x}}\mathbb{E}\left[\left\|\hat{\mathbf{D}}^k_{\mathbf{x}}\right\|^2\right].
$$
(205)

Combining the above inequalities and using the Cauchy–Schwarz inequality and Lemma 24, we have

$$\mathbb{E}\left[\left\|\mathbf{z}^{k+1} - \mathbf{z}_\star^{k+1}\right\|^2\right]$$

$$\leq (1 - \frac{\eta_{\mathbf{z}}\mu_g}{2})\mathbb{E}\left[\left\|\mathbf{z}^k - \mathbf{z}_\star^k\right\|^2\right] + \frac{4\eta_{\mathbf{z}}L_2^2}{\mu_g}\mathbb{E}\left[\left\|\mathbf{y}^k - \mathbf{y}_\star^k\right\|^2\right] + \left(\zeta_{\mathbf{z}}^k + \frac{2\eta_{\mathbf{z}}}{\mu_g}\right)\mathbb{E}\left[\left\|\xi_{\mathbf{z}}^k\right\|^2\right]$$

$$+ (4L_{\mathbf{z}^\star}^2\eta_{\mathbf{x}}^2 + 4L_{zx}\rho\eta_{\mathbf{x}}^2 + \frac{4\eta_{\mathbf{x}}^2}{\eta_{\mathbf{z}}\mu_g}L_{\mathbf{z}^\star}^2)\left(\mathbb{E}\left[\left\|\mathbb{E}(\mathbf{D}_{\mathbf{x}}^k)\right\|^2\right] + \mathbb{E}\left[\left\|\xi_{\mathbf{x}}^k\right\|^2\right]\right) + \eta_{\mathbf{z}}^2\mathbb{E}\left[\left\|\hat{\mathbf{D}}_{\mathbf{z}}^k\right\|^2\right]$$

$$\leq \left(1 - \frac{\eta_{\mathbf{z}}\mu_g}{2}\right)\mathbb{E}\left[\left\|\mathbf{z}^k - \mathbf{z}_\star^k\right\|^2\right] + \left(\frac{4\eta_{\mathbf{z}}L_2^2}{\mu_g} + 6\eta_{\mathbf{z}}^2L_2^2\right)\mathbb{E}\left[\left\|\mathbf{y}^k - \mathbf{y}_\star^k\right\|^2\right] + 6\eta_{\mathbf{z}}^2L_g^2\mathbb{E}\left[\left\|\mathbf{z}^k - \mathbf{z}_\star^k\right\|^2\right]$$

$$+ \left(4L_{\mathbf{z}^\star}^2\eta_{\mathbf{x}}^2 + 4L_{zx}\rho\eta_{\mathbf{x}}^2 + \frac{4\eta_{\mathbf{x}}^2}{\eta_{\mathbf{z}}\mu_g}L_{\mathbf{z}^\star}^2\right)\left(\mathbb{E}\left[\left\|\mathbb{E}(\mathbf{D}_{\mathbf{x}}^k)\right\|^2\right] + \mathbb{E}\left[\left\|\xi_{\mathbf{x}}^k\right\|^2\right]\right) + \left(3\eta_{\mathbf{z}}^2 + \frac{2\eta_{\mathbf{z}}}{\mu_g}\right)\mathbb{E}\left[\left\|\xi_{\mathbf{z}}^k\right\|^2\right]$$

$$+ \left(2\eta_{\mathbf{z}}^2 + 4L_{\mathbf{z}^\star}^2\eta_{\mathbf{x}}^2 + 4L_{zx}\rho\eta_{\mathbf{x}}^2\right)\frac{\sigma_1^2}{(1-\delta)n},$$

(206)

where the second inequality uses Lemma 24. □

### J.1.3 PROOF OF LEMMA 12

We now restate the lemma 12 and present its proof.

**Lemma 12.** *Recall that the upper-level loss function is $L_{\nabla\Phi}$-smooth, we have*

$$\mathbb{E}\left[\Phi_{\mathcal{H}}\left(\mathbf{x}^{k+1}\right)\right] \leq \mathbb{E}\left[\Phi_{\mathcal{H}}\left(\mathbf{x}^k\right)\right] - \frac{\eta_{\mathbf{x}}}{4}\mathbb{E}\left[\left\|\Phi_{\mathcal{H}}\left(\mathbf{x}^k\right)\right\|^2\right] + \frac{\eta_x}{2}\mathbb{E}\left[\left\|\nabla\Phi_{\mathcal{H}}(\mathbf{x}^k) - \mathbb{E}(\mathbf{D}_{\mathbf{x}}^k)\right\|^2\right]$$

$$+ \left(L_{\nabla\Phi}\eta_{\mathbf{x}}^2 - \frac{\eta_{\mathbf{x}}}{2}\right)\mathbb{E}\left[\left\|\mathbb{E}(\mathbf{D}_{\mathbf{x}}^k)\right\|^2\right] + \left(4\eta_{\mathbf{x}} + L_{\nabla\Phi}\eta_{\mathbf{x}}^2\right)\mathbb{E}\left[\left\|\xi_{\mathbf{x}}^k\right\|^2\right] + \frac{L_{\nabla\Phi}\eta_{\mathbf{x}}^2\sigma_1^2}{(1-\delta)n},$$

(207)

*where $\sigma_1^2 = \left(1 + \rho^2\right)\sigma^2$.*

*Proof.* By using $L_{\nabla\Phi}$-smoothness of upper-level loss function $\Phi$, we obtain

$$\mathbb{E}\left[\Phi_{\mathcal{H}}\left(\mathbf{x}^{k+1}\right)\right] \leq \mathbb{E}\left[\Phi_{\mathcal{H}}\left(\mathbf{x}^k\right)\right] + \mathbb{E}\left[\langle\nabla\Phi_{\mathcal{H}}\left(\mathbf{x}^k\right), \mathbf{x}^{k+1} - \mathbf{x}^k\rangle\right] + \frac{L_{\nabla\Phi}}{2}\mathbb{E}\left[\left\|\mathbf{x}^{k+1} - \mathbf{x}^k\right\|^2\right].$$

(208)

Using $\mathbf{x}^{k+1} - \mathbf{x}^k = -\eta_{\mathbf{x}}\mathbf{D}_{\mathbf{x}}^k - \eta_{\mathbf{x}}\xi_{\mathbf{x}}^k$ and Assumption 3, we obtain that

$$\mathbb{E}\left[\Phi_{\mathcal{H}}\left(\mathbf{x}^{k+1}\right)\right] \leq \mathbb{E}\left[\Phi_{\mathcal{H}}\left(\mathbf{x}^k\right)\right] - \eta_{\mathbf{x}}\mathbb{E}\left[\langle\nabla\Phi_{\mathcal{H}}\left(\mathbf{x}^k\right), \mathbf{D}_{\mathbf{x}}^k + \xi_{\mathbf{x}}^k\rangle\right] + \frac{L_{\nabla\Phi}\eta_{\mathbf{x}}^2}{2}\mathbb{E}\left[\left\|\hat{\mathbf{D}}_{\mathbf{x}}^k\right\|^2\right]$$

$$\leq \mathbb{E}\left[\Phi_{\mathcal{H}}\left(\mathbf{x}^k\right)\right] - \eta_{\mathbf{x}}\mathbb{E}\left[\langle\nabla\Phi_{\mathcal{H}}\left(\mathbf{x}^k\right), \mathbb{E}(\mathbf{D}_{\mathbf{x}}^k) + \xi_{\mathbf{x}}^k\rangle\right] + \frac{L_{\nabla\Phi}\eta_{\mathbf{x}}^2}{2}\mathbb{E}\left[\left\|\hat{\mathbf{D}}_{\mathbf{x}}^k\right\|^2\right]$$

$$\leq \mathbb{E}\left[\Phi_{\mathcal{H}}\left(\mathbf{x}^k\right)\right] - \eta_{\mathbf{x}}\mathbb{E}\left[\langle\nabla\Phi_{\mathcal{H}}\left(\mathbf{x}^k\right), \mathbb{E}(\mathbf{D}_{\mathbf{x}}^k)\rangle\right]$$

$$- \eta_{\mathbf{x}}\mathbb{E}\left[\langle\nabla\Phi_{\mathcal{H}}\left(\mathbf{x}^k\right), \xi_{\mathbf{x}}^k\rangle\right] + \frac{L_{\nabla\Phi}\eta_{\mathbf{x}}^2}{2}\mathbb{E}\left[\left\|\hat{\mathbf{D}}_{\mathbf{x}}^k\right\|^2\right],$$

(209)

where the first inequality uses the Assumption 3 that $\mathbf{D}_{\mathbf{x}}^k$ is the unbiased estimation of $\mathbb{E}\left(\mathbf{D}_{\mathbf{x}}^k\right)$.

Using uses $\langle a, b\rangle = \frac{1}{2}(a^2 + b^2 - (a+b)^2)$, we obtain that

$$- \eta_{\mathbf{x}}\mathbb{E}\left[\langle\nabla\Phi_{\mathcal{H}}\left(\mathbf{x}^k\right), \mathbb{E}(\mathbf{D}_{\mathbf{x}}^k)\rangle\right]$$

$$= \frac{\eta_x}{2}\left(\mathbb{E}\left[\left\|\nabla\Phi_{\mathcal{H}}(\mathbf{x}^k) - \mathbb{E}(\mathbf{D}_{\mathbf{x}}^k)\right\|^2\right] - \mathbb{E}\left[\left\|\nabla\Phi_{\mathcal{H}}(\mathbf{x}^k)\right\|^2\right] - \mathbb{E}\left[\left\|\mathbb{E}(\mathbf{D}_{\mathbf{x}}^k)\right\|^2\right]\right).$$

(210)

Using the fact that $2ab \leq \frac{1}{c}a^2 + cb^2$, we obtain that

$$-\eta_{\mathbf{x}}\mathbb{E}\left[\langle\nabla\Phi_{\mathcal{H}}\left(\mathbf{x}^k\right), \xi_{\mathbf{x}}^k\rangle\right] \leq \eta_{\mathbf{x}}\mathbb{E}\left[\left\|\nabla\Phi_{\mathcal{H}}\left(\mathbf{x}^k\right)\right\| \cdot \left\|\xi_{\mathbf{x}}^k\right\|\right]$$

$$\leq \frac{\eta_{\mathbf{x}}}{4}\mathbb{E}\left[\left\|\nabla\Phi_{\mathcal{H}}\left(\mathbf{x}^k\right)\right\|^2\right] + 4\eta_{\mathbf{x}}\mathbb{E}\left[\left\|\xi_{\mathbf{x}}^k\right\|^2\right].$$

(211)

Substituting from (211), (210) in (209), we obtain that

$$
\begin{aligned}
\mathbb{E}\left[\Phi_{\mathcal{H}}\left(\mathbf{x}^{k+1}\right)\right] \leq{} & \mathbb{E}\left[\Phi_{\mathcal{H}}\left(\mathbf{x}^{k}\right)\right] - \frac{\eta_{\mathbf{x}}}{4}\mathbb{E}\left[\left\|\Phi_{\mathcal{H}}\left(\mathbf{x}^{k}\right)\right\|^2\right] + \frac{\eta_x}{2}\mathbb{E}\left[\left\|\nabla\Phi_{\mathcal{H}}(\mathbf{x}^{k}) - \mathbb{E}(\mathbf{D}_{\mathbf{x}}^{k})\right\|^2\right] \\
& - \frac{\eta_{\mathbf{x}}}{2}\mathbb{E}\left[\left\|\mathbb{E}(\mathbf{D}_{\mathbf{x}}^{k})\right\|^2\right] + \frac{L_{\nabla\Phi}\eta_{\mathbf{x}}^2}{2}\mathbb{E}\left[\left\|\hat{\mathbf{D}}_{\mathbf{x}}^{k}\right\|^2\right] + 4\eta_{\mathbf{x}}\mathbb{E}\left[\left\|\xi_{\mathbf{x}}^{k}\right\|^2\right] \\
\leq{} & \mathbb{E}\left[\Phi_{\mathcal{H}}\left(\mathbf{x}^{k}\right)\right] - \frac{\eta_{\mathbf{x}}}{4}\mathbb{E}\left[\left\|\Phi_{\mathcal{H}}\left(\mathbf{x}^{k}\right)\right\|^2\right] + \frac{\eta_x}{2}\mathbb{E}\left[\left\|\nabla\Phi_{\mathcal{H}}(\mathbf{x}^{k}) - \mathbb{E}(\mathbf{D}_{\mathbf{x}}^{k})\right\|^2\right] \\
& + \left(L_{\nabla\Phi}\eta_{\mathbf{x}}^2 - \frac{\eta_{\mathbf{x}}}{2}\right)\mathbb{E}\left[\left\|\mathbb{E}(\mathbf{D}_{\mathbf{x}}^{k})\right\|^2\right] + \left(4\eta_{\mathbf{x}} + L_{\nabla\Phi}\eta_{\mathbf{x}}^2\right)\mathbb{E}\left[\left\|\xi_{\mathbf{x}}^{k}\right\|^2\right] + \frac{L_{\nabla\Phi}\eta_{\mathbf{x}}^2\sigma_1^2}{(1-\delta)n},
\end{aligned}
\tag{212}
$$

where the last inequality uses Assumption 3 and Cauchy–Schwarz inequality.

$\square$

### J.1.4 OTHER USEFUL LEMMAS

**Lemma 23.** *Under Assumptions 3, the following inequality holds for BR-FedBi:*

$$
\mathbb{E}\left[\left\|\mathbb{E}(\mathbf{D}_{\mathbf{y}}^{k}) - \hat{\mathbf{D}}_{\mathbf{y}}^{k}\right\|^2\right] \leq \mathbb{E}\left[\left\|\xi_{\mathbf{y}}^{k}\right\|^2\right] + \frac{1}{((1-\delta)n)}\sigma^2,
\tag{213}
$$

$$
\mathbb{E}\left[\left\|\mathbb{E}\left(\mathbf{D}_{\mathbf{z}}^{k}\right) - \hat{\mathbf{D}}_{\mathbf{z}}^{k}\right\|^2\right] \leq \mathbb{E}\left[\left\|\xi_{\mathbf{z}}^{k}\right\|^2\right] + \frac{1}{((1-\delta)n)}\sigma_1^2.
\tag{214}
$$

*Proof.* **Analysis for** $\mathbb{E}\left[\left\|\mathbb{E}(\mathbf{D}_{\mathbf{y}}^{k}) - \hat{\mathbf{D}}_{\mathbf{y}}^{k}\right\|^2\right]$. We decompose the term into two components: the error due to the approximation $\hat{\mathbf{D}}_{\mathbf{y}}^{k}$ and the variance of $\mathbf{D}_{\mathbf{y}}^{k}$. Specifically, we rewrite the term as follows:

$$
\begin{aligned}
\mathbb{E}\left[\left\|\mathbb{E}(\mathbf{D}_{\mathbf{y}}^{k}) - \hat{\mathbf{D}}_{\mathbf{y}}^{k}\right\|^2\right] &= \mathbb{E}\left[\left\|\left(\hat{\mathbf{D}}_{\mathbf{y}}^{k} - \mathbf{D}_{\mathbf{y}}^{k}\right) + \left(\mathbf{D}_{\mathbf{y}}^{k} - \mathbb{E}(\mathbf{D}_{\mathbf{y}}^{k})\right)\right\|^2\right] \\
&\leq \mathbb{E}\left[\left\|\hat{\mathbf{D}}_{\mathbf{y}}^{k} - \mathbf{D}_{\mathbf{y}}^{k}\right\|^2\right] + \mathbb{E}\left[\left\|\mathbf{D}_{\mathbf{y}}^{k} - \mathbb{E}(\mathbf{D}_{\mathbf{y}}^{k})\right\|^2\right] \\
&\leq \mathbb{E}\left[\left\|\xi_{\mathbf{y}}^{k}\right\|^2\right] + \frac{1}{(1-\delta)n}\sigma^2,
\end{aligned}
$$

where the first inequality uses the fact that $\mathbb{E}[\mathbf{D}_{\mathbf{y}}^{k} - \mathbb{E}(\mathbf{D}_{\mathbf{y}}^{k})] = 0$, ensuring the cross-term vanishes. The second inequality applies Lemma 7 to bound the variance term.

**Analysis for** $\mathbb{E}\left[\left\|\mathbb{E}(\mathbf{D}_{\mathbf{x}}^{k}) - \hat{\mathbf{D}}_{\mathbf{x}}^{k}\right\|^2\right]$. Similarly, we have

$$
\begin{aligned}
\mathbb{E}\left[\left\|\mathbb{E}(\mathbf{D}_{\mathbf{x}}^{k}) - \hat{\mathbf{D}}_{\mathbf{x}}^{k}\right\|^2\right] &= \mathbb{E}\left[\left\|\left(\hat{\mathbf{D}}_{\mathbf{x}}^{k} - \mathbf{D}_{\mathbf{x}}^{k}\right) + \left(\mathbf{D}_{\mathbf{x}}^{k} - \mathbb{E}(\mathbf{D}_{\mathbf{x}}^{k})\right)\right\|^2\right] \\
&\leq \mathbb{E}\left[\left\|\hat{\mathbf{D}}_{\mathbf{x}}^{k} - \mathbf{D}_{\mathbf{x}}^{k}\right\|^2\right] + \mathbb{E}\left[\left\|\mathbf{D}_{\mathbf{x}}^{k} - \mathbb{E}(\mathbf{D}_{\mathbf{x}}^{k})\right\|^2\right] \\
&\leq \mathbb{E}\left[\left\|\xi_{\mathbf{x}}^{k}\right\|^2\right] + \frac{1}{(1-\delta)n}\sigma_1^2,
\end{aligned}
$$

where the first inequality uses $\mathbb{E}[\mathbf{D}_{\mathbf{x}}^{k} - \mathbb{E}(\mathbf{D}_{\mathbf{x}}^{k})] = 0$ and the second inequality uses Lemma 7.

**Analysis for** $\mathbb{E}\left[\left\|\mathbb{E}(\mathbf{D}_{\mathbf{z}}^k) - \hat{\mathbf{D}}_{\mathbf{z}}^k\right\|^2\right]$. Similarly, we have

$$
\mathbb{E}\left[\left\|\mathbb{E}(\mathbf{D}_{\mathbf{z}}^k) - \hat{\mathbf{D}}_{\mathbf{z}}^k\right\|^2\right] = \mathbb{E}\left[\left\|\left(\hat{\mathbf{D}}_{\mathbf{z}}^k - \mathbf{D}_{\mathbf{z}}^k\right) + \left(\mathbf{D}_{\mathbf{z}}^k - \mathbb{E}(\mathbf{D}_{\mathbf{z}}^k)\right)\right\|^2\right]
$$

$$
\leq \mathbb{E}\left[\left\|\hat{\mathbf{D}}_{\mathbf{z}}^k - \mathbf{D}_{\mathbf{z}}^k\right\|^2\right] + \mathbb{E}\left[\left\|\mathbf{D}_{\mathbf{z}}^k - \mathbb{E}(\mathbf{D}_{\mathbf{z}}^k)\right\|^2\right]
$$

$$
\leq \mathbb{E}\left[\left\|\xi_{\mathbf{z}}^k\right\|^2\right] + \frac{1}{(1-\delta)n}\sigma_1^2,
$$

where the first inequality uses $\mathbb{E}[\mathbf{D}_{\mathbf{z}}^k - \mathbb{E}(\mathbf{D}_{\mathbf{z}}^k)] = 0$ and the second inequality uses Lemma 7. $\qquad\square$

**Lemma 24** (Bounded Second Moment). *Suppose Assumption 1, 2 and 3 hold, if $\rho \geq C_f/\mu_g$. For each step $k \in [K]$, we have:*

$$
\mathbb{E}\left[\left\|\hat{\mathbf{D}}_{\mathbf{x}}^k\right\|^2\right] \leq 2\mathbb{E}\left[\left\|\mathbb{E}\left(\mathbf{D}_{\mathbf{x}}^k\right)\right\|^2\right] + 2\mathbb{E}\left[\left\|\xi_{\mathbf{x}}^k\right\|^2\right] + 2\frac{\sigma_1^2}{(1-\delta)n},
$$

$$
\mathbb{E}\left[\left\|\hat{\mathbf{D}}_{\mathbf{y}}^k\right\|^2\right] \leq 2L_g^2\mathbb{E}\left[\left\|\mathbf{y}^k - \mathbf{y}_\star^k\right\|^2\right] + 2\mathbb{E}\left[\left\|\xi_{\mathbf{y}}^k\right\|^2\right] + 2\frac{\sigma^2}{(1-\delta)n}, \tag{215}
$$

$$
\mathbb{E}\left[\left\|\hat{\mathbf{D}}_{\mathbf{z}}^k\right\|^2\right] \leq 6L_2^2\mathbb{E}\left[\left\|\mathbf{y}^k - \mathbf{y}_\star^k\right\|^2\right] + 6L_g^2\mathbb{E}\left[\left\|\mathbf{z}^k - \mathbf{z}_\star^k\right\|^2\right] + 2\mathbb{E}\left[\left\|\xi_{\mathbf{z}}^k\right\|^2\right] + 2\frac{\sigma_1^2}{(1-\delta)n},
$$

*where $\sigma_1^2 = (1+\rho^2)\sigma^2$.*

*Proof.* **Analysis for** $\mathbb{E}\left[\left\|\hat{\mathbf{D}}_{\mathbf{x}}^k\right\|^2\right]$. We decompose the second moment into two components: the true gradient term $\mathbf{D}_{\mathbf{x}}^k$ and the error term $\xi_{\mathbf{x}}^k = \hat{\mathbf{D}}_{\mathbf{x}}^k - \mathbf{D}_{\mathbf{x}}^k$. Using the decomposition, we can write:

$$
\mathbb{E}\left[\left\|\hat{\mathbf{D}}_{\mathbf{x}}^k\right\|^2\right] = \mathbb{E}\left[\left\|\mathbf{D}_{\mathbf{x}}^k + \hat{\mathbf{D}}_{\mathbf{x}}^k - \mathbf{D}_{\mathbf{x}}^k\right\|^2\right]
$$

$$
\leq 2\mathbb{E}\left[\left\|\mathbf{D}_{\mathbf{x}}^k\right\|^2\right] + 2\mathbb{E}\left[\left\|\xi_{\mathbf{x}}^k\right\|^2\right] \tag{216}
$$

$$
\leq 2\mathbb{E}\left[\left\|\mathbb{E}\left(\mathbf{D}_{\mathbf{x}}^k\right)\right\|^2\right] + 2\mathbb{E}\left[\left\|\xi_{\mathbf{x}}^k\right\|^2\right] + 2\frac{\sigma_1^2}{(1-\delta)n},
$$

where the first inequality follows from the Cauchy-Schwarz inequality.

**Analysis for** $\mathbb{E}\left[\left\|\hat{\mathbf{D}}_{\mathbf{y}}^k\right\|^2\right]$. We decompose the second moment into two components: the true gradient term $\mathbf{D}_{\mathbf{y}}^k$ and the error term $\xi_{\mathbf{y}}^k = \hat{\mathbf{D}}_{\mathbf{y}}^k - \mathbf{D}_{\mathbf{y}}^k$. Using the decomposition, we can write:

$$
\mathbb{E}\left[\left\|\hat{\mathbf{D}}_{\mathbf{y}}^k\right\|^2\right] = \mathbb{E}\left[\left\|\hat{\mathbf{D}}_{\mathbf{y}}^k - \mathbf{D}_{\mathbf{y}}^k + \mathbf{D}_{\mathbf{y}}^k\right\|^2\right]
$$

$$
\leq 2\mathbb{E}\left[\left\|\mathbb{E}\left(\mathbf{D}_{\mathbf{y}}^k\right)\right\|^2\right] + 2\mathbb{E}\left[\left\|\xi_{\mathbf{y}}^k\right\|^2\right] + 2\frac{\sigma^2}{(1-\delta)n} \tag{217}
$$

$$
\leq 2L_g^2\mathbb{E}\left[\left\|\mathbf{y}^k - \mathbf{y}_\star^k\right\|^2\right] + 2\mathbb{E}\left[\left\|\xi_{\mathbf{y}}^k\right\|^2\right] + 2\frac{\sigma^2}{(1-\delta)n},
$$

where the first inequality uses the Cauchy-Schwarz inequality, and the second inequality uses

$$
\mathbb{E}\left[\left\|\mathbb{E}\left(\mathbf{D}_{\mathbf{y}}^k\right)\right\|^2\right] = \mathbb{E}\left[\left\|\nabla_{\mathbf{y}}g^k - \nabla_{\mathbf{y}}g_\star^k\right\|^2\right] \leq L_g^2\mathbb{E}\left[\left\|\mathbf{y}^k - \mathbf{y}_\star^k\right\|^2\right]. \tag{218}
$$

**Analysis for** $\mathbb{E}\left[\left\|\hat{\mathbf{D}}_{\mathbf{z}}^k\right\|^2\right]$. We decompose the second moment into two components: the true gradient term $\mathbf{D}_{\mathbf{z}}^k$ and the error term $\xi_{\mathbf{z}}^k = \hat{\mathbf{D}}_{\mathbf{z}}^k - \mathbf{D}_{\mathbf{z}}^k$. Using the decomposition, we can write:

$$
\begin{aligned}
\mathbb{E}\left[\left\|\hat{\mathbf{D}}_{\mathbf{z}}^k\right\|^2\right] &= \mathbb{E}\left[\left\|\hat{\mathbf{D}}_{\mathbf{z}}^k - \mathbf{D}_{\mathbf{z}}^k + \mathbf{D}_{\mathbf{z}}^k\right\|^2\right] \\
&\leq 2\mathbb{E}\left[\|\mathbb{E}\left(\mathbf{D}_{\mathbf{z}}^k\right)\|^2\right] + 2\mathbb{E}\left[\left\|\xi_{\mathbf{z}}^k\right\|^2\right] + 2\frac{\sigma_1^2}{(1-\delta)n} \\
&\leq 6L_2^2\mathbb{E}\left[\left\|\mathbf{y}^k - \mathbf{y}_\star^k\right\|^2\right] + 6L_g^2\mathbb{E}\left[\left\|\mathbf{z}^k - \mathbf{z}_\star^k\right\|^2\right] + 2\mathbb{E}\left[\left\|\xi_{\mathbf{z}}^k\right\|^2\right] + 2\frac{\sigma_1^2}{(1-\delta)n},
\end{aligned}
\tag{219}
$$

where the first inequality uses the Cauchy-Schwarz inequality, and the second inequality uses the following inequality,

$$
\begin{aligned}
\mathbb{E}\left[\|\mathbb{E}\left(\mathbf{D}_{\mathbf{z}}^k\right)\|^2\right] &= \mathbb{E}\left[\left\|\left(\nabla_{\mathbf{yy}}^2 g^k - \nabla_{\mathbf{yy}}^2 g_\star^k\right)\mathbf{z}^k + \nabla_{\mathbf{yy}}^2 g_\star^k\left(\mathbf{z}^k - \mathbf{z}_\star^k\right) + \left(\nabla_{\mathbf{y}} f^k - \nabla_{\mathbf{y}} f_\star^k\right)\right\|^2\right] \\
&\leq 3\left(L_f^2 + L_g^2\rho^2\right)\mathbb{E}\left[\left\|\mathbf{y}^k - \mathbf{y}_\star^k\right\|^2\right] + 3L_g^2\mathbb{E}\left[\left\|\mathbf{z}^k - \mathbf{z}_\star^k\right\|^2\right],
\end{aligned}
\tag{220}
$$

where the equality in (220) uses $\nabla_{\mathbf{yy}}^2 g_\star^k z_\star^k + \nabla_{\mathbf{y}} f_\star^k = 0$. $\qquad\square$

## J.2 PROOF OF SUPPORTING LEMMAS FOR BR-FEDBIM

### J.2.1 PROOF OF LEMMA 13

**Lemma 13.** Suppose Assumptions 1-4 hold,and the robust aggregation rule $AGG(\cdot)$ satisfies the $(\delta, \kappa)$-robustness criterion. The aggregation errors in the BR-FedBiM algorithm (Algorithm 1) satisfy the following inequalities:

$$
\begin{aligned}
(1)\quad & \mathbb{E}\left[\left\|\xi_{\mathbf{y}}^k\right\|^2\right] \leq 3\kappa\left(\frac{1-\beta_{\mathbf{y}}}{1+\beta_{\mathbf{y}}}\right)\left(1 + \frac{1}{(1-\delta)n}\right)\sigma^2 + 3\kappa\zeta_g^2, \\
(2)\quad & \mathbb{E}\left[\left\|\xi_{\mathbf{x}}^k\right\|^2\right] \leq 6\kappa\left(\frac{1-\beta_{\mathbf{x}}}{1+\beta_{\mathbf{x}}}\right)\left(1 + \frac{1}{(1-\delta)n}\right)\sigma_1^2 + 6\kappa\left(\rho^2\zeta_g^2 + \zeta_f^2\right), \\
(3)\quad & \mathbb{E}\left[\left\|\xi_{\mathbf{z}}^k\right\|^2\right] \leq 6\kappa\left(\frac{1-\beta_{\mathbf{z}}}{1+\beta_{\mathbf{z}}}\right)\left(1 + \frac{1}{(1-\delta)n}\right)\sigma_1^2 + 6\kappa\left(\rho^2\zeta_g^2 + \zeta_f^2\right)
\end{aligned}
$$

where $\sigma_1^2 = \left(1 + \rho^2\right)\sigma^2$ and $\rho = C_f/\mu_g$.

*Proof.* Using Lemma 25 and Definition 2 , we can finish the proof.

$\qquad\square$

**Lemma 25** (bounded aggregation drift). *Suppose Assumptions 1-4 hold. For each step $k \in [K]$, we have:*

$$
\mathbb{E}\left[\frac{1}{|\mathcal{H}|}\sum_{i\in\mathcal{H}}\left\|\mathbf{m}_{\mathbf{y},i}^k - \mathbf{m}_{\mathbf{y}}^k\right\|^2\right] \leq 3\left(\frac{1-\beta_{\mathbf{y}}}{1+\beta_{\mathbf{y}}}\right)\left(1 + \frac{1}{(1-\delta)n}\right)\sigma^2 + 3\zeta_g^2,
$$

$$
\mathbb{E}\left[\frac{1}{|\mathcal{H}|}\sum_{i\in\mathcal{H}}\left\|\mathbf{m}_{\mathbf{x},i}^k - \mathbf{m}_{\mathbf{x}}^k\right\|^2\right] \leq 6\left(\frac{1-\beta_{\mathbf{x}}}{1+\beta_{\mathbf{x}}}\right)\left(1 + \frac{1}{(1-\delta)n}\right)\sigma_1^2 + 6\left(\rho^2\zeta_g^2 + \zeta_f^2\right),
$$

$$
\mathbb{E}\left[\frac{1}{|\mathcal{H}|}\sum_{i\in\mathcal{H}}\left\|\mathbf{m}_{\mathbf{z},i}^k - \mathbf{m}_{\mathbf{z}}^k\right\|^2\right] \leq 6\left(\frac{1-\beta_{\mathbf{z}}}{1+\beta_{\mathbf{z}}}\right)\left(1 + \frac{1}{(1-\delta)n}\right)\sigma_1^2 + 6\left(\rho^2\zeta_g^2 + \zeta_f^2\right).
$$

*Proof.* **Analysis the inequality for y.**

Expanding the sum in momentum, we can obtain

$$
\mathbf{m}_{\mathbf{y},i}^k = \beta_{\mathbf{y}}\mathbf{m}_{\mathbf{y},i}^{k-1} + (1-\beta_{\mathbf{y}})\mathbf{D}_{\mathbf{y},i}^k = (1-\beta_{\mathbf{y}})\sum_{t=1}^k \beta_{\mathbf{y}}^{k-t}\mathbf{D}_{\mathbf{y},i}^k.
$$

Therefore, applying Jensen's inequality, we write

$$
\mathbb{E}\left[\frac{1}{|\mathcal{H}|}\sum_{i\in\mathcal{H}}\left\|\mathbf{m}_{\mathbf{y},i}^k - \mathbf{m}_{\mathbf{y}}^k\right\|^2\right] = (1-\beta_{\mathbf{y}})^2\mathbb{E}\left[\frac{1}{|\mathcal{H}|}\sum_{i\in\mathcal{H}}\left\|\sum_{t=1}^{k}\beta_{\mathbf{y}}^{k-t}\left(\mathbf{D}_{\mathbf{y},i}^t - \mathbf{D}_{\mathbf{y}}^t\right)\right\|^2\right]
$$

$$
\leq 3(1-\beta_{\mathbf{y}})^2\mathbb{E}\left[\frac{1}{|\mathcal{H}|}\sum_{i=H}\underbrace{\left\|\sum_{t=1}^{k}\beta_{\mathbf{y}}^{k-t}\left(\mathbf{D}_{\mathbf{y},i}^t - \mathbb{E}\left(\mathbf{D}_{\mathbf{y},i}^t\right)\right)\right\|^2}_{\Lambda_1^k}\right]
$$

$$
+ 3(1-\beta_{\mathbf{y}})^2\mathbb{E}\left[\frac{1}{|\mathcal{H}|}\sum_{i\in\mathcal{H}}\underbrace{\left\|\sum_{t=1}^{k}\beta_{\mathbf{y}}^{k-t}\left(\mathbf{D}_{\mathbf{y}}^t - \mathbb{E}\left(\mathbf{D}_{\mathbf{y}}^t\right)\right)\right\|^2}_{\Lambda_2^k}\right]
$$

$$
+ 3(1-\beta_{\mathbf{y}})^2\mathbb{E}\left[\frac{1}{|\mathcal{H}|}\sum_{t\in H}\underbrace{\left\|\sum_{t=1}^{k}\beta_{\mathbf{y}}^{k-t}\left(\mathbb{E}\left(\mathbf{D}_{\mathbf{y},i}^t\right) - \mathbb{E}\left(\mathbf{D}_{\mathbf{y}}^t\right)\right)\right\|^2}_{\Lambda_3^k}\right].
$$

(221)

For the first term on R.H.S., we have

$$
\Lambda_1^k = \mathbb{E}\left[\left\|\sum_{t=1}^{k}\beta_{\mathbf{y}}^{k-t}\left(\mathbf{D}_{\mathbf{y},i}^t - \mathbb{E}\left(\mathbf{D}_{\mathbf{y},i}^t\right)\right)\right\|^2\right]
$$

$$
= \mathbb{E}\left[\left\|\sum_{t=1}^{k-1}\beta_{\mathbf{y}}^{k-t}\left(\mathbf{D}_{\mathbf{y},i}^t - \mathbb{E}\left(\mathbf{D}_{\mathbf{y},i}^t\right)\right) + \left(\mathbf{D}_{\mathbf{y},i}^k - \mathbb{E}\left(\mathbf{D}_{\mathbf{y},i}^k\right)\right)\right\|^2\right]
$$

$$
= \mathbb{E}\left[\|\sum_{t=1}^{k-1}\beta_{\mathbf{y}}^{k-t}\left(\mathbf{D}_{\mathbf{y},i}^t\right) - \mathbb{E}\left(\mathbf{D}_{\mathbf{y},i}^t\right)\|^2\right] + \mathbb{E}\left[\|\mathbf{D}_{\mathbf{y},i}^k - \mathbb{E}\left(\mathbf{D}_{\mathbf{y},i}^k\right)\|^2\right]
$$

$$
\leq \beta_{\mathbf{y}}^2\Lambda_1^{k-1} + \sigma^2.
$$

By recursion, we have

$$
\Lambda_1^k \leq \beta_{\mathbf{y}}^{2(k-1)}\Lambda_1^1 + \sigma^2\sum_{l=1}^{k-2}\beta_{\mathbf{y}}^{2l}.
$$

As $\Lambda_1^1 = \mathbb{E}\left[\|\mathbf{D}_{\mathbf{y},i}^1 - \mathbb{E}\left(\mathbf{D}_{\mathbf{y},i}^1\right)\|^2\right] \leq \sigma^2$, we can obtain that

$$
\Lambda_1^k \leq \sigma^2\sum_{l=0}^{k-1}\beta_{\mathbf{y}}^{2l} \leq \frac{\sigma^2}{1-\beta_{\mathbf{y}}^2}.
$$

For the second term of the R.H.S in (221), we denote

$$
\Lambda_2^k := \mathbb{E}\left[\left\|\sum_{t=1}^{k}\beta_{\mathbf{y}}^{t-k}\left(\mathbf{D}_{\mathbf{y}}^t - \mathbb{E}\left(\mathbf{D}_{\mathbf{y}}^t\right)\right)\right\|^2\right].
$$

Similarly, we can obtain that

$$
\Lambda_2^k \leq \frac{\sigma^2}{\left(1-\beta_{\mathbf{y}}^2\right)\left((1-\delta)n\right)}.
$$

For the third term of the R.H.S. in (221), we denote.

$$
\Lambda_3^k := \mathbb{E}\left[\frac{1}{|\mathcal{H}|}\sum_{i\in\mathcal{H}}\|\sum_{t=1}^{k}\beta_{\mathbf{y}}^{k-t}\left(\mathbb{E}(\mathbf{D}_{\mathbf{y},i}^t) - \mathbb{E}\left(\mathbf{D}_{\mathbf{y}}^k\right)\right)\|^2\right].
$$

By applying Jensen's inequality, we have.

$$
\begin{aligned}
\Lambda_3^k &\leq \left(\sum_{t=1}^{k} \beta_{\mathbf{y}}^{k-t}\right) \mathbb{E}\left[\frac{1}{|\mathcal{H}|}\sum_{t\in t}\sum_{t=1}^{k}\beta_{\mathbf{y}}^{k-t}\|\mathbb{E}\left(\mathbf{D}_{\mathbf{y},i}^t\right) - \mathbb{E}\left(\mathbf{D}_{\mathbf{y}}^t\right)\|^2)\right] \\
&\leq \left(\sum_{i=1}^{k} \beta_{\mathbf{y}}^{k-t}\right) \mathbb{E}\left[\sum_{t=1}^{k}\beta_{\mathbf{y}}^{k-t}\frac{1}{|\mathcal{H}|}\sum_{t\in H}\|\mathbb{E}\left(\mathbf{D}_{\mathbf{y},i}^t\right) - \mathbb{E}\left(\mathbf{D}_{\mathbf{y}}^t\right)\|^2\right] \\
&\leq \left(\sum_{i=1}^{k} \beta_{\mathbf{y}}^{k-t}\right)^2 (\zeta_g^2) \leq \frac{\zeta_g^2}{(1-\beta_{\mathbf{y}})^2}.
\end{aligned}
$$

Combining the above results, we get

$$
\begin{aligned}
&\mathbb{E}\left[\frac{1}{|\mathcal{H}|}\sum_{i\in\mathcal{H}}\left\|\mathbf{m}_{\mathbf{y},i}^k - \mathbf{m}_{\mathbf{y}}^k\right\|^2\right] \\
&\leq 3(1-\beta_{\mathbf{y}})^2\frac{1}{|\mathcal{H}|}\sum_{i\in\mathcal{H}}\left(\Lambda_1^k + \Lambda_2^k + \Lambda_3^k\right) \\
&\leq 3(1-\beta_{\mathbf{y}})^2\frac{1}{|\mathcal{H}|}\sum_{i\in\mathcal{H}}\left(\frac{\sigma^2}{1-\beta_{\mathbf{y}}^2} + \frac{\sigma^2}{\left(1-\beta_{\mathbf{y}}^2\right)\left((1-\delta)n\right)} + \frac{\zeta_g^2}{(1-\beta_{\mathbf{y}})^2}\right) \\
&\leq 3\frac{1-\beta_{\mathbf{y}}}{1+\beta_{\mathbf{y}}}\left(1+\frac{1}{(1-\delta)n}\right)\sigma^2 + 3\zeta_g^2.
\end{aligned}
$$

Similarly, we can prove the inequality for $\mathbf{x}$ and $\mathbf{z}$ following the same proof of Lemma 9. $\qquad\square$

### J.2.2 PROOF OF LEMMA 14

We now restate the lemma 14 and present its proof.

**Lemma 14.** *Suppose Assumptions 1-4 hold, and assuming* $\eta_{\mathbf{y}} \leq \min\left\{\lambda_g, \frac{1}{\mu_g+L_g}\right\}, \eta_{\mathbf{z}} \leq \min\left\{\lambda_g, \frac{1}{\mu_g+L_g}\right\}$, *we have*

(1)
$$
\begin{aligned}
\mathbb{E}\left[\left\|\mathbf{y}^{k+1} - \mathbf{y}_\star^{k+1}\right\|^2\right] &\leq \left(1 - \frac{\lambda_g\eta_{\mathbf{y}}}{2}\right)\mathbb{E}\left[\left\|\mathbf{y}^k - \mathbf{y}_\star^k\right\|^2\right] - \frac{\eta_{\mathbf{y}}}{\mu_g+L_g}\mathbb{E}\left[\left\|\mathbb{E}(\mathbf{D}_{\mathbf{y}}^k)\right\|^2\right] \\
&\quad + \frac{12\eta_{\mathbf{y}}}{\lambda_g}\left(\mathbb{E}\left[\left\|\mathbf{m}_{\mathbf{y}}^k - \mathbb{E}(\mathbf{D}_{\mathbf{y}}^k)\right\|^2\right] + \mathbb{E}\left[\left\|\xi_{\mathbf{y}}^k\right\|^2\right]\right) \\
&\quad + \frac{2L_{\mathbf{y}^\star}^2\eta_{\mathbf{x}}^2}{\lambda_g\eta_{\mathbf{y}}}\mathbb{E}\left[\left\|\hat{\mathbf{m}}_{\mathbf{x}}^k\right\|^2\right],
\end{aligned}
$$

(222)

(2)
$$
\begin{aligned}
\mathbb{E}\left[\left\|\mathbf{z}^{k+1} - \mathbf{z}_\star^{k+1}\right\|^2\right] &\leq \left(1 - \frac{\lambda_g\eta_{\mathbf{z}}}{2}\right)\mathbb{E}\left[\left\|\mathbf{z}^k - \mathbf{z}_\star^k\right\|^2\right] - \frac{\eta_{\mathbf{z}}}{\mu_g+L_g}\mathbb{E}\left[\left\|\mathbb{E}(\mathbf{D}_{\mathbf{z}}^k)\right\|^2\right] \\
&\quad + \frac{12\eta_{\mathbf{z}}}{\lambda_g}\left(\mathbb{E}\left[\left\|\mathbf{m}_{\mathbf{z}}^k - \mathbb{E}(\mathbf{D}_{\mathbf{z}}^k)\right\|^2\right] + \mathbb{E}\left[\left\|\xi_{\mathbf{z}}^k\right\|^2\right]\right) \\
&\quad + \frac{2L_{\mathbf{z}^\star}^2\eta_{\mathbf{x}}^2}{\lambda_g\eta_{\mathbf{z}}}\mathbb{E}\left[\left\|\hat{\mathbf{m}}_{\mathbf{x}}^k\right\|^2\right],
\end{aligned}
$$

where $\lambda_g := \frac{\mu_g L_g}{\mu_g+L_g}$.

*Proof.* **Inequality for y.**

By utilizing the Young's inequality and the $L_{\mathbf{y}^\star}$-Lipschitz continuity of $\mathbf{y}^\star(\mathbf{x})$ (Lemma 4), we have

$$\mathbb{E}\left[\left\|\mathbf{y}^{k+1} - \mathbf{y}^{\star}\left(\mathbf{x}^{k+1}\right)\right\|^2\right]$$

$$= \mathbb{E}\left[\left\|\mathbf{y}^{k+1} - \mathbf{y}^{\star}\left(\mathbf{x}^{k}\right) + \mathbf{y}^{\star}\left(\mathbf{x}^{k}\right) - \mathbf{y}^{\star}\left(\mathbf{x}^{k+1}\right)\right\|^2\right]$$

$$\leq (1 + \nu_k)\,\mathbb{E}\left[\left\|\mathbf{y}^{k+1} - \mathbf{y}^{\star}\left(\mathbf{x}^{k}\right)\right\|^2\right] + \left(1 + \frac{1}{\nu_k}\right)\mathbb{E}\left[\left\|\mathbf{y}^{\star}\left(\mathbf{x}^{k}\right) - \mathbf{y}^{\star}\left(\mathbf{x}^{k+1}\right)\right\|^2\right] \cdot \tag{223}$$

$$\leq (1 + \nu_k)\,\mathbb{E}\left[\left\|\mathbf{y}^{k+1} - \mathbf{y}^{\star}\left(\mathbf{x}^{k}\right)\right\|^2\right] + \left(1 + \frac{1}{\nu_k}\right)L_{\mathbf{y}^{\star}}^2 \eta_{\mathbf{x}}^2 \mathbb{E}\left[\left\|\hat{\mathbf{m}}_{\mathbf{x}}^k\right\|^2\right].$$

For the first term, once again employing Young's inequality, we have

$$\mathbb{E}\left[\left\|\mathbf{y}^{k+1} - \mathbf{y}^{\star}\left(\mathbf{x}^{k}\right)\right\|^2\right]$$

$$= \mathbb{E}\left[\left\|\mathbf{y}^{k} - \mathbf{y}^{\star}\left(\mathbf{x}^{k}\right) - \eta_{\mathbf{y}}\hat{\mathbf{m}}_{\mathbf{y}}^k\right\|^2\right]$$

$$= \mathbb{E}\left[\left\|\mathbf{y}^{k} - \mathbf{y}^{\star}\left(\mathbf{x}^{k}\right) - \eta_{\mathbf{y}}\nabla_{\mathbf{y}}g^k - \eta_{\mathbf{y}}\left(\hat{\mathbf{m}}_{\mathbf{y}}^k - \nabla_{\mathbf{y}}g^k\right)\right\|^2\right]$$

$$\leq \left(1 + \frac{\nu_k}{2}\right)\mathbb{E}\left[\left\|\mathbf{y}^{k} - \eta_{\mathbf{y}}\nabla_{\mathbf{y}}g^k - \mathbf{y}^{\star}\left(\mathbf{x}^{k}\right)\right\|^2\right] + \left(1 + \frac{2}{\nu_k}\right)\eta_{\mathbf{y}}^2\mathbb{E}\left[\left\|\hat{\mathbf{m}}_{\mathbf{y}}^k - \nabla_{\mathbf{y}}g^k\right\|^2\right].$$

Utilizing Lemma 8 , we can thus establish the following inequality

$$\mathbb{E}\left[\left\|\mathbf{y}^{k} - \eta_{\mathbf{y}}\nabla_{\mathbf{y}}g^k - \mathbf{y}^{\star}\left(\mathbf{x}^{k}\right)\right\|^2\right]$$

$$= \mathbb{E}\left[\left\|\mathbf{y}^{k} - \mathbf{y}^{\star}\left(\mathbf{x}^{k}\right)\right\|^2\right] + \mathbb{E}\left[\left\|\eta_{\mathbf{y}}\nabla_{\mathbf{y}}g^k\right\|^2\right] - 2\mathbb{E}\left[\eta_{\mathbf{y}}\langle\nabla_{\mathbf{y}}g^k, \mathbf{y}^{k} - \mathbf{y}^{\star}\left(\mathbf{x}^{k}\right)\rangle\right]$$

$$\leq (1 - 2\eta_{\mathbf{y}}\lambda_g)\,\mathbb{E}\left[\left\|\mathbf{y}^{k} - \mathbf{y}^{\star}\left(\mathbf{x}^{k}\right)\right\|^2\right] + \left(\eta_{\mathbf{y}}^2 - 2\eta_{\mathbf{y}}\frac{1}{\mu_g + L_g}\right)\mathbb{E}\left[\left\|\nabla_{\mathbf{y}}g^k\right\|^2\right].$$

Plugging it into (223) and taking the total expectation, we have

$$\mathbb{E}\left[\left\|\mathbf{y}^{k+1} - \mathbf{y}_{*}^{k+1}\right\|^2\right] \leq (1 + \nu_k)\left(1 + \frac{\nu_k}{2}\right)(1 - 2\eta_{\mathbf{y}}\lambda_g)\,\mathbb{E}\left[\left\|\mathbf{y}^{k} - \mathbf{y}^{\star}\left(\mathbf{x}^{k}\right)\right\|^2\right]$$

$$+ (1 + \nu_k)\left(1 + \frac{\nu_k}{2}\right)\left(\eta_{\mathbf{y}}^2 - 2\eta_{\mathbf{y}}\frac{1}{\mu_g + L_g}\right)\mathbb{E}\left[\left\|\nabla_{\mathbf{y}}g^k\right\|^2\right]$$

$$+ (1 + \nu_k)\left(1 + \frac{2}{\nu_k}\right)\eta_{\mathbf{y}}^2\mathbb{E}\left[\left\|\hat{\mathbf{m}}_{\mathbf{y}}^k - \nabla_{\mathbf{y}}g^k\right\|^2\right]$$

$$+ \left(1 + \frac{1}{\nu_k}\right)L_{\mathbf{y}}^{\star}\eta_{\mathbf{x}}^2\mathbb{E}\left[\left\|\hat{\mathbf{m}}_{\mathbf{x}}^k\right\|^2\right].$$

We choose the parameter $\nu_k$ and the step size $\eta_{\mathbf{y}}$ to satisfy

$$\nu_k = \lambda_g\eta_{\mathbf{y}}, \quad \eta_{\mathbf{y}} \leq \min\left\{\lambda_g, \frac{1}{\mu_g + L_g}\right\}.$$

Then, we can obtain

$$\mathbb{E}\left[\left\|\mathbf{y}^{k} - \mathbf{y}_{\star}^{k}\right\|^2\right] \leq \left(1 - \frac{\lambda_g\eta_{\mathbf{y}}}{2}\right)\mathbb{E}\left[\left\|\mathbf{y}^{k} - \mathbf{y}^{\star}(\mathbf{x}^{k})\right\|^2\right] - \frac{1}{\mu_g + L_g}\eta_{\mathbf{y}}\mathbb{E}\left[\left\|\nabla_{\mathbf{y}}g^k\right\|^2\right]$$

$$+ \frac{6\eta_{\mathbf{y}}}{\lambda_g}\mathbb{E}\left[\left\|\hat{\mathbf{m}}_{\mathbf{y}}^k - \nabla_{\mathbf{y}}g^k\right\|^2\right] + \frac{2L_{\mathbf{y}^{\star}}^2\eta_{\mathbf{x}}^2}{\lambda_g\eta_{\mathbf{y}}}\mathbb{E}\left[\left\|\hat{\mathbf{m}}_{\mathbf{x}}^k\right\|^2\right]$$

$$\leq \left(1 - \frac{\lambda_g\eta_{\mathbf{y}}}{2}\right)\mathbb{E}\left[\left\|\mathbf{y}^{k} - \mathbf{y}^{\star}(\mathbf{x}^{k})\right\|^2\right] + \frac{12\eta_{\mathbf{y}}}{\lambda_g}\mathbb{E}\left[\left\|\mathbf{m}_{\mathbf{y}}^k - \mathbb{E}(\mathbf{D}_{\mathbf{y}}^k)\right\|^2\right]$$

$$+ \frac{12\eta_{\mathbf{y}}}{\lambda_g}\mathbb{E}\left[\left\|\xi_{\mathbf{y}}^k\right\|^2\right] + \frac{2L_{\mathbf{y}^{\star}}^2\eta_{\mathbf{x}}^2}{\lambda_g\eta_{\mathbf{y}}}\mathbb{E}\left[\left\|\hat{\mathbf{m}}_{\mathbf{x}}^k\right\|^2\right].$$

Similarly, we can get the result for $\mathbf{z}$. $\qquad\square$

### J.2.3 Proof of Lemma 15

We now restate the lemma 15 and present its proof.

**Lemma 15.** *Suppose Assumption 1, 2 and 3 hold, if $\rho \geq C_f/\mu_g$, we have:*

$$
\begin{aligned}
\mathbb{E}\left[\left\|\nabla\Phi_{\mathcal{H}}\left(\mathbf{x}^k\right) - \mathbf{m}_{\mathbf{x}}^k\right\|^2\right] \leq & 6L_2^2\mathbb{E}\left[\left\|\mathbf{y}^k - \mathbf{y}_\star^k\right\|^2\right] + 6L_g^2\mathbb{E}\left[\left\|\mathbf{z}^k - \mathbf{z}_\star^k\right\|^2\right] \\
& + 2\mathbb{E}\left[\left\|\mathbf{m}_{\mathbf{x}}^k - \mathbb{E}\left(\mathbf{D}_{\mathbf{x}}^k\right)\right\|^2\right],
\end{aligned}
\tag{224}
$$

*where $L_2^2 = L_f^2 + L_g^2\rho^2$.*

*Proof.* We begin by analyzing the expected squared norm of the difference between the gradient of the Hamiltonian and the estimate $\mathbf{m}_{\mathbf{x}}^k$. Starting with the decomposition:

$$
\begin{aligned}
\mathbb{E}\left[\left\|\nabla\Phi_{\mathcal{H}}\left(\mathbf{x}^k\right) - \mathbf{m}_{\mathbf{x}}^k\right\|^2\right] &= \mathbb{E}\left[\left\|\nabla\Phi_{\mathcal{H}}\left(\mathbf{x}^k\right) - \mathbb{E}\left(\mathbf{D}_{\mathbf{x}}^k\right) + \mathbb{E}\left(\mathbf{D}_{\mathbf{x}}^k\right) - \mathbf{m}_{\mathbf{x}}^k\right\|^2\right] \\
&\leq 2\mathbb{E}\left[\left\|\nabla\Phi_{\mathcal{H}}\left(\mathbf{x}^k\right) - \mathbb{E}\left(\mathbf{D}_{\mathbf{x}}^k\right)\right\|^2\right] + 2\mathbb{E}\left[\left\|\mathbb{E}\left(\mathbf{D}_{\mathbf{x}}^k\right) - \mathbf{m}_{\mathbf{x}}^k\right\|^2\right],
\end{aligned}
\tag{225}
$$

where the first inequality follows from the Cauchy-Schwarz inequality. By applying Lemma 11, we obtain:

$$
\mathbb{E}\left[\left\|\nabla\Phi_{\mathcal{H}}\left(\mathbf{x}^k\right) - \mathbf{m}_{\mathbf{x}}^k\right\|^2\right] \leq 6L_2^2\mathbb{E}\left[\left\|\mathbf{y}^k - \mathbf{y}_\star^k\right\|^2\right] + 6L_g^2\mathbb{E}\left[\left\|\mathbf{z}^k - \mathbf{z}_\star^k\right\|^2\right] + 2\mathbb{E}\left[\left\|\mathbb{E}\left(\mathbf{D}_{\mathbf{x}}^k\right) - \mathbf{m}_{\mathbf{x}}^k\right\|^2\right].
\tag{226}
$$

Thus, the lemma is proven. $\square$

### J.2.4 Proof of Lemma 16

We now restate the lemma 16 and present its proof.

**Lemma 16.** *Suppose Assumption 2 and 3 hold, consider Algorithm BR-FedBi in 1. For each step $k \in [K]$, we have*

(1) $\quad \mathbb{E}\left[\left\|\mathbf{m}_{\mathbf{y}}^{k+1} - \mathbb{E}\left(\mathbf{D}_{\mathbf{y}}^{k+1}\right)\right\|^2\right]$

$$
\begin{aligned}
\leq & \beta_{\mathbf{y}}^2\left(1 + L_g\eta_{\mathbf{y}}\right)\left(1 + 3\eta_{\mathbf{y}}L_g\right)\mathbb{E}\left[\left\|\mathbf{m}_{\mathbf{y}}^k - \mathbb{E}\left(\mathbf{D}_{\mathbf{y}}^k\right)\right\|^2\right] + L_g^2\eta_{\mathbf{x}}^2\beta_{\mathbf{y}}^2\left(1 + \frac{1}{\eta_{\mathbf{y}}L_g}\right)\mathbb{E}\left[\left\|\hat{\mathbf{m}}_{\mathbf{x}}^k\right\|^2\right] \\
& + 3\beta_{\mathbf{y}}^2\left(1 + \frac{1}{\eta_{\mathbf{y}}L_g}\right)\left(\eta_{\mathbf{y}}^2L_g^4\mathbb{E}\left[\left\|\mathbf{y}^k - \mathbf{y}^\star\left(\mathbf{x}^k\right)\right\|^2\right] + \eta_{\mathbf{y}}^2L_g^2\mathbb{E}\left[\left\|\xi_{\mathbf{y}}^k\right\|^2\right]\right) + (1 - \beta_{\mathbf{y}})^2\frac{\sigma^2}{(1 - \delta)n}.
\end{aligned}
$$

(2) $\quad \mathbb{E}\left[\left\|\mathbf{m}_{\mathbf{x}}^{k+1} - \mathbb{E}\left(\mathbf{D}_{\mathbf{x}}^{k+1}\right)\right\|^2\right]$

$$
\begin{aligned}
\leq & \beta_{\mathbf{x}}^2\left(1 + L_g\eta_{\mathbf{z}}\right)\mathbb{E}\left[\left\|\mathbf{m}_{\mathbf{x}}^k - \mathbb{E}\left(\mathbf{D}_{\mathbf{x}}^k\right)\right\|^2\right] + 4\beta_{\mathbf{x}}^2\left(1 + \frac{1}{\eta_{\mathbf{z}}L_1}\right)L_1^2\eta_{\mathbf{x}}^2\mathbb{E}\left[\left\|\hat{\mathbf{m}}_{\mathbf{x}}^k\right\|^2\right] \\
& + 12\beta_{\mathbf{x}}^2\left(1 + \frac{1}{\eta_{\mathbf{z}}L_1}\right)L_1^2\eta_{\mathbf{y}}^2\mathbb{E}\left[\left\|\mathbf{m}_{\mathbf{y}}^k - \mathbb{E}\left(\mathbf{D}_{\mathbf{y}}^k\right)\right\|^2\right] \\
& + 12\beta_{\mathbf{x}}^2\left(1 + \frac{1}{\eta_{\mathbf{z}}L_g}\right)L_g^2\eta_{\mathbf{z}}^2\mathbb{E}\left[\left\|\mathbf{m}_{\mathbf{z}}^k - \mathbb{E}\left(\mathbf{D}_{\mathbf{z}}^k\right)\right\|^2\right] \\
& + 12\beta_{\mathbf{x}}^2\left(1 + \frac{1}{\eta_{\mathbf{z}}L_g}\right)\left(L_1^2L_g^2\eta_{\mathbf{y}}^2 + 3L_g^2L_2^2\eta_{\mathbf{z}}^2\right)\mathbb{E}\left[\left\|\mathbf{y}^k - \mathbf{y}^\star\left(\mathbf{x}^k\right)\right\|^2\right] \\
& + 36\beta_{\mathbf{x}}^2\left(1 + \frac{1}{\eta_{\mathbf{z}}L_g}\right)L_g^4\eta_{\mathbf{z}}^2\mathbb{E}\left[\left\|\mathbf{z}^k - \mathbf{z}^\star\left(\mathbf{x}^k\right)\right\|^2\right] \\
& + 12\beta_{\mathbf{x}}^2\left(1 + \frac{1}{\eta_{\mathbf{z}}L_g}\right)L_g^2\eta_{\mathbf{z}}^2\mathbb{E}\left[\left\|\xi_{\mathbf{z}}^k\right\|^2\right] + 12\beta_{\mathbf{x}}^2\left(1 + \frac{1}{\eta_{\mathbf{z}}L_g}\right)L_1^2\eta_{\mathbf{y}}^2\mathbb{E}\left[\left\|\xi_{\mathbf{y}}^k\right\|^2\right] + (1 - \beta_{\mathbf{x}})^2\frac{\sigma_1^2}{(1 - \delta)n}.
\end{aligned}
$$

(3) $\quad \mathbb{E}\left[\left\|\mathbf{m}_{\mathbf{z}}^{k+1} - \mathbb{E}\left(\mathbf{D}_{\mathbf{z}}^{k+1}\right)\right\|^2\right]$

$$\leq \beta_{\mathbf{z}}^2 \left(1 + L_g \eta_{\mathbf{z}}\right)\left(1 + 12 L_g \eta_{\mathbf{z}}\right) \mathbb{E}\left[\left\|\mathbf{m}_{\mathbf{z}}^k - \mathbb{E}\left(\mathbf{D}_{\mathbf{z}}^k\right)\right\|^2\right] + 4\beta_{\mathbf{z}}^2 \left(1 + \frac{1}{\eta_{\mathbf{z}} L_g}\right) L_1^2 \eta_{\mathbf{x}}^2 \mathbb{E}\left[\left\|\hat{\mathbf{m}}_{\mathbf{x}}^k\right\|^2\right]$$

$$+ 12\beta_{\mathbf{z}}^2 \left(1 + \frac{1}{\eta_{\mathbf{z}} L_g}\right) L_1^2 \eta_{\mathbf{y}}^2 \mathbb{E}\left[\left\|\mathbf{m}_{\mathbf{y}}^k - \mathbb{E}\left(\mathbf{D}_{\mathbf{y}}^k\right)\right\|^2\right]$$

$$+ 12\beta_{\mathbf{z}}^2 \left(1 + \frac{1}{\eta_{\mathbf{z}} L_g}\right)\left(L_1^2 L_g^2 \eta_{\mathbf{y}}^2 + 3L_g^2 L_2^2 \eta_{\mathbf{z}}^2\right) \mathbb{E}\left[\left\|\mathbf{y}^k - \mathbf{y}^\star\left(\mathbf{x}^k\right)\right\|^2\right]$$

$$+ 36\beta_{\mathbf{z}}^2 \left(1 + \frac{1}{\eta_{\mathbf{z}} L_g}\right) L_g^4 \eta_{\mathbf{z}}^2 \mathbb{E}\left[\left\|\mathbf{z}^k - \mathbf{z}^\star\left(\mathbf{x}^k\right)\right\|^2\right]$$

$$+ 12\beta_{\mathbf{z}}^2 \left(1 + \frac{1}{\eta_{\mathbf{z}} L_g}\right) L_1^2 \eta_{\mathbf{y}}^2 \mathbb{E}\left[\left\|\xi_{\mathbf{y}}^k\right\|^2\right] + 12\beta_{\mathbf{z}}^2 \left(1 + \frac{1}{\eta_{\mathbf{z}} L_g}\right) L_g^2 \eta_{\mathbf{z}}^2 \mathbb{E}\left[\left\|\xi_{\mathbf{z}}^k\right\|^2\right] + (1 - \beta_{\mathbf{z}})^2 \frac{\sigma_1^2}{(1 - \delta)n}.$$

*Proof.* **Analysis for** $\mathbb{E}\left[\left\|\mathbf{m}_{\mathbf{y}}^{k+1} - \mathbb{E}\left(\mathbf{D}_{\mathbf{y}}^{k+1}\right)\right\|^2\right].$

The update of the momentum term $\mathbf{m}_{\mathbf{y}}^{k+1}$ is defined as:

$$\mathbf{m}_{\mathbf{y}}^k = \beta_{\mathbf{y}} \mathbf{m}_{\mathbf{y}}^{k-1} + (1 - \beta_{\mathbf{y}}) \mathbf{D}_{\mathbf{y}}^k.$$

To analyze the variance of $\mathbf{m}_{\mathbf{y}}^{k+1}$ relative to $\mathbb{E}\left(\mathbf{D}_{\mathbf{y}}^{k+1}\right)$, we consider an arbitrary step $k > 1$. The squared expectation of the deviation is given by:

$$\mathbb{E}\left[\left\|\mathbf{m}_{\mathbf{y}}^{k+1} - \mathbb{E}\left(\mathbf{D}_{\mathbf{y}}^{k+1}\right)\right\|^2\right]$$

$$= \mathbb{E}\left[\left\|\beta_{\mathbf{y}} \mathbf{m}_{\mathbf{y}}^k + (1 - \beta_{\mathbf{y}}) \mathbf{D}_{\mathbf{y}}^{k+1} - \beta_{\mathbf{y}} \mathbb{E}\left(\mathbf{D}_{\mathbf{y}}^k\right) - \mathbb{E}\left(\mathbf{D}_{\mathbf{y}}^{k+1}\right) + \beta_{\mathbf{y}} \mathbb{E}\left(\mathbf{D}_{\mathbf{y}}^k\right)\right\|^2\right]$$

$$= \mathbb{E}\left[\left\|\beta_{\mathbf{y}} \left(\mathbf{m}_{\mathbf{y}}^k - \mathbb{E}\left(\mathbf{D}_{\mathbf{y}}^k\right)\right) + (1 - \beta_{\mathbf{y}}) \left(\mathbf{D}_{\mathbf{y}}^{k+1} - \mathbb{E}\left(\mathbf{D}_{\mathbf{y}}^{k+1}\right)\right) - \beta_{\mathbf{y}} \left(\mathbb{E}\left(\mathbf{D}_{\mathbf{y}}^{k+1}\right) - \mathbb{E}\left(\mathbf{D}_{\mathbf{y}}^k\right)\right)\right\|^2\right]$$

$$\leq \beta_{\mathbf{y}}^2 \mathbb{E}\left[\left\|\mathbf{m}_{\mathbf{y}}^k - \mathbb{E}\left(\mathbf{D}_{\mathbf{y}}^k\right)\right\|^2\right] + (1 - \beta_{\mathbf{y}})^2 \mathbb{E}\left[\left\|\mathbf{D}_{\mathbf{y}}^{k+1} - \mathbb{E}\left(\mathbf{D}_{\mathbf{y}}^{k+1}\right)\right\|^2\right]$$

$$+ \beta_{\mathbf{y}}^2 \mathbb{E}\left[\left\|\mathbb{E}\left(\mathbf{D}_{\mathbf{y}}^{k+1}\right) - \mathbb{E}\left(\mathbf{D}_{\mathbf{y}}^k\right)\right\|^2\right] - 2\beta_{\mathbf{y}}^2 \langle \mathbf{m}_{\mathbf{y}}^k - \mathbb{E}\left(\mathbf{D}_{\mathbf{y}}^k\right), \mathbb{E}\left(\mathbf{D}_{\mathbf{y}}^{k+1}\right) - \mathbb{E}\left(\mathbf{D}_{\mathbf{y}}^k\right) \rangle, \qquad (227)$$

where the inequality uses unbiased estimation of $\mathbf{D}_{\mathbf{y}}^k$. For the fourth term, the inner product can be bounded using the Cauchy-Schwarz inequality:

$$- 2\beta_{\mathbf{y}}^2 \langle \mathbf{m}_{\mathbf{y}}^k - \mathbb{E}\left(\mathbf{D}_{\mathbf{y}}^k\right), \mathbb{E}\left(\mathbf{D}_{\mathbf{y}}^{k+1}\right) - \mathbb{E}\left(\mathbf{D}_{\mathbf{y}}^k\right) \rangle$$

$$\leq 2\beta_{\mathbf{y}}^2 \|\mathbf{m}_{\mathbf{y}}^k - \mathbb{E}\left(\mathbf{D}_{\mathbf{y}}^k\right)\| \cdot \|\mathbb{E}\left(\mathbf{D}_{\mathbf{y}}^{k+1}\right) - \mathbb{E}\left(\mathbf{D}_{\mathbf{y}}^k\right)\|$$

$$\leq 2\beta_{\mathbf{y}}^2 L_g \|\mathbf{m}_{\mathbf{y}}^k - \mathbb{E}\left(\mathbf{D}_{\mathbf{y}}^k\right)\| \cdot \sqrt{\eta_{\mathbf{y}}^2 \|\hat{\mathbf{m}}_{\mathbf{y}}^k\|^2 + \eta_{\mathbf{x}}^2 \|\hat{\mathbf{m}}_{\mathbf{x}}^k\|^2}$$

$$\leq 2\beta_{\mathbf{y}}^2 L_g \eta_{\mathbf{y}} \|\mathbf{m}_{\mathbf{y}}^k - \mathbb{E}\left(\mathbf{D}_{\mathbf{y}}^k\right)\| \cdot \sqrt{\|\hat{\mathbf{m}}_{\mathbf{y}}^k\|^2 + \frac{\|\hat{\mathbf{m}}_{\mathbf{x}}^k\|^2}{c_{\mathbf{y}}^2}}$$

$$\leq \beta_{\mathbf{y}}^2 L_g \eta_{\mathbf{y}} \left(\mathbb{E}\left[\left\|\mathbf{m}_{\mathbf{y}}^k - \mathbb{E}\left(\mathbf{D}_{\mathbf{y}}^k\right)\right\|^2\right] + \mathbb{E}\left[\left\|\hat{\mathbf{m}}_{\mathbf{y}}^k\right\|^2\right] + \frac{1}{c_{\mathbf{y}}^2} \mathbb{E}\left[\left\|\hat{\mathbf{m}}_{\mathbf{x}}^k\right\|^2\right]\right),$$

where the second inequality uses the lipschitz continuous $\|\nabla_{\mathbf{y}} G(\mathbf{x}^{k+1}, \mathbf{y}^{k+1}) - \nabla_{\mathbf{y}} G(\mathbf{x}^k, \mathbf{y}^k)\|^2 \leq L_g^2 \left(\|\mathbf{y}^{k+1} - \mathbf{y}^k\|^2 + \|\mathbf{x}^{k+1} - \mathbf{x}^k\|^2\right)$ and last inequality uses $2ab \leq a^2 + b^2$.

Combining (227) and (228), we have

$$\mathbb{E}\left[\left\|\mathbf{m}_{\mathbf{y}}^{k+1} - \mathbb{E}\left(\mathbf{D}_{\mathbf{y}}^{k+1}\right)\right\|^2\right]$$

$$\leq \beta_{\mathbf{y}}^2 \left(1 + L_g \eta_{\mathbf{y}}\right) \mathbb{E}\left[\left\|\mathbf{m}_{\mathbf{y}}^k - \mathbb{E}\left(\mathbf{D}_{\mathbf{y}}^k\right)\right\|^2\right] + (1 - \beta_{\mathbf{y}})^2 \mathbb{E}\left[\left\|\mathbf{D}_{\mathbf{y}}^{k+1} - \mathbb{E}\left(\mathbf{D}_{\mathbf{y}}^{k+1}\right)\right\|^2\right]$$

$$+ \beta_{\mathbf{y}}^2 \eta_{\mathbf{y}} L_g \left(1 + \eta_{\mathbf{y}} L_g\right) \left(\mathbb{E}\left[\left\|\hat{\mathbf{m}}_{\mathbf{y}}^k\right\|^2\right] + \frac{1}{c_{\mathbf{y}}^2} \mathbb{E}\left[\left\|\hat{\mathbf{m}}_{\mathbf{x}}^k\right\|^2\right]\right). \qquad (228)$$

Combining (228) with Lemma 7 and Lemma 26, we have

$$\mathbb{E}\left[\|\mathbf{m}_{\mathbf{y}}^{k+1} - \mathbb{E}\left(\mathbf{D}_{\mathbf{y}}^{k+1}\right)\|^2\right]$$

$$\leq \beta_{\mathbf{y}}^2 \left(1 + L_g \eta_{\mathbf{y}}\right)\left(1 + 3\eta_{\mathbf{y}} L_g\right)\mathbb{E}\left[\left\|\mathbf{m}_{\mathbf{y}}^k - \mathbb{E}\left(\mathbf{D}_{\mathbf{y}}^k\right)\right\|^2\right] + (1 - \beta_{\mathbf{y}})^2 \cdot \frac{\sigma^2}{(1-\delta)n}$$

$$+ \beta_{\mathbf{y}}^2 \eta_{\mathbf{y}} L_g \left(1 + \eta_{\mathbf{y}} L_g\right)\left(3L_g^2 \mathbb{E}\left[\left\|\mathbf{y}^k - \mathbf{y}^\star\left(\mathbf{x}^k\right)\right\|^2\right] + 3\mathbb{E}\left[\left\|\xi_{\mathbf{y}}^k\right\|^2\right]\right)$$

$$+ \frac{\beta_{\mathbf{y}}^2}{c_{\mathbf{y}}^2} \eta_{\mathbf{y}} L_g \left(1 + \eta_{\mathbf{y}} L_g\right)\mathbb{E}\left[\left\|\hat{\mathbf{m}}_{\mathbf{x}}^k\right\|^2\right].$$

**Analysis for** $\mathbb{E}\left[\left\|\mathbf{m}_{\mathbf{x}}^{k+1} - \mathbb{E}\left(\mathbf{D}_{\mathbf{x}}^{k+1}\right)\right\|^2\right]$**.** The update of the momentum term $\mathbf{m}_{\mathbf{x}}^{k+1}$ is defined as:

$$\mathbf{m}_{\mathbf{x}}^k = \beta_{\mathbf{x}} \mathbf{m}_{\mathbf{x}}^{k-1} + (1 - \beta_{\mathbf{x}})\mathbf{D}_{\mathbf{x}}^k.$$

To analyze the variance of $\mathbf{m}_{\mathbf{x}}^{k+1}$ relative to $\mathbb{E}\left(\mathbf{D}_{\mathbf{x}}^{k+1}\right)$, we consider an arbitrary step $k > 1$. The squared expectation of the deviation is given by:

$$\mathbb{E}\left[\left\|\mathbf{m}_{\mathbf{x}}^{k+1} - \mathbb{E}\left(\mathbf{D}_{\mathbf{x}}^k\right)\right\|^2\right]$$

$$= \mathbb{E}\left[\left\|\beta_{\mathbf{x}}\mathbf{m}_{\mathbf{x}}^k + (1 - \beta_{\mathbf{x}})\mathbf{D}_{\mathbf{x}}^{k+1} - \beta_{\mathbf{x}}\mathbb{E}\left(\mathbf{D}_{\mathbf{x}}^k\right) - \mathbb{E}\left(\mathbf{D}_{\mathbf{x}}^{k+1}\right) + \beta_{\mathbf{x}}\mathbb{E}\left(\mathbf{D}_{\mathbf{x}}^k\right)\right\|^2\right]$$

$$= \mathbb{E}\left[\left\|\beta_{\mathbf{x}}\left(\mathbf{m}_{\mathbf{x}}^k - \mathbb{E}\left(\mathbf{D}_{\mathbf{x}}^k\right)\right) + (1 - \beta_{\mathbf{x}})\left(\mathbf{D}_{\mathbf{x}}^k - \mathbb{E}\left(\mathbf{D}_{\mathbf{x}}^k\right)\right) + \beta_{\mathbf{x}}\left(\mathbb{E}\left(\mathbf{D}_{\mathbf{x}}^{k+1}\right) - \mathbb{E}\left(\mathbf{D}_{\mathbf{x}}^k\right)\right)\right\|^2\right] \quad (229)$$

$$\leq \beta_{\mathbf{x}}^2 \mathbb{E}\left[\left\|\mathbf{m}_{\mathbf{x}}^k - \mathbb{E}\left(\mathbf{D}_{\mathbf{x}}^k\right)\right\|^2\right] + (1 - \beta_{\mathbf{x}})^2 \mathbb{E}\left[\left\|\mathbf{D}_{\mathbf{x}}^{k+1} - \mathbb{E}\left(\mathbf{D}_{\mathbf{x}}^{k+1}\right)\right\|^2\right]$$

$$+ \beta_{\mathbf{x}}^2 \mathbb{E}\left[\left\|\mathbb{E}\left(\mathbf{D}_{\mathbf{x}}^{k+1}\right) - \mathbb{E}\left(\mathbf{D}_{\mathbf{x}}^k\right)\right\|^2\right] - 2\beta_{\mathbf{x}}^2\langle\mathbf{m}_{\mathbf{x}}^k - \mathbb{E}\left(\mathbf{D}_{\mathbf{x}}^k\right), \mathbb{E}\left(\mathbf{D}_{\mathbf{x}}^{k+1}\right) - \mathbb{E}\left(\mathbf{D}_{\mathbf{x}}^k\right)\rangle,$$

where the inequality uses unbiased estimation of $\mathbf{D}_{\mathbf{x}}^k$. For the fourth term, the inner product can be bounded using the Cauchy-Schwarz inequality:

$$- 2\beta_{\mathbf{x}}^2\langle\mathbf{m}_{\mathbf{x}}^k - \mathbb{E}\left(\mathbf{D}_{\mathbf{x}}^k\right), \mathbb{E}\left(\mathbf{D}_{\mathbf{x}}^{k+1}\right) - \mathbb{E}\left(\mathbf{D}_{\mathbf{x}}^k\right)\rangle$$

$$\leq 2\beta_{\mathbf{x}}^2\|\mathbf{m}_{\mathbf{x}}^k - \mathbb{E}\left(\mathbf{D}_{\mathbf{x}}^k\right)\| \cdot \|\mathbb{E}\left(\mathbf{D}_{\mathbf{x}}^{k+1}\right) - \mathbb{E}\left(\mathbf{D}_{\mathbf{x}}^k\right)\|$$

$$\leq 2\beta_{\mathbf{x}}^2 L_g \eta_{\mathbf{z}}\|\mathbf{m}_{\mathbf{x}}^k - \mathbb{E}\left(\mathbf{D}_{\mathbf{x}}^k\right)\| \cdot \frac{1}{L_g \eta_{\mathbf{z}}}\sqrt{\|\mathbb{E}\left(\mathbf{D}_{\mathbf{x}}^{k+1}\right) - \mathbb{E}\left(\mathbf{D}_{\mathbf{x}}^k\right)\|^2} \quad (230)$$

$$\leq \beta_{\mathbf{x}}^2 L_g \eta_{\mathbf{z}}\left(\mathbb{E}\left[\left\|\mathbf{m}_{\mathbf{x}}^k - \mathbb{E}\left(\mathbf{D}_{\mathbf{x}}^k\right)\right\|^2\right] + \frac{1}{L_g^2 \eta_{\mathbf{z}}^2}\mathbb{E}\left[\left\|\mathbb{E}\left(\mathbf{D}_{\mathbf{x}}^{k+1}\right) - \mathbb{E}\left(\mathbf{D}_{\mathbf{x}}^k\right)\right\|^2\right]\right),$$

where last inequality uses $2ab \leq a^2 + b^2$.

Then we use following inequality

$$\mathbb{E}\left[\left\|\mathbb{E}\left(\mathbf{D}_{\mathbf{x}}^{k+1}\right) - \mathbb{E}\left(\mathbf{D}_{\mathbf{x}}^k\right)\right\|^2\right]$$

$$\leq 2L_f^2\left(\eta_{\mathbf{x}}^2 \mathbb{E}\left[\left\|\hat{\mathbf{m}}_{\mathbf{x}}^k\right\|^2\right] + \eta_{\mathbf{y}}^2 \mathbb{E}\left[\left\|\hat{\mathbf{m}}_{\mathbf{y}}^k\right\|^2\right]\right) + 4L_g^2\rho^2\left(\eta_{\mathbf{x}}^2 \mathbb{E}\left[\left\|\hat{\mathbf{m}}_{\mathbf{x}}^k\right\|^2\right] + \eta_{\mathbf{y}}^2 \mathbb{E}\left[\left\|\hat{\mathbf{m}}_{\mathbf{y}}^k\right\|^2\right]\right)$$

$$+ 4L_g^2\eta_{\mathbf{z}}^2 \mathbb{E}\left[\left\|\hat{\mathbf{m}}_{\mathbf{z}}^k\right\|^2\right] \quad (231)$$

$$\leq 4L_1^2\eta_{\mathbf{x}}^2 \mathbb{E}\left[\left\|\hat{\mathbf{m}}_{\mathbf{x}}^k\right\|^2\right] + 4L_1^2\eta_{\mathbf{y}}^2 \mathbb{E}\left[\left\|\hat{\mathbf{m}}_{\mathbf{y}}^k\right\|^2\right] + 4L_g^2\eta_{\mathbf{z}}^2 \mathbb{E}\left[\left\|\hat{\mathbf{m}}_{\mathbf{z}}^k\right\|^2\right].$$

Combining above inequality and Lemma 26, we have

$$\mathbb{E}\left[\left\|\mathbf{m}_{\mathbf{x}}^{k+1} - \mathbb{E}\left(\mathbf{D}_{\mathbf{x}}^{k+1}\right)\right\|^2\right]$$

$$\leq \beta_{\mathbf{x}}^2 \left(1 + L_g \eta_{\mathbf{z}}\right) \mathbb{E}\left[\left\|\mathbf{m}_{\mathbf{x}}^k - \mathbb{E}\left(\mathbf{D}_{\mathbf{x}}^k\right)\right\|^2\right] + \left(1 - \beta_{\mathbf{x}}\right)^2 \frac{\sigma_1^2}{(1-\delta)n} + 4\beta_{\mathbf{x}}^2 \left(1 + \frac{1}{\eta_{\mathbf{z}} L_1}\right) L_1^2 \eta_{\mathbf{x}}^2 \mathbb{E}\left[\left\|\hat{\mathbf{m}}_{\mathbf{x}}^k\right\|^2\right]$$

$$+ 12\beta_{\mathbf{x}}^2 \left(1 + \frac{1}{\eta_{\mathbf{z}} L_1}\right) L_1^2 \eta_{\mathbf{y}}^2 \mathbb{E}\left[\left\|\mathbf{m}_{\mathbf{y}}^k - \mathbb{E}\left(\mathbf{D}_{\mathbf{y}}^k\right)\right\|^2\right] + 12\beta_{\mathbf{x}}^2 \left(1 + \frac{1}{\eta_{\mathbf{z}} L_g}\right) L_g^2 \eta_{\mathbf{z}}^2 \mathbb{E}\left[\left\|\mathbf{m}_{\mathbf{z}}^k - \mathbb{E}\left(\mathbf{D}_{\mathbf{z}}^k\right)\right\|^2\right]$$

$$+ 12\beta_{\mathbf{z}}^2 \left(1 + \frac{1}{\eta_{\mathbf{z}} L_g}\right) \left(L_1^2 L_g^2 \eta_{\mathbf{y}}^2 + 3 L_g^2 L_2^2 \eta_{\mathbf{z}}^2\right) \mathbb{E}\left[\left\|\mathbf{y}^k - \mathbf{y}^\star\left(\mathbf{x}^k\right)\right\|^2\right]$$

$$+ 36\beta_{\mathbf{x}}^2 \left(1 + \frac{1}{\eta_{\mathbf{z}} L_g}\right) L_g^4 \eta_{\mathbf{z}}^2 \mathbb{E}\left[\left\|\mathbf{z}^k - \mathbf{z}^\star\left(\mathbf{x}^k\right)\right\|^2\right]$$

$$+ 12\beta_{\mathbf{x}}^2 \left(1 + \frac{1}{\eta_{\mathbf{z}} L_g}\right) L_1^2 \eta_{\mathbf{y}}^2 \mathbb{E}\left[\left\|\xi_{\mathbf{y}}^k\right\|^2\right] + 12\beta_{\mathbf{x}}^2 \left(1 + \frac{1}{\eta_{\mathbf{z}} L_g}\right) L_g^2 \eta_{\mathbf{z}}^2 \mathbb{E}\left[\left\|\xi_{\mathbf{z}}^k\right\|^2\right].$$

Similarly, for $\mathbf{z}$, we have

$$\mathbb{E}\left[\left\|\mathbf{m}_{\mathbf{z}}^{k+1} - \mathbb{E}\left(\mathbf{D}_{\mathbf{z}}^{k+1}\right)\right\|^2\right] \tag{232}$$

$$\leq \beta_{\mathbf{z}}^2 \left(1 + L_g \eta_{\mathbf{z}}\right) \left(1 + 12 L_g \eta_{\mathbf{z}}\right) \mathbb{E}\left[\left\|\mathbf{m}_{\mathbf{z}}^k - \mathbb{E}\left(\mathbf{D}_{\mathbf{z}}^k\right)\right\|^2\right] + \left(1 - \beta_{\mathbf{z}}\right)^2 \frac{\sigma_1^2}{(1-\delta)n}$$

$$+ 4\beta_{\mathbf{z}}^2 \left(1 + \frac{1}{\eta_{\mathbf{z}} L_g}\right) L_1^2 \eta_{\mathbf{x}}^2 \mathbb{E}\left[\left\|\hat{\mathbf{m}}_{\mathbf{x}}^k\right\|^2\right] + 12\beta_{\mathbf{z}}^2 \left(1 + \frac{1}{\eta_{\mathbf{z}} L_g}\right) L_1^2 \eta_{\mathbf{y}}^2 \mathbb{E}\left[\left\|\mathbf{m}_{\mathbf{y}}^k - \mathbb{E}\left(\mathbf{D}_{\mathbf{y}}^k\right)\right\|^2\right]$$

$$+ 12\beta_{\mathbf{z}}^2 \left(1 + \frac{1}{\eta_{\mathbf{z}} L_g}\right) \left(L_1^2 L_g^2 \eta_{\mathbf{y}}^2 + 3 L_g^2 L_2^2 \eta_{\mathbf{z}}^2\right) \mathbb{E}\left[\left\|\mathbf{y}^k - \mathbf{y}^\star\left(\mathbf{x}^k\right)\right\|^2\right]$$

$$+ 36\beta_{\mathbf{z}}^2 \left(1 + \frac{1}{\eta_{\mathbf{z}} L_g}\right) L_g^4 \eta_{\mathbf{z}}^2 \mathbb{E}\left[\left\|\mathbf{z}^k - \mathbf{z}^\star\left(\mathbf{x}^k\right)\right\|^2\right]$$

$$+ 12\beta_{\mathbf{z}}^2 \left(1 + \frac{1}{\eta_{\mathbf{z}} L_g}\right) L_1^2 \eta_{\mathbf{y}}^2 \mathbb{E}\left[\left\|\xi_{\mathbf{y}}^k\right\|^2\right] + 12\beta_{\mathbf{z}}^2 \left(1 + \frac{1}{\eta_{\mathbf{z}} L_g}\right) L_g^2 \eta_{\mathbf{z}}^2 \mathbb{E}\left[\left\|\xi_{\mathbf{z}}^k\right\|^2\right]. \tag{233}$$

Thus, the lemma is proven. $\qquad\square$

**Lemma 26** (Bounded second moment for momentum). *Suppose Assumptions 1-4 hold. For each step $k \in [K]$, we have*

(1) $\quad \mathbb{E}\left[\left\|\hat{\mathbf{m}}_{\mathbf{x}}^k\right\|^2\right] \leq 2\mathbb{E}\left[\left\|\mathbf{m}_{\mathbf{x}}^k\right\|^2\right] + 2\mathbb{E}\left[\left\|\xi_{\mathbf{x}}^k\right\|^2\right]$

(2) $\quad \mathbb{E}\left[\left\|\hat{\mathbf{m}}_{\mathbf{y}}^k\right\|^2\right] \leq 3\mathbb{E}\left[\left\|\xi_{\mathbf{y}}^k\right\|^2\right] + 3\mathbb{E}\left[\left\|\mathbf{m}_{\mathbf{y}}^k - \mathbb{E}\left(\mathbf{D}_{\mathbf{y}}^k\right)\right\|^2\right] + 3L_g^2 \mathbb{E}\left[\left\|\mathbf{y}^k - \mathbf{y}_\star^k\right\|^2\right],$

$$\tag{234}$$

(3) $\quad \mathbb{E}\left[\left\|\hat{\mathbf{m}}_{\mathbf{z}}^k\right\|^2\right] \leq 3\mathbb{E}\left[\left\|\xi_{\mathbf{z}}^k\right\|^2\right] + 3\mathbb{E}\left[\left\|\mathbf{m}_{\mathbf{z}}^k - \mathbb{E}\left(\mathbf{D}_{\mathbf{z}}^k\right)\right\|^2\right] + 9L_g^2 \mathbb{E}\left[\left\|\mathbf{z}^k - \mathbf{z}_\star^k\right\|^2\right]$

$$+ 9L_2^2 \mathbb{E}\left[\left\|\mathbf{y}^k - \mathbf{y}_\star^k\right\|^2\right].$$

*Proof.* **Bounded second moment for $\mathbf{x}$.**

$$\mathbb{E}\left[\left\|\hat{\mathbf{m}}_{\mathbf{x}}^k\right\|^2\right] = \mathbb{E}\left[\left\|\hat{\mathbf{m}}_{\mathbf{x}}^k - \mathbf{m}_{\mathbf{x}}^k + \mathbf{m}_{\mathbf{x}}^k\right\|^2\right] \leq 2\mathbb{E}\left[\left\|\mathbf{m}_{\mathbf{x}}^k\right\|^2\right] + 2\mathbb{E}\left[\left\|\xi_{\mathbf{x}}^k\right\|^2\right]. \tag{235}$$

**Bounded second moment for $\mathbf{y}$.** Decomposing $\hat{\mathbf{m}}_{\mathbf{y}}^k$ into three components, we have

$$\mathbb{E}\left[\left\|\hat{\mathbf{m}}_{\mathbf{y}}^k\right\|^2\right] = \mathbb{E}\left[\left\|\hat{\mathbf{m}}_{\mathbf{y}}^k - \mathbf{m}_{\mathbf{y}}^k + \mathbf{m}_{\mathbf{y}}^k - \mathbb{E}\left(\mathbf{D}_{\mathbf{y}}^k\right) + \mathbb{E}\left(\mathbf{D}_{\mathbf{y}}^k\right)\right\|^2\right]$$

$$\leq 3\mathbb{E}\left[\left\|\xi_{\mathbf{y}}^k\right\|^2\right] + 3\mathbb{E}\left[\left\|\mathbf{m}_{\mathbf{y}}^k - \mathbb{E}\left(\mathbf{D}_{\mathbf{y}}^k\right)\right\|^2\right] + 3\mathbb{E}\left[\left\|\mathbb{E}\left(\mathbf{D}_{\mathbf{y}}^k\right)\right\|^2\right] \tag{236}$$

$$\leq 3\mathbb{E}\left[\left\|\xi_{\mathbf{y}}^k\right\|^2\right] + 3\mathbb{E}\left[\left\|\mathbf{m}_{\mathbf{y}}^k - \mathbb{E}\left(\mathbf{D}_{\mathbf{y}}^k\right)\right\|^2\right] + 3L_g^2 \mathbb{E}\left[\left\|\mathbf{y}^k - \mathbf{y}_k^\star\right\|^2\right],$$

where the first inequality uses Cauchy-Schwartz inequality and the second inequality uses Assumption 1.

**Bounded second moment for z.** Decomposing $\hat{\mathbf{m}}_{\mathbf{z}}^k$ into three components, we have

$$
\begin{aligned}
\mathbb{E}\left[\left\|\hat{\mathbf{m}}_{\mathbf{z}}^k\right\|^2\right] &= \mathbb{E}\left[\left\|\hat{\mathbf{m}}_{\mathbf{z}}^k - \mathbf{m}_{\mathbf{z}}^k + \mathbf{m}_{\mathbf{z}}^k - \mathbb{E}\left(\mathbf{D}_{\mathbf{z}}^k\right) + \mathbb{E}\left(\mathbf{D}_{\mathbf{z}}^k\right)\right\|^2\right] \\
&\leq 3\mathbb{E}\left[\left\|\xi_{\mathbf{z}}^k\right\|^2\right] + 3\mathbb{E}\left[\left\|\mathbf{m}_{\mathbf{z}}^k - \mathbb{E}\left(\mathbf{D}_{\mathbf{z}}^k\right)\right\|^2\right] + 3\mathbb{E}\left[\left\|\mathbb{E}\left(\mathbf{D}_{\mathbf{z}}^k\right)\right\|^2\right] \\
&\leq 3\mathbb{E}\left[\left\|\xi_{\mathbf{z}}^k\right\|^2\right] + 3\mathbb{E}\left[\left\|\mathbf{m}_{\mathbf{z}}^k - \mathbb{E}\left(\mathbf{D}_{\mathbf{z}}^k\right)\right\|^2\right] + 9L_g^2\mathbb{E}\left[\left\|\mathbf{z}^k - \mathbf{z}_\star^k\right\|^2\right] + 9L_2^2\mathbb{E}\left[\left\|\mathbf{y}^k - \mathbf{y}_\star^k\right\|^2\right],
\end{aligned}
\tag{237}
$$

where the first inequality uses Cauchy-Schwartz inequality and the second inequality uses Assumption 1. $\square$

### J.2.5 PROOF OF LEMMA 17

We now restate the lemma 17 and present its proof.

**Lemma 17.** *Recall that the upper-level loss function is $L_{\nabla\Phi}$-smooth, we have*

$$
\begin{aligned}
\mathbb{E}\left[\Phi_{\mathcal{H}}\left(\mathbf{x}^{k+1}\right)\right] &\leq \mathbb{E}\left[\Phi_{\mathcal{H}}\left(\mathbf{x}^k\right)\right] - \frac{\eta_{\mathbf{x}}}{2}\mathbb{E}\left[\left\|\nabla\Phi_{\mathcal{H}}\left(\mathbf{x}^k\right)\right\|^2\right] - \left(\frac{\eta_{\mathbf{x}}}{2} - \frac{L_{\nabla\Phi}\eta_{\mathbf{x}}^2}{2}\right)\mathbb{E}\left[\left\|\hat{\mathbf{m}}_{\mathbf{x}}^k\right\|^2\right] \\
&\quad + \eta_{\mathbf{x}}\mathbb{E}\left[\left\|\nabla\Phi_{\mathcal{H}}\left(\mathbf{x}^k\right) - \mathbf{m}_{\mathbf{x}}^k\right\|^2\right] + \eta_{\mathbf{x}}\mathbb{E}\left[\left\|\xi_{\mathbf{x}}^k\right\|^2\right].
\end{aligned}
\tag{238}
$$

*Proof.* By using $L_{\nabla\Phi}$-smoothness of upper-level loss function $\Phi$, we obtain

$$
\begin{aligned}
\mathbb{E}\left[\Phi_{\mathcal{H}}\left(\mathbf{x}^{k+1}\right)\right] &\leq \mathbb{E}\left[\Phi_{\mathcal{H}}\left(\mathbf{x}^k\right)\right] + \mathbb{E}\left[\left\langle\nabla\Phi_{\mathcal{H}}\left(\mathbf{x}^k\right), \mathbf{x}^{k+1} - \mathbf{x}^k\right\rangle\right] + \frac{L_{\nabla\Phi}}{2}\mathbb{E}\left[\left\|\mathbf{x}^{k+1} - \mathbf{x}^k\right\|^2\right] \\
&= \mathbb{E}\left[\Phi_{\mathcal{H}}\left(\mathbf{x}^k\right)\right] - \eta_{\mathbf{x}}\mathbb{E}\left[\left\langle\nabla\Phi_{\mathcal{H}}\left(\mathbf{x}^k\right), \hat{\mathbf{m}}_{\mathbf{x}}^k\right\rangle\right] + \frac{L_{\nabla\Phi}\eta_{\mathbf{x}}^2}{2}\mathbb{E}\left[\left\|\hat{\mathbf{m}}_{\mathbf{x}}^k\right\|^2\right] \\
&= \mathbb{E}\left[\Phi_{\mathcal{H}}\left(\mathbf{x}^k\right)\right] - \frac{\eta_{\mathbf{x}}}{2}\mathbb{E}\left[\left\|\nabla\Phi_{\mathcal{H}}\left(\mathbf{x}^k\right)\right\|^2\right] - \frac{\eta_{\mathbf{x}}}{2}\mathbb{E}\left[\left\|\hat{\mathbf{m}}_{\mathbf{x}}^k\right\|^2\right] \\
&\quad + \frac{\eta_{\mathbf{x}}}{2}\mathbb{E}\left[\left\|\nabla\Phi_{\mathcal{H}}\left(\mathbf{x}^k\right) - \hat{\mathbf{m}}_{\mathbf{x}}^k\right\|^2\right] + \frac{L_{\nabla\Phi}\eta_{\mathbf{x}}^2}{2}\mathbb{E}\left[\left\|\hat{\mathbf{m}}_{\mathbf{x}}^k\right\|^2\right],
\end{aligned}
\tag{239}
$$

where the last equality uses $\langle a, b\rangle = \frac{1}{2}(a^2 + b^2 - (a+b)^2)$.

$$
\begin{aligned}
\mathbb{E}\left[\Phi_{\mathcal{H}}\left(\mathbf{x}^{k+1}\right)\right] &\leq \mathbb{E}\left[\Phi_{\mathcal{H}}\left(\mathbf{x}^k\right)\right] - \frac{\eta_{\mathbf{x}}}{2}\mathbb{E}\left[\left\|\nabla\Phi_{\mathcal{H}}\left(\mathbf{x}^k\right)\right\|^2\right] - \left(\frac{\eta_{\mathbf{x}}}{2} - \frac{L_{\nabla\Phi}\eta_{\mathbf{x}}^2}{2}\right)\mathbb{E}\left[\left\|\hat{\mathbf{m}}_{\mathbf{x}}^k\right\|^2\right] \\
&\quad + \eta_{\mathbf{x}}\mathbb{E}\left[\left\|\nabla\Phi_{\mathcal{H}}\left(\mathbf{x}^k\right) - \mathbf{m}_{\mathbf{x}}^k\right\|^2\right] + \eta_{\mathbf{x}}\mathbb{E}\left[\left\|\xi_{\mathbf{x}}^k\right\|^2\right],
\end{aligned}
\tag{240}
$$

where the last inequality uses the Cauchy-Schwarz inequality. $\square$

### J.3   Proof of Supporting Lemmas for BR-FedBiP

#### J.3.1   Proof of Lemma 18

We now restate the lemma 18 and present its proof.

**Lemma 18.** *Suppose Assumptions 3 and 4 hold,and the robust aggregation rule $AGG(\cdot)$ satisfies the $(\delta, \kappa)$-robustness criterion. The aggregation errors in the BR-FedBi algorithm (Algorithm 1) satisfy the following inequalities:*

$$(1) \quad \mathbb{E}\left[\left\|\xi_{\mathbf{y}}^k\right\|^2\right] \leq 4\kappa\zeta_g^2 + 4\kappa\left(1 + \frac{1}{(1-\delta)n}\right)\sigma^2,$$

$$(2) \quad \mathbb{E}\left[\left\|\xi_{\mathbf{x}}^k\right\|^2\right] \leq 8\kappa\left(\rho^2\zeta_g^2 + \zeta_f^2\right) + 8\kappa\left(1 + \frac{1}{(1-\delta)n}\right)\sigma_1^2,$$

$$(3) \quad \mathbb{E}\left[\left\|\xi_{\mathbf{z}}^k\right\|^2\right] \leq 8\kappa\left(\rho^2\zeta_g^2 + \zeta_f^2\right) + 8\kappa\left(1 + \frac{1}{(1-\delta)n}\right)\sigma_1^2,$$

*where $\sigma_1^2 = (1 + \rho^2)\sigma^2$ and $\rho = \frac{C_f}{\mu_g}$.*

*Proof.* **Analysis for $\mathbb{E}\left[\left\|\xi_{\mathbf{y}}^k\right\|^2\right]$.** Recall that

$$\xi_{\mathbf{y}}^k = \hat{\mathbf{v}}_{\mathbf{y}}^k - \mathbf{v}_{\mathbf{y}}^k.$$

For an arbitrary step $k > 1$, we obtain that

$$\mathbb{E}\left[\left\|\hat{\mathbf{v}}_{\mathbf{y}}^k - \mathbf{v}_{\mathbf{y}}^k\right\|^2\right]$$

$$\leq \frac{\kappa}{(1-\delta)n}\sum_{i \in \mathcal{H}}\left(p\mathbb{E}\left[\left\|\nabla_{\mathbf{y}}G_i(\mathbf{x}^k, \mathbf{y}^k) - \nabla_{\mathbf{y}}G(\mathbf{x}^k, \mathbf{y}^k)\right\|^2\right]\right.$$

$$\left. + (1-p)\mathbb{E}\left[\left\|\nabla_{\mathbf{y}}G_i\left(\mathbf{x}^k, \mathbf{y}^k\right) - \nabla_{\mathbf{y}}G_i\left(\mathbf{x}^{k-1}, \mathbf{y}^{k-1}\right) - \left(\nabla_{\mathbf{y}}G\left(\mathbf{x}^k, \mathbf{y}^k\right) - \nabla_{\mathbf{y}}G\left(\mathbf{x}^{k-1}, \mathbf{y}^{k-1}\right)\right)\right\|^2\right]\right),$$

$$(241)$$

where the inequality uses the expectation over $c_k$,and the inequality holds due to the $(\delta, \kappa)$-robustness of $AGG(\cdot)$ as defined in definition 2.

Using the the Cauchy-Schwarz inequality, we obtain that

$$\mathbb{E}\left[\left\|\hat{\mathbf{v}}_{\mathbf{y}}^k - \mathbf{v}_{\mathbf{y}}^k\right\|^2\right]$$

$$\leq \frac{\kappa}{(1-\delta)n}\sum_{i \in \mathcal{H}}\left(2\mathbb{E}\left[\left\|\nabla_{\mathbf{y}}G_i(\mathbf{x}^k, \mathbf{y}^k) - \nabla_{\mathbf{y}}G(\mathbf{x}^k, \mathbf{y}^k)\right\|^2\right]\right.$$

$$\left. + 2\mathbb{E}\left[\left\|\nabla_{\mathbf{y}}G_i(\mathbf{x}^{k-1}, \mathbf{y}^{k-1}) - \nabla_{\mathbf{y}}G(\mathbf{x}^{k-1}, \mathbf{y}^{k-1})\right\|^2\right]\right) \qquad (242)$$

$$\leq \frac{\kappa}{(1-\delta)n}\sum_{i \in \mathcal{H}}4\kappa\zeta_g^2 + 4\kappa\left(1 + \frac{1}{((1-\delta)n)}\right)\sigma^2$$

$$\leq 4\kappa\zeta_g^2 + 4\kappa\left(1 + \frac{1}{((1-\delta)n)}\right)\sigma^2,$$

where the second inequality uses Assumptions 3 and 4.

Similarly, the inequality for $\mathbf{x}$ and $\mathbf{z}$ can be established by following the same proof as in Lemma 9. $\qquad\square$

#### J.3.2   Proof of Lemma 21

We now restate the lemma 21 and present its proof.

**Lemma 21.** *Suppose Assumptions 2 and 3 hold, consider Algorithm BR-FedBiP in 1. For each $k \in [K]$, we have*

$$
(1) \quad \mathbb{E}\left[\left\|\mathbf{v}_{\mathbf{y}}^{k+1} - \mathbb{E}\left(\mathbf{D}_{\mathbf{y}}^{k+1}\right)\right\|^2\right]
$$

$$
\leq p\frac{\sigma^2}{b'(n-f)} + (1-p)\mathbb{E}\left[\left\|\mathbf{v}_{\mathbf{y}}^k - \mathbb{E}(\mathbf{D}_{\mathbf{y}}^k)\right\|^2\right]
$$

$$
+ \frac{(1-p)L_g^2}{b}\left(\eta_{\mathbf{x}}^2\mathbb{E}\left[\left\|\hat{\mathbf{v}}_{\mathbf{x}}^k\right\|^2\right] + 3\eta_{\mathbf{y}}^2\mathbb{E}\left[\left\|\mathbf{v}_{\mathbf{y}}^k - \mathbb{E}\left(\mathbf{D}_{\mathbf{y}}^k\right)\right\|^2\right]\right)
$$

$$
+ \frac{(1-p)L_g^2}{b}\left(3L_g^2\eta_{\mathbf{y}}^2\mathbb{E}\left[\left\|\mathbf{y}^k - \mathbf{y}_\star^k\right\|^2\right] + 3\eta_{\mathbf{y}}^2\mathbb{E}\left[\left\|\xi_{\mathbf{y}}^k\right\|^2\right]\right),
$$

$$
(2) \quad \mathbb{E}\left[\left\|\mathbf{v}_{\mathbf{x}}^{k+1} - \mathbb{E}\left(\mathbf{D}_{\mathbf{x}}^{k+1}\right)\right\|^2\right]
$$

$$
\leq p\frac{\sigma_1^2}{b'(n-f)} + (1-p)\mathbb{E}\left[\left\|\mathbf{v}_{\mathbf{x}}^k - \mathbb{E}(\mathbf{D}_{\mathbf{x}}^k)\right\|^2\right] + 4\frac{(1-p)}{b}L_1^2\eta_{\mathbf{x}}^2\mathbb{E}\left[\left\|\hat{\mathbf{v}}_{\mathbf{x}}^k\right\|^2\right]
$$

$$
+ 12\frac{(1-p)}{b}L_1^2\eta_{\mathbf{y}}^2\left(\mathbb{E}\left[\left\|\xi_{\mathbf{y}}^k\right\|^2\right] + \mathbb{E}\left[\left\|\mathbf{v}_{\mathbf{y}}^k - \mathbb{E}\left(\mathbf{D}_{\mathbf{y}}^k\right)\right\|^2\right] + L_g^2\mathbb{E}\left[\left\|\mathbf{y}^k - \mathbf{y}_\star^k\right\|^2\right]\right)
$$

$$
+ \frac{12(1-p)}{b}L_g^2\eta_{\mathbf{z}}^2\left(\mathbb{E}\left[\left\|\xi_{\mathbf{z}}^k\right\|^2\right] + \mathbb{E}\left[\left\|\mathbf{v}_{\mathbf{z}}^k - \mathbb{E}\left(\mathbf{D}_{\mathbf{z}}^k\right)\right\|^2\right]\right)
$$

$$
+ \frac{12(1-p)}{b}L_g^2\eta_{\mathbf{z}}^2\left(3L_g^2\mathbb{E}\left[\left\|\mathbf{z}^k - \mathbf{z}_\star^k\right\|^2\right] + 3L_2^2\mathbb{E}\left[\left\|\mathbf{y}^k - \mathbf{y}_\star^k\right\|^2\right]\right),
$$

$$
(3) \quad \mathbb{E}\left[\left\|\mathbf{v}_{\mathbf{z}}^{k+1} - \mathbb{E}\left(\mathbf{D}_{\mathbf{z}}^{k+1}\right)\right\|^2\right]
$$

$$
\leq p\frac{\sigma_1^2}{b'(n-f)} + (1-p)\mathbb{E}\left[\left\|\mathbf{v}_{\mathbf{z}}^k - \mathbb{E}(\mathbf{D}_{\mathbf{z}}^k)\right\|^2\right] + 4\frac{(1-p)}{b}L_1^2\eta_{\mathbf{x}}^2\mathbb{E}\left[\left\|\hat{\mathbf{v}}_{\mathbf{x}}^k\right\|^2\right]
$$

$$
+ 12\frac{(1-p)}{b}L_1^2\eta_{\mathbf{y}}^2\left(\mathbb{E}\left[\left\|\xi_{\mathbf{y}}^k\right\|^2\right] + \mathbb{E}\left[\left\|\mathbf{v}_{\mathbf{y}}^k - \mathbb{E}\left(\mathbf{D}_{\mathbf{y}}^k\right)\right\|^2\right] + L_g^2\mathbb{E}\left[\left\|\mathbf{y}^k - \mathbf{y}_\star^k\right\|^2\right]\right)
$$

$$
+ \frac{12(1-p)}{b}L_g^2\eta_{\mathbf{z}}^2\left(\mathbb{E}\left[\left\|\xi_{\mathbf{z}}^k\right\|^2\right] + \mathbb{E}\left[\left\|\mathbf{v}_{\mathbf{z}}^k - \mathbb{E}\left(\mathbf{D}_{\mathbf{z}}^k\right)\right\|^2\right] + 3L_g^2\mathbb{E}\left[\left\|\mathbf{z}^k - \mathbf{z}_\star^k\right\|^2\right]\right)
$$

$$
+ \frac{12(1-p)}{b}L_g^2\eta_{\mathbf{z}}^2\left(3L_2^2\mathbb{E}\left[\left\|\mathbf{y}^k - \mathbf{y}_\star^k\right\|^2\right]\right).
$$

*Proof.* **Analysis for** $\mathbb{E}\left[\left\|\mathbf{v}_{\mathbf{y}}^{k+1} - \mathbb{E}\left(\mathbf{D}_{\mathbf{y}}^{k+1}\right)\right\|^2\right]$**.** Recall that

$$
\mathbf{v}_{\mathbf{y},i}^k = \begin{cases} \nabla G_i\left(\mathbf{u}^k; \boldsymbol{\varphi}_i^{k'}\right), & \text{w.p. } p, \\ \mathbf{v}_{\mathbf{y}}^{k-1} + \nabla G_i\left(\mathbf{u}^k; \boldsymbol{\varphi}_i^k\right) - \nabla G_i\left(\mathbf{u}^{k-1}; \boldsymbol{\varphi}_i^k\right), & \text{w.p. } 1-p. \end{cases}
$$

To analyze the variance of $\mathbf{v}_{\mathbf{y}}^{k+1}$ relative to $\mathbb{E}\left(\mathbf{D}_{\mathbf{y}}^{k+1}\right)$, we consider an arbitrary step $k > 1$. The squared expectation of the deviation is given by:

$$
\mathbb{E}\left[\left\|\mathbf{v}_{\mathbf{y}}^{k+1} - \mathbb{E}\left(\mathbf{D}_{\mathbf{y}}^{k+1}\right)\right\|^2\right]
$$

$$
\leq p\left(\mathbb{E}\left[\left\|\nabla_{\mathbf{y}}G(\mathbf{x}_{k+1}, \mathbf{y}_{k+1}) - \mathbb{E}\left(\mathbf{D}_{\mathbf{y}}^{k+1}\right)\right\|^2\right]\right) + (1-p)\mathbb{E}\left[\left\|\mathbf{v}_{\mathbf{y}}^k + \mathbf{D}_{\mathbf{y}}^{k+1} - \mathbf{D}_{\mathbf{y}}^k - \mathbb{E}\left(\mathbf{D}_{\mathbf{y}}^{k+1}\right)\right\|^2\right]
$$

$$
\leq p\frac{\sigma^2}{b'(n-f)} + (1-p)\mathbb{E}\left[\left\|\mathbf{v}_{\mathbf{y}}^k - \mathbb{E}\left(\mathbf{D}_{\mathbf{y}}^k\right)\right\|^2\right] + (1-p)\mathbb{E}\left[\left\|\mathbf{D}_{\mathbf{y}}^{k+1} - \mathbf{D}_{\mathbf{y}}^k - \mathbb{E}\left(\mathbf{D}_{\mathbf{y}}^{k+1}\right) + \mathbb{E}\left(\mathbf{D}_{\mathbf{y}}^k\right)\right\|^2\right],
$$

$$
(243)
$$

where the first inequality uses the expectation over $c_k$, the second inequality uses unbiased estimation in Assumption 3.

Using $\mathbb{E}\|X - \mathbb{E}[X]\|^2 \leq \mathbb{E}\left[X^2\right]$, we can imply that

$$
\mathbb{E}\left[\left\|\mathbf{v}_{\mathbf{y}}^k + \mathbf{D}_{\mathbf{y}}^{k+1} - \mathbf{D}_{\mathbf{y}}^k - \mathbb{E}\left(\mathbf{D}_{\mathbf{y}}^{k+1}\right)\right\|^2\right] \leq \mathbb{E}\left[\left\|\mathbf{D}_{\mathbf{y}}^{k+1} - \mathbf{D}_{\mathbf{y}}^k\right\|^2\right]. \qquad (244)
$$

Plugging (244) into (243), we obtain that

$$
\mathbb{E}\left[\left\|\mathbf{v}_{\mathbf{y}}^{k+1} - \mathbb{E}\left(\mathbf{D}_{\mathbf{y}}^{k+1}\right)\right\|^2\right]
$$
$$
\leq p\frac{\sigma^2}{b'(n-f)} + (1-p)\mathbb{E}\left[\left\|\mathbf{v}_{\mathbf{y}}^k - \mathbb{E}\left(\mathbf{D}_{\mathbf{y}}^k\right)\right\|^2\right] + (1-p)\mathbb{E}\left[\left\|\mathbf{D}_{\mathbf{y}}^{k+1} - \mathbf{D}_{\mathbf{y}}^k\right\|^2\right]
$$
$$
\leq p\frac{\sigma^2}{b'(n-f)} + (1-p)\mathbb{E}\left[\left\|\mathbf{v}_{\mathbf{y}}^k - \mathbb{E}\left(\mathbf{D}_{\mathbf{y}}^k\right)\right\|^2\right] + \frac{(1-p)L_g^2}{b}\left(\eta_{\mathbf{y}}^2\mathbb{E}\left[\left\|\hat{\mathbf{v}}_{\mathbf{y}}^k\right\|^2\right] + \eta_{\mathbf{x}}^2\mathbb{E}\left[\left\|\hat{\mathbf{v}}_{\mathbf{x}}^k\right\|^2\right]\right)
$$
$$
\leq p\frac{\sigma^2}{b'(n-f)} + (1-p)\mathbb{E}\left[\left\|\mathbf{v}_{\mathbf{y}}^k - \mathbb{E}\left(\mathbf{D}_{\mathbf{y}}^k\right)\right\|^2\right] + \frac{(1-p)L_g^2}{b}\left(\eta_{\mathbf{x}}^2\mathbb{E}\left[\left\|\hat{\mathbf{v}}_{\mathbf{x}}^k\right\|^2\right] + 3\eta_{\mathbf{y}}^2\mathbb{E}\left[\left\|\mathbf{v}_{\mathbf{y}}^k - \mathbb{E}\left(\mathbf{D}_{\mathbf{y}}^k\right)\right\|^2\right]\right.
$$
$$
\left. + 3L_g^2\eta_{\mathbf{y}}^2\mathbb{E}\left[\left\|\mathbf{y}^k - \mathbf{y}_\star^k\right\|^2\right] + 3\eta_{\mathbf{y}}^2\mathbb{E}\left[\left\|\xi_{\mathbf{y}}^k\right\|^2\right]\right),
$$

$$(245)$$

where the second inequality uses Assumption 7, and last inequality uses Lemma 27.

**Analysis for $\mathbb{E}\left[\|\mathbf{v}_{\mathbf{x}}^{k+1} - \mathbb{E}\left(\mathbf{D}_{\mathbf{x}}^{k+1}\right)\|^2\right]$.**

To analyze the variance of $\mathbf{v}_{\mathbf{x}}^{k+1}$ relative to $\mathbb{E}\left(\mathbf{D}_{\mathbf{y}}^{k+1}\right)$, we consider an arbitrary step $k > 1$. The squared expectation of the deviation is given by:

$$
\mathbb{E}\left[\left\|\mathbf{v}_{\mathbf{x}}^{k+1} - \mathbb{E}\left(\mathbf{D}_{\mathbf{x}}^{k+1}\right)\right\|^2\right]
$$
$$
\leq p\frac{\sigma_1^2}{b'(n-f)} + (1-p)\mathbb{E}\left[\left\|\mathbf{v}_{\mathbf{x}}^k + \mathbf{D}_{\mathbf{x}}^{k+1} - \mathbf{D}_{\mathbf{x}}^k - \mathbb{E}\left(\mathbf{D}_{\mathbf{x}}^{k+1}\right)\right\|^2\right]
$$
$$
\leq p\frac{\sigma_1^2}{b'(n-f)} + (1-p)\mathbb{E}\left[\left\|\mathbf{v}_{\mathbf{x}}^k - \mathbb{E}\left(\mathbf{D}_{\mathbf{x}}^k\right)\right\|^2\right] + (1-p)\mathbb{E}\left[\left\|\mathbf{D}_{\mathbf{x}}^{k+1} - \mathbf{D}_{\mathbf{x}}^k - \mathbb{E}\left(\mathbf{D}_{\mathbf{x}}^{k+1}\right) + \mathbb{E}\left(\mathbf{D}_{\mathbf{x}}^k\right)\right\|^2\right],
$$

$$(246)$$

where the first inequality uses the expectation over $c_k$, the second inequality uses unbiased estimation in Assumption 3.

Using $\mathbb{E}\|X - \mathbb{E}[X]\|^2 \leq \mathbb{E}\left[X^2\right]$, we can imply that

$$
\mathbb{E}\left[\left\|\mathbf{v}_{\mathbf{x}}^k + \mathbf{D}_{\mathbf{x}}^{k+1} - \mathbf{D}_{\mathbf{x}}^k - \mathbb{E}\left(\mathbf{D}_{\mathbf{x}}^{k+1}\right)\right\|^2\right] \leq \mathbb{E}\left[\left\|\mathbf{D}_{\mathbf{x}}^{k+1} - \mathbf{D}_{\mathbf{x}}^k\right\|^2\right]. \tag{247}
$$

Plugging (247) into (246), we obtain that

$$
\mathbb{E}\left[\left\|\mathbf{v}_{\mathbf{x}}^{k+1} - \mathbb{E}\left(\mathbf{D}_{\mathbf{x}}^{k+1}\right)\right\|^2\right]
$$
$$
\leq p\frac{\sigma_1^2}{b'(n-f)} + (1-p)\mathbb{E}\left[\left\|\mathbf{v}_{\mathbf{x}}^k - \mathbb{E}\left(\mathbf{D}_{\mathbf{x}}^k\right)\right\|^2\right] + (1-p)\mathbb{E}\left[\left\|\mathbf{D}_{\mathbf{x}}^{k+1} - \mathbf{D}_{\mathbf{x}}^k\right\|^2\right]
$$
$$
\leq p\frac{\sigma_1^2}{b'(n-f)} + (1-p)\mathbb{E}\left[\left\|\mathbf{v}_{\mathbf{x}}^k - \mathbb{E}\left(\mathbf{D}_{\mathbf{x}}^k\right)\right\|^2\right] + 4\frac{(1-p)}{b}L_1^2\eta_{\mathbf{x}}^2\mathbb{E}\left[\left\|\hat{\mathbf{v}}_{\mathbf{x}}^k\right\|^2\right]
$$
$$
+ 4\frac{(1-p)}{b}L_1^2\eta_{\mathbf{y}}^2\mathbb{E}\left[\left\|\hat{\mathbf{v}}_{\mathbf{y}}^k\right\|^2\right] + \frac{4(1-p)}{b}L_g^2\eta_{\mathbf{z}}^2\mathbb{E}\left[\left\|\hat{\mathbf{v}}_{\mathbf{z}}^k\right\|^2\right]
$$
$$
\leq p\frac{\sigma_1^2}{b'(n-f)} + (1-p)\mathbb{E}\left[\left\|\mathbf{v}_{\mathbf{x}}^k - \mathbb{E}\left(\mathbf{D}_{\mathbf{x}}^k\right)\right\|^2\right] + 4\frac{(1-p)}{b}L_1^2\eta_{\mathbf{x}}^2\mathbb{E}\left[\left\|\hat{\mathbf{v}}_{\mathbf{x}}^k\right\|^2\right]
$$
$$
+ 12\frac{(1-p)}{b}L_1^2\eta_{\mathbf{y}}^2\left(\mathbb{E}\left[\left\|\xi_{\mathbf{y}}^k\right\|^2\right] + \mathbb{E}\left[\left\|\mathbf{v}_{\mathbf{y}}^k - \mathbb{E}\left(\mathbf{D}_{\mathbf{y}}^k\right)\right\|^2\right] + L_g^2\mathbb{E}\left[\left\|\mathbf{y}^k - \mathbf{y}_\star^k\right\|^2\right]\right)
$$
$$
+ \frac{12(1-p)}{b}L_g^2\eta_{\mathbf{z}}^2\left(\mathbb{E}\left[\left\|\xi_{\mathbf{z}}^k\right\|^2\right] + \mathbb{E}\left[\left\|\mathbf{v}_{\mathbf{z}}^k - \mathbb{E}\left(\mathbf{D}_{\mathbf{z}}^k\right)\right\|^2\right] + 3L_g^2\mathbb{E}\left[\left\|\mathbf{z}^k - \mathbf{z}_\star^k\right\|^2\right] + 3L_2^2\mathbb{E}\left[\left\|\mathbf{y}^k - \mathbf{y}_\star^k\right\|^2\right]\right),
$$

$$(248)$$

where the second inequality uses Assumption 7, and last inequality uses Lemma 27.

**Analysis for $\mathbb{E}\left[\|\mathbf{v}_{\mathbf{z}}^{k+1} - \mathbb{E}\left(\mathbf{D}_{\mathbf{z}}^{k+1}\right)\|^2\right]$.**

To analyze the variance of $\mathbf{v}_{\mathbf{z}}^{k+1}$ relative to $\mathbb{E}\left(\mathbf{D}_{\mathbf{z}}^{k+1}\right)$, we consider an arbitrary step $k > 1$. The squared expectation of the deviation is given by:

$$\mathbb{E}\left[\left\|\mathbf{v}_{\mathbf{z}}^{k+1} - \mathbb{E}\left(\mathbf{D}_{\mathbf{z}}^{k+1}\right)\right\|^2\right]$$

$$\leq p\frac{\sigma_1^2}{b'(n-f)} + (1-p)\mathbb{E}\left[\left\|\mathbf{v}_{\mathbf{z}}^k + \mathbf{D}_{\mathbf{z}}^{k+1} - \mathbf{D}_{\mathbf{z}}^k - \mathbb{E}\left(\mathbf{D}_{\mathbf{z}}^{k+1}\right)\right\|^2\right]$$

$$\leq p\frac{\sigma_1^2}{b'(n-f)} + (1-p)\mathbb{E}\left[\left\|\mathbf{v}_{\mathbf{z}}^k - \mathbb{E}\left(\mathbf{D}_{\mathbf{z}}^k\right)\right\|^2\right] + (1-p)\mathbb{E}\left[\left\|\mathbf{D}_{\mathbf{z}}^{k+1} - \mathbf{D}_{\mathbf{z}}^k - \mathbb{E}\left(\mathbf{D}_{\mathbf{z}}^{k+1}\right) + \mathbb{E}\left(\mathbf{D}_{\mathbf{z}}^k\right)\right\|^2\right],$$
(249)

where the first inequality uses the expectation over $c_k$, the second inequality uses unbiased estimation in Assumption 3. Using $\mathbb{E}\|X - \mathbb{E}[X]\|^2 \leq \mathbb{E}\left[X^2\right]$, we can imply that

$$\mathbb{E}\left[\left\|\mathbf{v}_{\mathbf{z}}^k + \mathbf{D}_{\mathbf{z}}^{k+1} - \mathbf{D}_{\mathbf{z}}^k - \mathbb{E}\left(\mathbf{D}_{\mathbf{z}}^{k+1}\right)\right\|^2\right] \leq \mathbb{E}\left[\left\|\mathbf{D}_{\mathbf{z}}^{k+1} - \mathbf{D}_{\mathbf{z}}^k\right\|^2\right]. \tag{250}$$

Plugging (250) into (249), we obtain that

$$\mathbb{E}\left[\left\|\mathbf{v}_{\mathbf{z}}^{k+1} - \mathbb{E}\left(\mathbf{D}_{\mathbf{z}}^{k+1}\right)\right\|^2\right]$$

$$\leq p\frac{\sigma_1^2}{b'(n-f)} + (1-p)\mathbb{E}\left[\left\|\mathbf{v}_{\mathbf{z}}^k - \mathbb{E}\left(\mathbf{D}_{\mathbf{z}}^k\right)\right\|^2\right] + (1-p)\mathbb{E}\left[\left\|\mathbf{D}_{\mathbf{z}}^{k+1} - \mathbf{D}_{\mathbf{x}}^k\right\|^2\right]$$

$$\leq p\frac{\sigma_1^2}{b'(n-f)} + (1-p)\mathbb{E}\left[\left\|\mathbf{v}_{\mathbf{z}}^k - \mathbb{E}\left(\mathbf{D}_{\mathbf{z}}^k\right)\right\|^2\right] + 4\frac{(1-p)}{b}L_1^2\eta_{\mathbf{x}}^2\mathbb{E}\left[\left\|\hat{\mathbf{v}}_{\mathbf{x}}^k\right\|^2\right]$$

$$+ 4\frac{(1-p)}{b}L_1^2\eta_{\mathbf{y}}^2\mathbb{E}\left[\left\|\hat{\mathbf{v}}_{\mathbf{y}}^k\right\|^2\right] + \frac{4(1-p)}{b}L_g^2\eta_{\mathbf{z}}^2\mathbb{E}\left[\left\|\hat{\mathbf{v}}_{\mathbf{z}}^k\right\|^2\right]$$

$$\leq p\frac{\sigma_1^2}{b'(n-f)} + (1-p)\mathbb{E}\left[\left\|\mathbf{v}_{\mathbf{z}}^k - \mathbb{E}\left(\mathbf{D}_{\mathbf{z}}^k\right)\right\|^2\right] + 4\frac{(1-p)}{b}L_1^2\eta_{\mathbf{x}}^2\mathbb{E}\left[\left\|\hat{\mathbf{v}}_{\mathbf{x}}^k\right\|^2\right]$$

$$+ 12\frac{(1-p)}{b}L_1^2\eta_{\mathbf{y}}^2\left(\mathbb{E}\left[\left\|\xi_{\mathbf{y}}^k\right\|^2\right] + \mathbb{E}\left[\left\|\mathbf{v}_{\mathbf{y}}^k - \mathbb{E}\left(\mathbf{D}_{\mathbf{y}}^k\right)\right\|^2\right] + L_g^2\mathbb{E}\left[\left\|\mathbf{y}^k - \mathbf{y}_\star^k\right\|^2\right]\right)$$

$$+ \frac{12(1-p)}{b}L_g^2\eta_{\mathbf{z}}^2\left(\mathbb{E}\left[\left\|\xi_{\mathbf{z}}^k\right\|^2\right] + \mathbb{E}\left[\left\|\mathbf{v}_{\mathbf{z}}^k - \mathbb{E}\left(\mathbf{D}_{\mathbf{z}}^k\right)\right\|^2\right] + 3L_g^2\mathbb{E}\left[\left\|\mathbf{z}^k - \mathbf{z}_\star^k\right\|^2\right] + 3L_2^2\mathbb{E}\left[\left\|\mathbf{y}^k - \mathbf{y}_\star^k\right\|^2\right]\right),$$
(251)

where the second inequality uses Assumption 7, and last inequality uses Lemma 27.

$\square$

**Lemma 27** (Bounded second moment). *Suppose Assumptions 1-4 hold. For each step $k \in [K]$, we have*

$$(1) \quad \mathbb{E}\left[\left\|\hat{\mathbf{v}}_{\mathbf{y}}^k\right\|^2\right] \leq 2\mathbb{E}\left[\left\|\mathbf{v}_{\mathbf{x}}^k\right\|^2\right] + 2\mathbb{E}\left[\left\|\xi_{\mathbf{x}}^k\right\|^2\right],$$

$$(2) \quad \mathbb{E}\left[\left\|\hat{\mathbf{v}}_{\mathbf{y}}^k\right\|^2\right] \leq 3\mathbb{E}\left[\left\|\xi_{\mathbf{y}}^k\right\|^2\right] + 3\mathbb{E}\left[\left\|\mathbf{v}_{\mathbf{y}}^k - \mathbb{E}\left(\mathbf{D}_{\mathbf{y}}^k\right)\right\|^2\right] + 3L_g^2\mathbb{E}\left[\left\|\mathbf{y}^k - \mathbf{y}_\star^k\right\|^2\right],$$
(252)

$$(3) \quad \mathbb{E}\left[\left\|\hat{\mathbf{v}}_{\mathbf{z}}^k\right\|^2\right] \leq 3\mathbb{E}\left[\left\|\xi_{\mathbf{z}}^k\right\|^2\right] + 3\mathbb{E}\left[\left\|\mathbf{v}_{\mathbf{z}}^k - \mathbb{E}\left(\mathbf{D}_{\mathbf{z}}^k\right)\right\|^2\right] + 9L_g^2\mathbb{E}\left[\left\|\mathbf{z}^k - \mathbf{z}_\star^k\right\|^2\right]$$
$$+ 9L_2^2\mathbb{E}\left[\left\|\mathbf{y}^k - \mathbf{y}_\star^k\right\|^2\right].$$

*Proof.* **Bounded second moment for x.**

$$\mathbb{E}\left[\left\|\hat{\mathbf{v}}_{\mathbf{x}}^k\right\|^2\right] = \mathbb{E}\left[\left\|\hat{\mathbf{v}}_{\mathbf{x}}^k - \mathbf{v}_{\mathbf{x}}^k + \mathbf{v}_{\mathbf{x}}^k\right\|^2\right] \leq 2\mathbb{E}\left[\left\|\mathbf{v}_{\mathbf{x}}^k\right\|^2\right] + 2\mathbb{E}\left[\left\|\xi_{\mathbf{x}}^k\right\|^2\right]. \tag{253}$$

**Bounded second moment for y.** Decomposing $\hat{\mathbf{v}}_{\mathbf{y}}^k$ into three components, we have

$$\mathbb{E}\left[\left\|\hat{\mathbf{v}}_{\mathbf{y}}^k\right\|^2\right] = \mathbb{E}\left[\left\|\hat{\mathbf{v}}_{\mathbf{y}}^k - \mathbf{v}_{\mathbf{y}}^k + \mathbf{v}_{\mathbf{y}}^k - \mathbb{E}\left(\mathbf{D}_{\mathbf{y}}^k\right) + \mathbb{E}\left(\mathbf{D}_{\mathbf{y}}^k\right)\right\|^2\right]$$

$$\leq 3\mathbb{E}\left[\left\|\xi_{\mathbf{y}}^k\right\|^2\right] + 3\mathbb{E}\left[\left\|\mathbf{v}_{\mathbf{y}}^k - \mathbb{E}\left(\mathbf{D}_{\mathbf{y}}^k\right)\right\|^2\right] + 3\mathbb{E}\left[\left\|\mathbb{E}\left(\mathbf{D}_{\mathbf{y}}^k\right)\right\|^2\right] \tag{254}$$

$$\leq 3\mathbb{E}\left[\left\|\xi_{\mathbf{y}}^k\right\|^2\right] + 3\mathbb{E}\left[\left\|\mathbf{v}_{\mathbf{y}}^k - \mathbb{E}\left(\mathbf{D}_{\mathbf{y}}^k\right)\right\|^2\right] + 3L_g^2\mathbb{E}\left[\left\|\mathbf{y}^k - \mathbf{y}_\star^k\right\|^2\right].$$

where the first inequality uses Cauchy-Schwartz inequality and the second inequality uses Assumption 1 and (218).

**Bounded second moment for z.** Decomposing $\hat{\mathbf{v}}_{\mathbf{z}}^k$ into three components, we have

$$
\begin{aligned}
\mathbb{E}\left[\left\|\hat{\mathbf{v}}_{\mathbf{z}}^k\right\|^2\right] &= \mathbb{E}\left[\left\|\hat{\mathbf{v}}_{\mathbf{z}}^k - \mathbf{v}_{\mathbf{z}}^k + \mathbf{v}_{\mathbf{z}}^k - \mathbb{E}\left(\mathbf{D}_{\mathbf{z}}^k\right) + \mathbb{E}\left(\mathbf{D}_{\mathbf{z}}^k\right)\right\|^2\right] \\
&\leq 3\mathbb{E}\left[\left\|\xi_{\mathbf{z}}^k\right\|^2\right] + 3\mathbb{E}\left[\left\|\mathbf{v}_{\mathbf{z}}^k - \mathbb{E}\left(\mathbf{D}_{\mathbf{z}}^k\right)\right\|^2\right] + 3\mathbb{E}\left[\left\|\mathbb{E}\left(\mathbf{D}_{\mathbf{z}}^k\right)\right\|^2\right] \\
&\leq 3\mathbb{E}\left[\left\|\xi_{\mathbf{z}}^k\right\|^2\right] + 3\mathbb{E}\left[\left\|\mathbf{v}_{\mathbf{z}}^k - \mathbb{E}\left(\mathbf{D}_{\mathbf{z}}^k\right)\right\|^2\right] + 9L_g^2\mathbb{E}\left[\left\|\mathbf{z}^k - \mathbf{z}_\star^k\right\|^2\right] + 9L_2^2\mathbb{E}\left[\left\|\mathbf{y}^k - \mathbf{y}_\star^k\right\|^2\right],
\end{aligned}
$$
$$(255)$$

where the first inequality uses Cauchy-Schwartz inequality and the second inequality uses Assumption 1 and (220). $\qquad\square$