# OpenReview forum: "Single-Loop Byzantine-Resilient Federated Bilevel Optimization"
_ICLR.cc/2026/Conference — ICLR 2026 Poster_

### Official Review · Reviewer_h9q3 · 2025-10-25

**Soundness:** 3
**Presentation:** 3
**Contribution:** 3
**Rating:** 4
**Confidence:** 4

**Summary:**

The paper studies federated bilevel optimization (FBO) under Byzantine clients. It first presents an algorithm‑independent asymptotic lower bound. Algorithmically, the paper proposes a single‑loop Byzantine‑robust FBO framework BR‑FedBi, plus a Polyak‑momentum version (BR‑FedBiM) and a PAGE variance‑reduced version (BR‑FedBiP). With momentum, the method is shown to match the lower bound up to constants, claiming “optimal Byzantine robustness.” Experiments also validate the advantages of the proposed algorithm.

**Strengths:**

1. First Byzantine lower bound for FBO. Extends single‑level robust learning lower bounds to bilevel settings, clarifying that both upper‑ and lower‑level heterogeneity amplify error and that the breakdown depends on $B_f,B_g$.
2. Theory–algorithm closure. After giving the lower bound, BR‑FedBiM attains it (up to constants), which is convincing.
3. Robust empirical gains across diverse attacks/aggregators with sizeable margins.

**Weaknesses:**

1. External validity / representativeness. Experiments focus on hyper‑representation with small models and datasets. Add a more realistic, higher‑dimensional FBO (e.g., hyperparameter optimization of weight decay/augmentation, or meta‑learning / personalized FL) and report end‑to‑end communication volume, latency, and speedups to support the single‑loop efficiency claim.
2. Lacked related works. The related works section lacks coverage of distributed stochastic bilevel optimization, particularly decentralized bilevel optimization, which has gained an increasing focus recently. Therefore, the author should include a review of relevant works on decentralized bilevel optimization in this section.
3. Near‑critical $\delta$ behavior & aggregator choice. Theory constrains $\kappa$ vs. $\delta$, but the paper lacks a systematic study when $\delta$ approaches the breakdown. Provide $\delta$ sweeps and compare $\kappa$ across aggregators to probe the near‑bound regime.
4. Assumption strength & measurability. Mean‑square smoothness and $(\zeta,B)$‑heterogeneity are standard but not easily measurable in practice. Offer empirical estimates (from client gradient statistics) for $\zeta_f,\zeta_g,B_f,B_g$ to connect constants to practice.
5. Baseline tuning fairness. Has BILANTINE been tuned fairly for communication rounds, stepsizes, batch sizes? Include ablations under equal communication budget or equal wall‑clock, and disclose full tuning tables.
6. Engineering clarifications. Clarify PAGE trigger probability $p$, mini‑batch sizes $b$, momentum/stepsize coupling (theory ranges vs. actual values). Address practical FL issues (asynchrony, partial participation, client dropouts) experimentally if possible.

**Questions:**

See Weaknesses.

---

> ### Author Response · Authors · 2025-11-21
> **Response to Reviewer h9q3 (1/3)**
>
> **W1: Larger-scale and more diverse experiment.**
>
> Thank you for your concern. The datasets used in our work are **widely adopted** in prior studies on federated bi-level optimization [1,2]. Due to time and computational resource constraints, we were unable to use a larger dataset.  Nevertheless, we incorporated an additional **hyperparameter optimization task**, which is the loss function tuning task, to further validate the effectiveness of the proposed method, as detailed **in Appendix D.2.** Our experimental results demonstrate that the proposed methods outperform the baseline approaches by a significant margin.
>
> [1] Tarzanagh, Davoud Ataee, et al. "Fednest: Federated bilevel, minimax, and compositional optimization." *International Conference on Machine Learning*. PMLR, 2022.
>
> [2] Yang, Yifan, Peiyao Xiao, and Kaiyi Ji. "Simfbo: Towards simple, flexible and communication-efficient federated bilevel learning." *Advances in Neural Information Processing Systems* 36 (2023): 33027-33040.
>
> ---
>
> **W2: Lacked related works.**
>
> Thanks for the helpful advice. We have added more related works and discussion in **Appendix B.**
>
> ---
>
> **W3: Near-critical $\delta$ behavior \& aggregator choice. Theory constrains $\kappa$ vs. $\delta$, but the paper lacks a systematic study when $\delta$ approaches the breakdown. Provide $\delta$ sweeps and compare $\kappa$ across aggregators to probe the near-bound regime.**
>
> Thank you for your concern.
>
> - To address this issue, we have included additional experiments for the case where $\delta$ approaches the breakdown point, as presented in **Appendix D.3.2 in Figure 2**. The figure illustrates the trend as the fraction of attackers increases, revealing a particularly **sharp decline** in accuracy when $\delta$ nears the breakdown threshold.
> - We conducted experiments with varying values of $\delta$ and compared the corresponding $\kappa$ across different aggregators. Since both Median and Krum select only a single gradient, their $\kappa$ values are not strongly affected by changes in $\delta$. In contrast, the $\kappa$ values of other robust aggregators increase as the number of attackers grows.
>
> | Aggregator | δ=5.0% | δ=10.0% | δ=15.0% | δ=20.0% | δ=25.0% | δ=30.0% | δ=35.0% | δ=40.0% | δ=45.0% |
> | --- | --- | --- | --- | --- | --- | --- | --- | --- | --- |
> | Median | 0.7285 | 0.6125 | 0.7249 | 0.5311 | 0.7184 | 0.7280 | 0.6757 | 0.7019 | 0.6163 |
> | CW-Med | 0.0811 | 0.0890 | 0.0921 | 0.0939 | 0.1041 | 0.1141 | 0.1135 | 0.1311 | 0.1754 |
> | TM | 0.0058 | 0.0120 | 0.0269 | 0.0328 | 0.0877 | 0.0838 | 0.2101 | 0.1643 | 0.4625 |
> | CW-TM | 0.0115 | 0.0269 | 0.0447 | 0.0545 | 0.0763 | 0.0916 | 0.0985 | 0.1144 | 0.1616 |
> | Krum | 0.7093 | 0.6125 | 0.6421 | 0.5311 | 0.6873 | 0.7418 | 0.6325 | 0.7103 | 0.6319 |
> | Multi-Krum | 0.0058 | 0.0120 | 0.0269 | 0.0310 | 0.0877 | 0.0524 | 0.2101 | 0.1971 | 0.3593 |
>
> Table: Comparison of $\kappa$ for different values of $\delta$ under the BF attack on the MNIST dataset with non-iid data ($\alpha = 0.1$).

---

> ### Author Response · Authors · 2025-11-21
> **Response to Reviewer h9q3 (2/3)**
>
> **W4:  Assumption strength & measurability.**
>
> Thank you for your concern.  Although Mean‑square smoothness and $(\zeta,B)$-heterogeneity are not easily measured in practice, the proposed algorithm **does not rely on the values of these coefficients**. They are used solely to support the **theoretical convergence analysis**, and the method itself operates without the need to estimate them during training.
>
> To provide a clearer connection between theory and practice, we additionally report **empirical estimates** of these quantities derived from **client gradient statistics**. These empirical values help illustrate how the theoretical constants relate to observable behavior in federated systems. Our analysis incorporates the following considerations:
>
> - A given set of client loss functions can satisfy the $(\zeta, B)$-heterogeneity condition **under many different valid pairs $(\zeta, B)$**. For example, if the losses satisfy $(\zeta, B)$-gradient dissimilarity, then they also satisfy $(\zeta, 2B)$-gradient dissimilarity [3]. This illustrates that **increasing $B$** generally allows for a **smaller corresponding $\zeta$**, meaning that multiple feasible $(\zeta, B)$ pairs may describe the same level of gradient variability. In our empirical estimation of $(\zeta, B)$-heterogeneity, we report **one representative choice** from these valid pairs for clarity and consistency.
> - To offer practical insight, we compute empirical estimates of **$(\zeta, B)$-heterogeneity**, **$\zeta$-heterogeneity**, and **mean-square smoothness $L$** across several model architectures and datasets. These values are obtained by averaging **pairwise gradient deviations** across clients under various levels of non-IID data heterogeneity (controlled by the Dirichlet parameter $\alpha$).
>
> | Model & Dataset | $\alpha=0.1$ | $\alpha=0.3$ | $\alpha=0.5$ | $\alpha=1.0$ | $\alpha=5.0$ | $\alpha=10.0$ |
> | --- | --- | --- | --- | --- | --- | --- |
> | $B$ | 3 | 2.5 | 1.5 | 1.4 | 1 | 0.3 |
> | $\zeta$ | 3.04 | 2.79 | 1.98 | 1.51 | 1.29 | 1.23 |
>
> Table: Empirical estimates of $(\zeta,B)$-heterogeneity on MNIST dataset using logistic regression model.
>
> | Model & Dataset | $\alpha=0.1$ | $\alpha=0.3$ | $\alpha=0.5$ | $\alpha=1.0$ | $\alpha=5.0$ | $\alpha=10.0$ | **IID** |
> | --- | --- | --- | --- | --- | --- | --- | --- |
> | logistic regression model+MNIST | 11.43 | 8.96 | 7.03 | 5.37 | 2.25 | 1.65 | 0.35 |
> | MLP+MNIST | 1.05 | 0.858 | 0.669 | 0.508 | 0.207 | 0.149 | 0.033 |
> | CNN+CIFAR10 | 0.844 | 0.671 | 0.523 | 0.400 | 0.163 | 0.117 | 0.034 |
>
> Table: Empirical estimates of $\zeta$-heterogeneity.
>
> | Model & Dataset | logistic regression model+MNIST | MLP+MNIST | CNN+CIFAR10 |
> | --- | --- | --- | --- |
> | $L$ | 1.466 | 6.778 | 2.65 |
>
> Table: Empirical estimates of $L$ mean‑square smoothness.
>
> [3] Allouah, Youssef, et al. "Robust distributed learning: Tight error bounds and breakdown point under data heterogeneity." *Advances in Neural Information Processing Systems* 36 (2023): 45744-45776.
>
> ---
>
> **W5: Has BILANTINE been tuned fairly for communication rounds, stepsizes, and batch sizes? Include ablations under equal communication budget or equal wall‑clock, and disclose full tuning tables.**
>
> Yes.  We use the same batch size $b = 64$ and the same server-side update scheme across all methods, and we run each algorithm until its performance stabilizes and converges. All approaches are trained for an identical number of server updates—**200 rounds for MNIST** and **1000 rounds for CIFAR-10**. Although BILANTINE incorporates a sub-loop update, it incurs higher communication **cost** because each server round includes 5 communication rounds for the Neumann-series approximation and inner-parameter updates. In contrast, BR-FedBi updates ${x, y, z}$ in parallel within each server-side round, resulting in a lower communication burden.
> To further support the fairness of comparison, we additionally report the **wall-clock running time** required by each method to reach **50%**, **60%**, **75%**, and **85%** of the final top-1 test accuracy. These measurements are included in **Appendix D.3**.
>
> |  | communication cost | server-side update (epoch)  | clock (s) |
> | --- | --- | --- | --- |
> | BILANTINE | 1600 | 200 | 7701.67 |
> | BR-FedBi | 600 | 200 | 3671.85 |
>
> Table: Comparison of the communication cost of BILANTINE and BR-FedBi on MNIST  dataset using Krum under ALIE attack.
>
> | Algorithm | Krum | TM | Median | RFA | CWMed | CWTM |
> | --- | --- | --- | --- | --- | --- | --- |
> | BILANTINE | 7701.42 | 7202.51 | 7229.27 | 7458.81 | 6590.10 | 6523.29 |
> | BR-FedBi | 3138.29 | 3139.92 | 3102.36 | 3132.67 | 3138.48 | 3126.39 |
> | BR-FedBiM | 3200.38 | 3148.26 | 3136.17 | 3173.29 | 3169.27 | 3153.02 |
> | BR-FedBiP | 3158.74 | 3166.83 | 3165.91 | 3207.01 | 3238.28 | 3203.91 |
>
> Table: The wall clock running time (s) for different methods on MNIST dataset under ALIE attack.

---

> > ### Author Response · Authors · 2025-11-21
> > **Response to Reviewer h9q3 (3/3)**
> >
> > **W6: Engineering clarifications. Clarify PAGE trigger probability $p$, mini-batch sizes $b$, momentum/stepsize coupling (theory ranges vs. actual values). Address practical FL issues (asynchrony, partial participation, client dropouts) experimentally if possible.**
> >
> > Thank you for the concerns. We summarize our hyperparameter choices and practical considerations as follows:
> >
> > - **Regarding the choice of $p$ for BR-FedBiP.**  The optimal trigger probability $p = 0.2$ is consistent with our theoretical guideline $p = \frac{b}{b' + b},$ where we set $b' = 256$ and $b = 64$. As indicated in the theory, increasing $p$ improves the likelihood of using a larger batch size, which reduces gradient variance but increases computational cost. In practice, we find that a moderate value such as **$p = 0.2$–$0.3$** achieves a good trade-off between performance and efficiency, and we adopt **$p = 0.2$** in the main experiments.
> >
> > | Attack Type | p = 0.1 | p = 0.2 | p = 0.3 | p = 0.5 | p = 0.8 |
> > | --- | --- | --- | --- | --- | --- |
> > | ALIE | 62.79 | 62.63 | 62.35 | 62.91 | 62.58 |
> > | BF | 59.80 | 60.25 | 60.83 | 60.12 | 60.84 |
> > | IPM | 57.84 | 58.89 | 57.51 | 57.95 | 57.67 |
> > | Random Noise | 62.30 | 61.92 | 62.46 | 62.46 | 62.61 |
> > | Worst | 57.84 | **58.89** | 57.51 | 57.95 | 57.67 |
> >
> > *Table: Performance of BR-FedBiP with different choices of $p$ on the CIFAR10 dataset using TM as the robust aggregation method.*
> >
> > - **Regarding the choice of $\beta$ for BR-FedBiM.** The best-performing momentum coefficient is **$\beta = 0.8$**, which is close to $1$ and aligns well with the theoretical recommendation $\beta = \sqrt{1 - L \eta},$ when the stepsize $\eta$ is small.
> >
> > | Attack | $\beta=0$ | $\beta=0.5$ | $\beta=0.7$ | $\beta=0.8$ | $\beta=0.9$ |
> > | --- | --- | --- | --- | --- | --- |
> > | ALIE | 61.31 | 63.64 | 64.03 | 63.84 | 63.37 |
> > | BF | 62.92 | 63.34 | 63.49 | 64.44 | 63.04 |
> > | IPM | 63.11 | 63.18 | 63.90 | 63.81 | 63.15 |
> > | RN | 42.45 | 63.74 | 63.59 | 63.93 | 63.40 |
> > | Worst | 42.45 | 63.18 | 63.49 | **63.81** | 63.04 |
> >
> > *Table: Performance of BR-FedBiM with different choices of $\beta$ on the CIFAR10 dataset using TM as the robust aggregation method.*
> >
> > - **Address practical FL issues:** To address practical federated learning challenges, we have added an additional experiment that evaluates **varying client participation rates**. This experiment, included in **Appendix D.3.2**, demonstrates the robustness of our method under partial participation, which indirectly captures common FL issues such as temporary client unavailability or dropouts.

---

> ### Comment · Reviewer_h9q3 · 2025-11-27
>
> Thanks for the clarification and the revision for the manuscript. All the weaknesses have been solved and the reviewer will update the score accordingly.
>
> ----
>
> Minor:
> There is two 'Decentralized stochastic bilevel optimization with improved per-iteration complexity' in the reference and it should be revised.

---

> > ### Author Response · Authors · 2025-11-27
> >
> > We thank the reviewer for pointing this out. We have now corrected the reference issue accordingly.

---

> > ### Author Response · Authors · 2025-11-29
> >
> > Dear AC,
> >
> > I would like to kindly point out that the reviewer was satisfied with the rebuttal and increased the score from 4 to 6. However, the score was later reverted.
> >
> > Authors

---

### Official Review · Reviewer_jwjE · 2025-10-31

**Soundness:** 2
**Presentation:** 3
**Contribution:** 3
**Rating:** 6
**Confidence:** 3

**Summary:**

This paper proposes BR-FedBi, a single-loop Byzantine-resilient federated bilevel optimization framework. By introducing an auxiliary variable and integrating Polyak’s momentum with the probabilistic gradient estimator (PAGE), the method achieves efficient hypergradient estimation, optimal Byzantine resilience, and sample complexity. Both theoretical analysis and experiments show strong empirical performance under adversarial settings.

**Strengths:**

1. The paper addresses an important and challenging problem, Byzantine-resilient bilevel optimization in federated settings, which is timely and underexplored.

2. The proposed single-loop design is elegant and significantly reduces computation and communication overhead compared to two-loop approaches.

3. Theoretical analysis is comprehensive, providing convergence guarantees with optimal sample complexity, supported by consistent empirical results.

**Weaknesses:**

1. The paper’s presentation could be clearer, especially in explaining how the auxiliary variable interacts with upper and lower levels in the single-loop update.

2. Experimental evaluation is somewhat limited; it would be more convincing to include larger-scale or more diverse federated settings.

3. The robustness analysis mainly relies on existing aggregation rules; more ablation on the aggregation mechanism itself would strengthen the contribution.

4. While the proposed single-loop design is interesting, the experimental section lacks direct comparison with other single-loop bilevel approaches, which would help isolate the benefit of the proposed structure itself.

**Questions:**

see weakness.

---

> ### Author Response · Authors · 2025-11-21
> **Response to Reviewer jwjE  (1/2)**
>
> > **Weakness 1: Clarity of the interaction between the auxiliary variable and the bilevel updates**
> >
>
> Thank you for your concern. We have added more detail as presented in **Appendix C** of the revised manuscript.
>
> ---
>
> > **Weakness 2: Experimental evaluation is somewhat limited; it would be more convincing to include larger-scale or more diverse federated settings.**
> >
>
> Thank you for your concern. The datasets used in our work are **widely adopted** in prior studies on federated bi-level optimization [1,2].  Due to limitations in time and computational resources, we were unable to employ a larger dataset.  Nevertheless, we incorporated an additional **hyperparameter optimization task**, which is the loss function tuning task, to further validate the effectiveness of the proposed method, as detailed **in Appendix D.2.** Our experimental results demonstrate that the proposed methods outperform the baseline approaches by a significant margin.
>
> [1] Tarzanagh, Davoud Ataee, et al. "Fednest: Federated bilevel, minimax, and compositional optimization." *International Conference on Machine Learning*. PMLR, 2022.
>
> [2] Yang, Yifan, Peiyao Xiao, and Kaiyi Ji. "Simfbo: Towards simple, flexible and communication-efficient federated bilevel learning." *Advances in Neural Information Processing Systems* 36 (2023): 33027-33040.
>
> ---
>
> > **Weakness 3:  More ablations on the aggregation mechanism itself would strengthen the contribution.**
> >
>
> Thank you for this valuable suggestion. In response, we have expanded our empirical analysis to more thoroughly investigate the role of the aggregation mechanism within our framework. Specifically, we conducted additional ablation studies that systematically evaluate how different robust aggregation rules influence the optimization of each variable $\{x, y, z\}$ in the bilevel formulation. These experiments examine the **sensitivity** of our proposed algorithms to the choice of aggregator at each level and quantify how various aggregation strategies affect stability, convergence behavior, and robustness under adversarial conditions.
>
> In our framework, the variables $\{x, y, z\}$ play **distinct functional roles**:
>
> - **$x$** and **$y$** *directly* modify model parameters, whereas
> - **$z$** influences the *hypergradient estimation* and indirectly affects the model parameters.
>
> Based on our empirical observations, the sensitivities of these variables to different aggregation rules follow the trend: **$x > z$** and **$x > y$**. More concretely:
>
> - **$x > y$**: In our experiments, we partition CNN parameters such that $y$ contains a subset of layers while $x$ contains the remaining (and larger) portion. Because $x$ encompasses more layers, perturbations to $x$ have a more pronounced effect on model performance. We note, however, that this relative influence may vary depending on model architecture and parameter assignment.
> - **$x > z$**: Since $z$ only influences the *estimated* hypergradient and does not directly modify the model parameters, changes to $z$ generally have a weaker effect on performance compared to $x$.
> - The relative influence between **$y$** and **$z$** is more nuanced and depends strongly on the specific experimental setup, architecture, and the degree to which the inner-level updates interact with the outer-level hypergradient.
>
> This extended analysis provides a clearer understanding of how aggregation mechanisms interact with each component of the bilevel optimization process, thereby strengthening the contribution and interpretability of our framework.
>
> |  Aggregator of $z$ | Krum | RFA | Median | TM | CWMed | CWTM |
> | --- | --- | --- | --- | --- | --- | --- |
> | $x,y$ with  Krum | 63.49 | 63.74 | 64.65 | 62.67 | 63.04 | 62.08 |
> | $x,y$ with  CWTM | 48.09 | 48.81 | 49.59 | 48.26 | 48.27 | 48.84 |
>
> *Table: Performance of BR-FedBi on the CIFAR-10 dataset using different robust aggregation strategies for parameters $x$, $y$, and $z$.*
>
> |  Aggregator of $x$ | Krum | RFA | Median | TM | CWMed | CWTM |
> | --- | --- | --- | --- | --- | --- | --- |
> | $z,y$ with   Krum | 63.49 | 52.02 | 60.41 | 61.26 | 58.50 | 48.99 |
> | $z,y$ with   CWTM | 63.04 | 51.74 | 59.66 | 58.87 | 54.18 | 48.84 |
>
> *Table: Performance of BR-FedBi on the CIFAR-10 dataset using different robust aggregation strategies for parameters $x$, $y$, and $z$.*

---

> > ### Author Response · Authors · 2025-11-21
> > **Response to Reviewer jwjE (2/2)**
> >
> > > **Weakness 4:  Lack of direct comparison with other single-loop bilevel approaches.**
> > >
> >
> > Thank you for raising this important point. To directly compare with other single-loop bilevel optimization approaches, we have incorporated a new baseline into our experimental evaluation. Specifically, we compare our method with an alternative single-loop design in which the Hessian inverse is approximated using a single-step **Neumann series expansion**, rather than being estimated via the auxiliary variable $z$ through gradient descent to solve the associated linear system. This Neumann-based approximation preserves the single-loop update structure and therefore serves as an appropriate counterpart for isolating the contribution of our proposed design. As demonstrated in the newly added experiments, our method consistently outperforms this Neumann-series-based single-loop alternative.
> >
> > |  | ALIE | BF | IPM | RN | Worst |
> > | --- | --- | --- | --- | --- | --- |
> > | Krum | 58.87 | 57.07 | 54.49 | 58.60 | 54.49 |
> > | TM | 80.43 | 56.29 | 72.87 | 74.30 | 56.29 |
> > | Median | 81.12 | 64.08 | 61.22 | 76.87 | 61.22 |
> > | RFA | 79.84 | 18.81 | 25.45 | 76.82 | 18.81 |
> >
> > *Table: Performance of single-loop algorithm with **Neumann series** on the MNIST dataset using different robust aggregation strategies.*

---

### Official Review · Reviewer_aw31 · 2025-10-31

**Soundness:** 3
**Presentation:** 3
**Contribution:** 3
**Rating:** 6
**Confidence:** 4

**Summary:**

this paper studies the vulnerability of federated bilevel optimization against Byzantine attacks and the inefficiency of existing sub-loop-based defenses by proposing a family of single-loop Byzantine-resilient algorithms (FBO). The proposed algorithms leverage auxiliary variables, Polyak’s momentum, and variance reduction to balance robustness, computational and communication efficiencies.  The experiments on a set of hyper-representation tasks show the algorithms outperform existing tasks under four attacks and six robust aggregators. Authors also give theoretic analysis regarding algorithm-independent lower bound for FBO under Byzantine attacks.

**Strengths:**

The paper studies two important problems in federated bilevel optimization including Byzantine vulnerability and computational inefficiency of sub-loop defenses.   It also provides first-of-its-kind theoretical guarantees for Byzantine-resilient federated bilevel optimization.  Experimental results have shown consistently that the proposed method outperform the existing methods in the literature.

**Weaknesses:**

Experiments use fixed attacker ratios and static attack types, but real-world Byzantine attacks often involve dynamic attacker counts e.g., sudden spikes in faulty clients or adaptive attack strategies. The paper does not test how algorithms adapt to such changes.

**Questions:**

Can authors elaborate the potential applications of the proposed method in real-world scenarios ?

---

> ### Author Response · Authors · 2025-11-21
> **Response to Reviewer aw31 (1/2)**
>
> > **Weaknesses: Lack of study of sudden spikes in faulty clients and dynamic attacks.**
> >
>
> Thank you for raising this important concern. In response, we have conducted additional experiments and analyses to evaluate our method under **adaptive and dynamic attack strategies** and **sudden attacks**.
>
> **Experiments with Adaptive and Dynamic Attack Strategies**
>
> - We have **added new experiments to evaluate the adaptive attack strategies**. The adaptive attacks we consider are tailored to specific aggregation rules (e.g., Krum, Trimmed Mean). In each training round, the attacker adjusts its behavior by solving an optimization problem, making the attacks dynamic and more challenging than fixed-pattern Byzantine behaviors.
> - To defend against such adaptive and dynamic attacks, we also study ERR and LFR [1], which can handle fluctuating attacker counts and evolving adversarial strategies. We integrate these methods with existing aggregation rules to examine their combined robustness.
>
> Our experimental results show that simple aggregation rules alone are ineffective against these tailored, dynamic attacks. However, when augmented with ERR and LFR, they can effectively defend against such adversarial behavior. Moreover, incorporating ERR and LFR,  our **single-loop algorithm** achieves significantly better performance than baseline methods under dynamic attack scenarios.
>
> |  | Krum+ERR | Krum+ LFR | Krum  |
> | --- | --- | --- | --- |
> | BR-FedBi | 80.97% | 81.38% | 9.91% |
> | BR-FedBiM | 81.25% | 82.63% | 9.91% |
> | BR-FedBiP | 81.16% | 81.49% | 9.91% |
> | BILANTINE | 71.27% | 71.89% | 10.20% |
>
> *Table: Performance of the proposed algorithms and baseline under adaptive attack against Krum on the MNIST dataset.*
>
> |  | TM+ ERR | TM+LFR | TM |
> | --- | --- | --- | --- |
> | BR-FedBi | 80.35% | 80.92% | 9.91% |
> | BR-FedBiM | 81.18% | 81.27% | 9.91% |
> | BR-FedBiP | 80.46% | 80.99% | 9.91% |
> | BILANTINE | 66.26% | 66.48% | 10.20% |
>
> *Table: Performance of the proposed algorithms and baseline under adaptive attack against TM on the MNIST dataset.*
>
> [1] Fang, Minghong, et al. "Local model poisoning attacks to Byzantine-Robust federated learning." *29th USENIX security symposium (USENIX Security 20)*. 2020.
>
> ---
>
> **Sudden spikes in faulty clients or sudden attacks .**
>
> We consider a scenario in which, for every 10 global rounds, 6 of 10 clients are malicious in the first two rounds, while the remaining 8 rounds have only honest clients, simulating a sudden attack.
> This setting mimics a short-term spike in Byzantine behavior. Under such conditions, **standard robust aggregation methods are unable to provide adequate protection**. To enhance resilience, we integrate the robust aggregator with **client-side defense mechanisms**, including **LeadFL** [3].
> For clarity, we evaluate model robustness using two standard metrics:
>
> - Main Task Accuracy (MA):  The main task accuracy is the fraction of correctly classified samples of the model on test data without the trigger.
> - Backdoor Accuracy (BA): The backdoor accuracy  qualifies how successful the attacker is in integrating  a backdoor into the model.
>
> As shown in the following  table, **sub-loop algorithms increase the frequency of communication between clients and the server**, which inadvertently expands the opportunities for adversarial interference. During sudden attacks, this elevated communication creates more windows for malicious clients to exploit, resulting in **higher backdoor success rates and degraded overall performance.**
>
> |  | Server Defense | MA | BA Avg | BA Final |
> | --- | --- | --- | --- | --- |
> | BR-FedBi | Krum | 82.93 | 26.33% | 22.38% |
> | BR-FedBi | Krum + LeadFL | 82.35 | 22.34% | 19.34% |
> | BR-FedBi | TM | 82.38 | 27.01% | 23.49% |
> | BR-FedBi | TM +LeadFL | 81.97 | 21.20% | 20.94% |
> | BILANTINE | Krum | 54.71 | 32.30% | 30.57% |
> | BILANTINE | Krum + LeadFL | 52.28 | 30.49%  | 25.39% |
> | BILANTINE | TM | 65.28 | 37.45% | 36.28% |
> | BILANTINE | TM +LeadFL | 63.29 | 31.02% | 30.22% |
>
> *Table: Performance of algorithms combined with client-side defense under sudden attacks on the MNIST dataset.*
>
> [2] Sun, Jingwei, et al. "Fl-wbc: Enhancing robustness against model poisoning attacks in federated learning from a client perspective." *Advances in neural information processing systems* 34 (2021): 12613-12624.
>
> [3] Zhu, Chaoyi, Stefanie Roos, and Lydia Y. Chen. "LeadFL: Client self-defense against model poisoning in federated learning." *International Conference on Machine Learning*. PMLR, 2023.

---

> > ### Author Response · Authors · 2025-11-21
> > **Response to Reviewer aw31 (2/2)**
> >
> > > **Questions: Real-world application.**
> > >
> >
> > We thank the reviewer for this insightful question. We are happy to further clarify real-world scenarios where **Byzantine-robust federated bilevel learning** is particularly valuable. Bilevel optimization frequently appears in applications such as **hyperparameter optimization**, **personalized or few-shot meta-learning**, **model selection in resource-limited environments, etc**. In practice, the data required for both the inner- and outer-level problems are often **distributed across devices, institutions, or organizations**, and cannot be centralized due to privacy regulations, ownership constraints, or communication limits. Federated learning, therefore, offers a natural framework to perform these bilevel tasks **without sharing raw data**.
> >
> > At the same time, real-world federated systems are susceptible to **faulty or malicious (Byzantine) clients**, whose behaviors can significantly disrupt the bilevel optimization procedure. This makes robustness essential. Our approach directly addresses this challenge by enabling bilevel optimization to remain reliable even when a non-trivial fraction of clients behaves adversarially.

---

### Author Response · Authors · 2025-12-03
**Message to Reviewers and AC**

Dear reviewers and AC,

Thank you for your thoughtful comments, questions, and suggestions. We have addressed the individual reviewer feedback in our detailed responses. In this global response, we will highlight and address several issues that are common across all reviews. **Initially, the scores were 6, 6, and 4. After our rebuttal, the scores improved to 6, 6, and 6; however, the scores were later reverted.**

### **New experiments:**

We address the reviewers' concerns with the following updates and improvements, and submit an improved manuscript highlighted in blue:

- New study of sudden spikes in faulty clients and dynamic attacks.
- New experiment on additional hyperparameter optimization task **(Appendix D.2).**
- Additional ablation studies that systematically evaluate how different robust aggregation rules influence the optimization of each variable.
- New direct comparison with other single-loop bilevel approaches.
- Near-critical  $\delta$ behavior & aggregator choice **(Appendix D.3.2 in Figure 2)**.
- Provide $\delta$ sweeps and compare $\kappa$  across aggregators to probe the near-bound regime.
- Experiment on measurability of the assumptions.
- Clarification of the communication rounds, stepsizes, batch sizes and wall‑clock (**Appendix D.3).**
- Experiment regarding the choice of $p$ for BR-FedBiP and the choice of $\beta$ for BR-FedBiM.
- Additional experiment that evaluates varying client participation rates (**Appendix D.3.2**).

### **Other updates on the manuscript**

- Add more related work **(Appendix B).**
- Add more Clarity of the interaction between the auxiliary variable and the bilevel updates **(Appendix C).**
- Correct minor reference issues.

Thank you again for the time invested in reviewing our work.

Best regards,

Authors

---

### Meta-Review · Area_Chair_9hYe · 2026-01-04

**Summary:**

This paper studies the Byzantine-resilient federated bilevel optimization problem. It first establishes a lower bound for this problem and then develops three algorithms based on stochastic gradients, momentum, and variance-reduced gradients, along with their corresponding convergence rates. Experimental results are also provided to demonstrate the performance of the three algorithms.

The reviewers acknowledge the contribution that this work is the first to study Byzantine issues in federated bilevel optimization. At the same time, they raised several concerns regarding the evaluation. For example, the proposed algorithms were evaluated only in small-scale settings, lacking experiments on large-scale problems, and the influence of hyperparameters on performance was not fully explored. In the rebuttal, the authors provided additional experimental results to address these concerns.

Considering that this paper is the first effort in this direction and the fact that the authors have addressed most of the concerns in the rebuttal, I recommend acceptance.

**Reviewer Concerns:**

Most concerns are about the evaluation.  For example, the proposed algorithms were evaluated only in small-scale settings, lacking experiments on large-scale problems, and the influence of hyperparameters on performance was not fully explored. In the rebuttal, the authors provided additional experimental results to address these concerns.

**Reviewer Scores:**

The reviewer who initially gave a negative evaluation has decided to raise the initial score.

---

### Decision · Program_Chairs · 2026-01-26

Accept (Poster)